# MAS³AC: A Learning Framework for General Multi-Agent Safe and Stable Control with State-Wise Guarantees

## Abstract

Ensuring both safety and stability is essential when deploying reinforcement learning to control safety-critical systems, including multi-agent ones. However, many existing safe multi-agent reinforcement learning (MARL) studies focus only on cooperative tasks and adopt the constrained Markov decision process setting that only enforces expectation-based constraints, limiting their applicability to domains requiring strict state-wise guarantees, while stability remains largely under-explored. To address these challenges, we propose **M**ulti-**A**gent **S**afe and **S**table **S**oft **A**ctor-**C**ritic (**MAS³AC**), a model-free framework that incorporates state-wise safety and stability constraints into MARL for general multi-agent tasks where each agent has its own objective. Our approach uses neural barrier functions to enforce safety, supported by a theoretical analysis of its convergence to a feasible local Nash equilibrium. It then uses the concept of input-to-state stability to guarantee stability for the multi-agent system, together with an analysis of the issue of infeasibility arising from conflicting state-wise safety and stability requirements. Empirically, we introduce a suite, spanning both cooperative tasks with global information and non-cooperative tasks with local observation, for benchmarking safe and stable MARL algorithms. Experimental results show that MAS³AC consistently achieves a favorable balance between reward maximization and constraint satisfaction, delivering competitive or superior rewards while maintaining fewer safety violations compared to baselines on benchmarks.

## 1 Introduction

Multi-agent reinforcement learning (MARL) has attracted significant attention due to its many successes in solving decision-making tasks for multi-agent systems (MAS) across diverse domains, such as video games (Samvelyan et al., 2019). Encouraged by these achievements, researchers have started to focus on its application to real-world tasks such as autonomous vehicles (Shalev-Shwartz et al., 2016) and power systems (Biagioni et al., 2022), where agents must operate under certain constraints. Among previous studies, cooperative tasks that require agents to be controlled to accomplish complex tasks collaboratively while obeying local and/or system-level constraints have received great attention (Gu et al., 2023; Zhao et al., 2023c). To efficiently train policies to achieve the team goal under constraints, a single joint reward is typically required for each agent in the team, and many studies impose global state observability (Zhao et al., 2023c; Li & Azizan, 2024) to enable centralized training (CT) (Gronauer & Diepold, 2022; Peng et al., 2021). Thanks to recent studies by Zhang et al. (2024), Ying et al. (2023) and Zhang et al. (2025), research on decentralized constrained cooperative MARL has advanced; however, less attention has been paid to general MARL tasks (Lowe et al., 2017), where each agent can have its own reward rather than sharing a single joint team reward, and access to global information may no longer hold due to the possible competition among agents, even if many real-world domains are naturally aligned with this general setting.

Among all constraints, safety has received the greatest focus in recent years (Gu et al., 2024). However, drawing on single-agent safe RL methods, many safe MARL algorithms so far formulate the problem based on the constrained Markov decision process (CMDP) (Gu et al., 2023; Zhao et al., 2023c; Zhang et al., 2024; Ding et al., 2023; Lu et al., 2021) that only requires the expected cumulative discounted safety costs along trajectories to be smaller than one pre-defined threshold. In

addition, the traditional dual ascent update adopted in Gu et al. (2023); Zhang et al. (2024) may suffer from inefficient training, as described in Zhao et al. (2023c), which can result in poor convergence. Since real-world multi-agent safety-critical tasks usually require efficient training for zero safety violations at every step to avoid catastrophic consequences, rather than merely imposing constraints on expected values (Zhao et al., 2023a), more advanced methods remain in urgent demand.

Maintaining stability is also crucial in many tasks, and the Lyapunov framework has been applied (Chang et al., 2019; Yang et al., 2024). RL has been integrated with Lyapunov functions to help guarantee stability for single-agent tasks (Han et al., 2020), but this relies on a final unique stationary distribution, which may limit its extension to MARL tasks where non-stationarity is a well-known issue that may prevent the learning from converging to a final fixed joint policy. A small-gain theorem-based learning method has been proposed to develop stable controllers for networked systems (Zhang et al., 2023). However, this approach requires knowledge of system dynamics for gradient computation or one-step simulation, which may limit the applicability of the algorithm to tasks with sophisticated dynamics (Wang et al., 2023). Further, when using safety and stability conditions separately, the simultaneous enforcement of both raises the challenging issue of infeasibility (Tan & Dimarogonas, 2024), since the two types of constraints may conflict under certain circumstances.

**Contributions**: Here we address these challenges by proposing, to the best of our knowledge, the first framework that helps to obtain a joint controller with post-convergence state-wise safety and stability guarantees in general MARL tasks under unknown dynamics. The main contributions are :

- We develop a method to train neural barrier certificates for multi-agent systems, which is then integrated into general model-free MARL tasks with state-wise safety requirements to obtain safe and locally optimal controllers via a modified dual ascent update, whose convergence to a safe local Nash equilibrium is theoretically proved under certain assumptions.

- Based on the concept of input-to-state stability (ISS) (Sontag & Wang, 1995; Jiang & Liu, 2018), we establish the local condition for each agent to achieve stable control of general networked MAS, supported by theoretical proofs. In particular, we show that ISS can be enforced using only each agent's own state and a few scalars from others, providing a theoretical justification for decentralized training with minimal communication. We further study the issue of infeasibility under state-wise safety and stability constraints, showing that this issue does not prevent the framework from finding safe and stable controllers for MAS.

- We propose **M**ulti-**A**gent **S**afe and **S**table **S**oft **A**ctor-**C**ritic (**MAS**$^3$**AC**), a unified framework to solve model-free multi-agent safe and stable control problems under centralized or decentralized training. We also provide environments for benchmarking safe and stable MARL algorithms. Empirical evaluations demonstrate that MAS$^3$AC consistently and efficiently achieves a strong balance between reward maximization and constraint satisfaction, outperforming baselines in both cooperative and non-cooperative settings.

## 2 RELATED WORK

**Safe single-agent RL**: Safe single-agent RL has become a focal point in recent research (Gu et al., 2024), and there are mainly two genres of formulations. The first uses CMDP settings (Altman, 2021), and techniques, including trust region policy optimization (Achiam et al., 2017), and traditional dual ascent update (Paternain et al., 2019), have been proposed. However, these approaches enforce safety constraints as cumulative discounted costs along trajectories, whereas many real-world applications require enforcing safety at each state (Zhao et al., 2023b). This motivates the second formulation, which relies on state-wise constraints, and concepts like model predictive controller (Tian et al., 2024), and control barrier function (CBF) (Cheng et al., 2019; Emam et al., 2022; Chriat & Sun, 2023; Yang et al., 2023) have been employed to enable safety in single-agent RL.

**Safe multi-agent RL**: Safe MARL is a research area that aims to learn safe policies for MAS, and most studies so far either rely on reward engineering to design the objective function to impose safety constraints (Chen et al., 2017; Semnani et al., 2020), or formulate the problem based on the CMDP (Zhang et al., 2024; Ding et al., 2023; Ying et al., 2023; Lu et al., 2021) extended from single-agent cases and thus employ similar methods like trust region-based optimization, e.g., MAFOCOPS (Zhao et al., 2023c), and traditional dual ascent update like MAPPO-Lagrangian (Gu et al., 2023). To address state-wise safety constraints, several methods using neural certificates (Li & Azizan, 2024;

Zhang et al., 2025) have been proposed for multi-agent tasks. However, these approaches either assume access to system dynamics (Qin et al., 2021), focus on tasks with a joint team reward/cost (Zhang et al., 2025), or consider only the centralized training scheme (Li & Azizan, 2024).

**Stable control via learning and its integration with safety**: Compared to safety, stability in RL, particularly in multi-agent settings, remains relatively unexplored. Yao et al. (2025) apply Lyapunov functions in RL to provide stability guarantees; however, the requirement of a final unique stationary distribution may not easily be satisfied in MARL since the non-stationarity may make it difficult for the update to converge towards a final fixed joint policy. Beyond RL, some works propose stable neural network-based controllers for both single-agent settings (Chang et al., 2019; Yang et al., 2024) and multi-agent ones (Zhang et al., 2023), but they require system dynamics. To ensure both safety and stability, some studies like Dawson et al. (2022); Du et al. (2023) use control Lyapunov barrier functions. However, these methods still rely on system dynamics, or CMDP, highlighting the need for methods that can support automatic training in multi-agent tasks with state-wise constraints.

**Challenges**: Many methods applying pre-defined control barrier function for safe single-agent RL require a control-affine nominal model of the system (Emam et al., 2022), or further restrict the CBF to be affine with respect to the state (Cheng et al., 2019). Meanwhile, although neural barrier certificates have also shown promising results in reducing safety violations, they may also rely on system models (Dawson et al., 2022), which could limit their applicability. In contrast, the neural barrier certificates used in our work can be easily integrated into RL training in a principled way without demanding system dynamics. Further, our work differs from these existing safe multi-agent RL studies based on CMDP (Altman, 2021) and/or developed for cooperative tasks (Gu et al., 2023): first, our method considers state-wise safety requirements and can be applied to general MARL settings beyond cooperative ones. Second, we additionally incorporate stability conditions naturally into the training process, yielding a more comprehensive framework for learning safe and stable controllers for multi-agent tasks. Third, the application of multiple constraints can lead to the issue of infeasibility (Reis et al., 2020), which requires further study in learning-based settings. Our work provides a theoretical analysis of this issue for MARL. Empirical results demonstrate that our algorithm yields consistently high rewards with very few safety violations compared to baselines on both cooperative and non-cooperative MARL tasks with either centralized or decentralized training.

## 3 PROBLEM FORMULATION AND PRELIMINARIES

**Multi-Agent Reinforcement Learning**: In this study, we consider the MARL problem represented by a tuple $\mathcal{M} = (\mathcal{N}, \mathcal{X}, \mathcal{U}, f, \boldsymbol{R}, \mathcal{O}, \gamma)$. Here, $\mathcal{N} = \{1, \ldots, N\}$ denotes the set of $N > 1$ interacting agents, $\mathcal{X} = (\mathcal{X}^1, \mathcal{X}^2, \ldots, \mathcal{X}^N)$ and $\mathcal{U} = (\mathcal{U}^1, \mathcal{U}^2, \ldots, \mathcal{U}^N)$ denote the joint state and action spaces, with $\mathcal{O} = (\mathcal{O}^1, \mathcal{O}^2, \ldots, \mathcal{O}^N)$ where $\mathcal{O}^i$ is the local observation space of agent $i$. The system dynamics $f : \mathcal{X} \times \mathcal{U} \to \mathcal{X}$ is assumed to be unknown. $\boldsymbol{r} = (r^1, r^2, \ldots, r^N)$ represents the reward functions, with $r^i$ being the individual reward of agent $i$. In this work, we consider general settings without imposing specific requirements on $r^i, \forall i \in \mathcal{N}$. $\gamma \in (0, 1)$ is the discount factor. At each timestep $t$, every agent $i$ selects an action $u_t^i$ according to its policy $\pi^i$, and the joint action $\boldsymbol{u}_t = (u_t^1, u_t^2, \ldots, u_t^N)$ is generated under the joint policy $\boldsymbol{\pi} = \prod_{i=1}^N \pi^i \in \boldsymbol{\Pi}$, where $\boldsymbol{\Pi}$ denotes the set of all joint policies. $\boldsymbol{\pi}^{-i}$ denotes the joint policy of all agents except $i$. When global information is available, the joint state $\boldsymbol{x}_t = (x_t^1, x_t^2, \ldots, x_t^N)$ serves as the input to every $\pi^i$. For tasks where only local observations are available, we consider cases in which interconnections among agents are represented by a graph $\mathcal{G} = (\mathcal{N}, \mathcal{E})$, where each node corresponds to an agent. A directed edge $(j, i) \in \mathcal{E}$ indicates that agent $j$ directly influences agent $i$ through a class-$\mathcal{K}$ gain function $\delta_{ij} \in \mathcal{K}$; otherwise, no direct influence exists from agent $j$ to agent $i$, and $\delta_{ij} = 0$. In this setting, the local observation of agent $i$ at timestep $t$, denoted $o_t^i$, is used as the input to its individual policy $\pi^i$.

**Safety and Stability**: For each agent $i \in \mathcal{N}$, when global information is available [1], the state-wise safety requirement is $b^i(\boldsymbol{x}_t) \geq 0, \forall t \geq 0$, where $b^i(\boldsymbol{x}_t)$ is a barrier signal measuring the safety of agent $i$ at the state $\boldsymbol{x}_t$. Based on this notation, the following subspaces are defined.

---

[1]For convenience and clarity, we present the algorithm design and analysis for safety under global information. In practice, the algorithm can also be applied to tasks with only local observations, such as inter-agent collision avoidance, where observing neighbors suffices to maintain safety. Related analyses can be found in Zhang et al. (2024); Ying et al. (2023); Zhang et al. (2025). For stability, local observations are used, and theoretical analysis and empirical results under this setting are reported in Section 4.2 and Section 5, respectively.

**Definition 1** *The overall state space $\mathcal{X}$ can be partitioned into: (1). $\mathcal{X}_{vio}$: The set of states $\boldsymbol{x}_t$ where $\exists i \in \mathcal{N}$ such that $b^i(\boldsymbol{x}_t) < 0$. (2). $\mathcal{X}_{irrec}$: The set of states $\boldsymbol{x}_0 \notin \mathcal{X}_{vio}$, but under any joint controller $\boldsymbol{\pi} \in \boldsymbol{\Pi}$, the trajectory starting from $\boldsymbol{x}_0$ will enter $\mathcal{X}_{vio}$ at least once with a positive probability, i.e., $\forall \boldsymbol{\pi} \in \boldsymbol{\Pi}$, $\exists t > 0$ such that $Pr(\boldsymbol{x}_t \in \mathcal{X}_{vio}) = Pr(\exists i \in \mathcal{N}, b^i(\boldsymbol{x}_t) < 0) > 0$. (3). $\mathcal{X}_{unsafe} = \mathcal{X}_{vio} \cup \mathcal{X}_{irrec}$ is the unsafe set, and $\mathcal{X}_{safe} = \mathcal{X} \setminus \mathcal{X}_{unsafe}$ is the safe set.*

Here, we consider tasks where there exist control policies under which trajectories starting from any initial state do not enter $\mathcal{X}_{vio}$ with a positive probability. Otherwise, no valid controller can maintain safety for the MAS. In addition to safety, ensuring stability for networked MAS is also critical, as it guarantees convergence of the controlled MAS to the desired equilibrium $\boldsymbol{x}_{desired}$, thereby enabling successful completion of tasks like reach-and-avoid. Without loss of generality, we set $\boldsymbol{x}_{desired} = \boldsymbol{0}$, the zero vector with the same dimension as $\boldsymbol{x}$, since a simple translation can be applied whenever an equilibrium is specified for a given task. Then, for stabilization tasks, the condition $\lim_{t \to \infty} ||\boldsymbol{x}_t|| = 0$ should hold to ensure stability of the networked MAS. Based on all these definitions, here we define:

**Problem 1 (Stable and Safe Control Problem for MAS)** *For a MAS represented by $\mathcal{M}$, given a desired equilibrium $\boldsymbol{x}_{desired} \in \mathcal{X}_{safe}$ and an initial set $\mathcal{X}_0 \subseteq \mathcal{X}_{safe}$, find a joint policy $\boldsymbol{\pi} \in \boldsymbol{\Pi}$ producing a sequence $\{\boldsymbol{u}_t\}_{t \geq 0}$ such that the state sequence $\{\boldsymbol{x}_t\}_{t \geq 0}$ satisfies: (1). **Safety:** $\forall t \geq 0$, $Pr(\boldsymbol{x}_t \in \mathcal{X}_{vio}) = Pr(\exists i \in \mathcal{N}, b^i(\boldsymbol{x}_t) < 0) = 0$. (2). **Stability:** $\lim_{t \to \infty} ||\boldsymbol{x}_t - \boldsymbol{x}_{desired}|| = 0$. Here, $\mathcal{X}_{vio}$ and $\mathcal{X}_{safe}$ are as defined in Definition 1, and $b^i(\boldsymbol{x}_t)$ is the barrier signal of agent $i$ at $\boldsymbol{x}_t$.*

## 4 METHODOLOGY

### 4.1 STATE-WISE SAFETY VIA NEURAL BARRIER CERTIFICATES FOR GENERAL MARL

In this section, we investigate the state-wise safety for MAS. Since it is natural to update each individual policy $\pi^i$ to satisfy the state-wise safety requirement given by $b^i$, inspired by CBFs, we aim to construct a learned barrier certificate for each agent $i$, and update agent $i$'s policy jointly with this learned barrier certificate. We begin with the following definitions and lemma:

**Definition 2** *For the update of agent $i$, similar to Definition 1, the overall state space can be divided into the following subspaces: (1). $\mathcal{X}_{vio}^{\boldsymbol{\pi},i}$: The set of states $\boldsymbol{x}_t$ where $b^i(\boldsymbol{x}_t) < 0$. (2). $\mathcal{X}_{irrec}^{\boldsymbol{\pi},i}$: The set of states $\boldsymbol{x}_0$ where $b^i(\boldsymbol{x}_0) \geq 0$, but from which the trajectory will reach a state with $b^i(\boldsymbol{x}_t) < 0$ at least once with a positive probability under the current joint policy $\boldsymbol{\pi}$, i.e., $\exists t > 0$ such that $Pr(b^i(\boldsymbol{x}_t) < 0) > 0$ under the controller $\pi^i$ when all other agents follow $\boldsymbol{\pi}^{-i}$. (3). $\mathcal{X}_{unsafe}^{\boldsymbol{\pi},i} = \mathcal{X}_{vio}^{\boldsymbol{\pi},i} \cup \mathcal{X}_{irrec}^{\boldsymbol{\pi},i}$ is the unsafe set, and $\mathcal{X}_{safe}^{\boldsymbol{\pi},i} = \mathcal{X} \setminus \mathcal{X}_{unsafe}^{\boldsymbol{\pi},i}$ is the safe set for agent $i$. i.e., for every $\boldsymbol{x}_0 \in \mathcal{X}_{safe}^{\boldsymbol{\pi},i}$, under the joint policy $\boldsymbol{\pi}$, $Pr(b^i(\boldsymbol{x}_t) < 0) = 0$ for all $t \geq 0$, meaning the safety of agent $i$ is maintained under $\pi^i$ when other agents follow $\boldsymbol{\pi}^{-i}$.*

**Definition 3** *$H_{\boldsymbol{\pi}}^i$ is a barrier certificate for the agent $i$ if: (1). $\forall \boldsymbol{x}_t \in \mathcal{X}_{unsafe}^{\boldsymbol{\pi},i}$, $H_{\boldsymbol{\pi}}^i(\boldsymbol{x}_t) < 0$. (2). $\forall \boldsymbol{x}_t \in \mathcal{X}_{safe}^{\boldsymbol{\pi},i}$, $H_{\boldsymbol{\pi}}^i(\boldsymbol{x}_t) \geq 0$. (3). $\forall \boldsymbol{x}_t \in \mathcal{X}_{safe}^{\boldsymbol{\pi},i}$, the certificate value satisfies $H_{\boldsymbol{\pi}}^i(\boldsymbol{x}_{t+1}) - H_{\boldsymbol{\pi}}^i(\boldsymbol{x}_t) \geq -\epsilon H_{\boldsymbol{\pi}}^i(\boldsymbol{x}_t)$ where $\epsilon \in (0,1)$ and $\boldsymbol{x}_{t+1}$ is the next state induced by $\boldsymbol{\pi}$.*

**Informal Lemma 1** *Define the barrier signal $b^i$ as:*

$$b^i(\boldsymbol{x}_t) = \begin{cases} d & \text{if } \boldsymbol{x}_t \notin \mathcal{X}_{vio}^{\boldsymbol{\pi},i}, \\ D & \text{if } \boldsymbol{x}_t \in \mathcal{X}_{vio}^{\boldsymbol{\pi},i}, \end{cases} \tag{1}$$

*where $d = 0$ and $D < 0$ are constants. Under certain assumption, the function $H_{\boldsymbol{\pi}}^i(\boldsymbol{x}_t) = \mathbb{E}_{\tau \sim \boldsymbol{\pi}}\left[\sum_{k=0}^{\infty} \gamma^k b^i(\boldsymbol{x}_{t+k})\right]$, where $\gamma \in (0,1)$ is the discount factor and $\tau = \{\boldsymbol{x}_t, \boldsymbol{x}_{t+1}, \boldsymbol{x}_{t+2}, \dots\}$ is a trajectory under policy $\boldsymbol{\pi}$ starting from $\boldsymbol{x}_t$, is a barrier certificate for the agent $i$, given that all other agents follow $\boldsymbol{\pi}^{-i}$.*

The formal version and proof are provided in Appendix C.1. This result implies that the value function of $b^i$ is a barrier certificate for agent $i$. Consider the update process where, without loss of generality, agents are updated sequentially from $1$ to $N$, then, with its action-value function

$Q_{\boldsymbol{\pi}}^i(\boldsymbol{x}_t, \boldsymbol{u}_t) = \mathbb{E}_{\tau \sim \boldsymbol{\pi}}\left[\sum_{k=0}^{\infty} \gamma^k r^i(\boldsymbol{x}_{t+k}, \boldsymbol{u}_{t+k})\right]$, and barrier certificate $H_{\boldsymbol{\pi}}^i$, the optimization for agent $i$ for all $\boldsymbol{x}_t \in \mathcal{X}$ is formulated as:

$$\pi_{e+1}^i = \arg\min_{\pi^i} \quad -Q_{\hat{\boldsymbol{\pi}}_i}^i(\boldsymbol{x}_t, (\boldsymbol{\pi}_{e+1}^{1:i-1}(\boldsymbol{x}_t), \pi^i(\boldsymbol{x}_t), \boldsymbol{\pi}_e^{i+1:N}(\boldsymbol{x}_t))) \tag{2a}$$

$$\text{s.t.} \quad -H_{\hat{\boldsymbol{\pi}}_i}^i(\boldsymbol{x}_{t+1}) + H_{\hat{\boldsymbol{\pi}}_i}^i(\boldsymbol{x}_t) - \epsilon H_{\hat{\boldsymbol{\pi}}_i}^i(\boldsymbol{x}_t) \leq 0 \tag{2b}$$

where $\hat{\boldsymbol{\pi}}_i = (\boldsymbol{\pi}_{e+1}^{1:i-1}, \boldsymbol{\pi}_e^{i:N})$, $Q_{\hat{\boldsymbol{\pi}}_i}^i$ and $H_{\hat{\boldsymbol{\pi}}_i}^i$ are the action-value function and barrier certificate of agent $i$ under the joint policy $\hat{\boldsymbol{\pi}}_i$, $\boldsymbol{x}_{t+1}$ is the next state following $(\boldsymbol{\pi}_{e+1}^{1:i}, \boldsymbol{\pi}_e^{i+1:N})$, and $e$ is the number of updates. We now present a proposition formalizing a key desirable property of the update:

**Proposition 1** *For each $i \in \mathcal{N}$, suppose that all other agents follow the fixed policies $\boldsymbol{\pi}_{e+1}^{1:i-1}$ and $\boldsymbol{\pi}_e^{i+1:N}$, then $\mathcal{X}_{safe}^{(\boldsymbol{\pi}_{e+1}^{1:i-1}, \boldsymbol{\pi}_e^{i:N}), i} \subseteq \mathcal{X}_{safe}^{(\boldsymbol{\pi}_{e+1}^{1:i}, \boldsymbol{\pi}_e^{i+1:N}), i}$, namely, the safe set for agent $i$ will not shrink when $\pi_{e+1}^i$ is obtained as the solution to Problem 2.*

The proof is provided in Appendix C.2. This proposition suggests that, by solving Problem 2 when all other agents follow their certain policies, and agent $i$'s value function and barrier certificate are approximated accurately, the region in which safety is maintained for agent $i$ expands or at least remains unchanged. Therefore, given these requirements for updating agent $i$'s policy, and the fact that each agent should update to a good policy, which is also dependent on other agents' policies, during its turn, we adopt a multi-timescale approach to design an iterative process which can be theoretically shown to converge to a feasible local Nash equilibrium.

**Definition 4** *With $Q_{\boldsymbol{\pi}}^i$ denoting the action-value function of agent $i$, and $b^i(\boldsymbol{x}_t)$ denoting the state-wise safety for agent $i$, $\boldsymbol{\pi}^* = \{\pi_i^*\}_{i \in \mathcal{N}}$ is a local Nash equilibrium (Ratliff et al., 2016) if for each $i \in \mathcal{N}$, there exists a neighborhood $\mathcal{W}^i \subseteq \Pi^i$ of $\pi_i^*$ such that $Q_{(\pi_i^*, \boldsymbol{\pi}_{-i}^*)}^i(\boldsymbol{x}, \boldsymbol{u}) \geq Q_{(\pi^i, \boldsymbol{\pi}_{-i}^*)}^i(\boldsymbol{x}, \boldsymbol{u}), \quad \forall \pi^i \in \mathcal{W}^i$. If, in addition, for all $i \in \mathcal{N}$, the joint policy $\boldsymbol{\pi}^*$ satisfies the safety constraint $Pr(b^i(\boldsymbol{x}_t) < 0) = 0, \forall t \geq 0$, then it is called a **feasible local Nash equilibrium**.*

Since the actor-critic framework is usually applied for model-free RL training, the barrier certificate can thus be approximated by an action-value function $H_{\omega_i}(\boldsymbol{x}_t, \boldsymbol{u}_t)$ in practice. Accordingly, we define two loss functions, $J_{Q^i}$ and $J_{H^i}$, for updating the action-value function and barrier certificate of agent $i$, respectively, with their explicit forms provided in Appendix A. In a general case, for each $i \in \mathcal{N}$, we augment the Lagrangian function with additional squared terms, and sample data and $\tilde{\boldsymbol{u}}_{t+1}$ according to a joint policy $\bar{\boldsymbol{\pi}}_i = (\pi^i, \boldsymbol{\pi}^{-i})$ where $\pi^i$ is the policy of agent $i$ to be updated and $\boldsymbol{\pi}^{-i}$ is a certain joint policy for all other agents. If $\pi^i$ is parameterized by $\theta_i$ and $\boldsymbol{\pi}^{-i}$ by $\boldsymbol{\theta}_{-i}$, then, the modified Lagrangian function for agent $i$ is:

$$\mathcal{L}_{\boldsymbol{\theta}^{-i}}^i(\theta_i, \lambda_i) = \mathbb{E}_{\bar{\boldsymbol{\pi}}_i}\Big[ -Q_{\bar{\boldsymbol{\pi}}_i}^i(\boldsymbol{x}_t, \boldsymbol{u}_t) + \lambda_i \sum_{t=0}^{\infty} \gamma^t \max\{0, -H_{\bar{\boldsymbol{\pi}}_i}^i(\boldsymbol{x}_{t+1}, \tilde{\boldsymbol{u}}_{t+1}) + H_{\bar{\boldsymbol{\pi}}_i}^i(\boldsymbol{x}_t, \boldsymbol{u}_t)$$

$$-\epsilon H_{\bar{\boldsymbol{\pi}}_i}^i(\boldsymbol{x}_t, \boldsymbol{u}_t)\} + \frac{\lambda_i}{2} \sum_{t=0}^{\infty} \gamma^t [\max\{0, -H_{\bar{\boldsymbol{\pi}}_i}^i(\boldsymbol{x}_{t+1}, \tilde{\boldsymbol{u}}_{t+1}) + H_{\bar{\boldsymbol{\pi}}_i}^i(\boldsymbol{x}_t, \boldsymbol{u}_t) - \epsilon H_{\bar{\boldsymbol{\pi}}_i}^i(\boldsymbol{x}_t, \boldsymbol{u}_t)\}]^2\Big],$$

$$\tag{3}$$

where $\lambda_i$ is the Lagrangian multiplier for agent $i$. Compared to the standard Lagrangian commonly used for traditional dual ascent updates, here we add extra squared terms to more strongly penalize safety violations, thereby improving safe policy learning. Moreover, the constraint is formulated using the difference between consecutive barrier certificates, rather than relying on a single critic function to impose the safety constraint as in the CMDP setting. Finally, we denote $J_{\theta_i}$ and $J_{\lambda_i}$ as loss functions to update $\theta_i$ and $\lambda_i$, respectively, with their detailed forms in Appendix A. With these loss functions, we provide Algorithm 1, together with the following theorem:

**Informal Theorem 1** *Under certain assumptions, the joint policy updated in Algorithm 1 will almost surely converge to a feasible local Nash equilibrium $\boldsymbol{\pi}^* = \{\pi_i^*\}_{i \in \mathcal{N}}$, where $H_{\boldsymbol{\pi}^*}^i(\boldsymbol{x}_t, \boldsymbol{u}_t) \geq 0$ for all $(\boldsymbol{x}_t, \boldsymbol{u}_t)$ along the trajectory under $\boldsymbol{\pi}^*$, for every $i \in \mathcal{N}$ and $t \geq 0$. Further, define $H_{\boldsymbol{\pi}^*}^g(\boldsymbol{x}_t, \boldsymbol{u}_t) := \min_{i \in \mathcal{N}} H_{\boldsymbol{\pi}^*}^i(\boldsymbol{x}_t, \boldsymbol{u}_t)$, then $H_{\boldsymbol{\pi}^*}^g(\boldsymbol{x}_t, \boldsymbol{u}_t)$ is a global barrier certificate for the MAS.*

---

**Algorithm 1** State-Wise Safe MARL via Neural Barrier Certificates and Multi-Timescales

---

**Input:** Learning rates $\eta_1(e)$, $\eta_{2,i}(e)$, and $\eta_{3,i}(e)$. Initialized parameters $\{\phi_i^0\}_{i\in\mathcal{N}}$, $\{\omega_i^0\}_{i\in\mathcal{N}}$, $\{\theta_i^0\}_{i\in\mathcal{N}}$, $\{\lambda_i^0\}_{i\in\mathcal{N}}$ for corresponding neural networks and parameters, and maximum iterations $E$.

1: **for** $e = 0, 1, \ldots, E$ **do**
2:    **for** $i = 1, 2, \ldots, N$ **do**
3:       $\phi_i^{e+1} \leftarrow \phi_i^e - \eta_1(e)\nabla_{\phi_i}J_{Q^i}(Q^i)$, $\omega_i^{e+1} \leftarrow \omega_i^e - \eta_1(e)\nabla_{\omega_i}J_{H^i}(H^i)$.
4:       $\theta_i^{e+1} \leftarrow \theta_i^e - \eta_{2,i}(e)\nabla_{\theta_i}J_{\theta_i}(\theta_i)$.
5:       $\lambda_i^{e+1} \leftarrow [\lambda_i^e + \eta_{3,i}(e)\nabla_{\lambda_i}J_{\lambda_i}(\lambda_i)]_+$.
6:    **end for**
7: **end for**
**Output:** $\{\theta_i^E\}_{i\in\mathcal{N}}$

---

The formal version and proof are provided in Appendix C.3. This theorem applies to general MARL settings beyond purely cooperative tasks, and while the joint policy only converges to a feasible *local* Nash equilibrium, different initializations may lead to different local solutions; in practice, one can select solutions that yield better task performance. Hence, the proposed framework offers a principled way to obtain safe controllers.

### 4.2 STATE-WISE STABILITY VIA ISS AND INFEASIBILITY UNDER MULTIPLE CONSTRAINTS

In the previous section, we investigated the problem of ensuring state-wise safety for MAS. We now turn to stability, as shown in Problem 1, using local observations, since, unlike safe MARL with local observations, which has been well studied previously (Zhang et al., 2024; Ying et al., 2023; Zhang et al., 2025), stability in MARL under local observations remains relatively underexplored. Moreover, for large-scale control problems, decentralized training with local information is generally more favorable due to the curse of dimensionality and the inability to obtain global information. Based on the graph structure $\mathcal{G}$, the MDP is accordingly partitioned into $N$ subsystems, each corresponding to one agent. For each agent $i$, a cost signal is defined as $c^i(x_t^i) = ||x_t^i - x_{\text{desired}}^i||$, where, still without loss of generality, we set the desired equilibrium $x_{\text{desired}}^i$ to be $\mathbf{0}$ for the subsequent analysis. The value function associated with this cost is given by $V_{\boldsymbol{\pi}}^i(x_t^i) = \mathbb{E}_{\tau^i \sim \boldsymbol{\pi}}\left[\sum_{k=0}^{\infty}\gamma^k c^i(x_{t+k}^i)\right]$, where $\tau^i = \{x_t^i, x_{t+1}^i, x_{t+2}^i \ldots\}$ is the trajectory of agent $i$ under $\boldsymbol{\pi}$ starting from $x_t^i$. [2] With $\delta_{ij} \in \mathcal{K} \cup \{0\}, \forall i, j \in \mathcal{N}$ denoting the gain functions representing inter-agent couplings, and $\kappa_i(||\boldsymbol{d}_t||)$, where $\kappa_i \in \mathcal{K} \cup \{0\}$, denote the influence of exogenous signal $\boldsymbol{d}_t$ on agent $i$, we propose the following theorem by using the concept of input-to-state stability (ISS) (Sontag & Wang, 1995):

**Theorem 2** *Suppose for a set of $\mathcal{K}_\infty$ functions $\{\beta_i\}_{i\in\mathcal{N}}$, there exists a set of $\mathcal{K}_\infty$ functions $\{\alpha_i\}_{i\in\mathcal{N}}$ and $\{\mu_i\}_{i\in\mathcal{N}}$ satisfying $\sum_{j\neq i}\delta_{ij}(V^j(x_t^j)) + \beta_i(\sum_{j\neq i}\delta_{ij}(V^j(x_t^j))) \leq \alpha_i(V^i(x_t^i))$, and $V^i(x_t^i) \leq \mu_i(||x_t^i||), \forall i \in \mathcal{N}$, for each state sampled under certain joint policy $\boldsymbol{\pi}$. Then, if*

$$\dot{V}^i(x_t^i) \leq -\alpha_i(V^i(x_t^i)) + \sum_{j\neq i}\delta_{ij}(V^j(x_t^j)) + \kappa_i(||\boldsymbol{d}_t||), \forall i \in \mathcal{N}, \tag{4}$$

*also holds for each state sampled under $\boldsymbol{\pi}$, we have $||\boldsymbol{x}_t|| \leq \xi(||\boldsymbol{x}_0||, t) + \rho(||\boldsymbol{d}||_\infty)$. Here $\xi \in \mathcal{KL}$ is a function such that $\xi(\cdot, t) \in \mathcal{K}$ for all $t \geq 0$, and $\xi(r, \cdot) \in \mathcal{L}$ for all $r > 0$, where $\mathcal{L}$ is the set of strictly decreasing functions with $\lim_{t\to\infty}\xi(||\boldsymbol{x}_0||, t) = 0$. $\rho \in \mathcal{K}$ and $||\boldsymbol{d}||_\infty = \sup\{||\boldsymbol{d}_t||, t \geq 0\}$.*

The proof is provided in Appendix D.1. The above theorem indicates that if $\{\alpha_i\}_{i\in\mathcal{N}}$ are chosen sufficiently large, then under other mild conditions, the networked multi-agent system is input-to-state stable when (4) is satisfied. For simplicity and ease of implementation, here we assume a fixed list of appropriate gain functions $\delta_{ij}$ describing the coupling between agent $i$ and $j$, based on which there exist $\{\alpha_i\}_{i\in\mathcal{N}}$ that are large enough such that $\sum_{j\neq i}\delta_{ij}(V^j(x_t^j)) + \beta_i(\sum_{j\neq i}\delta_{ij}(V^j(x_t^j))) \leq \alpha_i(V^i(x_t^i))$ under joint policy $\boldsymbol{\pi} \in \boldsymbol{\Pi}$. Suppose there exists another set of functions $\{\alpha_i'\}_{i\in\mathcal{N}}$ satisfying $(\alpha_i - \alpha_i')(V^i(x_t^i)) \leq \sum_{j\neq i}\delta_{ij}(V^j(x_t^j)) \leq \sum_{j\neq i}\delta_{ij}(V^j(x_t^j)) + \beta_i(\sum_{j\neq i}\delta_{ij}(V^j(x_t^j))) \leq$

---

[2]For simplicity, we later use $V^i$ to denote $V_{\boldsymbol{\pi}}^i$ when the dependence on $\boldsymbol{\pi}$ is clear from context.

$\alpha_i(V^i(x_t^i))$, for all $i \in \mathcal{N}$ and $\boldsymbol{x}_t$ sampled under $\boldsymbol{\pi}$, then, the condition (4) can be further strengthened to $\dot{V}^i(x_t^i) \leq -\alpha_i'(V^i(x_t^i)) + \kappa_i(||\boldsymbol{d}_t||), \forall i \in \mathcal{N}$, since $-\alpha_i'(V^i(x_t^i)) \leq -\alpha_i(V^i(x_t^i)) + \sum_{j \neq i} \delta_{ij}(V^j(x_t^j)) \leq -\beta_i(\sum_{j \neq i} \delta_{ij}(V^j(x_t^j)))$. In practice, $\{\alpha_i'\}_{i \in \mathcal{N}}$ are treated as tunable hyperparameters, and Theorem 2 provides guidance for this tuning in a MAS: for agents that exhibit strong couplings with other agents via physical properties or communication designs, the corresponding $\alpha_i'$ should be chosen large to counteract the influence from other agents and realize dissipation. Notably, it is easy to have the natural extension where class-$\mathcal{K}$ functions are jointly trained together with the policy $\boldsymbol{\pi}$, rather than being pre-defined and fixed. In this way, some conditions in Theorem 2 need not be assumed to hold, but can instead be enforced as constraints during training. More details of an example extension where $\{\alpha_i\}_{i \in \mathcal{N}}$ are trained jointly are provided in Appendix F.

**Remark 1** *Theorem 2 shows that each agent can help enforce ISS of the MAS by using only its own state and a few exchanged scalar quantities, without requiring complete states of other agents. This provides a theoretical foundation and justification for designing decentralized training schemes with minimal communication, reducing both communication and computation burdens. More broadly, stability can serve as an implicit coordination mechanism: when rewards are coupled with convergence toward the desired equilibrium, stability directly improves returns by preventing significant divergence, therefore mitigating severe performance deterioration during learning and alleviating the conservatism introduced by safety constraints, if present. This insight opens new possibilities for scalable MARL algorithms that rely on limited information exchange (Foerster et al., 2016).*

Since the control signal is generated by certain joint policy $\boldsymbol{\pi}$ rather than by any exogenous generator during training, and the control policy takes the state as input, the networked system can be regarded as autonomous, and the term $\kappa_i(||\boldsymbol{d}_t||)$ can be discarded when other external disturbances are negligible. By replacing the derivative with the difference between consecutive steps, and extending Problem 2 to include stability constraints, we obtain the optimization problem for agent $i$:

$$\pi_{e+1}^i = \underset{\pi^i}{\arg\min} \quad -Q_{\hat{\boldsymbol{\pi}}_i}^i(\boldsymbol{x}_t, (\boldsymbol{\pi}_{e+1}^{1:i-1}(\boldsymbol{x}_t), \pi^i(\boldsymbol{x}_t), \boldsymbol{\pi}_e^{i+1:N}(\boldsymbol{x}_t))) \tag{5a}$$

$$s.t. -H_{\hat{\boldsymbol{\pi}}_i}^i(\boldsymbol{x}_{t+1}, \boldsymbol{u}_{t+1}) + H_{\hat{\boldsymbol{\pi}}_i}^i(\boldsymbol{x}_t, \boldsymbol{u}_t) - \epsilon H_{\hat{\boldsymbol{\pi}}_i}^i(\boldsymbol{x}_t, \boldsymbol{u}_t) \leq 0 \tag{5b}$$

$$V_{\hat{\boldsymbol{\pi}}_i}^i(x_{t+1}^i) - V_{\hat{\boldsymbol{\pi}}_i}^i(x_t^i) + \alpha_i' V_{\hat{\boldsymbol{\pi}}_i}^i(x_t^i) \leq 0 \tag{5c}$$

where $\hat{\boldsymbol{\pi}}_i = (\boldsymbol{\pi}_{e+1}^{1:i-1}, \boldsymbol{\pi}_e^{i:N})$, $\boldsymbol{u}_{t+1}$ is output by the updated joint policy $(\boldsymbol{\pi}_{e+1}^{1:i}, \boldsymbol{\pi}_e^{i+1:N})$, and data is sampled under $\hat{\boldsymbol{\pi}}_i$. It is worth noting that $V_{\hat{\boldsymbol{\pi}}_i}^i$ can be designed as an action-value network in practice.

Although it is natural to incorporate stability constraints into the training of each agent alongside safety constraints, as in Problem 5, one well-known challenge in control theory is that enforcing both state-wise safety and stability constraints simultaneously can lead to the issue of infeasibility (Reis et al., 2020). To formalize this, we introduce the following definition:

**Definition 5** $\boldsymbol{\mathcal{X}}_{infeasible}^{\boldsymbol{\pi},i}$ *is the set of states for which no controller $\pi^i$ exists such that agent $i$ can further approach towards its desired equilibrium $x_{desired}^i$ (stability constraint) without causing immediate safety violations, when other agents follow $\boldsymbol{\pi}^{-i}$.*

An illustration is shown in Figure 7 in the Appendix. To address this issue, we provide Proposition 2.

**Informal Proposition 2** *Under certain assumptions, suppose for each $i \in \mathcal{N}$, at its $K_i$-th update the initial state $\boldsymbol{x}_0$ is included within the safe set of agent $i$, then, when the number of updates for agent $i$ goes to infinity, for any trajectory that starts from $\boldsymbol{x}_0$ under the controller obtained by solving Problem 5, $Pr(\boldsymbol{x}_t \in \boldsymbol{\mathcal{X}}_{infeasible}^{\boldsymbol{\pi},i}) = 0, \forall t \geq 0$.*

The formal version and proof are provided in Appendix D.2. This proposition shows that, as the number of updates grows and goes to infinity, the system is prevented from visiting $\boldsymbol{\mathcal{X}}_{infeasible}^{\boldsymbol{\pi},i}$ for all $i \in \mathcal{N}$, under the current joint policy $\boldsymbol{\pi}$, due to the neural barrier certificate. This naturally fits the multi-timescale technique in Theorem 1, which guarantees that each individual policy receives sufficient updates in theory. Consequently, the issue of infeasibility becomes negligible when integrated with the multi-timescale method, and adding the extra stability constraint into Algorithm 1 does not alter the overall reasoning or structure of the relevant proof. By following Algorithm 1 with the added stability constraint for each agent, a locally optimal solution is finally obtained, and:

**Corollary 1** *Every agent $i$ will converge to its desired equilibrium (goal), $x^i_{desired}$, while satisfying safety constraints, under the policy $\boldsymbol{\pi}^* = \{\pi^*_i\}_{i \in \mathcal{N}}$ obtained when Algorithm 1 with extra stability constraint (5c) converges, in the absence of exogenous signal $\boldsymbol{d}_t$.*

The proof is in Appendix D.3. The complete framework, termed **M**ulti-**A**gent **S**afe and **S**table **S**oft **A**ctor-**C**ritic (**MAS**$^3$**AC**), is shown in Figure 4, with pseudocode (Algorithm 2) in Section E.

## 5 EXPERIMENTS

In this section, we answer **Q1**: In cooperative MARL tasks with global information, where high rewards are achieved when the MAS converges to the desired equilibrium (e.g., navigation), can MAS$^3$AC outperform baselines in reward maximization and safety constraint satisfaction? **Q2**: In non-cooperative MARL tasks with only local observations, can MAS$^3$AC consistently achieve high rewards while maintaining few safety violations compared to baselines, for each individual agent? **Q3**: How does the size of an agent's local observation affect MAS$^3$AC's performance? Can MAS$^3$AC with a smaller field of view outperform baselines having more informative observations?

**Baselines**: We compare MAS$^3$AC against standard MARL baselines: HASAC, HAPPO, MAPPO, MASAC, and IPPO, and safe MARL baselines: MACPO, MAPPO-Lagrangian, abbreviated as MAPPOL, and MAFOCOPS. Further details of these baselines are provided in Appendix G.

**Setup**: The safe and stable general MARL benchmarks that we introduce consist of two parts: 1. We modified three environments based on the widely used Safe Multi-Agent MuJoCo benchmark (Gu et al., 2023) to construct safe and stable cooperative MARL benchmarks. In addition, we design a new decentralized non-cooperative version of the Coupled HalfCheetah environment with local observations only. 2. We introduce a multi-agent electrical bidding task whose system dynamics is fundamentally different from robot control tasks as another decentralized non-cooperative benchmark. Further details of these benchmarks are provided in Appendix H.

**Main result**: To answer **Q1**, we first evaluate MAS$^3$AC on the three cooperative benchmarks (Ant, HalfCheetah, and Coupled HalfCheetah), where global information is available and centralized training is applied. Across all tasks, MAS$^3$AC consistently achieves the best trade-off between maximizing reward and minimizing safety costs. Since rewards are tightly coupled with approaching toward and maintaining the desired equilibrium, the strong reward performance of MAS$^3$AC indicates its ability to guarantee stability for the systems. Unlike baselines that either achieve high reward at the expense of frequent safety violations (e.g., HASAC, HAPPO) or enforce safety while greatly sacrificing task performance (e.g., MAFOCOPS), MAS$^3$AC combines both advantages. As shown in Figure 1 and Table 1, MAS$^3$AC demonstrates rapid learning progress from the start, stable convergence to high rewards, and sustained low safety costs without unrecoverable degradation. These results confirm that MAS$^3$AC can effectively coordinate agents in cooperative settings to achieve safe and stable control.

Table 1: Reward and safety cost comparison across cooperative benchmarks.

| Algorithm | Ant | | HalfCheetah | | Coupled HalfCheetah | |
|---|---|---|---|---|---|---|
| | **Reward** | **Cost** | **Reward** | **Cost** | **Reward** | **Cost** |
| **HAPPO** | $41.29 \pm 215.89$ | $311.96 \pm 67.96$ | $373.64 \pm 200.58$ | $2131.21 \pm 483.13$ | $2197.22 \pm 112.77$ | $230.28 \pm 67.11$ |
| **HASAC** | $993.56 \pm 87.49$ | $107.49 \pm 22.64$ | $\mathbf{2229.41 \pm 335.17}$ | $211.07 \pm 300.58$ | $2410.71 \pm 106.28$ | $84.77 \pm 65.91$ |
| **MACPO** | $-1381.66 \pm 37.86$ | $105.24 \pm 50.66$ | $443.50 \pm 265.72$ | $134.58 \pm 142.09$ | $1104.05 \pm 356.15$ | $9.32 \pm 8.53$ |
| **MAFOCOPS** | $-3965.01 \pm 720.78$ | $156.47 \pm 77.84$ | $-781.55 \pm 302.59$ | $581.41 \pm 494.46$ | $-1865.72 \pm 117.10$ | $0.14 \pm 0.97$ |
| **MAPPO** | $-354.58 \pm 203.39$ | $322.31 \pm 64.85$ | $418.49 \pm 194.12$ | $1834.08 \pm 513.17$ | $480.86 \pm 195.30$ | $1764.05 \pm 431.53$ |
| **MAPPOL** | $-903.98 \pm 67.67$ | $142.52 \pm 58.00$ | $672.25 \pm 143.48$ | $898.40 \pm 379.02$ | $1968.86 \pm 118.37$ | $3.67 \pm 3.39$ |
| **MAS**$^3$**AC** | $\mathbf{1221.79 \pm 50.45}$ | $\mathbf{2.10 \pm 6.92}$ | $2177.90 \pm 714.48$ | $\mathbf{6.22 \pm 25.28}$ | $\mathbf{2596.56 \pm 105.95}$ | $\mathbf{0.03 \pm 0.17}$ |

To answer **Q2**, we first evaluate MAS$^3$AC on the decentralized non-cooperative Coupled HalfCheetah benchmark, where each agent has access only to local observations within its neighborhood. The observation size is treated as a tunable hyperparameter (see the ablation study and Appendix I.2.1 for details). Each agent is trained with its own reward, safety, and stability cost signals, and rewards remain closely tied to the multi-agent system's ability to converge to the desired equilibrium. This setting is different from the cooperative benchmarks since, instead of sharing a joint objective and global state, agents must make decisions independently, which increases the complexity due to partial observability and potential conflicts among individual goals. Despite these challenges, MAS$^3$AC

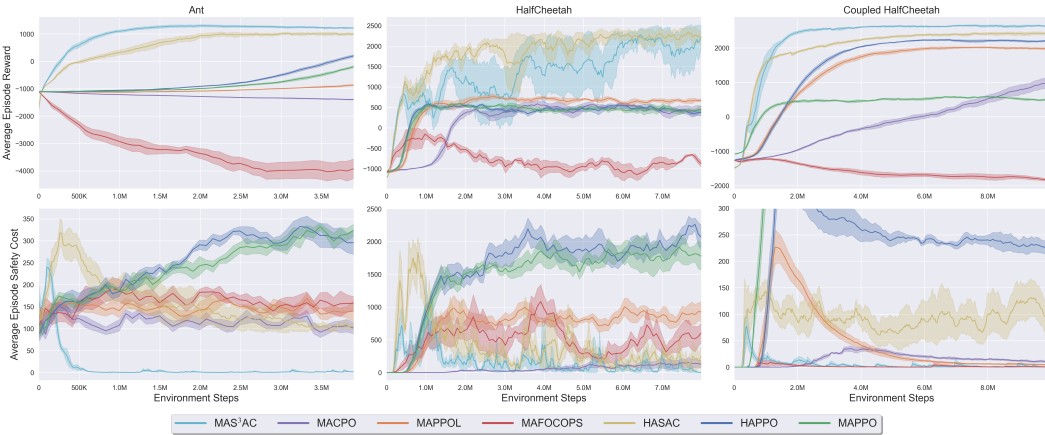

Figure 1: Average episode reward and average episode safety cost in the three cooperative tasks, comparing MAS³AC with baselines. Each curve illustrates the average across 5 experiments employing different random seeds, with the shaded area denoting the standard error.

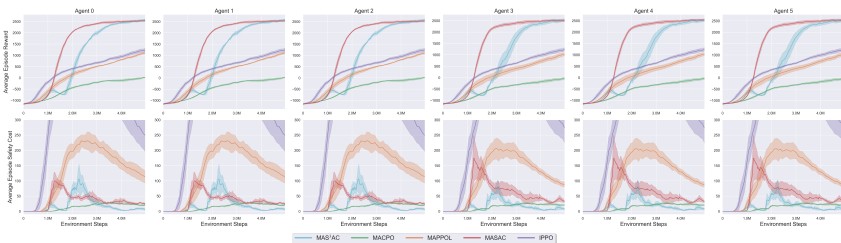

Figure 2: Average episode reward and average episode safety cost in the non-cooperative Coupled HalfCheetah task, comparing MAS³AC against baselines. Each curve illustrates the average across 5 experiments employing different random seeds, with the shaded area denoting the standard error.

consistently achieves high final rewards and low safety violations for *each* agent, outperforming the baselines. As shown in Figure 2 and Table 2, while some baselines may show strong performance for certain agents in either reward or safety costs, they either fail to maintain this across all agents, or often trade off stability for safety or vice versa. In contrast, MAS³AC exhibits superior learning process, ensuring that every agent benefits from both the high final reward and strong constraint satisfaction. The evaluation of MAS³AC on the decentralized electrical bidding task is in Appendix I.1.1. These results highlight the ability of MAS³AC to generalize beyond cooperative tasks and effectively handle decentralized, non-cooperative MARL settings.

Table 2: Reward and safety cost comparison for the non-cooperative Coupled HalfCheetah task across agents.

| | **Agent 0** | | **Agent 1** | | **Agent 2** | |
|---|---|---|---|---|---|---|
| **Algorithm** | **Reward** | **Cost** | **Reward** | **Cost** | **Reward** | **Cost** |
| **IPPO** | $902.29 \pm 386.41$ | $451.38 \pm 305.37$ | $904.99 \pm 389.86$ | $451.38 \pm 305.37$ | $905.91 \pm 388.62$ | $451.38 \pm 305.37$ |
| **MACPO** | $-140.28 \pm 276.72$ | $21.48 \pm 25.26$ | $-141.91 \pm 276.68$ | $21.48 \pm 25.26$ | $-142.75 \pm 276.65$ | $21.48 \pm 25.26$ |
| **MAPPOL** | $735.92 \pm 416.97$ | $174.96 \pm 143.68$ | $735.77 \pm 417.75$ | $174.96 \pm 143.68$ | $732.35 \pm 417.00$ | $174.96 \pm 143.68$ |
| **MASAC** | $2499.75 \pm 78.45$ | $25.25 \pm 20.76$ | $2503.90 \pm 84.95$ | $25.25 \pm 20.76$ | $2489.93 \pm 81.61$ | $25.25 \pm 20.76$ |
| **MAS³AC** | $\mathbf{2531.32 \pm 136.89}$ | $\mathbf{6.35 \pm 14.12}$ | $\mathbf{2533.62 \pm 141.96}$ | $\mathbf{6.35 \pm 14.12}$ | $\mathbf{2529.90 \pm 142.89}$ | $\mathbf{6.35 \pm 14.12}$ |

| | **Agent 3** | | **Agent 4** | | **Agent 5** | |
|---|---|---|---|---|---|---|
| **Algorithm** | **Reward** | **Cost** | **Reward** | **Cost** | **Reward** | **Cost** |
| **IPPO** | $898.48 \pm 395.57$ | $432.92 \pm 273.22$ | $900.26 \pm 397.68$ | $432.92 \pm 273.22$ | $903.87 \pm 397.26$ | $432.92 \pm 273.22$ |
| **MACPO** | $-215.51 \pm 277.43$ | $21.14 \pm 27.75$ | $-216.04 \pm 277.56$ | $21.14 \pm 27.75$ | $-215.42 \pm 277.29$ | $21.14 \pm 27.75$ |
| **MAPPOL** | $646.70 \pm 464.02$ | $150.76 \pm 134.02$ | $648.57 \pm 465.17$ | $150.76 \pm 134.02$ | $648.17 \pm 466.71$ | $150.76 \pm 134.02$ |
| **MASAC** | $2479.16 \pm 102.04$ | $35.51 \pm 26.71$ | $\mathbf{2502.30 \pm 96.44}$ | $35.51 \pm 26.71$ | $\mathbf{2501.68 \pm 108.95}$ | $35.51 \pm 26.71$ |
| **MAS³AC** | $\mathbf{2486.43 \pm 172.32}$ | $\mathbf{7.48 \pm 20.11}$ | $2488.44 \pm 171.39$ | $\mathbf{7.48 \pm 20.11}$ | $2489.25 \pm 175.59$ | $\mathbf{7.48 \pm 20.11}$ |

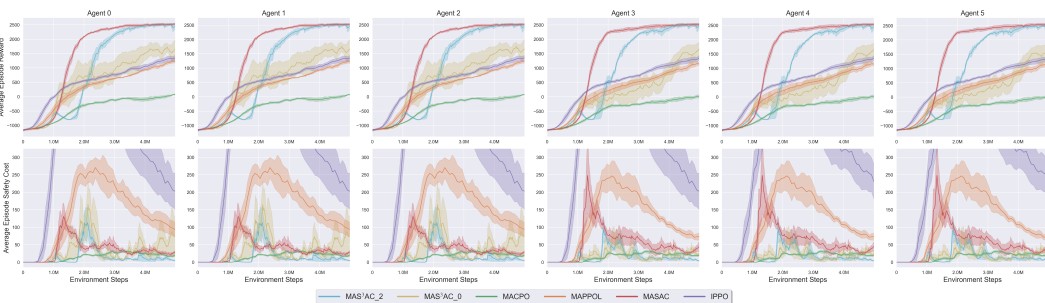

Figure 3: Average episode reward and average episode safety cost in the non-cooperative Coupled HalfCheetah task, comparing MAS$^3$AC with different neighbor sizes against baselines.

**Ablation studies**: To answer **Q3**, we conduct an ablation study on the decentralized Coupled HalfCheetah benchmark to examine how the **neighbor size**, i.e., the radius of local observations, affects performance. Figure 3 and Table 6 (deferred to the Appendix) compare MAS$^3$AC with neighbor size 0 (self-state only) and 2, against baselines that use neighbor size 3. The results show a clear trend: with more information available to all critics, certificates, and policies, MAS$^3$AC achieves stronger performance. In particular, MAS$^3$AC with neighbor size 2 matches the reward level of MASAC using neighbor size 3, which purely maximizes rewards, while maintaining substantially lower safety costs, outperforming all baselines in balancing safety and performance. By contrast, MAS$^3$AC with neighbor size 0 suffers from higher safety violations, highlighting that barrier certificates for safety must incorporate information from neighbors to effectively enforce inter-agent safety requirements. Nevertheless, even with neighbor size 0, MAS$^3$AC remains competitive with some baselines, demonstrating the robustness of the framework even under severely limited information. Additional results with intermediate neighbor sizes $(0, 1, 3)$ are provided in Appendix I.2.1, which confirm the consistent advantage of MAS$^3$AC across different observation ranges.

We next examine the **effect of Lyapunov function inputs** on MAS$^3$AC by using the decentralized Coupled HalfCheetah benchmark. Detailed results and analysis are provided in Appendix I.2.2, where we show how different Lyapunov certificate designs influence the overall performance of MAS$^3$AC, in alignment with our theoretical ISS analysis.

More ablations are provided in Appendix J, including but not limited to robustness to external disturbances, barrier versus reachability for safety, the impact of input-to-state stability conditions on MAS$^3$AC, additional benchmarks, and an extra baseline, further supporting the claims.

## 6 CONCLUSION AND LIMITATIONS

In this work, we propose MAS$^3$AC, a framework that integrates state-wise constraints with model-free RL for safe and stable control of general multi-agent tasks. Theoretical proof is provided to show the convergence of the algorithm to a feasible locally optimal solution, and a specific investigation is conducted on the infeasibility issue. Empirical results across all proposed benchmarks demonstrate that our algorithm yields consistently high rewards with very few safety violations compared to baselines on both cooperative and non-cooperative MARL tasks with either centralized or decentralized training. We hope this MAS$^3$AC algorithm, together with new benchmarks, can facilitate future research and serve as a baseline for safe and stable MARL algorithm development.

Despite the promising results obtained by the MAS$^3$AC algorithm proposed in this study, several limitations remain. First, theoretically, the multi-timescale training scheme requires that at each timescale, sufficient data should be available to update the corresponding networks, which may result in high sample complexity. Second, for non-cooperative tasks, it would be valuable to establish an upper bound on the error between updates using global state information and local observations to provide a stronger theoretical justification for decentralized training. Third, a detailed theoretical analysis for extending the algorithm to have adaptive class-$\mathcal{K}$ functions in stability constraints remains an important open direction. We plan to address these questions in future work.

## REPRODUCIBILITY STATEMENT

To facilitate reproducibility, we provide an anonymized implementation of MAS$^3$AC as the supplementary material. The submitted code is for one **cooperative multi-agent MuJoCo task**. A full version of the code, which covers general-sum tasks (with different task structures and training code) and additional cooperative tasks, will be released publicly upon paper acceptance. For theoretical results, all assumptions, together with clear explanations, are summarized in Section B.

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

## A    LOSS FUNCTIONS FOR UPDATES

In this part, we provide the explicit forms of loss functions used in the update. The loss functions to update the action-value function $Q^i$, parameterized by $\phi_i$, and the barrier certificate $H_{\omega_i}$ for agent $i$ are:

$$J_{Q^i}(Q^i) = \mathbb{E}_{(\boldsymbol{x}_t, \boldsymbol{u}_t, r_t^i, \boldsymbol{x}_{t+1}) \sim \mathcal{D}} \big[ r_t^i + \gamma Q_{\phi_i, targ}(\boldsymbol{x}_{t+1}, \tilde{\boldsymbol{u}}_{t+1}) - Q_{\phi_i}(\boldsymbol{x}_t, \boldsymbol{u}_t) \big]^2, \qquad (6)$$

$$J_{H^i}(H^i) = \mathbb{E}_{(\boldsymbol{x}_t, \boldsymbol{u}_t, b_t^i, \boldsymbol{x}_{t+1}) \sim \mathcal{D}} \big[ b_t^i + \gamma H_{\omega_i, targ}(\boldsymbol{x}_{t+1}, \tilde{\boldsymbol{u}}_{t+1}) - H_{\omega_i}(\boldsymbol{x}_t, \boldsymbol{u}_t) \big]^2, \qquad (7)$$

where $\tilde{\boldsymbol{u}}_{t+1}$ is generated by the current joint policy $\hat{\boldsymbol{\pi}}_i = (\boldsymbol{\pi}_{e+1}^{1:i-1}, \boldsymbol{\pi}_e^{i:N})$, and $\mathcal{D}$ denotes the replay buffer. Target networks $Q_{\phi_i, targ}$ and $H_{\omega_i, targ}$ are used in practical training. Since both the action-value function and the barrier certificate are represented by neural networks, the loss function to train the parameter $\theta_i$ for policy $\pi^i$, and its corresponding Lagrangian multiplier $\lambda_i$, derived from (3), is:

$$J_{\theta_i}(\theta_i) = J_{\lambda_i}(\lambda_i) = \mathbb{E}_{\mathcal{D}} \big[ -Q^i(\boldsymbol{x}_t, \hat{\boldsymbol{u}}_t; \phi_i) + \lambda_i \sum_{t=0}^{\infty} \gamma^t \max\{0, -H^i(\boldsymbol{x}_{t+1}, \hat{\boldsymbol{u}}_{t+1}; \omega_i) + H^i(\boldsymbol{x}_t, \boldsymbol{u}_t; \omega_i)$$

$$- \epsilon H^i(\boldsymbol{x}_t, \boldsymbol{u}_t; \omega_i)\} + \frac{\lambda_i}{2} \sum_{t=0}^{\infty} \gamma^t [\max\{0, -H^i(\boldsymbol{x}_{t+1}, \hat{\boldsymbol{u}}_{t+1}; \omega_i) + H^i(\boldsymbol{x}_t, \boldsymbol{u}_t; \omega_i) - \epsilon H^i(\boldsymbol{x}_t, \boldsymbol{u}_t; \omega_i)\}]^2 \big],$$

$$(8)$$

where $\hat{\boldsymbol{u}}_t$ and $\hat{\boldsymbol{u}}_{t+1}$ are actions output by the policy network with gradient information for update $\theta_i$. When stability constraints are also incorporated, we first train the function $V^i$, parameterized by $\chi_i$ for agent $i$, using the following loss function:

$$J_{V^i}(V^i) = \mathbb{E}_{(x_t^i, u_t^i, c_t^i, x_{t+1}^i) \sim \mathcal{D}} \big[ c_t^i + \gamma V_{\chi_i, targ}(x_{t+1}^i, \tilde{u}_{t+1}^i) - V_{\chi_i}(x_t^i, u_t^i) \big]^2, \qquad (9)$$

when $V^i$ is implemented as an action-value network, and target function $V_{\chi_i, targ}$ is applied. Subsequently, the loss function used to train the policy $\pi^i$, and its corresponding Lagrangian multipliers, is given as

$$J_{\theta_i}^{\text{full}}(\theta_i) = J_{\lambda_i}^{\text{full}}(\lambda_i) = \mathbb{E}_{\mathcal{D}} \big[ -Q^i(\boldsymbol{x}_t, \hat{\boldsymbol{u}}_t; \phi_i) + \lambda_{i,1} \sum_{t=0}^{\infty} \gamma^t \max\{0, -H^i(\boldsymbol{x}_{t+1}, \hat{\boldsymbol{u}}_{t+1}; \omega_i)$$

$$+ H^i(\boldsymbol{x}_t, \boldsymbol{u}_t; \omega_i) - \epsilon H^i(\boldsymbol{x}_t, \boldsymbol{u}_t; \omega_i)\} + \frac{\lambda_{i,1}}{2} \sum_{t=0}^{\infty} \gamma^t [\max\{0, -H^i(\boldsymbol{x}_{t+1}, \hat{\boldsymbol{u}}_{t+1}; \omega_i)$$

$$+ H^i(\boldsymbol{x}_t, \boldsymbol{u}_t; \omega_i) - \epsilon H^i(\boldsymbol{x}_t, \boldsymbol{u}_t; \omega_i)\}]^2 + \lambda_{i,2} \sum_{t=0}^{\infty} \gamma^t \max\{0, V^i(x_{t+1}^i, \hat{u}_{t+1}^i, \chi_i) \qquad (10)$$

$$- V^i(x_t^i, u_t^i, \chi_i) + \alpha_i' V^i(x_t^i, u_t^i, \chi_i)\} + \frac{\lambda_{i,2}}{2} \sum_{t=0}^{\infty} \gamma^t [\max\{0, V^i(x_{t+1}^i, \hat{u}_{t+1}^i, \chi_i)$$

$$- V^i(x_t^i, u_t^i, \chi_i) + \alpha_i' V^i(x_t^i, u_t^i, \chi_i)\}]^2 \big].$$

Here $\hat{\boldsymbol{u}}_t$, $\hat{\boldsymbol{u}}_{t+1}$ and $\tilde{u}_{t+1}^i$ are actions output by the policy network with gradient information for update $\theta_i$. $\lambda_{i,1}$ and $\lambda_{i,2}$ are Lagrangian multipliers for safety and stability constraints, respectively. **When global information is not available, all inputs are replaced by local observations**.

## B    ASSUMPTIONS

This section summarizes the assumptions used in the theoretical analysis. Some notations are formally introduced later in the proof sections, but are still included here for completeness.

**Assumption 1** *There exists an upper bound $T$ such that $\forall i \in \mathcal{N}$, all trajectories starting from $\boldsymbol{x}_0 \in \boldsymbol{\mathcal{X}}_{\text{irrec}}^{\boldsymbol{\pi},i}$ will result in $\Pr(b^i(\boldsymbol{x}_t) < 0) > 0$ at least once within $T$ steps under the controller $\pi^i$ when all other agents follow the fixed policy $\boldsymbol{\pi}^{-i}$.*

This assumes that states within the irrecoverable unsafe set $\boldsymbol{\mathcal{X}}_{irrec}^{\pi,i}$ lead to a safety violation with non-zero probability within a bounded number of steps. This is a mild assumption since it does not require deterministic violations, but only requires that safety violations occur with positive probability under the given policy.

**Assumption 2** *Define $\eta_1(e)$ as the learning rate for updating the action-value functions and barrier certificates of all agents. For each agent $i \in \mathcal{N}$, let $\eta_{2,i}(e)$ and $\eta_{3,i}(e)$ denote the learning rates for updating the policy and the Lagrangian multiplier, respectively. The following conditions hold:*

*1. $\sum_e \eta_1(e) = \sum_e \eta_{2,i}(e) = \sum_e \eta_{3,i}(e) = \infty, \quad \forall i \in \mathcal{N}$*

*2. $\sum_e \eta_1(e)^2 < \infty, \quad \sum_e \eta_{2,i}(e)^2 < \infty, \quad \sum_e \eta_{3,i}(e)^2 < \infty, \quad \forall i \in \mathcal{N}$*

*3. $\eta_{2,1}(e) = \mathcal{O}(\eta_1(e)); \quad \eta_{3,i}(e) = \mathcal{O}(\eta_{2,i}(e)), \quad \forall i \in \mathcal{N}; \eta_{2,i}(e) = \mathcal{O}(\eta_{3,i-1}(e)), \forall i \in \{2,\dots,N\}.$*

This assumption adopts a multi-timescale learning framework widely used in prior constrained reinforcement learning studies. It ensures that the critic and barrier functions for all agent are updated on the fastest timescale, while for each agent, the policy is updated on a faster timescale than its associated Lagrangian multiplier; and the update of the Lagrangian multiplier of agent $i$ is faster than the update of the policy of the following agent $i+1$. This hierarchy enables effective tracking of inter-agent constraints, as in our case, and addresses non-stationarity, thereby improving learning stability in multi-agent systems.

Theoretically, step-size schedules such as $\eta_1(e) = \frac{1}{e}$, $\eta_{2,i}(e) = \frac{1}{e^{2i}}, \eta_{3,i}(e) = \frac{1}{e^{2i+1}}$ can be used to satisfy multi-timescale requirements. However, in practice, since the number of update steps is limited, it is difficult to satisfy the first point in Assumption 2 with any schedule. In implementation, we apply linearly decaying learning rates with different initial values to ensure $\eta_1(e) > \eta_{2,i}(e) > \eta_{3,i}(e), \forall e, \forall i \in \mathcal{N}$ and $\eta_{3,i-1}(e) > \eta_{2,i}(e), \forall e, \forall i \in \{2,\dots,N\}$ to approximate the required relationships among the learning rates, which has been empirically proved to be sufficient to support stable and efficient training across all benchmarks compared to all baselines.

To mitigate potential slow convergence due to timescale separation, we incorporate additional squared penalty terms into the constraint loss, which encourage and accelerate the update toward feasible solutions. Based on the experimental results, our algorithm has a better sample efficiency compared to baselines in quickly achieving higher rewards while maintaining very few or near-zero safety violations with fewer samples, allowing early stopping to reduce computational burden. Moreover, for homogeneous agents, the parameter sharing method can be applied to reduce the number of trainable parameters and distinct learning rates, which further improves the training efficiency. However, further investigation under the multi-timescale framework, including more techniques for enhancing sample efficiency, remains an important direction for future work.

**Assumption 3** *For each agent $i$, the defined signals $\hat{r}_t^i$ and $\hat{r}_t^{i,*}$, the function $Q_H^i(\boldsymbol{x}_t, \boldsymbol{u}_t; \hat{\omega}_i)$ and the space $\hat{\Omega}_i$ are bounded. Besides, $\nabla_{\hat{\omega}_i} Q_H^i(\boldsymbol{x}_t, \boldsymbol{u}_t; \hat{\omega}_i)$ is a Lipschitz function in $\hat{\omega}_i$ for all $\boldsymbol{x}_t$ and $\boldsymbol{u}_t$.*

This assumption guarantees the boundedness of signals and critic functions for agents, which is a standard requirement in actor-critic methods to ensure stable training and convergence. The Lipschitz condition, which is also commonly used in previous RL studies, on the gradient promotes well-behaved updates. For neural networks, approximate Lipschitz continuity can be induced through techniques such as weight decay and clipping, making this assumption realistic in practice.

*Note: The terms $\hat{r}_t^i$, $\hat{r}_t^{i,*}$, $Q_H^i$, $\hat{\omega}_i$, and $\hat{\Omega}_i$ are formally introduced in Section C.3. They are referenced here to provide a complete overview of all theoretical assumptions, even though they are not defined in the main body of the submission.*

**Assumption 4** *For each agent $i$, there always exists a policy $\pi^i$ such that*

$$H_{\boldsymbol{\pi}}^i(\boldsymbol{x}_{t+1}, \boldsymbol{u}_{t+1}) - H_{\boldsymbol{\pi}}^i(\boldsymbol{x}_t, \boldsymbol{u}_t) \geq -\epsilon H_{\boldsymbol{\pi}}^i(\boldsymbol{x}_t, \boldsymbol{u}_t)$$

*for all consecutive pairs $(\boldsymbol{x}_t, \boldsymbol{u}_t)$ and $(\boldsymbol{x}_{t+1}, \boldsymbol{u}_{t+1})$, $\forall t \geq 0$, along with trajectories generated under any joint policy $\boldsymbol{\pi}^{-i}$. Further, all initial states are within the safe set $\boldsymbol{\mathcal{X}}_{safe}$.*

This assumption rules out scenarios where the environment, including other agents' fixed policies, renders safety of agent $i$ impossible, namely situations where no admissible action can help to satisfy safety requirements for agent $i$. Since the other agents' policies $\boldsymbol{\pi}^{-i}$ are treated as part of the environment in this formulation, this feasibility condition is effectively an assumption for single-agent tasks. It ensures the problem is well-posed and that a safe policy is learnable and achievable under the given dynamics. The requirement that all initial states lie within the safe set is also standard in constrained control setups, as starting from unsafe states would inevitably lead to safety constraint violations.

**Assumption 5** *The state space and action space for each agent are bounded. Besides, the parameter space $\Theta_i$ and Lagrangian multiplier $\lambda_i$ for each agent $i$ are also bounded. Further, $\nabla_{u_t^i} Q^i(\boldsymbol{x}_t, \boldsymbol{u}_t; \phi_i)$ and $\nabla_{u_t^i} Q_H^i(\boldsymbol{x}_t, \boldsymbol{u}_t; \hat{\omega}_i)$ are Lipschitz functions in $u_t^i$ for all $\boldsymbol{x}_t$, $\phi_i$ and $\hat{\omega}_i$, and are also Lipschitz functions in $\phi_i$ and $\hat{\omega}_i$ for all $\boldsymbol{x}_t$ and $\boldsymbol{u}_t$. $\nabla_{\theta_i} \pi^i(\boldsymbol{x}_t; \theta_i)$ is a Lipschitz function in $\theta_i$.*

This assumption ensures the boundedness of the relevant spaces and the smoothness of gradients used in the policy update. Bounded state and action spaces are standard in RL and control theory to avoid unbounded optimization landscapes. Bounding the policy parameter and Lagrange multiplier spaces helps the dual updates and prevents divergence. The Lipschitz continuity of the critic gradients and policy gradient guarantees well-behaved gradient-based updates during policy training. In practice, neural networks with smooth activations and regularization techniques can effectively lead to approximate Lipschitz properties, which makes the assumption mild and practical.

**Assumption 6** *For each agent $i$, the function $Q_H^i(\boldsymbol{x}_t, \boldsymbol{u}_t; \hat{\omega}_i)$ is a Lipschitz function in $u_t^i$ for all $\boldsymbol{x}_t$ and $\hat{\omega}_i$. Moreover, the gradients $\nabla_{u_t^i} Q_{\bar{\boldsymbol{\pi}}_i}^i(\boldsymbol{x}_t, \boldsymbol{u}_t)$ and $\nabla_{u_t^i} Q_{H_{\bar{\boldsymbol{\pi}}_i}}^i(\boldsymbol{x}_t, \boldsymbol{u}_t)$ are Lipschitz functions in the joint policy $\bar{\boldsymbol{\pi}}_i$.*

This assumption ensures local smoothness of the value function and its gradients with respect to the agent's action. It is essential for bounding the approximation error during policy updates, and reflects a standard smoothness assumption in theoretical RL, capturing the idea that small changes in the policy lead to proportionally bounded changes in value estimates. In practice, the approximate Lipschitz property of value networks can also be induced through methods like regularization techniques, as stated in previous explanations.

*Note: Related terms and gradients are formally introduced in Section C.3. They are included here for completeness, as this appendix summarizes all theoretical assumptions supporting the theoretical analysis.*

**Assumption 7** *During the update of agent $i$'s policy while other agents follow certain $\boldsymbol{\pi}^{-i}$, denote $k_i$ as the number of updates for $\pi^i$, then, for any $k_i \geq 0$ and any $\boldsymbol{x}_t \in \mathcal{X}_{infeasible}^{\boldsymbol{\pi},i}$, the control signal $u_t^i$ generated by the new policy $\pi_{k_i+1}^i$, obtained by solving*

$$\pi_{k_i+1}^i = \arg\min_{\pi^i} \; -Q_{\boldsymbol{\pi}_{i,k_i}}^i\left(\boldsymbol{x}_t, (\pi^i(\boldsymbol{x}_t), \boldsymbol{\pi}^{-i}(\boldsymbol{x}_t))\right) \tag{11a}$$

$$s.t. \; -H_{\boldsymbol{\pi}_{i,k_i}}^i(\boldsymbol{x}_{t+1}, \boldsymbol{u}_{t+1}) + H_{\boldsymbol{\pi}_{i,k_i}}^i(\boldsymbol{x}_t, \boldsymbol{u}_t) - \epsilon H_{\boldsymbol{\pi}_{i,k_i}}^i(\boldsymbol{x}_t, \boldsymbol{u}_t) \leq 0 \tag{11b}$$

$$V_{\boldsymbol{\pi}_{i,k_i}}^i(x_{t+1}^i) - V_{\boldsymbol{\pi}_{i,k_i}}^i(x_t^i) + \alpha_i' V_{\boldsymbol{\pi}_{i,k_i}}^i(x_t^i) \leq 0 \tag{11c}$$

*where $\boldsymbol{\pi}_{i,k_i} = (\pi_{k_i}^i, \boldsymbol{\pi}^{-i})$ denotes the joint policy in which agent $i$'s policy is being updated for the $k_i$-th time while others follow $\boldsymbol{\pi}^{-i}$, satisfies the stability constraint with a positive probability.*

This assumption states that when the state lies within $\mathcal{X}_{infeasible}^{\boldsymbol{\pi},i}$, where no controller exists to simultaneously maintain safety and drive the agent closer toward its goal according to the stability constraint, for each iteration, there is a positive probability that the control signal, output by the updated $\pi_{k_i+1}^i$, satisfies the stability constraint, thereby driving the agent closer to the goal and causing immediate safety violations.

## C  APPENDIX FOR STATE-WISE SAFETY VIA NEURAL BARRIER CERTIFICATES

### C.1  FORMAL VERSION AND PROOF FOR LEMMA 1

**Lemma 2** *Define the barrier signal $b^i$ as:*

$$b^i(\boldsymbol{x}_t) = \begin{cases} d & \text{if } \boldsymbol{x}_t \notin \boldsymbol{\mathcal{X}}_{vio}^{\boldsymbol{\pi},i}, \\ D & \text{if } \boldsymbol{x}_t \in \boldsymbol{\mathcal{X}}_{vio}^{\boldsymbol{\pi},i}, \end{cases} \tag{12}$$

*where $d = 0$ and $D < 0$ are constants. Under Assumption 1, the function $H_{\boldsymbol{\pi}}^i(\boldsymbol{x}_t) = \mathbb{E}_{\tau \sim \boldsymbol{\pi}}\left[\sum_{k=0}^{\infty} \gamma^k b^i(\boldsymbol{x}_{t+k})\right]$, where $\gamma \in (0,1)$ is the discount factor and $\tau = \{\boldsymbol{x}_t, \boldsymbol{x}_{t+1}, \boldsymbol{x}_{t+2}, \dots\}$ is a trajectory under controller $\boldsymbol{\pi}$ starting from $\boldsymbol{x}_t$, is a barrier certificate for the agent $i$, given that all other agents following $\boldsymbol{\pi}^{-i}$.*

**Proof 1** *When $\boldsymbol{x}_t \in \boldsymbol{\mathcal{X}}_{safe}^{\boldsymbol{\pi},i}$, by Definition 2, under the controller $\boldsymbol{\pi} = (\pi^i, \boldsymbol{\pi}^{-i})$, for all $k \geq 0$, $Pr(b^i(\boldsymbol{x}_{t+k})) = 0$, and therefore, $H_{\boldsymbol{\pi}}^i(\boldsymbol{x}_t) = H_{\boldsymbol{\pi}}^i(\boldsymbol{x}_{t+1}) = \mathbb{E}_{\tau \sim \boldsymbol{\pi}}\left[\sum_{k=0}^{\infty} \gamma^k b^i(\boldsymbol{x}_{t+k})\right] = 0$ given $d = 0$. Hence, the second property in Definition 3 is satisfied. When $\boldsymbol{x}_t \in \boldsymbol{\mathcal{X}}_{unsafe}^{\boldsymbol{\pi},i}$, according to Definition 2, there exists $n \geq 0$ such that $Pr(b^i(\boldsymbol{x}_{t+n}) < 0) > 0$ under the joint controller $\boldsymbol{\pi}$, and therefore $\mathbb{E}_{\boldsymbol{x}_{t+n}}\left[b^i(\boldsymbol{x}_{t+n})\right] < 0$. Furthermore, under Assumption 1, we have $n \leq T$. Thus, we have:*

$$H_{\boldsymbol{\pi}}^i(\boldsymbol{x}_t) = \mathbb{E}_{\tau \sim \boldsymbol{\pi}}\left[\sum_{k=0}^{\infty} \gamma^k b^i(\boldsymbol{x}_{t+k})\right] \leq \gamma^n \mathbb{E}_{\boldsymbol{x}_{t+n}}\left[b^i(\boldsymbol{x}_{t+n})\right] \leq \gamma^T \mathbb{E}_{\boldsymbol{x}_{t+n}}\left[b^i(\boldsymbol{x}_{t+n})\right] < 0,$$

*since $\gamma \in (0,1)$. Therefore, the first property in Definition 3 is satisfied. Also, when $\boldsymbol{x}_t \in \boldsymbol{\mathcal{X}}_{safe}^{\boldsymbol{\pi},i}$, by Definition 2, $\boldsymbol{x}_{t+1} \in \boldsymbol{\mathcal{X}}_{safe}^{\boldsymbol{\pi},i}$, and thus $H_{\boldsymbol{\pi}}^i(\boldsymbol{x}_{t+1}) - H_{\boldsymbol{\pi}}^i(\boldsymbol{x}_t) = 0 - 0 = -\epsilon H_{\boldsymbol{\pi}}^i(\boldsymbol{x}_t)$. Therefore, the third property in Definition 3 is satisfied. In sum, all three properties for being a barrier certificate are satisfied, and thus, the function $H_{\boldsymbol{\pi}}^i(\boldsymbol{x}_t)$ is a barrier certificate of the agent $i$ with all other agents following $\boldsymbol{\pi}^{-i}$.*

### C.2  PROOF FOR PROPOSITION 1

**Proof 2** *For all $\boldsymbol{x}_t \in \mathcal{X}_{safe}^{(\boldsymbol{\pi}_{e+1}^{1:i-1}, \boldsymbol{\pi}_e^{i:N}),i}$, the condition $H_{\hat{\boldsymbol{\pi}}_i}^i(\boldsymbol{x}_{t+1}) \geq (1-\epsilon)H_{\hat{\boldsymbol{\pi}}_i}^i(\boldsymbol{x}_t) = 0$ in Problem (2) for $\boldsymbol{x}_{t+1}$ following the new joint policy $(\boldsymbol{\pi}_{e+1}^{1:i}, \boldsymbol{\pi}_e^{i+1:N})$. Therefore, $\boldsymbol{x}_{t+1} \in \mathcal{X}_{safe}^{(\boldsymbol{\pi}_{e+1}^{1:i-1}, \boldsymbol{\pi}_e^{i:N}),i}$ and similarly, $\forall \boldsymbol{x}_t \in \mathcal{X}_{safe}^{(\boldsymbol{\pi}_{e+1}^{1:i-1}, \boldsymbol{\pi}_e^{i:N}),i}$. the trajectory $\{\boldsymbol{x}_t, \boldsymbol{x}_{t+1}, \dots\}$ under the policy $(\boldsymbol{\pi}_{e+1}^{1:i}, \boldsymbol{\pi}_e^{i+1:N})$ remains in $\mathcal{X}_{safe}^{(\boldsymbol{\pi}_{e+1}^{1:i-1}, \boldsymbol{\pi}_e^{i:N}),i}$, without causing $b^i(\boldsymbol{x}_{t+k}) < 0$ at any $k \geq 0$ with a positive probability. Thus, $\boldsymbol{x}_t \in \mathcal{X}_{safe}^{(\boldsymbol{\pi}_{e+1}^{1:i}, \boldsymbol{\pi}_e^{i+1:N}),i}$ since $\mathbb{E}_{(\boldsymbol{\pi}_{e+1}^{1:i}, \boldsymbol{\pi}_e^{i+1:N})}\left[\sum_{k=0}^{\infty} \gamma^k b^i(\boldsymbol{x}_{t+k})\right] = 0$.*

*Since $\epsilon \in (0,1)$, if $\exists \boldsymbol{x}_t \in \boldsymbol{\mathcal{X}}_{irrec}^{(\boldsymbol{\pi}_{e+1}^{1:i-1}, \boldsymbol{\pi}_e^{i:N}),i}$, such that under the policy $(\boldsymbol{\pi}_{e+1}^{1:i}, \boldsymbol{\pi}_e^{i+1:N})$, $H_{\hat{\boldsymbol{\pi}}_i}^i(\boldsymbol{x}_{t+1}) = 0 > (1-\epsilon)H_{\hat{\boldsymbol{\pi}}_i}^i(\boldsymbol{x}_t)$, where $\boldsymbol{x}_{t+1}$ follows the new joint policy $(\boldsymbol{\pi}_{e+1}^{1:i}, \boldsymbol{\pi}_e^{i+1:N})$, then, according to the previous analysis, the following states $\{\boldsymbol{x}_{t+2}, \boldsymbol{x}_{t+3}, \dots\}$ will stay within $\mathcal{X}_{safe}^{(\boldsymbol{\pi}_{e+1}^{1:i-1}, \boldsymbol{\pi}_e^{i:N}),i}$ under $(\boldsymbol{\pi}_{e+1}^{1:i}, \boldsymbol{\pi}_e^{i+1:N})$, and thus $\boldsymbol{x}_t \in \mathcal{X}_{safe}^{(\boldsymbol{\pi}_{e+1}^{1:i}, \boldsymbol{\pi}_e^{i+1:N}),i}$ given $b^i(\boldsymbol{x}_t) = 0$.*

*Based on the aforementioned analysis, it can be concluded that $\mathcal{X}_{safe}^{(\boldsymbol{\pi}_{e+1}^{1:i-1}, \boldsymbol{\pi}_e^{i:N}),i} \subseteq \mathcal{X}_{safe}^{(\boldsymbol{\pi}_{e+1}^{1:i}, \boldsymbol{\pi}_e^{i+1:N}),i}$, namely the safe set for agent $i$ will not shrink when other agents follow the fixed policy $\boldsymbol{\pi}_{e+1}^{1:i-1}$ and $\boldsymbol{\pi}_e^{i+1:N}$.*

### C.3  FORMAL VERSION AND PROOF FOR THEOREM 1

**Theorem 3** *Under Assumptions 1-6, the joint policy updated in Algorithm 1 will almost surely converge to a feasible local Nash equilibrium $\boldsymbol{\pi}^* = \{\pi_i^*\}_{i \in \mathcal{N}}$, where $H_{\boldsymbol{\pi}^*}^i(\boldsymbol{x}_t, \boldsymbol{u}_t) \geq 0$ for all $(\boldsymbol{x}_t, \boldsymbol{u}_t)$ along the trajectory induced by $\boldsymbol{\pi}^*$, for every $i \in \mathcal{N}$ and $t \geq 0$. Furthermore, define $H_{\boldsymbol{\pi}^*}^g(\boldsymbol{x}_t, \boldsymbol{u}_t) := \min_{i \in \mathcal{N}} H_{\boldsymbol{\pi}^*}^i(\boldsymbol{x}_t, \boldsymbol{u}_t)$, then $H_{\boldsymbol{\pi}^*}^g(\boldsymbol{x}_t, \boldsymbol{u}_t)$ is a global barrier certificate for the multi-agent system.*

**Proof 3** *This proof is extended from Yu et al. (2022b); Chow et al. (2018), but considers a differ-ent problem with additional squared terms to better achieve and maintain the satisfaction of the constraints, and include multiple agents, which necessitates additional analysis when updating the policy of agent $i$ to account for the inaccuracy caused by the updates of previous agents (namely from 1 to $i - 1$). The structure of the proof is as follows:*

*1. For the first updated agent, we first demonstrate that its critic and barrier certificate will converge to the fixed solution corresponding to the policy parameterized by $\{\theta_i^k\}_{i \in \mathcal{N}}$.*

*2. We show that the policy $\theta_1$ and the corresponding Lagrangian multiplier $\lambda_1$ will con-verge to a local saddle point with respect to $(\{\theta_i^k\}_{i \in \{2,...,N\}}, \{\lambda_i^k\}_{i \in \{2,...,N\}})$.*

*3. For the second updated agent, we similarly prove that the policy $\theta_2$ and $\lambda_2$ will converge to a local saddle point with respect to $(\{\theta_i^k\}_{i \in \{3,...,N\}}, \{\lambda_i^k\}_{i \in \{3,...,N\}})$. By induction, finally, the algorithm converge to a feasible local Nash equilibrium asymptotically.*

*We first consider the update of the first agent: When update the first agent, since all other agents are updated on slower timescales, here we treat $\{\theta_i^k\}_{i \in \{2,...,N\}}$ (namely $\boldsymbol{\theta}_{-1}$) and $\{\lambda_i^k\}_{i \in \{2,...,N\}}$ as cer-tain fixed values, and thus let $\bar{\boldsymbol{\pi}}_1 = (\theta_1^k, \theta_2^k, \ldots, \theta_N^k)$ where $\theta_1^k$ is being updated and $\{\theta_i^k\}_{i \in \{2,...,N\}}$ are fixed constants. Then, we have the following different timescales:*

***Convergence of*** $\phi_1$ ***and*** $\omega_1$***:*** *The Assumption 2 indicates that the updates of $\phi_1^k$ and $\omega_1^k$ converge on a faster timescale than policies $\{\theta_i^k\}_{i \in \mathcal{N}}$ and $\{\lambda^k\}_{i \in \mathcal{N}}$. According to Borkar (2008), we can deem $\{\theta_i^k\}_{i \in \mathcal{N}}$ and $\{\lambda^k\}_{i \in \mathcal{N}}$ as arbitrary fixed values when updating $\phi_1^k$ and $\omega_1^k$, and therefore, with the standard convergence results in Sutton & Barto (2018), we can easily know that for policy $\bar{\boldsymbol{\pi}}_1$,*

$$\mathcal{B}^{\bar{\boldsymbol{\pi}}_1}[Q^1](\boldsymbol{x}_t, \boldsymbol{u}_t) = r^1 + \gamma \mathbb{E}_{\boldsymbol{u}_{t+1} \sim \bar{\boldsymbol{\pi}}_1}\left[Q^1(\boldsymbol{x}_{t+1}, \boldsymbol{u}_{t+1})\right]$$
$$\mathcal{B}^{\bar{\boldsymbol{\pi}}_1}[H^1](\boldsymbol{x}_t, \boldsymbol{u}_t) = b^1 + \gamma \mathbb{E}_{\boldsymbol{u}_{t+1} \sim \bar{\boldsymbol{\pi}}_1}\left[H^1(\boldsymbol{x}_{t+1}, \boldsymbol{u}_{t+1})\right] \tag{13}$$

*are $\gamma-$contraction mappings, and therefore when $k \to \infty$, we have $\phi_1^k \to \phi_1^*(\{\theta_i^k\}_{i \in \mathcal{N}})$ where $Q^1(\boldsymbol{x}_t, \boldsymbol{u}_t; \phi_1^k) \to Q^1(\boldsymbol{x}_t, \boldsymbol{u}_t; \phi_1^*) = Q_{\bar{\boldsymbol{\pi}}_1}^1(\boldsymbol{x}_t, \boldsymbol{u}_t)$, and $\omega_1^k \to \omega_1^*(\{\theta_i^k\}_{i \in \mathcal{N}})$ where $H^1(\boldsymbol{x}_t, \boldsymbol{u}_t; \omega_1^k) \to H^1(\boldsymbol{x}_t, \boldsymbol{u}_t; \omega_1^*) = H_{\bar{\boldsymbol{\pi}}_1}^1(\boldsymbol{x}_t, \boldsymbol{u}_t)$. Therefore, the convergence of this timescale is proved. Furthermore, define*

$$\hat{r}_t^1(\boldsymbol{x}_t, \boldsymbol{u}_t) = \mathbb{E}_{\tilde{\boldsymbol{u}}_{t+1} \sim \bar{\boldsymbol{\pi}}_1}\left[max\{0, -H^1(\boldsymbol{x}_{t+1}, \tilde{\boldsymbol{u}}_{t+1}; \omega_1^k) + H^1(\boldsymbol{x}_t, \boldsymbol{u}_t; \omega_1^k) - \epsilon H^1(\boldsymbol{x}_t, \boldsymbol{u}_t; \omega_1^k)\}\right.$$
$$\left. + \frac{1}{2}[max\{0, -H^1(\boldsymbol{x}_{t+1}, \tilde{\boldsymbol{u}}_{t+1}; \omega_1^k) + H^1(\boldsymbol{x}_t, \boldsymbol{u}_t; \omega_1^k) - \epsilon H^1(\boldsymbol{x}_t, \boldsymbol{u}_t; \omega_1^k)\}]^2\right] \tag{14}$$

*and*

$$Q_{H_{\bar{\boldsymbol{\pi}}_1}}^1(\boldsymbol{x}, \boldsymbol{u}) = \mathbb{E}_{\tau \sim \bar{\boldsymbol{\pi}}_1}\left[\sum_{t=0}^{\infty} \gamma^t \hat{r}_t^1(\boldsymbol{x}_t, \boldsymbol{u}_t) | \boldsymbol{x}_0 = \boldsymbol{x}, \boldsymbol{u}_0 = \boldsymbol{u}\right], \tag{15}$$

*then, given a step size $\eta_{Q_H}(e)$ such that $\sum_e \eta_{Q_H}(e) = \infty$, $\sum_e \eta_{Q_H}^2 < \infty$, $\eta_{Q_H}(e) = \mathcal{O}(\eta_1(e))$, $\eta_{2,1}(e) = \mathcal{O}(\eta_{Q_H}(e))$, if parameterize this $Q_{H_{\bar{\boldsymbol{\pi}}_1}}^1$ with $\hat{\omega}_1$, we have the update by us-ing some smapled data:*

$$\hat{\omega}_1^{k+1} = \Gamma_{\hat{\Omega}_1}\left[\hat{\omega}_1^k - \eta_{Q_H} \nabla_{\hat{\omega}_1}(\hat{r}_t^1 + \gamma Q_H^1(\boldsymbol{x}_{t+1}, \boldsymbol{u}_{t+1}; \hat{\omega}_1) - Q_H^1(\boldsymbol{x}_t, \boldsymbol{u}_t; \hat{\omega}_1))^2|_{\hat{\omega}_1 = \hat{\omega}_1^k}\right]. \tag{16}$$

*Let*

$$\hat{r}_t^{1,*}(\boldsymbol{x}_t, \boldsymbol{u}_t) = \mathbb{E}_{\tilde{\boldsymbol{u}}_{t+1} \sim \bar{\boldsymbol{\pi}}_1}\left[max\{0, -H_{\bar{\boldsymbol{\pi}}_1}^1(\boldsymbol{x}_{t+1}, \tilde{\boldsymbol{u}}_{t+1}) + H_{\bar{\boldsymbol{\pi}}_1}^1(\boldsymbol{x}_t, \boldsymbol{u}_t) - \epsilon H_{\bar{\boldsymbol{\pi}}_1}^1(\boldsymbol{x}_t, \boldsymbol{u}_t)\}\right.$$
$$\left. + \frac{1}{2}[max\{0, -H_{\bar{\boldsymbol{\pi}}_1}^1(\boldsymbol{x}_{t+1}, \tilde{\boldsymbol{u}}_{t+1}) + H_{\bar{\boldsymbol{\pi}}_1}^1(\boldsymbol{x}_t, \boldsymbol{u}_t) - \epsilon H_{\bar{\boldsymbol{\pi}}_1}^1(\boldsymbol{x}_t, \boldsymbol{u}_t)\}]^2\right], \tag{17}$$

*we further have an ideal update*

$$\hat{\omega}_1^{k+1} = \Gamma_{\hat{\Omega}_1}\left[\hat{\omega}_1^k - \eta_{Q_H} \nabla_{\hat{\omega}_1}(\hat{r}_t^{1,*} + \gamma Q_H^1(\boldsymbol{x}_{t+1}, \boldsymbol{u}_{t+1}; \hat{\omega}_1) - Q_H^1(\boldsymbol{x}_t, \boldsymbol{u}_t; \hat{\omega}_1))^2|_{\hat{\omega}_1 = \hat{\omega}_1^k}\right]. \tag{18}$$

*The difference between (16) and (18) is the use of signal ($\hat{r}_t^1$ or $\hat{r}_t^{1,*}$), which reveals the inaccuracy caused in the update of $\omega_1$ in the previous timescale. Then, we can rewrite the real update when the*

*sampled batch D follows the distribution caused by $\bar{\pi}_1$ to be:*

$$\hat{\omega}_1^{k+1} = \Gamma_{\hat{\Omega}_1} \left[ \hat{\omega}_1^k - \eta_{Q_H} \nabla_{\hat{\omega}_1} \left( (\hat{r}_t^{1,*} + \gamma Q_H^1(\boldsymbol{x}_{t+1}, \boldsymbol{u}_{t+1}; \hat{\omega}_1) - Q_H^1(\boldsymbol{x}_t, \boldsymbol{u}_t; \hat{\omega}_1))^2 |_{\hat{\omega}_1 = \hat{\omega}_1^k} \right. \right.$$
$$\left. \left. + \delta\hat{\omega}_1^{k+1} + \delta\hat{\omega}_{1,\varepsilon}) \right]. \tag{19}$$

*where*

$$\delta\hat{\omega}_1^{k+1} = -\mathbb{E}_{\bar{\pi}_1} \left[ \nabla_{\hat{\omega}_1}(\hat{r}_t^1 + \gamma Q_H^1(\boldsymbol{x}_{t+1}, \boldsymbol{u}_{t+1}; \hat{\omega}_1) - Q_H^1(\boldsymbol{x}_t, \boldsymbol{u}_t; \hat{\omega}_1))^2 |_{\hat{\omega}_1 = \hat{\omega}_1^k} \right]$$
$$+ \nabla_{\hat{\omega}_1}(\hat{r}_t^1 + \gamma Q_H^1(\boldsymbol{x}_{t+1}, \boldsymbol{u}_{t+1}; \hat{\omega}_1) - Q_H^1(\boldsymbol{x}_t, \boldsymbol{u}_t; \hat{\omega}_1))^2 |_{\hat{\omega}_1 = \hat{\omega}_1^k} \tag{20}$$

*and*

$$\delta\hat{\omega}_{1,\varepsilon} = \mathbb{E}_{\bar{\pi}_1} \left[ \nabla_{\hat{\omega}_1}(\hat{r}_t^1 + \gamma Q_H^1(\boldsymbol{x}_{t+1}, \boldsymbol{u}_{t+1}; \hat{\omega}_1) - Q_H^1(\boldsymbol{x}_t, \boldsymbol{u}_t; \hat{\omega}_1))^2 |_{\hat{\omega}_1 = \hat{\omega}_1^k} \right.$$
$$\left. - \nabla_{\hat{\omega}_1}(\hat{r}_t^{1,*} + \gamma Q_H^1(\boldsymbol{x}_{t+1}, \boldsymbol{u}_{t+1}; \hat{\omega}_1) - Q_H^1(\boldsymbol{x}_t, \boldsymbol{u}_t; \hat{\omega}_1))^2 |_{\hat{\omega}_1 = \hat{\omega}_1^k} \right] \tag{21}$$

*are two terms representing the difference between the real update and the ideal update.*

*Under Assumtion 3:*

*1. We show that $\delta\hat{\omega}_1^{k+1}$ is square integrable first. To be specific,*

$$\mathbb{E}\left[ ||\delta\hat{\omega}_1^{k+1}||^2 | \mathcal{F}_{\hat{\omega}_1, k} \right] \leq C_1 ||\nabla_{\hat{\omega}_1} Q_H^1(\boldsymbol{x}_t, \boldsymbol{u}_t; \hat{\omega}_1^k)||_\infty^2 \times (||\hat{r}_t^1 + \gamma Q_H^1(\boldsymbol{x}_{t+1}, \boldsymbol{u}_{t+1}; \hat{\omega}_1^k)||_\infty^2$$
$$+ ||Q_H^1(\boldsymbol{x}_t, \boldsymbol{u}_t; \hat{\omega}_1^k)||_\infty^2) \tag{22}$$

*where $\mathcal{F}_{\hat{\omega}_1, k} = \sigma(\hat{\omega}_1^m, \delta\hat{\omega}_1^m, m \leq k)$ is the filtration of $\hat{\omega}_1^k$ generated by different independent trajectories (Chow et al., 2018). Furthermore, Assumption 3 requires that $||\nabla_{\hat{\omega}_1} Q_H^1(\boldsymbol{x}_t, \boldsymbol{u}_t; \hat{\omega}_1)||_\infty^2 \leq K_1(1 + ||\hat{\omega}_1||_\infty^2)$ and other terms are bounded, therefore, $\mathbb{E}\left[ ||\delta\hat{\omega}_1^{k+1}||^2 | \mathcal{F}_{\hat{\omega}_1, k} \right] < \infty$, and $\delta\hat{\omega}_1^{k+1}$ is square integrable.*

*2. We then show that $\delta\hat{\omega}_{1,\varepsilon} \to 0$:*

$$||\delta\hat{\omega}_{1,\varepsilon}|| \leq C_2 ||\nabla_{\hat{\omega}_1} Q_H^1(\boldsymbol{x}_t, \boldsymbol{u}_t; \hat{\omega}_1)||_\infty ||\mathbb{E}_{\bar{\pi}_1} \left[ 2(\hat{r}_t^{1,*} + \gamma Q_H^1(\boldsymbol{x}_{t+1}, \boldsymbol{u}_{t+1}; \hat{\omega}_1^k) - Q_H^1(\boldsymbol{x}_t, \boldsymbol{u}_t; \hat{\omega}_1^k)) \right.$$
$$\left. - 2(\hat{r}_t^1 + \gamma Q_H^1(\boldsymbol{x}_{t+1}, \boldsymbol{u}_{t+1}; \hat{\omega}_1^k) - Q_H^1(\boldsymbol{x}_t, \boldsymbol{u}_t; \hat{\omega}_1^k)) \right]|| \tag{23}$$

*Since $||\nabla_{\hat{\omega}_1} Q_H^1(\boldsymbol{x}_t, \boldsymbol{u}_t; \hat{\omega}_1)||$ is upper bounded, and we have $\hat{r}_t^1 \to \hat{r}_t^{1,*}$ given $H^1(\boldsymbol{x}_t, \boldsymbol{u}_t; \omega_1^k) \to H_{\bar{\pi}_1}^1(\boldsymbol{x}_t, \boldsymbol{u}_t)$ according to the analysis in the previous timescale, we have $||\delta\hat{\omega}_{1,\varepsilon}|| \to 0$, and therefore $\delta\hat{\omega}_{1,\varepsilon} \to 0$.*

*3. Because the $\nabla_{\hat{\omega}_1}(\hat{r}_t^1 + \gamma Q_H^1(\boldsymbol{x}_{t+1}, \boldsymbol{u}_{t+1}; \hat{\omega}_1) - Q_H^1(\boldsymbol{x}_t, \boldsymbol{u}_t; \hat{\omega}_1))^2 |_{\hat{\omega}_1 = \hat{\omega}_1^k}$ is a sample of $\mathbb{E}_{\bar{\pi}_1} \left[ \nabla_{\hat{\omega}_1}(\hat{r}_t^1 + \gamma Q_H^1(\boldsymbol{x}_{t+1}, \boldsymbol{u}_{t+1}; \hat{\omega}_1) - Q_H^1(\boldsymbol{x}_t, \boldsymbol{u}_t; \hat{\omega}_1))^2 |_{\hat{\omega}_1 = \hat{\omega}_1^k} \right]$, we can conclude that $\mathbb{E}\left[ \delta\hat{\omega}_1^{k+1} | \mathcal{F}_{\hat{\omega}_1, k} \right] = 0$.*

*When $\mathbb{E}_{\bar{\pi}_1} \left[ \nabla_{\hat{\omega}_1}(\hat{r}_t^1 + \gamma Q_H^1(\boldsymbol{x}_{t+1}, \boldsymbol{u}_{t+1}; \hat{\omega}_1) - Q_H^1(\boldsymbol{x}_t, \boldsymbol{u}_t; \hat{\omega}_1))^2 |_{\hat{\omega}_1 = \hat{\omega}_1^k} \right]$ is a Lipschitz function in $\hat{\omega}_1$, following a similar way which will be detailed in the update of policy $\theta_1$ in the next step, we conclude that $\hat{\omega}_1^k$ converges to a minimizer $\hat{\omega}_1^*$. Following the analysis of policy evaluation in previous studies, assume that a sufficiently expressive function approximator is used such that there is only one local minimum, within the space $\hat{\Omega}_1$, which minimizes the TD error to 0, then we have shown that $\hat{\omega}_1^k \to \hat{\omega}_1^*(\{\theta_i^k\}_{i \in \mathcal{N}})$ where $Q_H^1(\boldsymbol{x}_t, \boldsymbol{u}_t; \hat{\omega}_1^k) \to Q_H^1(\boldsymbol{x}_t, \boldsymbol{u}_t; \hat{\omega}_1^*) = Q_{H_{\bar{\pi}_1}}^1(\boldsymbol{x}_t, \boldsymbol{u}_t)$.*

***Convergence of $\theta_1$:*** *Since $\{\theta_i^k\}_{i \in \{2,...,N\}}$ and $\{\lambda_i^k\}_{i \in \{1,...,N\}}$ are updated on slower timescales than $\theta_1$, again, according to Borkar (2008),we treat $\{\theta_i^k\}_{i \in \{2,...,N\}}$ and $\{\lambda_i^k\}_{i \in \{1,...,N\}}$ as fixed values when update $\theta_1$, and additionally, according to aforementioned analysis we have $||Q^1(\boldsymbol{x}_t, \boldsymbol{u}_t; \phi_1^k) - Q_{\bar{\pi}_1}^1(\boldsymbol{x}_t, \boldsymbol{u}_t)|| \to 0$ and $||Q_H^1(\boldsymbol{x}_t, \boldsymbol{u}_t; \hat{\omega}_1^k) - Q_{H_{\bar{\pi}_1}}^1(\boldsymbol{x}_t, \boldsymbol{u}_t)|| \to 0$ almost surely. With the function $Q_{H_{\bar{\pi}_1}}^1(\boldsymbol{x}_t, \boldsymbol{u}_t)$, by using data following $\bar{\pi}_1$ and extending the constraint from being based on $Q_{H_{\bar{\pi}_1}}^1(\boldsymbol{x}_0, \boldsymbol{u}_0)$ to being based on general $Q_{H_{\bar{\pi}_1}}^1(\boldsymbol{x}_t, \boldsymbol{u}_t)$, the ideal Lagranian function can be written as:*

$$\mathcal{L}_{\boldsymbol{\theta}^{-1}}(\theta_1, \lambda_1) = \mathbb{E}_{\bar{\pi}_1} \left[ -Q_{\bar{\pi}_1}^1(\boldsymbol{x}_t, \boldsymbol{u}_t) + \lambda_1 Q_{H_{\bar{\pi}_1}}^1(\boldsymbol{x}_t, \boldsymbol{u}_t) \right]. \tag{24}$$

*It is noteworthy that, although looks like constraints commonly used in CMDP settings, due to the application of the max function, only the states where constraints are violated will be taken into consideration, and therefore safety constraints are imposed on each state. Besides, because of the use of barrier certificate $H_{\bar{\pi}_1}^1$, one violation of safety constraints happening at timestep $t$ can be recorded and reflected in all $\hat{r}_k^{1,*}, k \leq t$ through the calculation of difference (17), and therefore this violation can be sufficiently suppressed during the update. Later, by using the sampled data to update $\theta_1$, we have:*

$$\theta_1^{k+1} = \Gamma_{\Theta_1}\big[\theta_1^k - \eta_{2,1}\nabla_{u_t^1}(-Q^1(\boldsymbol{x}_t, \boldsymbol{u}_t; \phi_1^k) + \lambda_1^k Q_H^1(\boldsymbol{x}_t, \boldsymbol{u}_t; \hat{\omega}_1^k))|_{u_t^1 = \pi^1(\boldsymbol{x}_t; \theta_1^k)}\nabla_{\theta_1}\pi^1(\boldsymbol{x}_t; \theta_1)|_{\theta_1 = \theta_1^k}\big]$$

$$= \Gamma_{\Theta_1}\big[\theta_1^k - \eta_{2,1}[\nabla_{\theta_1}\mathcal{L}_{\boldsymbol{\theta}^{-1}}(\theta_1, \lambda_1)|_{\theta_1 = \theta_1^k} + \delta\theta_1^{k+1} + \delta\theta_{1,\varepsilon}]\big] \tag{25}$$

*where*

$$\delta\theta_1^{k+1} = -\mathbb{E}_{\bar{\boldsymbol{\pi}}_1}\big[\nabla_{u_t^1}(-Q^1(\boldsymbol{x}_t, \boldsymbol{u}_t; \phi_1^k) + \lambda_1^k Q_H^1(\boldsymbol{x}_t, \boldsymbol{u}_t; \hat{\omega}_1^k))|_{u_t^1 = \pi^1(\boldsymbol{x}_t; \theta_1^k)}\nabla_{\theta_1}\pi^1(\boldsymbol{x}_t; \theta_1)|_{\theta_1 = \theta_1^k}\big]$$

$$+ \nabla_{u_t^1}(-Q^1(\boldsymbol{x}_t, \boldsymbol{u}_t; \phi_1^k) + \lambda_1^k Q_H^1(\boldsymbol{x}_t, \boldsymbol{u}_t; \hat{\omega}_1^k))|_{u_t^1 = \pi^1(\boldsymbol{x}_t; \theta_1^k)}\nabla_{\theta_1}\pi^1(\boldsymbol{x}_t; \theta_1)|_{\theta_1 = \theta_1^k} \tag{26}$$

*and*

$$\delta\theta_{1,\varepsilon} = \mathbb{E}_{\bar{\boldsymbol{\pi}}_1}\big[\nabla_{u_t^1}(-Q^1(\boldsymbol{x}_t, \boldsymbol{u}_t; \phi_1^k) + \lambda_1^k Q_H^1(\boldsymbol{x}_t, \boldsymbol{u}_t; \hat{\omega}_1^k))|_{u_t^1 = \pi^1(\boldsymbol{x}_t; \theta_1^k)}\nabla_{\theta_1}\pi^1(\boldsymbol{x}_t; \theta_1)|_{\theta_1 = \theta_1^k}$$

$$- \nabla_{u_t^1}(-Q_{\bar{\boldsymbol{\pi}}_1}^1(\boldsymbol{x}_t, \boldsymbol{u}_t) + \lambda_1^k Q_{H_{\bar{\boldsymbol{\pi}}_1}}^1(\boldsymbol{x}_t, \boldsymbol{u}_t))|_{u_t^1 = \pi^1(\boldsymbol{x}_t; \theta_1^k)}\nabla_{\theta_1}\pi^1(\boldsymbol{x}_t; \theta_1)|_{\theta_1 = \theta_1^k}\big] \tag{27}$$

*Under Assumption 5:*

*1. We show that $\delta\theta_1^{k+1}$ is square integrable first. To be specific,*

$$\mathbb{E}\big[||\delta\theta_1^{k+1}||^2|\mathcal{F}_{\theta_1, k}\big] \leq C_3 ||\nabla_{\theta_1}\pi^1(\boldsymbol{x}_t; \theta_1)|_{\theta_1 = \theta_1^k}||_\infty^2 \times (||\nabla_{u_t^1}Q^1(\boldsymbol{x}_t, \boldsymbol{u}_t; \phi_1^k)||_\infty^2$$

$$+ ||\lambda_1||_\infty^2 ||\nabla_{u_t^1}Q_H^1(\boldsymbol{x}_t, \boldsymbol{u}_t; \hat{\omega}_1^k)||_\infty^2) \tag{28}$$

*where $\mathcal{F}_{\theta_1, k} = \sigma(\theta_1^m, \delta\theta_1^m, m \leq k)$ is the filtration of $\theta_1^k$ generated by different independent trajectories. Based on Assumption 5, we know:*

$$||\nabla_{\theta_1}\pi^1(\boldsymbol{x}_t; \theta_1)|_{\theta_1 = \theta_1^k}||_\infty^2 \leq K_2(1 + ||\theta_i^k||_\infty^2)$$

$$||\nabla_{u_t^1}Q^1(\boldsymbol{x}_t, \boldsymbol{u}_t; \phi_1^k)||_\infty^2 \leq K_3(1 + ||u_t^1||_\infty^2) \tag{29}$$

$$||\nabla_{u_t^1}Q_H^1(\boldsymbol{x}_t, \boldsymbol{u}_t; \hat{\omega}_1^k)||_\infty^2 \leq K_4(1 + ||u_t^1||_\infty^2)$$

*where $K_2, K_3, K_4$ are constants and $||\lambda_1||_\infty^2$ is bounded. Therefore, $\mathbb{E}\big[||\delta\theta_1^{k+1}||^2|\mathcal{F}_{\theta_1, k}\big] \leq C_3 K_2(1 + ||\theta_i^k||_\infty^2)[K_3(1 + ||u_t^1||_\infty^2) + ||\lambda_1||_\infty^2 K_4(1 + ||u_t^1||_\infty^2)] < \infty$, and $\delta\theta_1^{k+1}$ is square integrable.*

*2. Later, we have:*

$$\delta\theta_{1,\varepsilon} = \mathbb{E}_{\bar{\boldsymbol{\pi}}_1}\big[\nabla_{u_t^1}(Q_{\bar{\boldsymbol{\pi}}_1}^1(\boldsymbol{x}_t, \boldsymbol{u}_t) - Q^1(\boldsymbol{x}_t, \boldsymbol{u}_t; \phi_1^k) + \lambda_1^k Q_H^1(\boldsymbol{x}_t, \boldsymbol{u}_t; \hat{\omega}_1^k) - \lambda_1^k Q_{H_{\bar{\boldsymbol{\pi}}_1}}^1(\boldsymbol{x}_t, \boldsymbol{u}_t))|_{u_t^1 = \pi^1(\boldsymbol{x}_t; \theta_1^k)}$$

$$\nabla_{\theta_1}\pi^1(\boldsymbol{x}_t; \theta_1)|_{\theta_1 = \theta_1^k} = \mathbb{E}_{\bar{\boldsymbol{\pi}}_1}\big[\nabla_{u_t^1}(Q^1(\boldsymbol{x}_t, \boldsymbol{u}_t; \phi_1^*(\{\theta_i^k\}_{i \in \mathcal{N}})) - Q^1(\boldsymbol{x}_t, \boldsymbol{u}_t; \phi_1^k)$$

$$+ \lambda_1^k Q_H^1(\boldsymbol{x}_t, \boldsymbol{u}_t; \hat{\omega}_1^k) - \lambda_1^k Q_H^1(\boldsymbol{x}_t, \boldsymbol{u}_t; \hat{\omega}_1^*(\{\theta_i^k\}_{i \in \mathcal{N}})))|_{u_t^1 = \pi^1(\boldsymbol{x}_t; \theta_1^k)}\nabla_{\theta_1}\pi^1(\boldsymbol{x}_t; \theta_1)|_{\theta_1 = \theta_1^k}$$

$$\leq \mathbb{E}_{\bar{\boldsymbol{\pi}}_1}\big[\nabla_{\theta_1}\pi^1(\boldsymbol{x}_t; \theta_1)|_{\theta_1 = \theta_1^k}\big](K_5||\phi_1^k - \phi_1^*||_\infty + ||\lambda_1||_\infty^2 K_6||\hat{\omega}_1^k - \hat{\omega}_1^*||_\infty) \to 0 \tag{30}$$

*since in the previous timescale, we showed $\phi_1^k \to \phi_1^*(\{\theta_i^k\}_{i \in \mathcal{N}})$ and $\hat{\omega}_1^k \to \hat{\omega}_1^*(\{\theta_i^k\}_{i \in \mathcal{N}})$ when $k \to \infty$.*

*3. Since $\nabla_{\theta_1}(-Q^1(\boldsymbol{x}_t, \boldsymbol{u}_t; \phi_1^k) + \lambda_1^k Q_H^1(\boldsymbol{x}_t, \boldsymbol{u}_t; \hat{\omega}_1^k))$ is a sample of $\nabla_{\theta_1}\mathcal{L}_{\boldsymbol{\theta}^{-1}}(\theta_1, \lambda_1)|_{\theta_1 = \theta_1^k}$, we have that $\mathbb{E}\big[\delta\theta_1^{k+1}|\mathcal{F}_{\theta_1, k}\big] = 0$*

*Due to these three facts, the policy $\theta_1$ update is a stochastic approximation of a continuous system $\theta_i(t)$ (Borkar, 2008) represented by:*

$$\dot{\theta}_1(t) = \Upsilon_{\Theta_1}\big[-\nabla_{\theta_1}\mathcal{L}_{\boldsymbol{\theta}^{-1}}(\theta_1, \lambda_1)\big] \tag{31}$$

*where*

$$\Upsilon_{\Theta_1}\big[F(\theta_1)\big] \triangleq \lim_{\eta \to 0^+} \frac{\Gamma_{\Theta_1}(\theta_1 + \eta F(\theta_1)) - \Gamma_{\Theta_1}(\theta_1)}{\eta} \tag{32}$$

*is the left directional derivative of the function $\Gamma_{\Theta_1}(\theta_1)$ in the direction of $F(\theta_1)$. According to Chow et al. (2018), we have:*

$$\frac{d\mathcal{L}_{\boldsymbol{\theta}^{-1}}(\theta_1, \lambda_1)}{dt} = \nabla_{\theta_1}\mathcal{L}_{\boldsymbol{\theta}^{-1}}(\theta_1, \lambda_1)^T \Upsilon_{\Theta_1}\big[ -\nabla_{\theta_1}\mathcal{L}_{\boldsymbol{\theta}^{-1}}(\theta_1, \lambda_1)\big] \leq 0 \tag{33}$$

*which is non-zero if $||\Upsilon_{\Theta_1}\big[-\nabla_{\theta_1}\mathcal{L}_{\boldsymbol{\theta}^{-1}}(\theta_1, \lambda_1)\big]|| \neq 0$. Therefore, for given fixed $\{\theta_i^k\}_{i\in\{2,...,N\}}$ and $\{\lambda_i^k\}_{i\in\mathcal{N}}$, define a Lyapunov function:*

$$L_{\{\theta_i^k\}_{i\in\{2,...,N\}},\{\lambda_i^k\}_{i\in\mathcal{N}}}(\theta_1) = \mathcal{L}_{\boldsymbol{\theta}^{-1}}(\theta_1, \lambda_1^k) - \mathcal{L}_{\boldsymbol{\theta}^{-1}}(\theta_1^*, \lambda_1^k) \tag{34}$$

*where $\theta_1^*$ is a local minimum point. Then, there exists a ball centered at $\theta_1^*$ with a radius $r_{\theta_1}$ such that $\forall \theta_1 \in B_{r_{\theta_1}}(\theta_1^*) = \{\theta_1 \mid ||\theta_1 - \theta_1^*|| \leq r_{\theta_1}\}$, $L_{\{\theta_i^k\}_{i\in\{2,...,N\}},\{\lambda_i^k\}_{i\in\mathcal{N}}}(\theta_1) \geq 0$. Given $d\mathcal{L}_{\boldsymbol{\theta}^{-1}}(\theta_1, \lambda_1)/dt \leq 0$, according to the Lyapunov theory for stable systems (Khalil & Grizzle, 2002), for any initial $\theta_1(0) \in B_{r_{\theta_1}}(\theta_1^*)$, the trajectory $\theta_1(t)$ converges to $\theta_1^*$, and thus $\mathcal{L}_{\boldsymbol{\theta}^{-1}}(\theta_1^*, \lambda_1) \leq \mathcal{L}_{\boldsymbol{\theta}^{-1}}(\theta_1(t), \lambda_1) \leq \mathcal{L}_{\boldsymbol{\theta}^{-1}}(\theta_1(0), \lambda_1), \forall t \geq 0$. Based on all aforementioned analysis and Chow et al. (2018, Proposition 17), as well as the fact that $\theta_1^k$ is bounded in set $\Theta_1$, according to Borkar (2008, Theorem 2, Chapter 6), the sequence $\{\theta_1^k\}$ converges almost surely to the locally minimum point $\theta_1^*$.*

***Convergence of $\lambda_1$:*** *Since $\lambda_1$ is updated at a slower timescale than $\phi_1^k$, $\hat{\omega}_1^k$ and $\theta_1^k$, we have that, given fixed $\{\theta_i^k\}_{i\in\{2,...,N\}}$ and $\{\lambda_i^k\}_{i\in\{2,...,N\}}$, $||\theta_1^k - \theta_1^*(\lambda_1^k)|| = 0$, $||Q^1(\boldsymbol{x}_t, \boldsymbol{u}_t; \phi_1^k) - Q_{\bar{\pi}_1}^1(\boldsymbol{x}_t, \boldsymbol{u}_t)|| = 0$ and $||Q_H^1(\boldsymbol{x}_t, \boldsymbol{u}_t; \hat{\omega}_1^k) - Q_{H_{\bar{\pi}_1}}^1(\boldsymbol{x}_t, \boldsymbol{u}_t)|| = 0$. With the continuity of $\nabla_{\lambda_1}\mathcal{L}_{\boldsymbol{\theta}^{-1}}(\theta_1, \lambda_1)$, we have $||\nabla_{\lambda_1}\mathcal{L}_{\boldsymbol{\theta}^{-1}}(\theta_1, \lambda_1)|_{\theta_1=\theta_1^k, \lambda_1=\lambda_1^k} - \nabla_{\lambda_1}\mathcal{L}_{\boldsymbol{\theta}^{-1}}(\theta_1, \lambda_1)|_{\theta_1=\theta_1^*(\lambda_1^k), \lambda_1=\lambda_1^k}|| = 0$ almost surely. Thus, the update of $\lambda_1$ based on the sampled data is:*

$$\begin{aligned}
\lambda_1^{k+1} &= \Gamma_{\Lambda_1}\big[\lambda_1^k + \eta_{3,1}Q_H^1(\boldsymbol{x}_t, \boldsymbol{u}_t; \hat{\omega}_1^k)\big] \\
&= \Gamma_{\Lambda_1}\big[\lambda_1^k + \eta_{3,1}[\nabla_{\lambda_1}\mathcal{L}_{\boldsymbol{\theta}^{-1}}(\theta_1, \lambda_1)|_{\theta_1=\theta_1^*(\lambda_1^k), \lambda_1=\lambda_1^k} + \delta\lambda_1^{k+1}]\big]
\end{aligned} \tag{35}$$

*where*

$$\begin{aligned}
\delta\lambda_1^{k+1} &= -\mathbb{E}\big[Q_{H_{\bar{\pi}_1^*}}^1(\boldsymbol{x}_t, \bar{\pi}_1^*(\boldsymbol{x}_t))\big] + Q_H^1(\boldsymbol{x}_t, \bar{\pi}_1(\boldsymbol{x}_t); \hat{\omega}_1^k) \\
&= -\mathbb{E}\big[Q_{H_{\bar{\pi}_1^*}}^1(\boldsymbol{x}_t, \bar{\pi}_1^*(\boldsymbol{x}_t))\big] + Q_{H_{\bar{\pi}_1}}^1(\boldsymbol{x}_t, \bar{\pi}_1(\boldsymbol{x}_t)) - Q_{H_{\bar{\pi}_1}}^1(\boldsymbol{x}_t, \bar{\pi}_1(\boldsymbol{x}_t)) + Q_H^1(\boldsymbol{x}_t, \bar{\pi}_1(\boldsymbol{x}_t); \hat{\omega}_1^k)
\end{aligned} \tag{36}$$

*Here, $\bar{\pi}_1^*$ is the joint policy parameterized by $\{\theta_1^*(\lambda_1^k), \theta_2^k, \ldots, \theta_N^k\}$ where $\theta_1^*(\lambda_1^k)$ is the locally minimum solution given a fixed $\lambda_1^k$, and $\{\theta_i^k\}_{i\in\{2,...,N\}}$ are certain policies for other agents. Similarly, here we have:*

$$\mathbb{E}\big[||\delta\lambda_1^{k+1}||^2|\mathcal{F}_{\lambda_1,k}\big] \leq C_4 \max_{\boldsymbol{x}_t, \boldsymbol{u}_t, \hat{\omega}_1^k} ||Q_H^1(\boldsymbol{x}_t, \boldsymbol{u}_t; \hat{\omega}_1^k)||_\infty^2 < \infty \tag{37}$$

*according to Assumption 3, where $\mathcal{F}_{\lambda_1,k} = \sigma(\lambda_1^m, \delta\lambda_1^m, m \leq k)$ is the filtration of $\lambda_1^k$ generated by different independent trajectories, and since $||Q_H^1(\boldsymbol{x}_t, \bar{\pi}_1(\boldsymbol{x}_t); \hat{\omega}_1^k) - Q_{H_{\bar{\pi}_1}}^1(\boldsymbol{x}_t, \bar{\pi}_1(\boldsymbol{x}_t))|| = 0$, and $Q_{H_{\bar{\pi}_1}}^1(\boldsymbol{x}_t, \bar{\pi}_1(\boldsymbol{x}_t))$ is a sample of $\nabla_{\lambda_1}\mathcal{L}_{\boldsymbol{\theta}^{-1}}(\theta_1, \lambda_1)|_{\theta_1=\theta_1^*(\lambda_1^k), \lambda_1=\lambda_1^k}$, we have $\mathbb{E}\big[\delta\lambda_1^{k+1}|\mathcal{F}_{\lambda_1,k}\big] = 0$ almost surely. Thus, similar to previous analysis, the $\lambda_1$'s update is a stochastic approximation of the following system:*

$$\dot{\lambda}_1(t) = \Upsilon_{\Lambda_1}\big[\nabla_{\lambda_1}\mathcal{L}_{\boldsymbol{\theta}^{-1}}(\theta_1, \lambda_1)|_{\theta_1=\theta_1^*(\lambda_1)}\big]. \tag{38}$$

*Similarly, we have:*

$$\frac{d\mathcal{L}_{\boldsymbol{\theta}^{-1}}(\theta_1^*(\lambda_1), \lambda_1)}{dt} = \nabla_{\lambda_1}\mathcal{L}_{\boldsymbol{\theta}^{-1}}(\theta_1, \lambda_1)|_{\theta_1=\theta_1^*(\lambda_1)}^T \Upsilon_{\Lambda_1}\big[\nabla_{\lambda_1}\mathcal{L}_{\boldsymbol{\theta}^{-1}}(\theta_1, \lambda_1)|_{\theta_1=\theta_1^*(\lambda_1)}\big] \geq 0 \tag{39}$$

For a local maximum point $\lambda_1^*$, define the following Lyapunov function:

$$L_{\{\theta_i^k\}_{i\in\{2,\ldots,N\}},\{\lambda_i^k\}_{i\in\{2,\ldots,N\}}}(\lambda_1) = \mathcal{L}_{\boldsymbol{\theta}^{-1}}(\theta_1^*(\lambda_1^*),\lambda_1^*) - \mathcal{L}_{\boldsymbol{\theta}^{-1}}(\theta_1^*(\lambda_1),\lambda_1),\qquad(40)$$

there exists a ball centered at $\lambda_1^*$ with a radius $r_{\lambda_1}$ such that $\forall \lambda_1 \in B_{r_{\lambda_1}}(\lambda_1^*) = \{\lambda_1 \mid ||\lambda_1 - \lambda_1^*|| \leq r_{\lambda_1}\}$, $L_{\{\theta_i^k\}_{i\in\{2,\ldots,N\}},\{\lambda_i^k\}_{i\in\{2,\ldots,N\}}}(\lambda_1) \geq 0$. Furthermore,

$$dL_{\{\theta_i^k\}_{i\in\{2,\ldots,N\}},\{\lambda_i^k\}_{i\in\{2,\ldots,N\}}}(\lambda_1)/dt = -d\mathcal{L}_{\boldsymbol{\theta}^{-1}}(\theta_1^*(\lambda_1),\lambda_1)/dt \leq 0.\qquad(41)$$

Therefore, starting from $\lambda_1(0) \in B_{r_{\lambda_1}}(\lambda_1^*)$, the trajectory $\lambda_1(t)$ converges to $\lambda_1^*$, which is a locally maximum point. Following the similar process in the update of $\theta_1$, we conclude that $\{\lambda_1^k\}$ converges almost surely to $\lambda_1^*$, with $\mathcal{L}_{\boldsymbol{\theta}^{-1}}(\theta_1^*(\lambda_1^*),\lambda_1^*) \geq \mathcal{L}_{\boldsymbol{\theta}^{-1}}(\theta_1^*(\lambda_1),\lambda_1)$, $\forall t \geq 0$.

From the aforementioned analysis, we have $\mathcal{L}_{\boldsymbol{\theta}^{-1}}(\theta_1^*,\lambda_1) \leq \mathcal{L}_{\boldsymbol{\theta}^{-1}}(\theta_1^*,\lambda_1^*) \leq \mathcal{L}_{\boldsymbol{\theta}^{-1}}(\theta_1,\lambda_1^*)$. Therefore, $(\theta_1^*,\lambda_1^*)$ is a locally saddle point for the certain fixed $\{\theta_i^k\}_{i\in\{2,\ldots,N\}}$ and $\{\lambda_i^k\}_{i\in\{2,\ldots,N\}}$, and the update of $\theta_1$ and $\lambda_1$ will converge to the feasible locally optimal policy and multiplier with respect to its value function for the certain fixed set of policies $\{\theta_i^k\}_{i\in\{2,\ldots,N\}}$ and multipliers $\{\lambda_i^k\}_{i\in\{2,\ldots,N\}}$ of other agents that have not been updated.

Later we analysis the update of the second agent. For the agent 2, according to the previous analysis, it is already obtained that $\lambda_1^k \to \lambda_1^*(\{\theta_i^k\}_{i\in\{2,\ldots,N\}},\{\lambda_i^k\}_{i\in\{2,\ldots,N\}})$ and $\theta_1^k \to \theta_1^*(\{\theta_i^k\}_{i\in\{2,\ldots,N\}},\{\lambda_i^k\}_{i\in\{2,\ldots,N\}})$ when we update $\theta_2^k$ and $\lambda_2^k$. Since agent 2 is updated at a quicker timescale than agent 3 to agent $N$, here we still have $\{\theta_i^k\}_{i\in\{3,\ldots,N\}}$ and $\{\lambda_i^k\}_{i\in\{3,\ldots,N\}}$ as fixed sets. Let $\bar{\boldsymbol{\pi}}_2 = (\theta_1^k, \theta_2^k, \ldots, \theta_N^k)$ where $\theta_1^k \to \theta_1^*(\theta_2^k, \lambda_2^k)$ and $\lambda_1^k \to \lambda_1^*(\theta_2^k, \lambda_2^k)$, then, similar to the previous proof, we have $\phi_2^k \to \phi_2^*(\{\theta_i^k\}_{i\in\mathcal{N}})$ where $Q^2(\boldsymbol{x}_t, \boldsymbol{u}_t; \phi_2^k) \to Q^2(\boldsymbol{x}_t, \boldsymbol{u}_t; \phi_2^*) = Q_{\bar{\boldsymbol{\pi}}_2}^2(\boldsymbol{x}_t, \boldsymbol{u}_t)$, and $\omega_2^k \to \omega_2^*(\{\theta_i^k\}_{i\in\mathcal{N}})$ where $H^2(\boldsymbol{x}_t, \boldsymbol{u}_t; \omega_2^k) \to H^2(\boldsymbol{x}_t, \boldsymbol{u}_t; \omega_2^*) = H_{\bar{\boldsymbol{\pi}}_2}^2(\boldsymbol{x}_t, \boldsymbol{u}_t)$. Further, define $\hat{r}_t^2$ and $Q_{H_{\bar{\boldsymbol{\pi}}_2}}^2$ similarly to $\hat{r}_t^1$ and $Q_{H_{\bar{\boldsymbol{\pi}}_1}}^1$, and we use $\hat{\omega}_2$ to parameterize $Q_{H_{\bar{\boldsymbol{\pi}}_2}}^2$, under Assumption 3, we can conclude that $\hat{\omega}_2^k \to \hat{\omega}_2^*(\{\theta_i^k\}_{i\in\mathcal{N}})$ where $Q_H^2(\boldsymbol{x}_t, \boldsymbol{u}_t; \hat{\omega}_2^k) \to Q_H^2(\boldsymbol{x}_t, \boldsymbol{u}_t; \hat{\omega}_2^*) = Q_{H_{\bar{\boldsymbol{\pi}}_2}}^2(\boldsymbol{x}_t, \boldsymbol{u}_t)$. Then, we can analyze the convergence of $\theta_2$ and $\lambda_2$.

**Convergence of $\theta_2$:** For the convergence of $\theta_2$, define $\bar{\boldsymbol{\pi}}_2^* = (\theta_1^*, \{\theta_i^k\}_{i\in\{2,\ldots,N\}})$ which differs from $\bar{\boldsymbol{\pi}}_2 = (\theta_1^k, \theta_2^k, \ldots, \theta_N^k)$ by $\theta_1^*$ and $\theta_1^k$. Based on previous analysis, here we have $\theta_1^k \to \theta_1^*$ and therefore $\bar{\boldsymbol{\pi}}_2 \to \bar{\boldsymbol{\pi}}_2^*$ for $\theta_2^k$ which is updated in this timescale under certain fixed $\{\theta_i^k\}_{i\in\{3,\ldots,N\}}$. The ideal Lagrangian function to use now is:

$$\mathcal{L}_{\boldsymbol{\theta}^{-2}}(\theta_2,\lambda_2) = \mathbb{E}_{\bar{\boldsymbol{\pi}}_2^*}\left[-Q_{\bar{\boldsymbol{\pi}}_2^*}^2(\boldsymbol{x}_t, \boldsymbol{u}_t) + \lambda_2 Q_{H_{\bar{\boldsymbol{\pi}}_2^*}}^2(\boldsymbol{x}_t, \boldsymbol{u}_t)\right].\qquad(42)$$

where $\boldsymbol{\theta}^{-2}$ means $\theta_2$ is updated, and during its update, $\theta_1^k = \theta_1^*(\{\theta_i^k\}_{i\in\{2,\ldots,N\}},\{\lambda_i^k\}_{i\in\{2,\ldots,N\}})$ with $\{\theta_i^k\}_{i\in\{3,\ldots,N\}}$ and $\{\lambda_i^k\}_{i\in\{3,\ldots,N\}}$ being certain fixed values. Then, by sampling data from the distribution to update $\theta_2$, we have:

$$\theta_2^{k+1} = \Gamma_{\Theta_2}\left[\theta_2^k - \eta_{2,2}\nabla_{u_t^2}(-Q^2(\boldsymbol{x}_t, \boldsymbol{u}_t; \phi_2^k) + \lambda_2^k Q_H^2(\boldsymbol{x}_t, \boldsymbol{u}_t; \hat{\omega}_2^k))|_{u_t^2=\pi^2(\boldsymbol{x}_t;\theta_2^k)}\nabla_{\theta_2}\pi^2(\boldsymbol{x}_t;\theta_2)|_{\theta_2=\theta_2^k}\right]$$

$$= \Gamma_{\Theta_2}\left[\theta_2^k - \eta_{2,2}[\nabla_{\theta_2}\mathcal{L}_{\boldsymbol{\theta}^{-2}}(\theta_2,\lambda_2)|_{\theta_2=\theta_2^k} + \delta\theta_2^{k+1} + \delta\theta_{2,\varepsilon}]\right]$$

$$(43)$$

where

$$\delta\theta_2^{k+1} = -\mathbb{E}_{\bar{\boldsymbol{\pi}}_2^*}\left[\nabla_{u_t^2}(-Q^2(\boldsymbol{x}_t, \boldsymbol{u}_t; \phi_2^k) + \lambda_2^k Q_H^2(\boldsymbol{x}_t, \boldsymbol{u}_t; \hat{\omega}_2^k))|_{u_t^2=\pi^2(\boldsymbol{x}_t;\theta_2^k)}\nabla_{\theta_2}\pi^2(\boldsymbol{x}_t;\theta_2)|_{\theta_2=\theta_2^k}\right]$$

$$+ \nabla_{u_t^2}(-Q^2(\boldsymbol{x}_t, \boldsymbol{u}_t; \phi_2^k) + \lambda_2^k Q_H^2(\boldsymbol{x}_t, \boldsymbol{u}_t; \hat{\omega}_2^k))|_{u_t^2=\pi^2(\boldsymbol{x}_t;\theta_2^k)}\nabla_{\theta_2}\pi^2(\boldsymbol{x}_t;\theta_2)|_{\theta_2=\theta_2^k}$$

$$(44)$$

and

$$\delta\theta_{2,\varepsilon} = \mathbb{E}_{\bar{\boldsymbol{\pi}}_2^*}\left[\nabla_{u_t^2}(-Q^2(\boldsymbol{x}_t, \boldsymbol{u}_t; \phi_2^k) + \lambda_2^k Q_H^2(\boldsymbol{x}_t, \boldsymbol{u}_t; \hat{\omega}_2^k))|_{u_t^2=\pi^2(\boldsymbol{x}_t;\theta_2^k)}\nabla_{\theta_2}\pi^2(\boldsymbol{x}_t;\theta_2)|_{\theta_2=\theta_2^k}\right.$$

$$\left.- \nabla_{u_t^2}(-Q_{\bar{\boldsymbol{\pi}}_2^*}^2(\boldsymbol{x}_t, \boldsymbol{u}_t) + \lambda_2^k Q_{H_{\bar{\boldsymbol{\pi}}_2^*}}^2(\boldsymbol{x}_t, \boldsymbol{u}_t))|_{u_t^2=\pi^2(\boldsymbol{x}_t;\theta_2^k)}\nabla_{\theta_2}\pi^2(\boldsymbol{x}_t;\theta_2)|_{\theta_2=\theta_2^k}\right].$$

$$(45)$$

1. We have the following:

$$\mathbb{E}\left[||\delta\theta_2^{k+1}||^2|\mathcal{F}_{\theta_2,k}\right] \leq C_5||\nabla_{\theta_2}\pi^2(\boldsymbol{x}_t;\theta_2)|_{\theta_2=\theta_2^k}||_\infty^2 \times (||\nabla_{u_t^2}Q^2(\boldsymbol{x}_t, \boldsymbol{u}_t; \phi_2^k)||_\infty^2$$

$$+ ||\lambda_2||_\infty^2||\nabla_{u_t^2}Q_H^2(\boldsymbol{x}_t, \boldsymbol{u}_t; \hat{\omega}_2^k)||_\infty^2) < \infty,\qquad(46)$$

where $\mathcal{F}_{\theta_2,k} = \sigma(\theta_2^m, \delta\theta_2^m, m \leq k)$ is the filtration of $\theta_2^k$ generated by different independent trajectories, and thus $\delta\theta_2^{k+1}$ is square integrable.

2. Under Assumption 6, we have:

$$
\begin{aligned}
\delta\theta_{2,\varepsilon} =& \, \mathbb{E}_{\bar{\pi}_2^*}\big[\nabla_{u_t^2}(Q_{\bar{\pi}_2^*}^2(\boldsymbol{x}_t, \boldsymbol{u}_t) - Q_{\bar{\pi}_2}^2(\boldsymbol{x}_t, \boldsymbol{u}_t) + Q_{\bar{\pi}_2}^2(\boldsymbol{x}_t, \boldsymbol{u}_t) - Q^2(\boldsymbol{x}_t, \boldsymbol{u}_t; \phi_2^k) + \lambda_2^k(Q_H^2(\boldsymbol{x}_t, \boldsymbol{u}_t; \hat{\omega}_2^k) \\
&- Q_{H_{\bar{\pi}_2}}^2(\boldsymbol{x}_t, \boldsymbol{u}_t) + Q_{H_{\bar{\pi}_2}}^2(\boldsymbol{x}_t, \boldsymbol{u}_t) - Q_{H_{\bar{\pi}_2^*}}^2(\boldsymbol{x}_t, \boldsymbol{u}_t)))|_{u_t^2=\pi^2(\boldsymbol{x}_t;\theta_2^k)}\nabla_{\theta_2}\pi^2(\boldsymbol{x}_t;\theta_1)|_{\theta_2=\theta_2^k} \\
\leq& \, \mathbb{E}_{\bar{\pi}_2^*}\big[\nabla_{\theta_2}\pi^2(\boldsymbol{x}_t;\theta_2)|_{\theta_2=\theta_2^k}\big](K_7||\bar{\pi}_2^* - \bar{\pi}_2||_\infty + K_8||\phi_2^k - \phi_2^*||_\infty + K_9||\hat{\omega}_2^k - \hat{\omega}_2^*||_\infty) \to 0
\end{aligned}
\tag{47}
$$

3. Since $\nabla_{\theta_2}(-Q^2(\boldsymbol{x}_t, \boldsymbol{u}_t; \phi_2^k) + \lambda_2^k Q_H^2(\boldsymbol{x}_t, \boldsymbol{u}_t; \hat{\omega}_2^k))$ is a sample of $\nabla_{\theta_2}\mathcal{L}_{\boldsymbol{\theta}^{-2}}(\theta_2, \lambda_2)|_{\theta_2=\theta_2^k}$ given $\bar{\pi}_2 \to \bar{\pi}_2^*$, one can obtain that $\mathbb{E}\big[\delta\theta_2^{k+1}|\mathcal{F}_{\theta_2,k}\big] = 0$ almost surely.

Then, similar to the previous section using Lyapunov function, we conclude that the sequence $\{\theta_2^k\}$ converges to a locally minimum point $\theta_2^*$ such that $\mathcal{L}_{\boldsymbol{\theta}^{-2}}(\theta_2^*, \lambda_2) \leq \mathcal{L}_{\boldsymbol{\theta}^{-2}}(\theta_2(t), \lambda_2) \leq \mathcal{L}_{\boldsymbol{\theta}^{-2}}(\theta_2(0), \lambda_2), \forall t \geq 0$.

***Convergence of*** $\lambda_2$: Similar to the previous section on $\lambda_1$, we can treat $\{\theta_i^k\}_{i\in\{3,...,N\}}$ and $\{\lambda_i^k\}_{i\in\{3,...,N\}}$ as fixed when update $\lambda_2$, and $||\theta_2^k - \theta_2^*|| = 0$. Then, the update of $\lambda_2$ can be written as:

$$
\begin{aligned}
\lambda_2^{k+1} =& \, \Gamma_{\Lambda_2}\big[\lambda_2^k + \eta_{3,2}Q_H^2(\boldsymbol{x}_t, \boldsymbol{u}_t; \hat{\omega}_2^k)\big] \\
=& \, \Gamma_{\Lambda_2}\big[\lambda_2^k + \eta_{3,2}[\nabla_{\lambda_2}\mathcal{L}_{\boldsymbol{\theta}^{-2}}(\theta_2, \lambda_2)|_{\theta_2=\theta_2^*(\lambda_2^k),\lambda_2=\lambda_2^k} + \delta\lambda_2^{k+1}]\big]
\end{aligned}
\tag{48}
$$

where

$$
\begin{aligned}
\delta\lambda_2^{k+1} =& -\mathbb{E}\big[Q_{H_{\bar{\pi}_{2,*}^*}}^2(\boldsymbol{x}_t, \bar{\pi}_{2,*}^*(\boldsymbol{x}_t))\big] + Q_H^2(\boldsymbol{x}_t, \bar{\pi}_2(\boldsymbol{x}_t); \hat{\omega}_2^k) \\
=& -\mathbb{E}\big[Q_{H_{\bar{\pi}_{2,*}^*}}^2(\boldsymbol{x}_t, \bar{\pi}_{2,*}^*(\boldsymbol{x}_t))\big] + Q_{H_{\bar{\pi}_2}}^2(\boldsymbol{x}_t, \bar{\pi}_2(\boldsymbol{x}_t)) - Q_{H_{\bar{\pi}_2}}^2(\boldsymbol{x}_t, \bar{\pi}_2(\boldsymbol{x}_t)) + Q_H^2(\boldsymbol{x}_t, \bar{\pi}_2(\boldsymbol{x}_t); \hat{\omega}_2^k)
\end{aligned}
\tag{49}
$$

where $\bar{\pi}_{2,*}^* = (\theta_1^*, \theta_2^*(\lambda_2^k), \{\theta_i^k\}_{i\in\{3,...,N\}})$, $\bar{\pi}_2 = (\theta_1^k, \theta_2^k, \{\theta_i^k\}_{i\in\{3,...,N\}})$ where $\theta_1^k \to \theta_1^*$ and $\theta_2^k \to \theta_2^*(\lambda_2^k)$ according to previous analysis. Therefore,

$$
\mathbb{E}\big[||\delta\lambda_2^{k+1}||^2|\mathcal{F}_{\lambda_2,k}\big] \leq C_6 \max_{\boldsymbol{x}_t, \boldsymbol{u}_t, \hat{\omega}_2^k} ||Q_H^2(\boldsymbol{x}_t, \boldsymbol{u}_t; \hat{\omega}_2^k)||_\infty^2 < \infty,
\tag{50}
$$

according to Assumption 3. Since based on aforementioned analysis, $||Q_H^2(\boldsymbol{x}_t, \bar{\pi}_2(\boldsymbol{x}_t); \hat{\omega}_2^k) - Q_{H_{\bar{\pi}_2}}^2(\boldsymbol{x}_t, \bar{\pi}_2(\boldsymbol{x}_t))|| \to 0$, and $Q_{H_{\bar{\pi}_2}}^2(\boldsymbol{x}_t, \bar{\pi}_2(\boldsymbol{x}_t))$ is a sample of $\nabla_{\lambda_2}\mathcal{L}_{\boldsymbol{\theta}^{-2}}(\theta_2, \lambda_2)|_{\theta_2=\theta_2^*(\lambda_2^k),\lambda_2=\lambda_2^k}$, we have $\mathbb{E}\big[\delta\lambda_2^{k+1}|\mathcal{F}_{\lambda_2,k}\big] = 0$ almost surely.

Then, similar to the previous section, we have that $\lambda_2^k$ converges to $\lambda_2^*$ which is a local maximum point, with $\mathcal{L}_{\boldsymbol{\theta}^{-2}}(\theta_2^*(\lambda_2^*), \lambda_2^*) \geq \mathcal{L}_{\boldsymbol{\theta}^{-2}}(\theta_2^*(\lambda_2), \lambda_2), \forall t \geq 0$. Therefore, $(\theta_2^*, \lambda_2^*)$ is a locally saddle point for the certain fixed $\{\theta_i^k\}_{i\in\{3,...,N\}}$ and $\{\lambda_i^k\}_{i\in\{3,...,N\}}$ of other agents that have not been updated, while $\theta_1$ and $\lambda_1$ have already converged to the local optimal policy and multiplier for the agent 1 with respect to $\{\theta_i^k\}_{i\in\{2,...,N\}}$ and $\{\lambda_i^k\}_{i\in\{2,...,N\}}$. Namely, we finally obtain $(\theta_1^*, \theta_2^*)$ and $(\lambda_1^*, \lambda_2^*)$ with respect to $\{\theta_i^k\}_{i\in\{3,...,N\}}$ and $\{\lambda_i^k\}_{i\in\{3,...,N\}}$. This process will iterate until the agent $N$ finishes its update, and we have a feasible joint policy $\{\theta_i^*\}_{i\in\mathcal{N}}$, where for each $i$, $\theta_i^*$ is a feasible policy. Furthermore, considering this obtained joint policy $\boldsymbol{\pi}^* = \{\pi_i^*\}_{i\in\mathcal{N}}$ parameterized by $\{\theta_i^*\}_{i\in\mathcal{N}}$, denote $\boldsymbol{\theta}^{-i,*}$ as the obtained joint policy except agent $i$, then, consider another joint policy $\boldsymbol{\pi}_{-i}^*$ parameterized by $\{\theta_i, \boldsymbol{\theta}^{-i,*}\}$ where $\theta_i \in B_{r_{\theta_i}}(\theta_i^*)\backslash\theta_i^*$ is a feasible policy, we have :

$$
\begin{aligned}
\mathbb{E}_{\boldsymbol{\pi}^*}\big[-Q_{\boldsymbol{\pi}^*}^i(\boldsymbol{x}_t, \boldsymbol{u}_t)\big] =& \, \mathbb{E}_{\boldsymbol{\pi}^*}\big[-Q_{\boldsymbol{\pi}^*}^i(\boldsymbol{x}_t, \boldsymbol{u}_t) + \lambda_i^* Q_{H_{\boldsymbol{\pi}^*}}^i(\boldsymbol{x}_t, \boldsymbol{u}_t)\big] = \mathcal{L}_{\boldsymbol{\theta}^{-i,*}}(\theta_i^*, \lambda_i^*) \\
\leq& \, \mathcal{L}_{\boldsymbol{\theta}^{-i,*}}(\theta_i, \lambda_i^*) = \mathbb{E}_{\boldsymbol{\pi}_{-i}^*}\big[-Q_{\boldsymbol{\pi}_{-i}^*}^i(\boldsymbol{x}_t, \boldsymbol{u}_t) + \lambda_i^* Q_{H_{\boldsymbol{\pi}_{-i}^*}}^i(\boldsymbol{x}_t, \boldsymbol{u}_t)\big] = \mathbb{E}_{\boldsymbol{\pi}_{-i}^*}\big[-Q_{\boldsymbol{\pi}_{-i}^*}^i(\boldsymbol{x}_t, \boldsymbol{u}_t)\big],
\end{aligned}
\tag{51}
$$

since $\theta_i^* = \arg\min_{\theta_i \in B_{r_{\theta_i}}(\theta_i^*)} \mathcal{L}_{\boldsymbol{\theta}^{-i,*}}(\theta_i, \lambda_i^*)$. Thus, $\mathbb{E}_{\boldsymbol{\pi}^*}\big[Q_{\boldsymbol{\pi}^*}^i(\boldsymbol{x}_t, \boldsymbol{u}_t)\big] \geq \mathbb{E}_{\boldsymbol{\pi}_{-i}^*}\big[Q_{\boldsymbol{\pi}_{-i}^*}^i(\boldsymbol{x}_t, \boldsymbol{u}_t)\big]$ for all feasible policy $\theta_i \in B_{r_{\theta_i}}(\theta_i^*)\backslash\theta_i^*, \forall i \in \mathcal{N}$. Therefore, the obtained joint policy $\boldsymbol{\pi}^* = \{\pi_i^*\}_{i\in\mathcal{N}}$ is a feasible local Nash equilibrium.

*Now we show that $H^i_{\boldsymbol{\pi}^*}(\boldsymbol{x}_t, \boldsymbol{u}_t) \geq 0$ for $(\boldsymbol{x}_t, \boldsymbol{u}_t)$ along the trajectory under $\boldsymbol{\pi}^*$, $\forall t \geq 0$ and $\forall i \in \mathcal{N}$: According to the definition, $H^i_{\boldsymbol{\pi}^*}(\boldsymbol{x}_0, \boldsymbol{u}_0) \leq 0$, $\forall i \in \mathcal{N}$. Consider any $i \in \mathcal{N}$ and let $H^i_{\boldsymbol{\pi}^*}(\boldsymbol{x}_0, \boldsymbol{u}_0) = p \leq 0$ with $\boldsymbol{u}_0$ being any control signal generated by the joint policy at $\boldsymbol{x}_0$, according to the barrier certificate, we have $H^i_{\boldsymbol{\pi}^*}(\boldsymbol{x}_1, \boldsymbol{u}_1) \geq (1 - \epsilon) H^i_{\boldsymbol{\pi}^*}(\boldsymbol{x}_0, \boldsymbol{u}_0)$ for all following $(\boldsymbol{x}_1, \boldsymbol{u}_1)$. Furthermore, according to the definition of barrier certificate, $H^i_{\boldsymbol{\pi}^*}(\boldsymbol{x}_0, \boldsymbol{u}_0) = b^i_0 + \gamma \mathbb{E}_{\boldsymbol{x}_1, \boldsymbol{u}_1}[H^i_{\boldsymbol{\pi}^*}(\boldsymbol{x}_1, \boldsymbol{u}_1)] \geq (1 - \epsilon) \gamma p$ given $b^i_0 = 0$. Thus, we have $H^i_{\boldsymbol{\pi}^*}(\boldsymbol{x}_0, \boldsymbol{u}_0) = p \geq (1 - \epsilon) \gamma p$ at convergence. Since $p \leq 0$ and $1 - \epsilon \in (0, 1)$, $\gamma \in (0, 1)$, we have $p = 0$, for all $i \in \mathcal{N}$ and all possible generated control signal $\boldsymbol{u}_0$. The similar process applies for all $t \geq 0$, and we have $H^i_{\boldsymbol{\pi}^*}(\boldsymbol{x}_t, \boldsymbol{u}_t) \geq 0$, $\forall t \geq 0$ and $\forall i \in \mathcal{N}$.*

*Furthermore, for a state $\boldsymbol{x}_t$ along the trajectory under the control policy $\boldsymbol{\pi}^*$, let $\boldsymbol{x}_{t+1}$ represent the next state following $\boldsymbol{u}_t$ generated by $\boldsymbol{\pi}^*$, and let $i_A$ and $i_B$ denote the index of the maximum $H_{\boldsymbol{\pi}^*}$ at $(\boldsymbol{x}_t, \boldsymbol{u}_t)$ and $(\boldsymbol{x}_{t+1}, \boldsymbol{u}_{t+1})$, respectively, namely:*

$$H^g_{\boldsymbol{\pi}^*}(\boldsymbol{x}_t, \boldsymbol{u}_t) = \min_{i \in \mathcal{N}} H^i_{\boldsymbol{\pi}^*}(\boldsymbol{x}_t, \boldsymbol{u}_t) = H^{i_A}_{\boldsymbol{\pi}^*}(\boldsymbol{x}_t, \boldsymbol{u}_t) \tag{52}$$

$$H^g_{\boldsymbol{\pi}^*}(\boldsymbol{x}_{t+1}, \boldsymbol{u}_{t+1}) = \min_{i \in \mathcal{N}} H^i_{\boldsymbol{\pi}^*}(\boldsymbol{x}_{t+1}, \boldsymbol{u}_{t+1}) = H^{i_B}_{\boldsymbol{\pi}^*}(\boldsymbol{x}_{t+1}, \boldsymbol{u}_{t+1}) \tag{53}$$

*Then, given $1 - \epsilon \in \mathcal{K}_\infty$, we have:*

$$\begin{aligned} H^g_{\boldsymbol{\pi}^*}(\boldsymbol{x}_{t+1}, \boldsymbol{u}_{t+1}) = H^{i_B}_{\boldsymbol{\pi}^*}(\boldsymbol{x}_{t+1}, \boldsymbol{u}_{t+1}) &\geq (1 - \epsilon) H^{i_B}_{\boldsymbol{\pi}^*}(\boldsymbol{x}_t, \boldsymbol{u}_t) \\ &\geq (1 - \epsilon) H^{i_A}_{\boldsymbol{\pi}^*}(\boldsymbol{x}_t, \boldsymbol{u}_t) = (1 - \epsilon) H^g_{\boldsymbol{\pi}^*}(\boldsymbol{x}_t, \boldsymbol{u}_t), \end{aligned} \tag{54}$$

*since $\forall i \in \mathcal{N}$ and $\forall t \geq 0$, $H^i_{\boldsymbol{\pi}^*}(\boldsymbol{x}_{t+1}, \boldsymbol{u}_{t+1}) \geq (1 - \epsilon) H^i_{\boldsymbol{\pi}^*}(\boldsymbol{x}_t, \boldsymbol{u}_t)$ for all $\boldsymbol{u}_t$ generated by $\boldsymbol{\pi}^*$. Besides, it is easily known that $H^g_{\boldsymbol{\pi}^*}(\boldsymbol{x}_{t+1}, \boldsymbol{u}_{t+1}) \geq 0$ implies that $\forall i \in \mathcal{N}$, $H^i_{\boldsymbol{\pi}^*}(\boldsymbol{x}_{t+1}, \boldsymbol{u}_{t+1}) \geq 0$, which means safety is maintained for all agents. Additionally, $H^g_{\boldsymbol{\pi}^*}(\boldsymbol{x}_{t+1}, \boldsymbol{u}_{t+1}) < 0$ if $\exists i \in \mathcal{N}$ such that $H^i_{\boldsymbol{\pi}^*}(\boldsymbol{x}_t, \boldsymbol{u}_t) < 0$, which means at least one agent will experience safety violation with a positive probability. Therefore, $H^g_{\boldsymbol{\pi}^*}(\boldsymbol{x}_t, \boldsymbol{u}_t)$ is a global barrier certificate for the whole multi-agent system, and the corresponding safe set is given as:*

$$\mathcal{X}^g := \{\boldsymbol{x} \mid H^g_{\boldsymbol{\pi}^*}(\boldsymbol{x}, \boldsymbol{\pi}^*(\boldsymbol{x})) \geq 0\} = \cap_{i \in \mathcal{N}} \{\boldsymbol{x} \mid H^i_{\boldsymbol{\pi}^*}(\boldsymbol{x}, \boldsymbol{\pi}^*(\boldsymbol{x})) \geq 0\}. \tag{55}$$

# D APPENDIX FOR STATE-WISE STABILITY VIA ISS AND INFEASIBILITY UNDER MULTIPLE CONSTRAINTS

## D.1 PROOF FOR THEOREM 2

**Proof 4** *This proof for Theorem 2 is developed based on previous papers on input-to-state stability (Sontag & Wang, 1995; Ito et al., 2012). Define operators $A(\boldsymbol{V}(\boldsymbol{x}_t)) = [\alpha_1 V^1(x^1_t), \alpha_2 V^2(x^2_t), \ldots, \alpha_N V^1(x^N_t)]^T$ where $\boldsymbol{V}(\boldsymbol{x}_t) = [V^1(x^1_t), V^2(x^2_t), \ldots, V^N(x^N_t)]^T$ and $\dot{\boldsymbol{V}}(\boldsymbol{x}_t) = [\dot{V}^1(x^1_t), \dot{V}^2(x^2_t), \ldots, \dot{V}^N(x^N_t)]^T$, and $\{\alpha_i\}_{i \in \mathcal{N}}$ is the set of $\mathcal{K}_\infty$ functions. Further, define the operator*

$$\Delta(\boldsymbol{V}(\boldsymbol{x}_t)) = [\sum_{j \neq 1} \delta_{1j}(V^j(x^j_t)), \sum_{j \neq 2} \delta_{2j}(V^j(x^j_t)), \ldots, \sum_{j \neq N} \delta_{Nj}(V^j(x^j_t))]^T,$$

*and the operator $K(||\boldsymbol{d}_t||) = [\kappa_1(||\boldsymbol{d}_t||), \kappa_2(||\boldsymbol{d}_t)||), \ldots, \kappa_N(||\boldsymbol{d}_t||)]^T$ to represent the gain terms describing the influence from agent $j \neq i$ to the agent $i$, $\forall i \in \mathcal{N}$, and the terms describing the influence from the external disturbance $\boldsymbol{d}_t$. Then, the constraint (4) can be written as:*

$$\dot{\boldsymbol{V}}(\boldsymbol{x}_t) \leq (-A + \Delta)(\boldsymbol{V}(\boldsymbol{x}_t)) + K(||\boldsymbol{d}_t||) \tag{56}$$

*for all $\boldsymbol{x}_t$ sampled under the joint policy $\boldsymbol{\pi}$. Based on Sontag & Wang (1995), we need to first find a function $V_{MAS}$ for the whole multi-agent system satisfying:*

$$\dot{V}_{MAS}(\boldsymbol{x}_t) \leq -\alpha_{MAS}(V_{MAS}(\boldsymbol{x}_t)) + K_{MAS}(||\boldsymbol{d}_t||). \tag{57}$$

*Thus, define $Y^i(x^i_t) = C \times V^i(x^i_t)$, $i = 1, 2, \ldots, n$, where $C > 0$ is a constant. Let $V_{MAS}(\boldsymbol{x}_t) = \max_{i \in \mathcal{N}} Y^i(x^i_t)$, then the following properties are satisfied:*

*1. $Y^i(x^i_t)$ is strictly increasing, which means when Lyapunov function $V^i(x^i_t)$ grows, its contribution to the $V_{MAS}$ will not shrink.*

2. *Since $Y^i(x_t^i) = C \times V^i(x_t^i) = \int_0^{V^i(x_t^i)} C(y)dy$ where $C(y) = C > 0$, we have $\int_1^\infty C(y)dy = \infty$ because $C(y)$ does not approach zero asymptotically as $V^i(x_t^i)$ goes to infinity, which means at large states, each $V^i(x_t^i)$ can make non-negligible contributions to $V_{MAS}(\boldsymbol{x}_t)$ if it is selected, and help to analyse the stability at states with large norms.*

*Define $\boldsymbol{Y}(\boldsymbol{x}_t) = [Y^1(x_t^1), Y^2(x_t^2)\ldots, Y^N(x_t^N)]^T$, and $\dot{\boldsymbol{Y}}(\boldsymbol{x}_t) = [\dot{Y}^1(x_t^1), \dot{Y}^2(x_t^2), \ldots, \dot{Y}^N(x_t^N)]^T$, with a set of $\mathcal{K}_\infty$ functions $\{\beta_i\}_{i \in \mathcal{N}}$, define*

$$D(\boldsymbol{V}(\boldsymbol{x}_t)) = [\sum_{j \neq 1} \delta_{1j}(V^j(x_t^j)) + \beta_1(\sum_{j \neq 1} \delta_{1j}(V^j(x_t^j))), \ldots, \sum_{j \neq N} \delta_{Nj}(V^j(x_t^j)) + \beta_N(\sum_{j \neq N} \delta_{Nj}(V^j(x_t^j)))]^T \tag{58}$$

*and let $\iota_i$ be defined according to $Id - \iota_i = (Id + \beta_i)^{-1}$ where $Id$ denotes the identity map, then $\iota_i \in \mathcal{K}_\infty$. Pick $\zeta_i \in \mathcal{K}_\infty$ satisfying $\iota_i - \zeta_i \in \mathcal{K}_\infty$, $\zeta_i \in \mathcal{K}_\infty$ and $Id - \zeta_i \in \mathcal{K}_\infty$, and construct the operator $F$ to be $F(\boldsymbol{s}) = [\zeta_1(s_1), \ldots, \zeta_N(s_N)]^T$ where $\boldsymbol{s} = [s_1, s_2, \ldots, s_N]$. Then, we have:*

$$\begin{aligned}
\dot{\boldsymbol{Y}}(\boldsymbol{x}_t) &\leq C[(-A + \Delta)(\boldsymbol{V}(\boldsymbol{x}_t)) + K(||\boldsymbol{d}_t||)] \\
&= C[\{-(Id - F + F) \circ A + \Delta\}(\boldsymbol{V}(\boldsymbol{x}_t)) + K(||\boldsymbol{d}_t||)] \\
&= C\{-(Id - F) \circ A + \Delta\}(\boldsymbol{V}(\boldsymbol{x}_t)) + C[-F \circ A(\boldsymbol{V}(\boldsymbol{x}_t)) + K(||\boldsymbol{d}_t||)] \\
&\leq C\{-(Id - F) \circ A + \Delta\}(\boldsymbol{V}(\boldsymbol{x}_t)) + CK(||\boldsymbol{d}_t||)
\end{aligned} \tag{59}$$

*Suppose at the state $\boldsymbol{x}_t$, the maximization of $Y^i(x_t^i)$ is attained at $i = p \in \mathcal{N}$, then, $V_{MAS}(\boldsymbol{x}_t) = Y^p(x_t^p) = C \times V^p(x_t^p)$ and the above inequality yields:*

$$\begin{aligned}
\dot{V}_{MAS}(\boldsymbol{x}_t) &= C \times \dot{V}^p(x_t^p) \\
&\leq C\{\big(-(Id - \zeta_p) + (Id + \beta_p)^{-1} - (Id + \beta_p)^{-1}\big) \circ \alpha_p\}(V^p(x_t^p)) + C[\Delta(\boldsymbol{V}(\boldsymbol{x}_t))]_p + C[K(||\boldsymbol{d}_t||)]_p \\
&= C\{-(\iota_p - \zeta_p) \circ \alpha_p\}(V^p(x_t^p)) + C\{-(Id + \beta_p)^{-1} \circ \alpha_p\}(V^p(x_t^p)) + C[\Delta(\boldsymbol{V}(\boldsymbol{x}_t))]_p + C[K(||\boldsymbol{d}_t||)]_p.
\end{aligned} \tag{60}$$

*Given*

$$(Id + \beta_p)([\Delta(\boldsymbol{V}(\boldsymbol{x}_t))]_p) = \sum_{j \neq p} \delta_{pj}(V^j(x_t^j)) + \beta_p(\sum_{j \neq p} \delta_{pj}(V^j(x_t^j))) \leq \alpha_p(V^p(x_t^p)) \tag{61}$$

*according to the assumption in Theorem 2, we have $(Id + \beta_p)^{-1}\alpha_p(V^p(x_t^p)) \geq [\Delta(\boldsymbol{V}(\boldsymbol{x}_t))]_p$, and therefore*

$$\dot{V}_{MAS}(\boldsymbol{x}_t) \leq -C(\iota_p - \zeta_p) \circ \alpha_p(V^p(x_t^p)) + C\kappa_p(||\boldsymbol{d}_t||), \tag{62}$$

*where $C(\iota_p - \zeta_p) \circ \alpha_p$ and $C\kappa_p$ are $\mathcal{K}_\infty$ functions. Therefore, there exists $\hat{\alpha}_p \in \mathcal{K}_\infty$ and $\hat{\kappa}_p \in \mathcal{K}_\infty$ such that $\dot{V}_{MAS}(\boldsymbol{x}_t) \leq -\hat{\alpha}_p(V^p(x_t^p)) + \hat{\kappa}_p(||\boldsymbol{d}_t||)$. Furthermore, repeating the above process for $p \in \mathcal{N}$, and since $V_{MAS}(\boldsymbol{x}_t) = C \times V^p(x_t^p)$, we have $\dot{V}_{MAS}(\boldsymbol{x}_t) \leq -\alpha_{MAS}(V_{MAS}(\boldsymbol{x}_t)) + K_{MAS}(||\boldsymbol{d}_t||)$ where $\alpha_{MAS}$ and $K_{MAS}$ are defined by:*

$$\begin{aligned}
\alpha_{MAS}(s) &= \min_i \hat{\alpha}_i(\frac{s}{C}), \\
K_{MAS}(s) &= \max_i \hat{\kappa}_i(s).
\end{aligned} \tag{63}$$

*Later, we prove that $V_{MAS}(\boldsymbol{x}_t)$ can be lower-bounded and upper-bounded by two $\mathcal{K}_\infty$ functions. For each $V^i(x_t^i)$, it is natural that $||x_t^i|| \leq \mathbb{E}_{\tau^i \sim \pi^i}\big[\sum_{k=0}^\infty \gamma^k c^i(x_{t+k}^i)\big] = V^i(x_t^i)$ given $\tau^i$ starts from $x_t^i$ and $c^i(x_t^i) = ||x_t^i||$ when the desired goal is, without loss of generality, at the origin. Then, under the assumption in Theorem 2 we have that $V^i(x_t^i) \leq \mu_i(||x_t^i||)$, for $\mu_i \in \mathcal{K}_\infty$ under the policy. This is not unreasonable since given the state space is bounded, $V^i(x_t^i) = \mathbb{E}_{\tau^i \sim \pi^i}\big[\sum_{k=0}^\infty \gamma^k c^i(x_{t+k}^i)\big] \leq \frac{||x^i||_\infty}{1-\gamma}$ where $||x^i||_\infty$ is the maximum norm of $x_t^i$ in the bounded state space, and therefore $V^i(x_t^i)$ is upper bounded. Given*

$$||x_t^i|| \leq V^i(x_t^i) \leq \mu_i(||x_t^i||), \tag{64}$$

*we have the following properties:*

1. *Since $V_{MAS}(\boldsymbol{x}_t) = \max_{i \in \mathcal{N}} C \times V^i(x_t^i)$, suppose $p = \arg\max_i C \times V^i(x_t^i)$, we have*

$$V_{MAS}(\boldsymbol{x}_t) = C \times V^p(x_t^p) \leq C \times \mu_p(||x_t^p||) \leq C \times \mu_p(||\boldsymbol{x}_t||) \leq C \times (\sum_{k=1}^N \mu_i)(||\boldsymbol{x}_t||) = \hat{\mu}_{MAX}(||\boldsymbol{x}_t||).$$

2. $V_{MAS}(\boldsymbol{x}_t) = C \times V^p(x_t^p) \geq C \times V^i(x_t^i) = C \times \sum_{k=0}^\infty \gamma_c^k ||x_{t+k}^i|| \geq C \times ||x_t^i||, \forall i \in \mathcal{N}$.
*Select $p' = \max_i ||x_t^i||$, we thus have $V_{MAS}(\boldsymbol{x}_t) \geq C \times ||x_t^{p'}|| \geq C\frac{||\boldsymbol{x}_t||}{\sqrt{N}} \geq \hat{\mu}_{MIN}(||\boldsymbol{x}_t||)$*
*since $||\boldsymbol{x}_t|| = \sqrt{\sum_{i=1}^N ||x_t^i||^2} \leq \sqrt{N \times ||x_t^{p'}||^2} = \sqrt{N}||x_t^{p'}||$.*

*In sum, we have*

$$\hat{\mu}_{MIN}(||\boldsymbol{x}_t||) \leq V_{MAS}(\boldsymbol{x}_t) \leq \hat{\mu}_{MAX}(||\boldsymbol{x}_t||) \tag{65}$$

*and $\dot{V}_{MAS}(\boldsymbol{x}_t) \leq -\alpha_{MAS}(V_{MAS}(\boldsymbol{x}_t)) + K_{MAS}(||\boldsymbol{d}_t||) \leq -\hat{\alpha}_{MAS}(||\boldsymbol{x}_t||) + K_{MAS}(||\boldsymbol{d}_t||)$ where $\hat{\alpha}_{MAS}(s) = \alpha_{MAS} \circ \hat{\mu}_{MIN}(s)$. Based on Sontag & Wang (1995, Remark 2.4), we have $||\boldsymbol{x}_t|| \leq \xi(||\boldsymbol{x}_0||, t) + \rho(||\boldsymbol{d}||_\infty)$.*

## D.2   Formal version and proof for Proposition 2

**Proposition 3** *Under the multi-timescale framework detailed in Theorem 1 and Assumption 7, suppose for each $i \in \mathcal{N}$, at its $K_i$-th update the initial state $\boldsymbol{x}_0$ is included within the safe set of agent $i$, then, when the number of updates for agent $i$ goes to infinity, for any trajectory that starts from $\boldsymbol{x}_0$ under the controller obtained by solving Problem (5), $Pr(\boldsymbol{x}_t \in \mathcal{X}_{infeasible}^{\boldsymbol{\pi},i}) = 0, \forall t \geq 0$.*

**Proof 5** *For simplicity, the barrier certificate taking the state only as input is used in the proof, and here we use induction to conduct the proof.*

*For the first agent, when update $\pi^1$, under Assumption 2, all other agents are following fixed policies $\{\pi^m\}_{m \in \{2,\ldots,N\}}$, and the update can be represented by Problem (11) by letting $i = 1$. Then, according to Proposition 1, we have the safe set $\mathcal{X}_{safe}^{\boldsymbol{\pi}_{1,k_1},1}$ for the agent 1 is non-shrinking. Since $\boldsymbol{x}_0 \in \mathcal{X}_{safe}^{\boldsymbol{\pi}_{1,K_1},1}$, we have $\boldsymbol{x}_0 \in \mathcal{X}_{safe}^{\boldsymbol{\pi}_{1,k_1},1}$ for $k_1 \geq K_1$. Therefore, when $k_1 \to \infty$, according to Lemma 1, we have $H_{\boldsymbol{\pi}_{1,k_1}}^1(\boldsymbol{x}_0) = 0$. Because $\forall k_1 \geq 0$ and $\forall \boldsymbol{x}_t' \in \mathcal{X}_{infeasible}^{\boldsymbol{\pi},1}$, $Pr(\boldsymbol{x}_{t+1}' \in \mathcal{X}_{vio}^{\boldsymbol{\pi},1}) > 0$ according to Assumption 7 and the definition of $\mathcal{X}_{infeasible}^{\boldsymbol{\pi},1}$, it can be concluded that*

$$H_{\boldsymbol{\pi}_{1,k_1+1}}^1(\boldsymbol{x}_t') = \mathbb{E}_{\tau \sim \boldsymbol{\pi}_{1,k_1+1}}\Big[\sum_{j=0}^\infty \gamma^k b^1(\boldsymbol{x}_{t+j}')\Big] < 0 \tag{66}$$

*since $Pr(b^1(\boldsymbol{x}_{t+1}') < 0) > 0$. Because $\forall k_1 \geq K_1$, $H_{\boldsymbol{\pi}_{1,k_1}}^1(\boldsymbol{x}_0) = 0 > H_{\boldsymbol{\pi}_{1,k_1}}^1(\boldsymbol{x}_t')$ for any $\boldsymbol{x}_t' \in \mathcal{X}_{infeasible}^{\boldsymbol{\pi},1}$, any trajectory starting from $\boldsymbol{x}_0$ will not enter $\mathcal{X}_{infeasible}^{\boldsymbol{\pi},1}$ with a positive probability under $\boldsymbol{\pi}_{1,k_1}$ when $k_1 \to \infty$ due to the application of the neural barrier certificate.*

*Then, assume for any $i - 1 \in \{1, \ldots, N-1\}$, $\forall m \in \{1, \ldots, i-1\}$, $Pr(\boldsymbol{x}_t \in \mathcal{X}_{infeasible}^{\boldsymbol{\pi},m}) = 0$, $\forall t \geq 0$, then, same as the proof of Theorem 1, under Assumption 2, when update the control policy $\pi^i$ for agent $i$, $\{\pi^m\}_{m \in \{1,\ldots,i-1\}}$ for agent 1 to $i-1$ have converged to their locally optimal policies without entering their infeasible sets at a positive probability, and will not result in the shrink of the safe set of agent $i$ since they avoid introducing safety violations between agent $i$ and agent $m$, $\forall m \in \{1, \ldots, i-1\}$, almost surely when the policy for the agent $i$ is updated from $\pi_{k_i}^i$ to $\pi_{k_i+1}^i$. Additionally, since $\{\pi^m\}_{m \in \{i+1,\ldots,N\}}$ are fixed policies under Assumption 2, the update for agent $i$ can be represented by Problem (11), and thus based on analysis similar to Proposition 1, we have the safe set $\mathcal{X}_{safe}^{\boldsymbol{\pi}_{i,k_i},i}$ for the agent $i$ is non-shrinking. Since $\boldsymbol{x}_0 \in \mathcal{X}_{safe}^{\boldsymbol{\pi}_{i,K_i},i}$, $\boldsymbol{x}_0 \in \mathcal{X}_{safe}^{\boldsymbol{\pi}_{i,k_i},i}$ for $k_i \geq K_i$, and $H_{\boldsymbol{\pi}_{i,k_i}}^i(\boldsymbol{x}_0) = 0$. Furthermore, under Assumption 7, the set $\mathcal{X}_{infeasible}^{\boldsymbol{\pi},i}$ depends on agents from $i+1$ to $N$ and the environment, given $\{\pi^m\}_{m \in \{1,\ldots,i-1\}}$ have converged to feasible solutions leading to safety violations with agent $i$ with probability 0. Thus, this infeasible set is fixed during the update for agent $i$ when $\{\pi^m\}_{m \in \{i+1,\ldots,N\}}$ are fixed. Similar to the previous analysis*

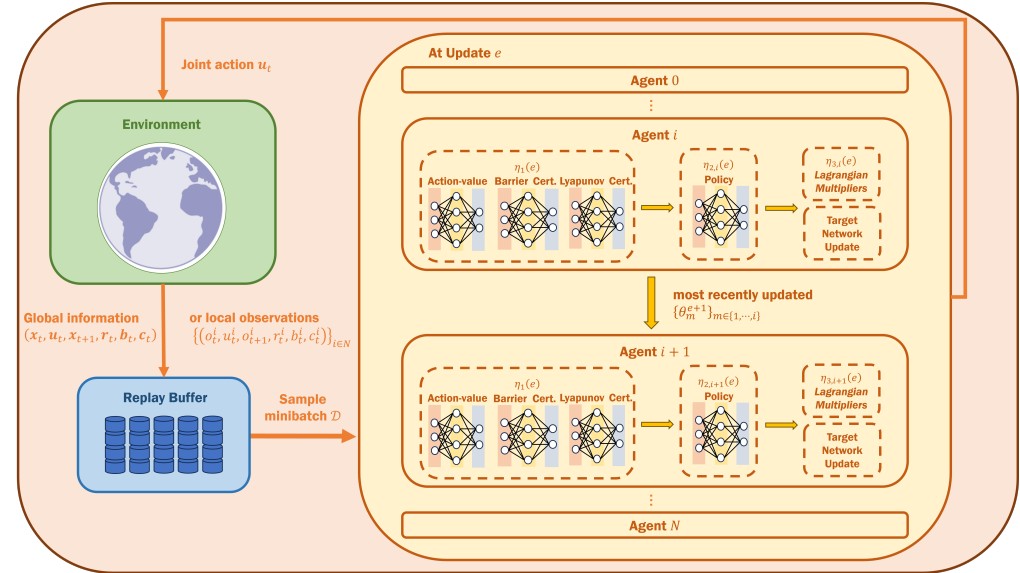

Figure 4: Overview of the **M**ulti-**A**gent **S**afe and **S**table **S**oft **A**ctor-**C**ritic (**MAS**$^3$**AC**) framework. Depending on the task setting, either global information or local observations are collected. Agents are updated sequentially, as detailed in Algorithm 2

.

*for agent 1, given $\forall k_i \geq 0$ and $\forall \boldsymbol{x}'_t \in \boldsymbol{\mathcal{X}}^{\boldsymbol{\pi},i}_{infeasible}$, $Pr(\boldsymbol{x}'_{t+1} \in \boldsymbol{\mathcal{X}}^{\boldsymbol{\pi},i}_{vio}) > 0$, we have*

$$H^i_{\boldsymbol{\pi}_{i,k_i+1}}(\boldsymbol{x}'_t) = \mathbb{E}_{\tau \sim \boldsymbol{\pi}_{i,k_i+1}}\Big[\sum_{j=0}^{\infty} \gamma^j b^i(\boldsymbol{x}'_{t+j})\Big] < 0. \tag{67}$$

*Since $\forall k_i \geq K_i$, $H^i_{\boldsymbol{\pi}_{i,k_i}}(\boldsymbol{x}_0) = 0 > H^i_{\boldsymbol{\pi}_{i,k_i}}(\boldsymbol{x}'_t)$ for any $\boldsymbol{x}'_t \in \boldsymbol{\mathcal{X}}^{\boldsymbol{\pi},i}_{infeasible}$, any trajectory from $\boldsymbol{x}_0$ will not enter $\boldsymbol{\mathcal{X}}^{\boldsymbol{\pi},i}_{infeasible}$ with a positive probability under $\boldsymbol{\pi}_{i,k_i}$ when $k_i \to \infty$ because of the neural barrier certificate, and thus $Pr(\boldsymbol{x}_t \in \boldsymbol{\mathcal{X}}^{\boldsymbol{\pi},i}_{infeasible}) = 0$, $\forall t \geq 0$.*

*Given the aforementioned analysis, it can be obtained that $\forall i \in \mathcal{N}$, $Pr(\boldsymbol{x}_t \in \boldsymbol{\mathcal{X}}^{\boldsymbol{\pi},i}_{infeasible}) = 0$, $\forall t \geq 0$ when $\{\pi^m\}_{m \in \{i+1,...,N\}}$ are fixed, and $k_i \to \infty$. Since the multi-timescale framework is applied, the joint policy updated will converge to $\boldsymbol{\pi}^* = \{\pi^*_i\}_{i \in \mathcal{N}}$, and thus when the number of updates for agent $i$ goes to infinity, $Pr(\boldsymbol{x}_t \in \boldsymbol{\mathcal{X}}^{\boldsymbol{\pi},i}_{infeasible}) = 0$, $\forall t \geq 0$.*

### D.3 PROOF FOR COROLLARY 1

**Proof 6** *For each agent $i$, the policy converges to a feasible policy $\pi^*_i$ that satisfies both the safety and stability constraints. Hence, the joint control signal $\boldsymbol{u}_t$, generated by $\boldsymbol{\pi}^*$, is a function of states, and therefore the whole multi-agent system becomes an autonomous system without external control input. In the absence of exogenous signal $\boldsymbol{d}_t$, we have $||\boldsymbol{x}_t|| \leq \xi(||\boldsymbol{x}_0||, t)$, which further implies that $\lim_{t \to \infty} ||\boldsymbol{x}_t|| = 0$. Therefore, the whole multi-agent system converges to the desired equilibrium (goal), and so does each subsystem (agent).*

## E PSEUDOCODE FOR MAS$^3$AC

Here we provide the detailed Algorithm 2, which essentially requires that, within each update $e$, the most recently updated $\{\theta^{e+1}_m\}_{m \in \{1,...,i-1\}}$, instead of $\{\theta^e_m\}_{m \in \{1,...,i-1\}}$, are used when updating $\theta_i$ for agent $i$. Other techniques, such as double-critic architectures, can also be incorporated in the real implementation for achieving better performance.

---

**Algorithm 2** Multi-Agent Safe and Stable Soft Actor-Critic (**MAS$^3$AC**)

---

**Input:** $\eta_1(e)$ for updating the action-value networks, barrier certificates, and Lyapunov certificates; $\eta_{2,i}(e)$ for updating the individual control policy $\pi^i$; and $\eta_{3,i}(e)$ for updating Lagrangian multipliers for agent $i$. Initialized parameters $\{\phi_i^0\}_{i \in \mathcal{N}}$, $\{\omega_i^0\}_{i \in \mathcal{N}}$, $\{\chi_i^0\}_{i \in \mathcal{N}}$, $\{\theta_i^0\}_{i \in \mathcal{N}}$, $\{\lambda_i^0\}_{i \in \mathcal{N}}$ for corresponding neural networks; maximum iterations $E$; policy update delay step $n_p$; Polyak coefficient $\tau$, and target neural networks $\{\phi_{i,targ}^0\}_{i \in \mathcal{N}} \leftarrow \{\phi_i^0\}_{i \in \mathcal{N}}$, $\{\omega_{i,targ}^0\}_{i \in \mathcal{N}} \leftarrow \{\omega_i^0\}_{i \in \mathcal{N}}$, $\{\chi_{i,targ}^0\}_{i \in \mathcal{N}} \leftarrow \{\chi_i^0\}_{i \in \mathcal{N}}$. $\eta_1(e) > \eta_{2,1}(e), \eta_{2,i}(e) > \eta_{3,i}(e), \forall i \in \mathcal{N}$, and $\eta_{3,i-1}(e) > \eta_{2,i}(e), \forall i \in \{2, \ldots, N\}$.

1: **for** $e = 0, 1, \ldots, E$ **do**
2:     Collect transitions $(\boldsymbol{x}_t, \boldsymbol{u}_t, \boldsymbol{x}_{t+1}, \boldsymbol{r}_t, \boldsymbol{b}_t, \boldsymbol{c}_t)$ when global information is available, or $\{(o_t^i, u_t^i, o_{t+1}^i, r_t^i, b_t^i, c_t^i)\}_{i \in \mathcal{N}}$ when only local observations are available, by using the joint policy $\boldsymbol{\pi}$ parameterized by $\{\theta_i^e\}_{i \in \mathcal{N}}$.
3:     Store transitions into the replay buffer.
4:     Sample a random minibatch $\mathcal{D}$ of transitions from the replay buffer.
5:     **for** $i = 1, 2, \ldots, N$ **do**
6:         Update agent $i$'s action-value network, barrier certificate, and Lyapunov certificate with the learning rate $\eta_1(e)$ and loss functions detailed in Section A:
        $\phi_i^{e+1} \leftarrow \phi_i^e - \eta_1(e)\nabla_{\phi_i} J_{Q^i}(Q^i)$;
        $\omega_i^{e+1} \leftarrow \omega_i^e - \eta_1(e)\nabla_{\omega_i} J_{H^i}(H^i)$;
        $\chi_i^{e+1} \leftarrow \chi_i^e - \eta_1(e)\nabla_{\chi_i} J_{V^i}(V^i)$.
7:         **if** $e \bmod n_p = 0$ **then**
8:             Update agent $i$'s control policy with the learning rate $\eta_{2,i}(e)$ and the loss function detailed in Section A:
            $\theta_i^{e+1} \leftarrow \theta_i^e - \eta_{2,i}(e)\nabla_{\theta_i} J_{\theta_i}^{\text{full}}(\theta_i)$.
9:             Update agent $i$'s Lagrangian multipliers with the learning rate $\eta_{3,i}(e)$ and the loss function detailed in Section A:
            $\lambda_{i,1}^{e+1} \leftarrow \left[\lambda_{i,1}^e + \eta_{3,i}(e)\nabla_{\lambda_{i,1}} J_{\lambda_i}^{\text{full}}(\lambda_i)\right]_+$;
            $\lambda_{i,2}^{e+1} \leftarrow \left[\lambda_{i,2}^e + \eta_{3,i}(e)\nabla_{\lambda_{i,2}} J_{\lambda_i}^{\text{full}}(\lambda_i)\right]_+$.
10:           Update agent $i$'s target networks:
            $\phi_{i,targ}^{e+1} \leftarrow (1-\tau)\phi_{i,targ}^e + \tau\phi_i^{e+1}$;
            $\omega_{i,targ}^{e+1} \leftarrow (1-\tau)\omega_{i,targ}^e + \tau\omega_i^{e+1}$;
            $\chi_{i,targ}^{e+1} \leftarrow (1-\tau)\chi_{i,targ}^e + \tau\chi_i^{e+1}$.
11:         **end if**
12:     **end for**
13:     Apply linear decay to all learning rates.
14: **end for**
**Output:** $\{\theta_i^E\}_{i \in \mathcal{N}}$

---

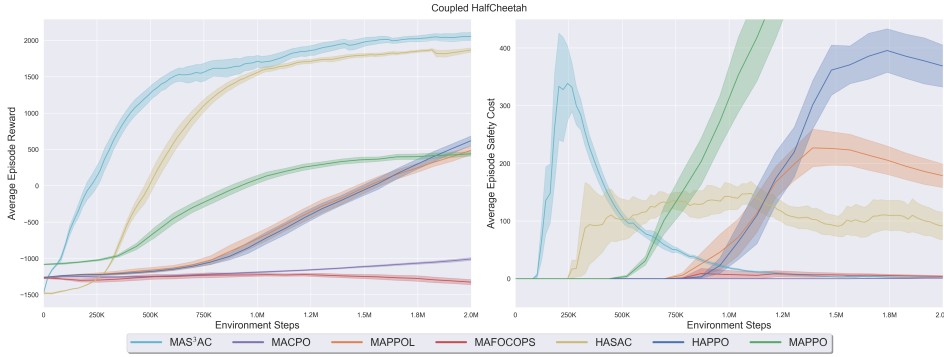

Figure 5: Average episode reward and average episode safety cost in the cooperative Coupled HalfCheetah. Here, MAS$^3$AC employs adaptive $\alpha_i$ instead of pre-defined values.

## F  FURTHER INVESTIGATION ON ISS CONDITIONS

In the main body, the inter-agent gain condition used in Theorem 2 is conservatively absorbed into the simplified dissipation term $\alpha_i'(V^i(x_t^i))$ to facilitate easy implementation, which has demonstrated better performance compared to baselines in experiments across benchmarks. However, in this section, we revisit the original form of the conditions for a more detailed investigation.

The original condition,

$$\sum_{j \neq i} \delta_{ij}(V^j(x_t^j)) + \beta_i \left( \sum_{j \neq i} \delta_{ij}(V^j(x_t^j)) \right) \leq \alpha_i(V^i(x_t^i)), \quad \forall i \in \mathcal{N},$$

is not treated as a fixed assumption in this section. Instead, we investigate its practical enforcement by integrating it as a constraint into the training loss. Specifically, it is possible to pre-define a system-wide set of gain weights $\{\delta_{ij}\}$ and functions $\{\beta_i\}$, and introduce the following inequality-based penalty into the total loss: $\left[ \sum_{j \neq i} \delta_{ij}(V^j(x_t^j)) + \beta_i \left( \sum_{j \neq i} \delta_{ij}(V^j(x_t^j)) \right) - \alpha_i(V^i(x_t^i)) \right]_+$, alongside a second constraint encouraging the original stability inequality: $\left[ \dot{V}^i(x_t^i) + \alpha_i(V^i(x_t^i)) - \sum_{j \neq i} \delta_{ij}(V^j(x_t^j)) \right]_+$, where $[\cdot]_+ = \max(0, \cdot)$ and we assume no exogenous signal $\boldsymbol{d}_t$ exists. Both $\alpha_i$ and the policy network are treated as trainable parameters. The full loss can include these terms to enforce the ISS for the multi-agent system.

As a preliminary validation of the above formulation, where the inter-agent gain condition is enforced via adaptive $\alpha_i$ through additional terms in the total training loss, we implemented a different actor network structure with the additional penalty terms in the training loss and evaluated the algorithm on the cooperative Coupled HalfCheetah benchmark. As shown in Figure 5, this version of MAS$^3$AC still achieves strong performance compared to baselines, demonstrating both high sample efficiency and a favorable balance between reward maximization and safety cost reduction. This result indicates that the explicit incorporation of inter-agent gain conditions can be practically effective, providing further evidence for the robustness of the MAS$^3$AC framework.

Nevertheless, we emphasize that this section presents only a preliminary investigation on integrating the inter-agent gain conditions as constraints. A more rigorous theoretical analysis and extensive empirical validation are left for future work.

## G  BASELINES

We compare MAS$^3$AC against both safe MARL algorithms and standard model-free MARL baselines: 1. **Cooperative tasks (global information)** As safe MARL baselines, we include MACPO, MAPPO-Lagrangian (Gu et al., 2023), which is abbreviated as MAPPOL in figures and tables, and MAFOCOPS (Zhao et al., 2023c). HASAC (Zhong et al., 2024), HAPPO (Kuba et al., 2022),

and MAPPO (Yu et al., 2022a) are used as standard MARL baselines. 2. **Non-cooperative tasks (local observations)** To ensure fair comparisons, we construct decentralized versions of MACPO and MAPPO-Lagrangian, restricting them to local information without coordination, as safe MARL baselines. For standard MARL baselines, we employ MASAC (the decentralized version of HASAC) and IPPO (De Witt et al., 2020). These baselines are widely adopted in prior studies and cover a broad spectrum of approaches, ranging from trust region-based methods to traditional dual ascent update, and from centralized to decentralized training.

## H BENCHMARK ENVIRONMENTS

To evaluate our proposed framework, we propose a suite of multi-agent benchmarks that capture diverse aspects of constrained decision-making. These environments are built upon MuJoCo-based robotic control tasks and an electrical bidding task with explicit state-wise safety and stability requirements, in addition to different reward structures. Each benchmark specifies: (i) the reward design, (ii) the safety constraints that agents need to satisfy, and (iii) desired equilibrium states associated with stability requirements. Each tasks involves multiple agents, with either global information or local observations to support centralized or decentralized training. Together, these benchmarks provide a comprehensive testbed for studying safe and stable MARL, spanning cooperative and non-cooperative settings, physics-based robotic control, and optimization-driven domains.

### H.1 COOPERATIVE ANT

We adopt the standard partitioning from the Safe Multi-Agent MuJoCo benchmark, where the Ant is divided into four agents, each controlling two joints ("4x2" configuration). The joint action is the concatenation of all agent actions, and all agents receive the full global state augmented with an agent identifier, and shared signals. The task is fully cooperative.

The reward encourages locomotion at a desired forward velocity $v^* = 0.7$ along the x-axis, while penalizing control effort. Formally, the reward at time $t$ is given by

$$r_t = -|v_t - v^*| - 0.5\|a_t\|^2 + \mathbf{1}_{[0.5 \le v_t \le 0.9]} \times 2.0,$$

where $v_t$ is the torso's forward velocity along the x-axis and $a_t$ is the joint action vector, and $\mathbf{1}_{[\cdot]}$ denotes the indicator function

Safety constraints are imposed by the presence of slanted walls forming a narrowing corridor. A safety violation occurs if the torso's lateral position $y_t$ is within 1.8 units of either wall, in which case the instantaneous safety cost (barrier signal) $b_t = -1.0$ is triggered; otherwise, $b_t = 0$.

To capture stability, a Lyapunov-inspired deviation cost signal can be defined as

$$c_t = |v_t - v^*|,$$

which penalizes deviation from the desired velocity. In practice, we further add an additional part to $c_t$ to avoid excessive actuation. Notably, the issue of infeasibility arises when the stability constraint conflicts with the safety constraint: maintaining the desired forward velocity along the x-axis when Ant is close to the slanted walls can inevitably lead the ant into unsafe regions that are too close to the wall in subsequent states. An illustration can be found in Figure 7.

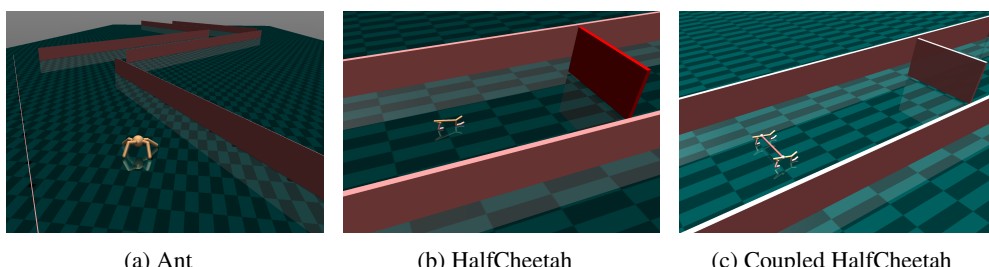

(a) Ant        (b) HalfCheetah        (c) Coupled HalfCheetah

Figure 6: Three cooperative tasks with global information.

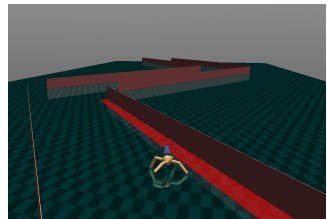

Figure 7: Illustration of the issue of infeasibility in the Ant task. Each agent is required to maintain a desired velocity along the x-axis (blue arrow). From certain states near the wall, satisfying this stability requirement inevitably leads to safety violations: the agent enters the danger region (red) that lies too close to the wall.

In sum, in this cooperative task, agents must coordinate their leg motions to maximize locomotion reward while simultaneously maintaining safety within the corridor and stability towards the equilibrium state.

## H.2 COOPERATIVE HALFCHEETAH

We adopt the standard Safe Multi-Agent MuJoCo partitioning, splitting the HalfCheetah into three agents, each controlling two joints ("3x2" configuration). The joint action is the concatenation of per-agent actions. All agents receive the full global state augmented with an agent identifier, and shared signals, yielding a fully cooperative setting.

Let $x_t$ denote the cheetah torso position along the x-axis and $w_t$ the position of the movable wall along the x-axis. Define the distance $d_t := w_t - x_t$. The objective is to maintain a desired distance $d^* = 12.0$ while minimizing control effort. The per-timestep reward is

$$r_t = -0.1 \times |d_t - d^*| - 0.1 \|a_t\|^2 + 3.0 \times \mathbf{1}_{[7.0 \le d_t \le 14.0]},$$

where $a_t$ is the joint action vector and $\mathbf{1}_{[\cdot]}$ denotes the indicator function.

A safety violation occurs when the cheetah is too close to the moving wall: $d_t < d_{\min}$ with $d_{\min} = 9.0$. We expose a barrier signal

$$b_t = -\max\{0, d_{\min} - d_t\},$$

which is zero when the constraint is satisfied and strictly negative otherwise.

Stability cost can also be defined to penalize departures from the equilibrium distance:

$$c_t = 0.1 \times |d_t - d^*|$$

and can practically be added an extra term to avoid excessive actuation. Similarly, this environment exhibits conflicting scenarios: an additional reward encouraging stability is granted for $7 \le d_t \le 14$, which partially overlaps with the unsafe region $d_t < 9$. In sum, agents must coordinate to track the movable boundary at the desired distance while ensuring state-wise constraints.

## H.3 COOPERATIVE COUPLED HALFCHEETAH

We adopt the standard Safe Multi-Agent MuJoCo partitioning for the Coupled HalfCheetah, yielding two agents each controlling one HalfCheetah. Two HalfCheetahs are coupled by a tendon with a finite length range, enforcing coordination between the bodies. The joint action is the concatenation of all per-agent actions. All agents can access the full global state augmented with an agent identifier, and shared signals, resulting in a fully cooperative task.

Let $x_t^{(1)}$ and $x_t^{(2)}$ denote the positions of the two cheetah torsos along the x-axis, and $w_t$ the position of the movable wall along the x-axis. Define the midpoint $m_t := \frac{1}{2}\left(x_t^{(1)} + x_t^{(2)}\right)$ and the distance $d_t := w_t - m_t$. The objective is to maintain a desired distance $d^* = 12.0$ while minimizing control effort. The per-timestep reward is

$$r_t = -0.1 \times |d_t - d^*| - 0.1 \|a_t\|^2 + 3.0 \times \mathbf{1}_{[10 \le d_t \le 14]},$$

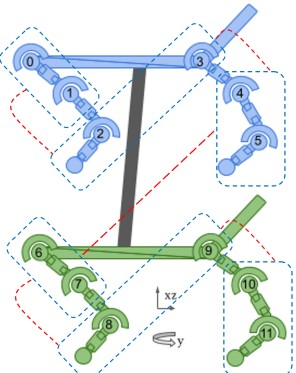

Figure 8: Communication in the decentralized non-cooperative Coupled HalfCheetah task. Each dashed rectangle represents one agent, which controls two joints. Information can be communicated through the red dashed line to neighbor agents. The number of red lines that the information can go through is dependent on the defined neighbor size.

where $a_t$ is the joint action vector and $\mathbf{1}_{[\cdot]}$ is the indicator function.

State-wise safety requires each cheetah to keep a minimum distance to the wall. With $d_t^{(i)} := w_t - x_t^{(i)}$ and $d_{\min} = 11.0$, a violation occurs when $d_t^{(i)} < d_{\min}$. We expose a barrier signal

$$b_t = -\sum_{i=1}^{2} \max\{0,\, d_{\min} - d_t^{(i)}\},$$

which is zero when both cheetahs are safe and strictly negative otherwise.

The cost penalizing departures from the equilibrium distance is:

$$c_t = 0.1 \times |d_t - d^*| \,.$$

An extra term can also be added to avoid excessive activation. This environment also exhibits conflicting cases: the stability bonus is granted for $10 \leq d_t \leq 14$, which can partially overlap with unsafe regions where either $d_t^{(1)}$ or $d_t^{(2)}$ falls below 11. In sum, the Coupled HalfCheetah benchmark challenges agents to coordinate locomotion across two physically coupled bodies, maintain a desired distance from a movable wall, and balance reward maximization with both state-wise safety and stability requirements.

### H.4  NON-COOPERATIVE COUPLED HALFCHEETAH

This environment comprises two tendon-coupled HalfCheetah and six agents; each agent controls two actuators: *(1) bthigh, bshin, (2) bfoot, fthigh, (3) fshin, ffoot, (4) bthigh2, bshin2, (5) bfoot2, fthigh2, (6) fshin2, ffoot2*. Agents observe only *local* information with optional $k$-neighbor augmentation (here, $k=3$ with zero-padding at the boundaries). For each agent $i$, the observation includes its own and $k$-neighbor's joint positions/velocities and a compact set of environment features (wall velocity, etc).

Let $x_t^{(1)}$ and $x_t^{(2)}$ be the $x$-positions of the two torsos and $w_t$ the wall position. Define per-cheetah distances $d_t^{(1)}:=w_t-x_t^{(1)}$, $d_t^{(2)}:=w_t-x_t^{(2)}$, the desired standoff $d^*=12.0$, and the stability interval $[d_{\min}^{\mathrm{stab}}, d_{\max}^{\mathrm{stab}}]=[10, 14]$. For agents $i \in \{1, 2, 3\}$ (first cheetah),

$$r_t^{(i)} = -0.1 \times |d_t^{(1)} - d^*| - 0.1\,\|a_t^{(i)}\|^2 + 3.0 \times \mathbf{1}_{[\,10 \leq d_t^{(1)} \leq 14\,]} + \rho_t^{(1)},$$

and for agents $i \in \{4, 5, 6\}$ (second cheetah),

$$r_t^{(i)} = -0.1 \times |d_t^{(2)} - d^*| - 0.1\,\|a_t^{(i)}\|^2 + 3.0 \times \mathbf{1}_{[\,10 \leq d_t^{(2)} \leq 14\,]} + \rho_t^{(2)},$$

where $a_t^{(i)}$ is agent $i$'s action vector, and $\rho_t^{(1)}, \rho_t^{(2)}$ are small competition terms that reward being ahead of the other HalfCheetah (zero if leading, $-0.05$ otherwise).

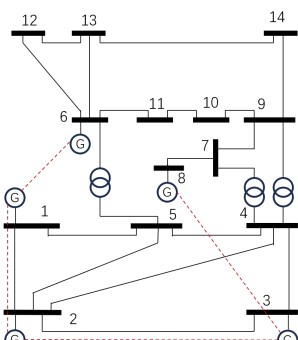

Figure 9: The network topology for the non-cooperative electrical bidding task. Each generator, marked with a 'G', represents an independent agent competing in the electricity market. The numbered nodes are buses interconnected by transmission lines. The red dashed lines indicate the communication lines between specific generator agents.

Each HalfCheetah must keep a minimum clearance $d_{\min}=11.0$ from the wall. We expose a per-halfcheetah barrier signal and feed it to each agent on that body:

$$b_t^{(1)} = -\max\{0,\, d_{\min} - d_t^{(1)}\}, \qquad b_t^{(2)} = -\max\{0,\, d_{\min} - d_t^{(2)}\}.$$

A safe state yields $b_t^{(\cdot)}=0$; otherwise it is strictly negative.

Stability is encouraged by the per-cheetah cost that each local agent inherits, for distance tracking:

$$c_t^{(1)} = 0.1 \times |d_t^{(1)} - d^*|, \qquad c_t^{(2)} = 0.1 \times |d_t^{(2)} - d^*|.$$

Extra terms can also be added to these costs to avoid excessive activation. Conflicts may arise since the stability reward interval $[10, 14]$ partially overlaps the unsafe region $d_t^{(\cdot)}<11$. In sum, this non-cooperative benchmark evaluates decentralized control with local sensing and per-agent objectives across two physically coupled half cheetahs; each agent must balance its own performance against state-wise constraints under partial observability. The communication among agents in this multi-agent system is described in Figure 8.

### H.5 NON-COOPERATIVE ELECTRICAL BIDDING TASK

This environment models a competitive electricity market based on the IEEE 14-bus system, featuring five generator nodes that function as agents, indexed by $i \in \{1, \ldots, 5\}$. Each agent participates in the market by submitting a bid price, $k_{i,t}$, at each time step $t$.

Agents operate under partial observability and observe only *local* and neighboring information. For each agent $i$, the observation includes the load demand $P_{\text{load},j,t}$ from its neighboring nodes $j$, the previous power outputs $P_{\text{out},j,t-1}$ of neighboring generators, and its own previous market outcomes—the clearing price $\lambda_{i,t-1}$ and power output $P_{\text{out},i,t-1}$.

At each time step, a central system operator collects the bid prices $\mathbf{K_t}$ and solves a DC Optimal Power Flow (DC-OPF) problem to clear the market. This process determines the power output $\mathbf{P}_{\text{out},\mathbf{t}}$ and the market clearing price (locational marginal price) $\boldsymbol{\lambda_t}$ for each generator. The optimization is defined as:

$$\min_{\mathbf{P}_{\text{out},\mathbf{t}}} \quad \mathbf{K_t^\top P_{\text{out},t}},$$

$$\text{s.t.} \quad \mathbf{P_n = B\theta} : \boldsymbol{\lambda_t},$$

$$-\mathbf{P_{\text{branchlimit}}} \leq \mathbf{P_{b,t}} \leq \mathbf{P_{\text{branchlimit}}},$$

$$\mathbf{P_{lb}} \leq \mathbf{P_{\text{out},t}} \leq \mathbf{P_{ub}},$$

$$\mathbf{P_{\text{out},t-1} - \Delta P_{\text{out}}} \leq \mathbf{P_{\text{out},t}} \leq \mathbf{P_{\text{out},t-1}} + \Delta P_{out}$$

Here, the constraints represent the DC power flow balance, transmission line capacity limits, generator output limits, and ramp-rate limits, respectively.

The reward for each agent $i$ is a composite signal designed to balance economic revenue with operational stability. It consists of the scaled revenue, a bonus for operating within a desired output range, and a penalty for deviating from a specific target output:

$$r_t^{(i)} = \alpha_r(\lambda_{i,t} P_{\mathrm{out},i,t} - c_{g,i,t}) + \beta_s \cdot \mathbf{1}_{[\,P_{\mathrm{des,low},i} \leq P_{\mathrm{out},i,t} \leq P_{\mathrm{des,up},i}\,]} - \gamma_d \cdot |P_{\mathrm{out},i,t} - P_{\mathrm{des},i}|,$$

$$\text{where} \quad c_{g,i,t} = a_i P_{\mathrm{out},i,t}^2 + b_i P_{\mathrm{out},i,t} + c_i.$$

Here, $c_{g,i,t}$ is the quadratic generation cost, $\mathbf{1}_{[\cdot]}$ is the indicator function, $[P_{\mathrm{des,low},i}, P_{\mathrm{des,up},i}]$ defines the stable operating range, and $P_{\mathrm{des},i}$ is the single ideal output point. The terms $\alpha_r, \beta_s$, and $\gamma_d$ are positive weighting coefficients for revenue, stability bonus, and deviation penalty, respectively.

To enforce environmental constraints, a safety barrier signal $b_t^{(i)}$ is provided to each agent $i$. This signal penalizes any generation that causes carbon emissions to exceed a predefined allowance, $Carbon_{\mathrm{allow},i}$:

$$b_t^{(i)} = -\max\{0, \phi_i P_{\mathrm{out},i,t} - Carbon_{\mathrm{allow},i}\},$$

where $\phi_i$ is the carbon emission intensity of generator $i$. A safe state yields $b_t^{(i)} = 0$; otherwise, it is strictly negative.

Operational stability is encouraged via a cost signal $c_t^{(i)}$, which penalizes the absolute deviation from a single desired power output point $P_{\mathrm{des},i}$:

$$c_t^{(i)} = \gamma_d \cdot |P_{\mathrm{out},i,t} - P_{\mathrm{des},i}|.$$

In sum, this environment serves as a non-cooperative benchmark for evaluating decentralized bidding strategies under partial observability. Conflicts may arise since the stability reward interval partially overlaps the unsafe region. Each agent must learn to balance maximizing its own revenue against system-wide constraints, including carbon emission limits (safety) and operational stability.

## H.6 NON-COOPERATIVE ANT

In addition to the decentralized version of Coupled HalfCheetah, here we propose another decentralized setting for robot control. This non-cooperative Ant environment comprises eight agents, with each agent controlling one actuator: *(1) hip1, (2) ankle1, (3) hip2, (4) ankle2, (5) hip3, (6) ankle3, (7) hip4, (8) ankle4*. Agents observe only *local* information with optional $k$-neighbor augmentation (here, $k=3$ with zero-padding at the boundaries). For each agent $i$, the observation includes its own and $k$-neighbor's joint positions/velocities and a compact set of environment features (lateral offset, etc).

Let $v_t$ be the torso $x$-velocity, and $v^*=0.7$ be the desired forward velocity along the x-axis. Define the deviation $\delta_t := |v_t - v^*|$ and the stability interval $[v_{\min}^{\mathrm{stab}}, v_{\max}^{\mathrm{stab}}] = [0.5, 0.9]$. Each agent $i$ controls a single actuator $a_t^{(i)}$ and belongs to one of four legs $\ell(i) \in \{1, 2, 3, 4\}$. The reward for agent $i$ is given by

$$r_t^{(i)} = -0.5||a_t^{(i)}||^2 - |v_t - v^*| + 2.0 \times \mathbf{1}_{[\,0.5 \leq v_t \leq 0.9\,]} + \rho_t^{(\ell(i))},$$

where $\rho_t^{(\ell)}$ is a competition term based on the leg's forward rank (by $x$-position of the leg's center of mass): $\rho_t^{(\ell)} \in \{-0.15, -0.09, 0.09, 0.15\}$ from trailing to leading leg, and shared by the two actuators on that leg.

The Ant moves through a piecewise-slanted corridor bounded by two walls at lateral offsets $\pm 5$ from a segmented centerline. For each leg $\ell$ with position $(x_t^{(\ell)}, y_t^{(\ell)})$, a per-agent barrier signal is computed as

$$b_t^{(i)} = -\mathbf{1}\left\{\min\left(|y_t^{(\ell(i))} - \mathrm{wall}^+(x_t^{(\ell(i))})|, |y_t^{(\ell(i))} - \mathrm{wall}^-(x_t^{(\ell(i))})|\right) < 1.0\right\},$$

which equals 0 in safe states and $-1$ when the leg is within $1.0\,\mathrm{m}$ of either wall.

To capture stability, a Lyapunov-inspired deviation cost signal for each agent $i$ can be defined as

$$c_t^{(i)} = |v_t - v^*|,$$

which penalizes deviation from the desired velocity. In practice, we further add an additional part to $c_t^{(i)}$ to avoid excessive actuation. In summary, this non-cooperative benchmark evaluates decentralized control with local sensing and per-agent objectives across the four-legged Ant. Each agent must coordinate its own motion to maintain the target velocity, stay within the slanted corridor, and compete for forward advancement under partial observability.

## H.7 Code Availability and Hyperparameters Setting

To facilitate reproducibility, we provide the anonymized implementation of MAS$^3$AC as supplementary material. The submitted code is for one **cooperative multi-agent MuJoCo task**. This implementation corresponds to the setup described and includes relevant configurations and training scripts. While the current submission contains only one task for review purposes, a full version of the code, which covers general-sum tasks (with different task structures and training code) and additional cooperative environments, will be released publicly upon paper acceptance.

More information, including hyperparameter settings, environment configurations, and implementation details, is clearly specified within the provided code files. For convenience, we provide the setting for some hyperparameters in Table 3. Notably, although linear decay is always applied to all learning rates, different initial values can be assigned. Besides the values provided in Table 3, we additionally try this set of initial values where differences are larger such that learning rates during training are more different from each other: Critics learning rate initial value 0.004, Policies learning rates initial values [0.002, 0.0005, 0.000125, 0.00003125] Lagrangian multipliers learning rates initial values [0.001, 0.00025, 0.0000625, 0.000015625]. The corresponding results are still good, as shown in Figure 22.

Table 3: Parameters setting for MAS$^3$AC.

| Hyperparameter | Value |
|---|---|
| Discount factor | 0.99 |
| Optimizer | Adam |
| Target smoothing coefficient | 0.005 |
| Initial entropy coefficient | 0.2 |
| Entropy coefficient learning rate | $3 \times 10^{-4}$ |
| Policy update interval | 2 |
| Target update interval | 2 |
| Activation function | ReLU |
| Final activation function | Tanh |
| Initialization method | Orthogonal |
| Neural Network (MLP) | [256, 256] |
| Critics learning rate | Initial value 0.002, linear decay applied |
| Policies learning rates | Initial values [0.001, 0.0008, 0.0006, 0.0004] , linear decay applied |
| Lagrangian multipliers learning rates | Initial values [0.0009, 0.0007, 0.0005, 0.0003] , linear decay applied |
| Batch size | 1000 |
| Buffer size | 1000000 |

# I Additional Experimental Results

## I.1 Additional Main Results

### I.1.1 The evaluation on the electrical bidding task

We further evaluate MAS$^3$AC on the decentralized electrical bidding environment, where the system dynamics is governed by a constrained optimization problem rather than physical robot dynamics, providing a qualitatively different benchmark. In this setting, each generator agent optimizes its

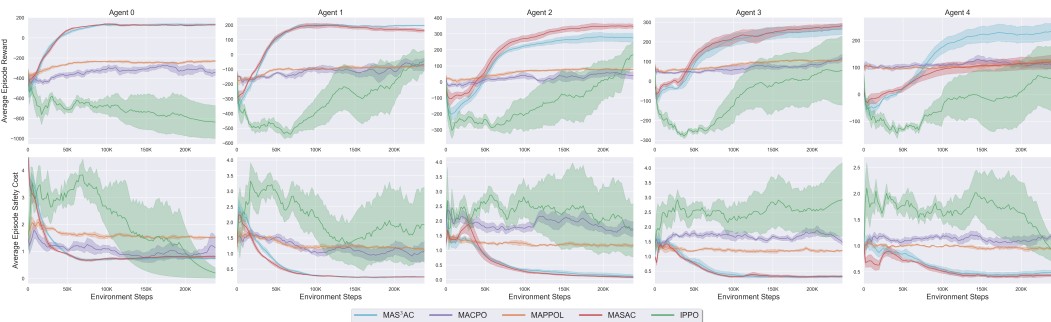

Figure 10: Average episode reward and average episode safety cost in the non-cooperative electrical bidding task, comparing MAS³AC against baselines. Each curve illustrates the average across 5 experiments employing different random seeds, with the shaded area denoting the standard error.

Table 4: Reward and safety cost comparison for the non-cooperative electrical bidding task across agents.

| | Agent 0 | | Agent 1 | | Agent 2 | |
|---|---|---|---|---|---|---|
| **Algorithm** | **Reward** | **Cost** | **Reward** | **Cost** | **Reward** | **Cost** |
| **IPPO** | $-805.15 \pm 406.63$ | $\mathbf{0.67 \pm 2.41}$ | $-141.19 \pm 420.37$ | $1.87 \pm 3.29$ | $58.84 \pm 405.21$ | $2.01 \pm 3.26$ |
| **MACPO** | $-315.37 \pm 211.80$ | $1.00 \pm 1.27$ | $-87.69 \pm 179.42$ | $1.02 \pm 1.02$ | $44.77 \pm 110.30$ | $1.81 \pm 1.18$ |
| **MAPPOL** | $-238.07 \pm 85.68$ | $1.51 \pm 0.51$ | $-83.65 \pm 54.88$ | $1.17 \pm 0.40$ | $78.57 \pm 35.76$ | $1.18 \pm 0.43$ |
| **MASAC** | $127.49 \pm 35.71$ | $0.80 \pm 0.15$ | $169.07 \pm 41.26$ | $\mathbf{0.24 \pm 0.07}$ | $\mathbf{348.67 \pm 48.67}$ | $\mathbf{0.10 \pm 0.07}$ |
| **MAS³AC** | $\mathbf{131.42 \pm 38.09}$ | $0.75 \pm 0.16$ | $\mathbf{194.20 \pm 31.61}$ | $\mathbf{0.24 \pm 0.07}$ | $276.81 \pm 70.11$ | $0.15 \pm 0.16$ |

| | Agent 3 | | Agent 4 | |
|---|---|---|---|---|
| **Algorithm** | **Reward** | **Cost** | **Reward** | **Cost** |
| **IPPO** | $38.08 \pm 358.01$ | $2.78 \pm 2.73$ | $27.36 \pm 273.47$ | $1.33 \pm 1.52$ |
| **MACPO** | $93.70 \pm 73.92$ | $1.67 \pm 0.84$ | $111.83 \pm 63.72$ | $1.11 \pm 0.55$ |
| **MAPPOL** | $106.33 \pm 35.23$ | $1.20 \pm 0.39$ | $116.92 \pm 44.45$ | $0.94 \pm 0.25$ |
| **MASAC** | $\mathbf{270.74 \pm 59.21}$ | $0.32 \pm 0.17$ | $115.16 \pm 77.97$ | $\mathbf{0.42 \pm 0.15}$ |
| **MAS³AC** | $258.59 \pm 67.66$ | $\mathbf{0.31 \pm 0.12}$ | $\mathbf{231.22 \pm 75.31}$ | $0.46 \pm 0.17$ |

own objective under local safety and stability constraints. Results show that MAS³AC still mostly outperforms baselines across rewards and safety costs, except for MASAC (see Figure 10 and Table 4). To further investigate this, we conducted extended training runs with a longer horizon. As shown in Figure 11 and Table 5, the difference between the two algorithms becomes obvious with more training steps, and MAS³AC succeeds in obtaining better performance in most cases. This extended evaluation confirms that MAS³AC maintains a long-term advantage in both reward maximization and safety constraint satisfaction, highlighting its robustness in complex, optimization-driven multi-agent environments.

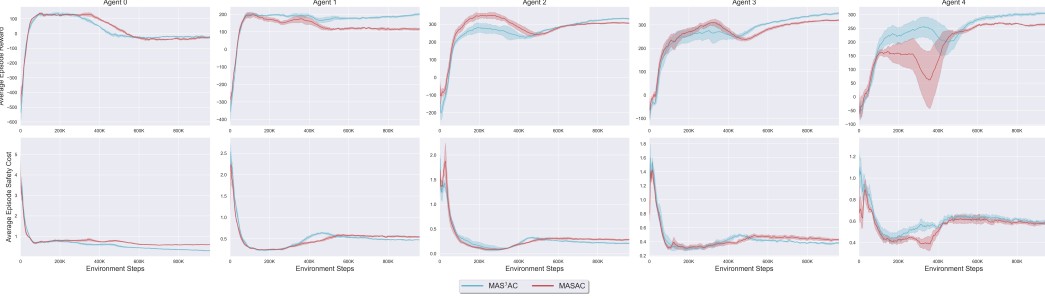

Figure 11: Average episode reward and average episode safety cost in the non-cooperative electrical bidding task, comparing MAS³AC and MASAC over an extended training horizon.

Table 5: Reward and safety cost comparison for the non-cooperative electrical bidding task with extended training steps.

| | Agent 0 | | Agent 1 | | Agent 2 | |
|---|---|---|---|---|---|---|
| Algorithm | Reward | Cost | Reward | Cost | Reward | Cost |
| MASAC | $-29.93 \pm 24.78$ | $0.59 \pm 0.07$ | $116.23 \pm 20.19$ | $0.54 \pm 0.07$ | $307.58 \pm 11.91$ | $0.28 \pm 0.05$ |
| MAS$^3$AC | $\mathbf{-21.56 \pm 23.98}$ | $\mathbf{0.31 \pm 0.05}$ | $\mathbf{197.39 \pm 18.30}$ | $\mathbf{0.47 \pm 0.05}$ | $\mathbf{331.98 \pm 13.67}$ | $\mathbf{0.21 \pm 0.04}$ |

| | Agent 3 | | Agent 4 | |
|---|---|---|---|---|
| Algorithm | Reward | Cost | Reward | Cost |
| MASAC | $319.78 \pm 11.31$ | $0.43 \pm 0.06$ | $262.92 \pm 15.75$ | $\mathbf{0.57 \pm 0.07}$ |
| MAS$^3$AC | $\mathbf{347.71 \pm 13.34}$ | $\mathbf{0.37 \pm 0.05}$ | $\mathbf{303.22 \pm 16.13}$ | $0.58 \pm 0.08$ |

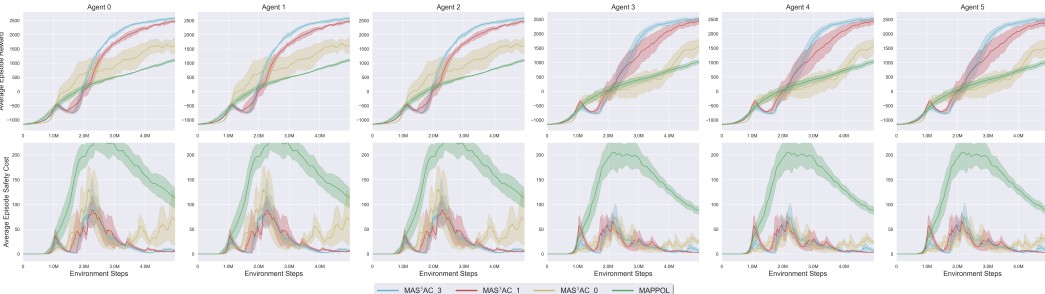

Figure 12: Average episode reward and average episode safety cost in the non-cooperative Coupled HalfCheetah task, comparing MAS$^3$AC with more different neighbor sizes (1 and 3).

## I.2 ADDITIONAL ABLATION STUDIES

### I.2.1 ADDITIONAL RESULTS ON NEIGHBOR SIZE

Table 6: Reward and safety cost comparison for the non-cooperative Coupled HalfCheetah task with different neighbor sizes.

| | Agent 0 | | Agent 1 | | Agent 2 | |
|---|---|---|---|---|---|---|
| Algorithm | Reward | Cost | Reward | Cost | Reward | Cost |
| IPPO | $1379.02 \pm 226.55$ | $165.11 \pm 107.22$ | $1387.27 \pm 225.99$ | $165.11 \pm 107.22$ | $1387.49 \pm 223.97$ | $165.11 \pm 107.22$ |
| MACPO | $193.36 \pm 152.44$ | $25.36 \pm 29.17$ | $191.02 \pm 152.34$ | $25.36 \pm 29.17$ | $189.91 \pm 152.60$ | $25.36 \pm 29.17$ |
| MAPPOL | $1288.79 \pm 230.43$ | $80.14 \pm 49.00$ | $1289.21 \pm 235.01$ | $80.14 \pm 49.00$ | $1287.07 \pm 230.98$ | $80.14 \pm 49.00$ |
| MASAC | $2486.44 \pm 65.09$ | $35.17 \pm 21.86$ | $2485.82 \pm 77.74$ | $35.17 \pm 21.86$ | $2468.01 \pm 62.02$ | $35.17 \pm 21.86$ |
| MAS$^3$AC_0 | $1743.53 \pm 716.81$ | $80.66 \pm 107.23$ | $1764.67 \pm 733.83$ | $80.66 \pm 107.23$ | $1769.68 \pm 722.35$ | $80.66 \pm 107.23$ |
| MAS$^3$AC_2 | $\mathbf{2502.49 \pm 86.37}$ | $\mathbf{8.54 \pm 17.46}$ | $\mathbf{2508.21 \pm 95.13}$ | $\mathbf{8.54 \pm 17.46}$ | $\mathbf{2503.10 \pm 91.08}$ | $\mathbf{8.54 \pm 17.46}$ |

| | Agent 3 | | Agent 4 | | Agent 5 | |
|---|---|---|---|---|---|---|
| Algorithm | Reward | Cost | Reward | Cost | Reward | Cost |
| IPPO | $1425.27 \pm 286.82$ | $165.11 \pm 107.22$ | $1429.73 \pm 285.75$ | $183.70 \pm 98.07$ | $1434.62 \pm 287.63$ | $183.70 \pm 98.07$ |
| MACPO | $80.03 \pm 250.42$ | $20.82 \pm 15.67$ | $80.26 \pm 250.40$ | $20.82 \pm 15.67$ | $80.64 \pm 250.74$ | $20.82 \pm 15.67$ |
| MAPPOL | $1205.69 \pm 186.90$ | $76.28 \pm 42.80$ | $1213.27 \pm 187.39$ | $76.28 \pm 42.80$ | $1211.63 \pm 185.34$ | $76.28 \pm 42.80$ |
| MASAC | $2423.02 \pm 63.20$ | $70.67 \pm 49.07$ | $2451.57 \pm 64.97$ | $70.67 \pm 49.07$ | $2447.62 \pm 72.79$ | $70.67 \pm 49.07$ |
| MAS$^3$AC_0 | $1833.44 \pm 403.83$ | $43.64 \pm 38.40$ | $1830.81 \pm 408.90$ | $43.64 \pm 38.40$ | $1837.18 \pm 402.57$ | $43.64 \pm 38.40$ |
| MAS$^3$AC_2 | $\mathbf{2523.87 \pm 71.62}$ | $\mathbf{2.21 \pm 4.16}$ | $\mathbf{2521.11 \pm 69.39}$ | $\mathbf{2.21 \pm 4.16}$ | $\mathbf{2521.40 \pm 71.64}$ | $\mathbf{2.21 \pm 4.16}$ |

We further provide extended results on the effect of neighbor size in the decentralized Coupled HalfCheetah benchmark, shown in Figure 12 and Table 7. Across all cases, MAS$^3$AC outperforms the baseline MAPPOL, confirming its robustness under varying information availability. Within MAS$^3$AC itself, we observe clear differences: with neighbor size 0, the algorithm performs poorly in terms of safety, as relying only on self-information fails to capture the inherently inter-agent nature of the safety constraints. Comparing neighbor size 1 and 3, larger neighborhoods do not always translate into uniformly better results. While size 3 tends to yield higher rewards, size 1 achieves lower final safety violations for Agent 3, 4, and 5, suggesting that adding more information

Table 7: Reward and safety cost comparison for the non-cooperative Coupled HalfCheetah task with more different neighbor sizes (1 and 3).

| | Agent 0 | | Agent 1 | | Agent 2 | |
|---|---|---|---|---|---|---|
| Algorithm | Reward | Cost | Reward | Cost | Reward | Cost |
| MAPPOL | $1229.41 \pm 246.37$ | $96.94 \pm 68.86$ | $1230.19 \pm 247.36$ | $96.94 \pm 68.86$ | $1226.87 \pm 245.02$ | $96.94 \pm 68.86$ |
| MAS$^3$AC_0 | $1587.19 \pm 777.46$ | $85.27 \pm 164.09$ | $1610.15 \pm 788.09$ | $85.27 \pm 164.09$ | $1610.81 \pm 778.27$ | $85.27 \pm 164.09$ |
| MAS$^3$AC_1 | $2491.39 \pm 136.21$ | $7.09 \pm 14.27$ | $2492.96 \pm 130.55$ | $7.09 \pm 14.27$ | $2489.45 \pm 132.87$ | $7.09 \pm 14.27$ |
| MAS$^3$AC_3 | $\mathbf{2576.03 \pm 124.63}$ | $\mathbf{6.86 \pm 13.78}$ | $\mathbf{2581.35 \pm 125.65}$ | $\mathbf{6.86 \pm 13.78}$ | $\mathbf{2579.96 \pm 125.70}$ | $\mathbf{6.86 \pm 13.78}$ |

| | Agent 3 | | Agent 4 | | Agent 5 | |
|---|---|---|---|---|---|---|
| Algorithm | Reward | Cost | Reward | Cost | Reward | Cost |
| MAPPOL | $1147.33 \pm 271.41$ | $69.14 \pm 39.50$ | $1151.22 \pm 273.06$ | $69.14 \pm 39.50$ | $1152.16 \pm 273.03$ | $69.14 \pm 39.50$ |
| MAS$^3$AC_0 | $1662.30 \pm 504.71$ | $21.93 \pm 25.29$ | $1658.89 \pm 515.04$ | $21.93 \pm 25.29$ | $1666.24 \pm 503.59$ | $21.93 \pm 25.29$ |
| MAS$^3$AC_1 | $2449.61 \pm 362.57$ | $\mathbf{4.02 \pm 6.72}$ | $2444.83 \pm 362.42$ | $\mathbf{4.02 \pm 6.72}$ | $2438.13 \pm 364.22$ | $\mathbf{4.02 \pm 6.72}$ |
| MAS$^3$AC_3 | $\mathbf{2522.42 \pm 142.71}$ | $5.70 \pm 8.02$ | $\mathbf{2525.37 \pm 138.01}$ | $5.70 \pm 8.02$ | $\mathbf{2528.33 \pm 143.15}$ | $5.70 \pm 8.02$ |

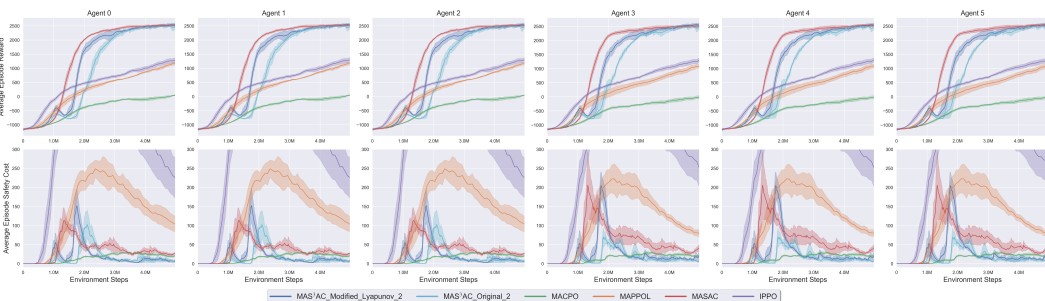

Figure 13: Average episode reward and average episode safety cost in the non-cooperative Coupled HalfCheetah task, comparing MAS$^3$AC with different Lyapunov certificate designs. The modified one represents the case where the input to the Lyapunov certificate only includes the agent's own state and action, while the original one represents the case where the input also includes information from all neighbor agents.

does not monotonically improve performance for all agents. A possible explanation is that larger neighborhoods introduce more complex inputs that require more sophisticated learning, or that each task may have an upper bound on useful neighbor size, beyond which additional information does not provide proportional benefits.

### I.2.2 ADDITIONAL RESULTS ON LYAPUNOV FUNCTION INPUT

Table 8: Reward and safety cost comparison for the non-cooperative Coupled HalfCheetah task with different Lyapunov certificate designs

| | Agent 0 | | Agent 1 | | Agent 2 | |
|---|---|---|---|---|---|---|
| Algorithm | Reward | Cost | Reward | Cost | Reward | Cost |
| IPPO | $902.29 \pm 386.41$ | $451.38 \pm 305.37$ | $904.99 \pm 389.86$ | $451.38 \pm 305.37$ | $905.91 \pm 388.62$ | $451.38 \pm 305.37$ |
| MACPO | $-140.28 \pm 276.72$ | $21.48 \pm 25.26$ | $-141.91 \pm 276.68$ | $21.48 \pm 25.26$ | $-142.75 \pm 276.65$ | $21.48 \pm 25.26$ |
| MAPPOL | $735.92 \pm 416.97$ | $174.96 \pm 143.68$ | $735.77 \pm 417.75$ | $174.96 \pm 143.68$ | $732.35 \pm 417.00$ | $174.96 \pm 143.68$ |
| MASAC | $2501.57 \pm 86.21$ | $28.06 \pm 28.15$ | $2508.60 \pm 90.78$ | $28.06 \pm 28.15$ | $2496.33 \pm 89.72$ | $28.06 \pm 28.15$ |
| MAS$^3$AC_2 | $2481.78 \pm 217.30$ | $\mathbf{8.20 \pm 19.13}$ | $2485.79 \pm 217.27$ | $\mathbf{8.20 \pm 19.13}$ | $2481.58 \pm 224.22$ | $\mathbf{8.20 \pm 19.13}$ |
| MAS$^3$AC | $\mathbf{2511.67 \pm 164.10}$ | $9.06 \pm 16.43$ | $\mathbf{2510.17 \pm 161.60}$ | $9.06 \pm 16.43$ | $\mathbf{2505.89 \pm 166.87}$ | $9.06 \pm 16.43$ |

| | Agent 3 | | Agent 4 | | Agent 5 | |
|---|---|---|---|---|---|---|
| Algorithm | Reward | Cost | Reward | Cost | Reward | Cost |
| IPPO | $898.48 \pm 395.57$ | $432.92 \pm 273.22$ | $900.26 \pm 397.68$ | $432.92 \pm 273.22$ | $903.87 \pm 397.26$ | $432.92 \pm 273.22$ |
| MACPO | $-215.51 \pm 277.43$ | $21.14 \pm 27.75$ | $-216.04 \pm 277.56$ | $21.14 \pm 27.75$ | $-215.42 \pm 277.29$ | $21.14 \pm 27.75$ |
| MAPPOL | $646.70 \pm 464.02$ | $150.76 \pm 134.02$ | $648.57 \pm 465.17$ | $150.76 \pm 134.02$ | $648.17 \pm 466.71$ | $150.76 \pm 134.02$ |
| MASAC | $2485.39 \pm 102.35$ | $37.73 \pm 35.27$ | $\mathbf{2502.09 \pm 97.16}$ | $37.73 \pm 35.27$ | $\mathbf{2505.65 \pm 108.50}$ | $37.73 \pm 35.27$ |
| MAS$^3$AC_2 | $2471.78 \pm 226.85$ | $\mathbf{8.67 \pm 19.91}$ | $2466.98 \pm 226.57$ | $\mathbf{8.67 \pm 19.91}$ | $2464.22 \pm 227.26$ | $\mathbf{8.67 \pm 19.91}$ |
| MAS$^3$AC | $\mathbf{2515.31 \pm 175.23}$ | $14.58 \pm 24.06$ | $2502.86 \pm 179.74$ | $14.58 \pm 24.06$ | $2509.51 \pm 169.11$ | $14.58 \pm 24.06$ |

We next examine the effect of Lyapunov function inputs in the decentralized Coupled HalfCheetah benchmark. Specifically, we compare two versions of MAS³AC: (*i*) the *modified* version, where each agent's Lyapunov certificate only uses its own state and action as input; and (*ii*) the *original* version, where Lyapunov certificates also take the states and actions of neighboring agents as input. As shown in Figure 13 and Table 8, both versions of MAS³AC maintain a strong balance between reward and safety, clearly outperforming all baselines by achieving significantly fewer safety violations while still obtaining high rewards.

Comparing the two versions, we find that the modified Lyapunov version is more sample efficient: its reward curve rises faster than that of the original version, while both lead to similar final performance. This observation aligns with our theoretical ISS analysis, which requires only an agent's own state (plus at most a few scalar signals from neighbors) rather than full neighbor states and actions. Including more information in the Lyapunov input may increase learning complexity without providing proportional benefits. Importantly, these results demonstrate that MAS³AC can still perform well even with restricted Lyapunov inputs, offering a practical and theoretically justified approach to decentralized training with minimal inter-agent information exchange. We further note that in this setting, MAS³AC uses neighbor size 2 for all other components (critics, barrier certificates, policies), which provides strictly less information than the baselines (neighbor size 3), yet it still outperforms them.

## J  ADDITIONAL EXPERIMENTS AND ABLATION STUDIES

In this section, we conduct a comprehensive set of additional experiments and ablation studies to evaluate the robustness, scalability, and specific component contributions of the proposed MAS³AC algorithm.

### J.1  ROBUSTNESS AGAINST EXTERNAL DISTURBANCES

We first consider the scenario where an external disturbance $d_t$ is applied to the system. To account for this, we incorporate the disturbance into the Lyapunov stability constraints by adding the term $\kappa_i(||d_t||)$ to the stability requirement and re-running the experiments. We evaluated the influence of different disturbance magnitudes (Small, Medium, and Strong) on the Ant task.

It can be found that as the disturbance becomes stronger, the algorithm's performance inevitably deteriorates in both reward maximization and safety satisfaction. However, even under the strong disturbances, the proposed algorithm still achieves better rewards and fewer safety violations compared to commonly-used baselines under no disturbance, which demonstrates the good performance of MAS³AC.

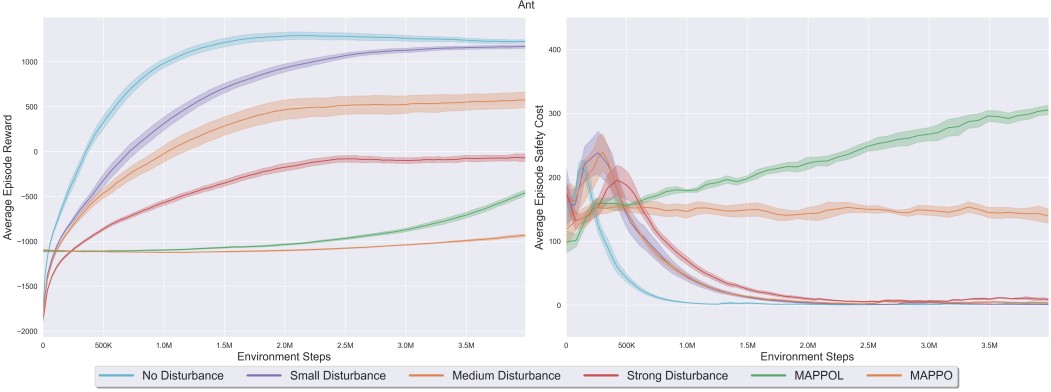

Figure 14: Performance influence of external disturbances with varying magnitudes (No, Small, Medium, Strong) on the Ant task.

Table 9: Reward and cost comparison (Disturbance in Ant)

| Condition | Ant | |
|---|---|---|
| | **Reward** | **Cost** |
| MAPPO | $-834.56 \pm 317.29$ | $249.46 \pm 87.03$ |
| MAPPOL | $-1039.23 \pm 95.85$ | $145.60 \pm 58.37$ |
| Strong Disturbance (MAS$^3$AC) | $-73.35 \pm 100.57$ | $7.86 \pm 12.35$ |
| Medium Disturbance (MAS$^3$AC) | $550.07 \pm 194.18$ | $4.02 \pm 8.27$ |
| Small Disturbance (MAS$^3$AC) | $1152.68 \pm 58.21$ | $1.69 \pm 5.90$ |
| No Disturbance (MAS$^3$AC) | $1232.87 \pm 57.45$ | $2.02 \pm 5.45$ |

## J.2 SCALING TO LARGER $N$ WITH PARTIAL OBSERVABILITY

To test scalability, we scaled the system to $N \geq 10$ agents using the Coupled HalfCheetah task, where the system is composed of 12 agents ($12 \times 1$ setting) operating under partial observability. We ran the non-cooperative version of the experiment, ensuring all baselines were also rerun on this new 12-agent setting, demonstrating the effectiveness of MA$^3$SAC when scaling to larger multi-agent systems.

Figure 15 and Table 10 illustrate the results. Despite the existence of more agents, MAS$^3$AC successfully scales, providing stable reward improvement and safety satisfaction where baselines fail.

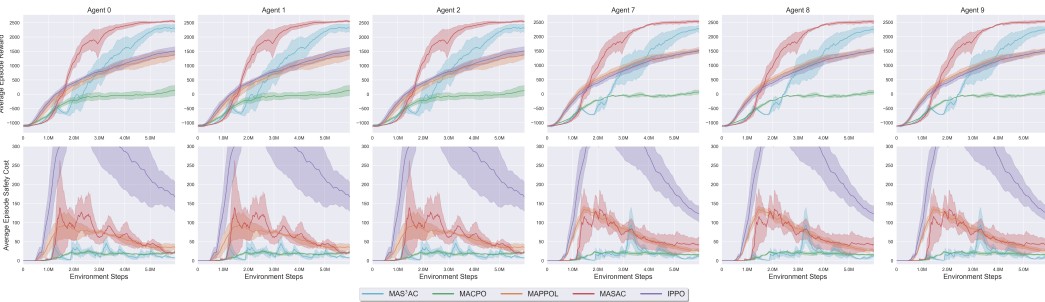

Figure 15: Results on the 12-agent non-cooperative decentralized Coupled HalfCheetah task. We report the training results of agent 0, 1, 2, 7, 8 and 9 for simplicity

Table 10: Reward and cost per agent (12-agent non-cooperative decentralized Coupled HalfCheetah task, report the training results of agent 0, 1, 2, 7, 8 and 9 for simplicity)

| Algorithm | Agent 0 | | Agent 1 | | Agent 2 | |
|---|---|---|---|---|---|---|
| | **Reward** | **Cost** | **Reward** | **Cost** | **Reward** | **Cost** |
| IPPO | $1255.74 \pm 372.60$ | $242.76 \pm 175.59$ | $1246.24 \pm 366.04$ | $242.76 \pm 175.59$ | $1255.41 \pm 374.03$ | $242.76 \pm 175.59$ |
| MACPO | $31.32 \pm 313.10$ | $18.64 \pm 22.76$ | $31.77 \pm 313.63$ | $18.64 \pm 22.76$ | $27.10 \pm 312.90$ | $18.64 \pm 22.76$ |
| MAPPOL | $1154.25 \pm 397.81$ | $46.82 \pm 32.72$ | $1152.97 \pm 398.61$ | $46.82 \pm 32.72$ | $1153.45 \pm 398.27$ | $46.82 \pm 32.72$ |
| MASAC | $\mathbf{2541.37 \pm 63.98}$ | $26.23 \pm 21.01$ | $\mathbf{2542.25 \pm 63.80}$ | $26.23 \pm 21.01$ | $\mathbf{2541.03 \pm 65.82}$ | $26.23 \pm 21.01$ |
| MAS$^3$AC | $2312.98 \pm 227.15$ | $\mathbf{10.07 \pm 22.83}$ | $2318.55 \pm 234.76$ | $\mathbf{10.07 \pm 22.83}$ | $2317.08 \pm 233.75$ | $\mathbf{10.07 \pm 22.83}$ |

| Algorithm | Agent 7 | | Agent 8 | | Agent 9 | |
|---|---|---|---|---|---|---|
| | **Reward** | **Cost** | **Reward** | **Cost** | **Reward** | **Cost** |
| IPPO | $1277.15 \pm 332.06$ | $218.90 \pm 168.98$ | $1290.05 \pm 337.91$ | $218.90 \pm 168.98$ | $1289.84 \pm 335.09$ | $218.90 \pm 168.98$ |
| MACPO | $0.75 \pm 268.65$ | $17.85 \pm 21.58$ | $-2.33 \pm 268.52$ | $17.85 \pm 21.58$ | $2.16 \pm 268.55$ | $17.85 \pm 21.58$ |
| MAPPOL | $1318.95 \pm 346.96$ | $43.45 \pm 34.27$ | $1321.36 \pm 347.10$ | $43.45 \pm 34.27$ | $1318.80 \pm 346.23$ | $43.45 \pm 34.27$ |
| MASAC | $\mathbf{2513.88 \pm 88.55}$ | $43.84 \pm 33.44$ | $\mathbf{2510.71 \pm 94.22}$ | $43.84 \pm 33.44$ | $\mathbf{2515.29 \pm 89.45}$ | $43.84 \pm 33.44$ |
| MAS$^3$AC | $2219.22 \pm 240.95$ | $\mathbf{8.63 \pm 19.38}$ | $2208.60 \pm 238.10$ | $\mathbf{8.63 \pm 19.38}$ | $2220.00 \pm 238.44$ | $\mathbf{8.63 \pm 19.38}$ |

## J.3 COMPARISON WITH THE ADDITIONAL BASELINE (MODIFIED SCAL-MAPPO-L)

We incorporated the modified Scal-MAPPO-L as an additional baseline to compare against the proposed algorithm. Figure 16 and Tables 11 and 12 show the comparison on new baseline cooperative

Ant, cooperative HalfCheetah, and non-cooperative Coupled HalfCheetah tasks. MAS³AC consistently demonstrates superior performance.

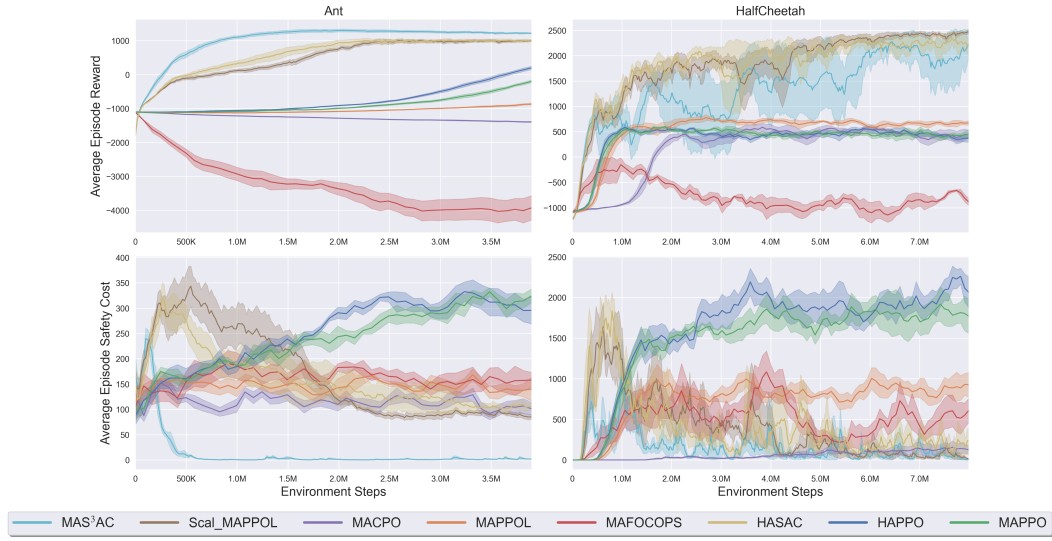

(a) Comparison in Cooperative Ant and HalfCheetah tasks.

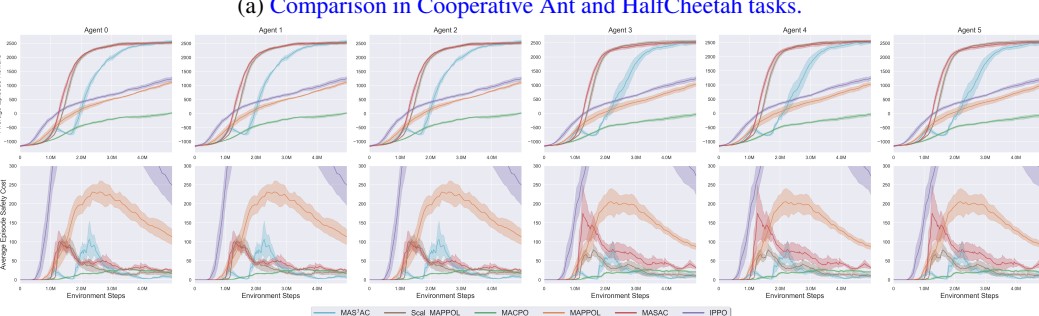

(b) Comparison in the Non-cooperative Coupled HalfCheetah task.

Figure 16: Comparison with the additional modified Scal-MAPPO-L baseline.

Table 11: Reward and cost comparison with the additional modified Scal-MAPPO-L baseline (cooperative tasks).

| Algorithm | Ant | | HalfCheetah | |
|---|---|---|---|---|
| | Reward | Cost | Reward | Cost |
| HAPPO | $41.29 \pm 215.89$ | $311.96 \pm 67.96$ | $373.64 \pm 200.58$ | $2131.21 \pm 483.13$ |
| HASAC | $993.56 \pm 87.49$ | $107.49 \pm 22.64$ | $2229.41 \pm 335.17$ | $211.07 \pm 300.58$ |
| MACPO | $-1381.66 \pm 37.86$ | $105.24 \pm 50.66$ | $443.50 \pm 265.72$ | $134.58 \pm 142.09$ |
| MAFOCOPS | $-3965.01 \pm 720.78$ | $156.47 \pm 77.84$ | $-781.55 \pm 302.59$ | $581.41 \pm 494.46$ |
| MAPPO | $-354.58 \pm 203.39$ | $322.31 \pm 64.85$ | $418.49 \pm 194.12$ | $1834.08 \pm 513.17$ |
| MAPPOL | $-903.98 \pm 67.67$ | $142.52 \pm 58.00$ | $672.25 \pm 143.48$ | $898.40 \pm 379.02$ |
| Scal_MAPPOL | $997.24 \pm 64.07$ | $86.24 \pm 12.39$ | $\mathbf{2460.87 \pm 95.22}$ | $22.44 \pm 67.28$ |
| MAS³AC | $\mathbf{1221.79 \pm 50.45}$ | $\mathbf{2.10 \pm 6.92}$ | $2177.90 \pm 714.48$ | $\mathbf{6.22 \pm 25.28}$ |

### J.4 PERFORMANCE ON NEW DECENTRALIZED ANT TASK

We introduced an additional benchmark, the **decentralized Ant** task. As depicted in Figure 17 and Table 13, MA³SAC almost achieves the highest rewards and lowest final safety violations compared to baselines for all agents, which demonstrates the efficacy of the proposed algorithm on more decentralized non-cooperative task.

Table 12: Reward and cost comparison with the additional modified Scal-MAPPO-L baseline (non-cooperative Coupled HalfCheetah task)

| Algorithm | Agent 0 Reward | Agent 0 Cost | Agent 1 Reward | Agent 1 Cost | Agent 2 Reward | Agent 2 Cost |
|---|---|---|---|---|---|---|
| IPPO | $902.29 \pm 386.41$ | $451.38 \pm 305.37$ | $904.99 \pm 389.86$ | $451.38 \pm 305.37$ | $905.91 \pm 388.62$ | $451.38 \pm 305.37$ |
| MACPO | $-140.28 \pm 276.72$ | $21.48 \pm 25.26$ | $-141.91 \pm 276.68$ | $21.48 \pm 25.26$ | $-142.75 \pm 276.65$ | $21.48 \pm 25.26$ |
| MAPPOL | $735.92 \pm 416.97$ | $174.96 \pm 143.68$ | $735.77 \pm 417.75$ | $174.96 \pm 143.68$ | $732.35 \pm 417.00$ | $174.96 \pm 143.68$ |
| MASAC | $2499.75 \pm 78.45$ | $25.25 \pm 20.76$ | $2503.90 \pm 84.95$ | $25.25 \pm 20.76$ | $2489.93 \pm 81.61$ | $25.25 \pm 20.76$ |
| Scal_MAPPOL | $2524.31 \pm 84.52$ | $18.51 \pm 17.73$ | $2516.04 \pm 88.89$ | $18.51 \pm 17.73$ | $2512.03 \pm 83.29$ | $18.51 \pm 17.73$ |
| MAS$^3$AC | $\mathbf{2531.32 \pm 136.89}$ | $\mathbf{6.35 \pm 14.12}$ | $\mathbf{2533.62 \pm 141.96}$ | $\mathbf{6.35 \pm 14.12}$ | $\mathbf{2529.90 \pm 142.89}$ | $\mathbf{6.35 \pm 14.12}$ |

| Algorithm | Agent 3 Reward | Agent 3 Cost | Agent 4 Reward | Agent 4 Cost | Agent 5 Reward | Agent 5 Cost |
|---|---|---|---|---|---|---|
| IPPO | $898.48 \pm 395.57$ | $432.92 \pm 273.22$ | $900.26 \pm 397.68$ | $432.92 \pm 273.22$ | $903.87 \pm 397.26$ | $432.92 \pm 273.22$ |
| MACPO | $-215.51 \pm 277.43$ | $21.14 \pm 27.75$ | $-216.04 \pm 277.56$ | $21.14 \pm 27.75$ | $-215.42 \pm 277.29$ | $21.14 \pm 27.75$ |
| MAPPOL | $646.70 \pm 464.02$ | $150.76 \pm 134.02$ | $648.57 \pm 465.17$ | $150.76 \pm 134.02$ | $648.17 \pm 466.71$ | $150.76 \pm 134.02$ |
| MASAC | $2479.16 \pm 102.04$ | $35.51 \pm 26.71$ | $2502.30 \pm 96.44$ | $35.51 \pm 26.71$ | $2501.68 \pm 108.95$ | $35.51 \pm 26.71$ |
| Scal_MAPPOL | $\mathbf{2564.30 \pm 89.97}$ | $13.98 \pm 19.73$ | $\mathbf{2554.35 \pm 81.43}$ | $13.98 \pm 19.73$ | $\mathbf{2577.36 \pm 87.96}$ | $13.98 \pm 19.73$ |
| MAS$^3$AC | $2486.43 \pm 172.32$ | $\mathbf{7.48 \pm 20.11}$ | $2488.44 \pm 171.39$ | $\mathbf{7.48 \pm 20.11}$ | $2489.25 \pm 175.59$ | $\mathbf{7.48 \pm 20.11}$ |

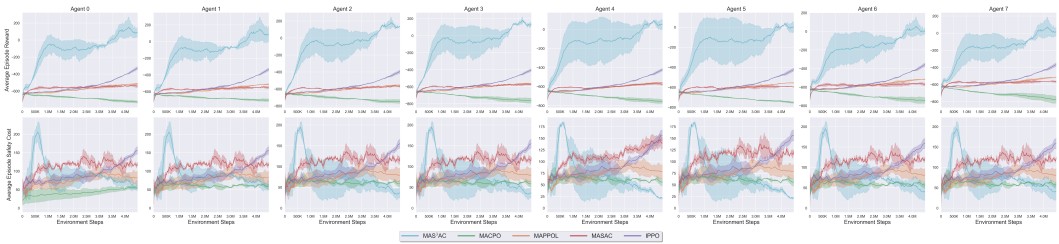

Figure 17: Performance on the Non-cooperative Decentralized Ant task.

Table 13: Reward and cost per agent (Non-cooperative Decentralized Ant task)

| Algorithm | Agent 0 Reward | Agent 0 Cost | Agent 1 Reward | Agent 1 Cost | Agent 2 Reward | Agent 2 Cost |
|---|---|---|---|---|---|---|
| IPPO | $-473.73 \pm 127.13$ | $111.25 \pm 56.83$ | $-474.56 \pm 124.97$ | $111.25 \pm 56.83$ | $-514.63 \pm 106.13$ | $109.26 \pm 56.65$ |
| MACPO | $-704.12 \pm 33.29$ | $\mathbf{47.89 \pm 27.50}$ | $-690.74 \pm 26.51$ | $\mathbf{59.28 \pm 32.32}$ | $-729.84 \pm 34.31$ | $60.08 \pm 32.47$ |
| MAPPOL | $-552.92 \pm 50.26$ | $83.31 \pm 40.09$ | $-552.76 \pm 51.65$ | $83.31 \pm 40.09$ | $-587.23 \pm 42.12$ | $80.83 \pm 38.76$ |
| MASAC | $-539.92 \pm 30.43$ | $116.42 \pm 50.14$ | $-551.35 \pm 29.67$ | $116.42 \pm 50.14$ | $-573.06 \pm 26.74$ | $114.57 \pm 49.26$ |
| MAS$^3$AC | $\mathbf{106.95 \pm 147.84}$ | $65.33 \pm 33.32$ | $\mathbf{102.38 \pm 164.69}$ | $65.33 \pm 33.32$ | $\mathbf{143.49 \pm 130.09}$ | $\mathbf{40.42 \pm 26.09}$ |

| Algorithm | Agent 3 Reward | Agent 3 Cost | Agent 4 Reward | Agent 4 Cost | Agent 5 Reward | Agent 5 Cost |
|---|---|---|---|---|---|---|
| IPPO | $-519.11 \pm 102.94$ | $109.26 \pm 56.65$ | $-531.28 \pm 97.42$ | $110.10 \pm 57.07$ | $-529.98 \pm 95.81$ | $110.10 \pm 57.07$ |
| MACPO | $-742.70 \pm 42.06$ | $60.08 \pm 32.47$ | $-733.88 \pm 35.31$ | $59.72 \pm 34.65$ | $-730.70 \pm 29.61$ | $59.72 \pm 34.65$ |
| MAPPOL | $-590.74 \pm 41.15$ | $80.83 \pm 38.76$ | $-590.14 \pm 41.02$ | $80.37 \pm 37.84$ | $-587.67 \pm 39.76$ | $80.37 \pm 37.84$ |
| MASAC | $-575.84 \pm 26.83$ | $114.57 \pm 49.26$ | $-573.30 \pm 33.81$ | $139.44 \pm 57.73$ | $-593.74 \pm 23.85$ | $115.38 \pm 50.61$ |
| MAS$^3$AC | $\mathbf{139.48 \pm 128.53}$ | $\mathbf{40.42 \pm 26.09}$ | $\mathbf{56.88 \pm 119.08}$ | $\mathbf{28.69 \pm 22.59}$ | $\mathbf{4.71 \pm 115.05}$ | $\mathbf{28.69 \pm 22.59}$ |

| Algorithm | Agent 6 Reward | Agent 6 Cost | Agent 7 Reward | Agent 7 Cost |
|---|---|---|---|---|
| IPPO | $-488.03 \pm 116.87$ | $112.29 \pm 56.79$ | $-493.59 \pm 113.75$ | $112.29 \pm 56.79$ |
| MACPO | $-721.77 \pm 39.74$ | $58.83 \pm 34.20$ | $-741.17 \pm 51.10$ | $58.83 \pm 34.20$ |
| MAPPOL | $-554.36 \pm 48.57$ | $83.16 \pm 38.62$ | $-558.00 \pm 50.22$ | $83.16 \pm 38.62$ |
| MASAC | $-565.41 \pm 26.94$ | $116.93 \pm 50.39$ | $-564.21 \pm 25.39$ | $116.93 \pm 50.39$ |
| MAS$^3$AC | $\mathbf{16.19 \pm 129.88}$ | $\mathbf{46.70 \pm 32.85}$ | $\mathbf{30.22 \pm 137.99}$ | $\mathbf{46.70 \pm 32.85}$ |

## J.5 ABLATION: MAX-OVER-TIME VS. SUM-OVER-TIME CERTIFICATES

We replaced the safety constraints from the sum-over-time neural barrier-based certificates to the max-over-time neural reachability-based certificates, and compared the performance. Figure 18 and

Tables 14 and 15 show the results. Based on the figures and statistics, it can be observed that for safety, using barrier certificates usually achieves better safety satisfaction than the counterpart with reachability-based certificates. The author think the possible reason is the constraint to apply: barrier certificates enforce safety in an exponential manner (the barrier certificate value, if negative, is explicitly required to increase at an exponential rate to achieve safety), enabling stronger and faster safety enforcement, while the constraints using barrier certificates only require the value to be non-positive and updated via standard gradient descent; the additional exponential mechanism for rapid safety enforcement is missing.

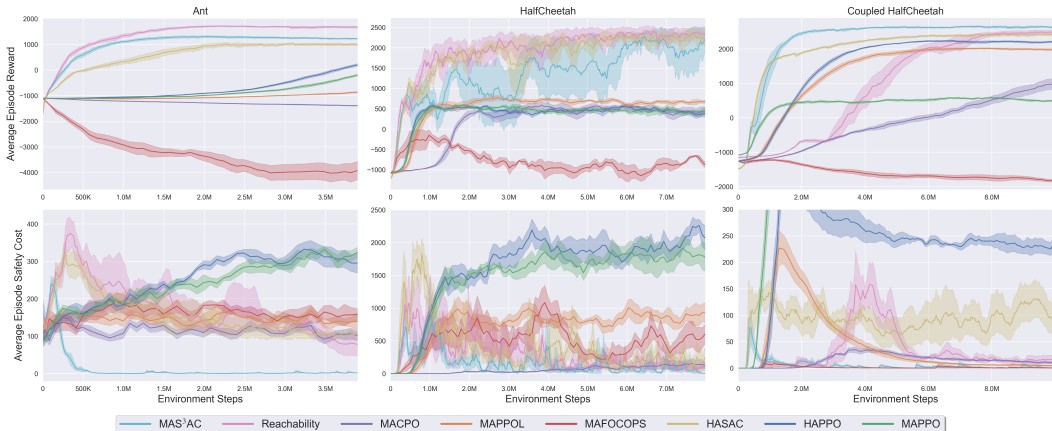

(a) Comparisons between different certificates on cooperative tasks.

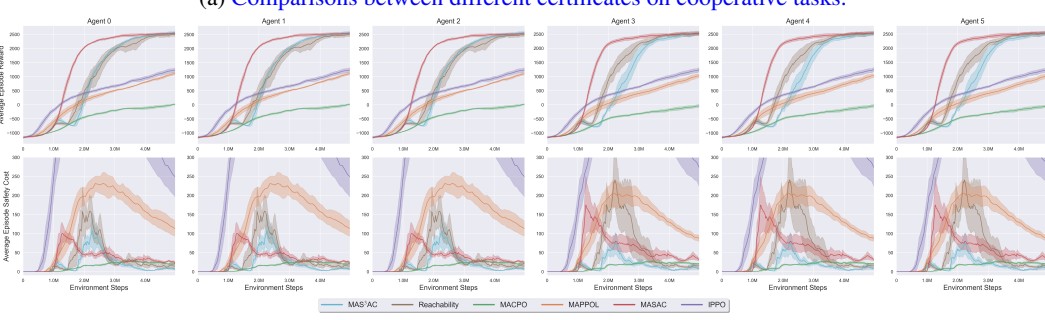

(b) Comparisons between different certificates on non-cooperative Coupled HalfCheetah task

Figure 18: Ablation study comparing the sum-over-time barrier-based certificates against the max-over-time reachability-based certificates.

Table 14: Reward and cost comparisons between different certificates on cooperative tasks.

| Algorithm | Ant | | HalfCheetah | | Coupled HalfCheetah | |
|---|---|---|---|---|---|---|
| | Reward | Cost | Reward | Cost | Reward | Cost |
| HAPPO | $41.29 \pm 215.89$ | $311.96 \pm 67.96$ | $373.64 \pm 200.58$ | $2131.21 \pm 483.13$ | $2197.22 \pm 112.77$ | $230.28 \pm 67.11$ |
| HASAC | $993.56 \pm 87.49$ | $107.49 \pm 22.64$ | $\mathbf{2229.41 \pm 335.17}$ | $211.07 \pm 300.58$ | $2410.71 \pm 106.28$ | $84.77 \pm 65.91$ |
| MACPO | $-1381.66 \pm 37.86$ | $105.24 \pm 50.66$ | $443.50 \pm 265.72$ | $134.58 \pm 142.09$ | $1104.05 \pm 356.15$ | $9.32 \pm 8.53$ |
| MAFOCOPS | $-3965.01 \pm 720.78$ | $156.47 \pm 77.84$ | $-781.55 \pm 302.59$ | $581.41 \pm 494.46$ | $-1865.72 \pm 117.10$ | $0.14 \pm 0.97$ |
| MAPPO | $-354.58 \pm 203.39$ | $322.31 \pm 64.85$ | $418.49 \pm 194.12$ | $1834.08 \pm 513.17$ | $480.86 \pm 195.30$ | $1764.08 \pm 431.53$ |
| MAPPOL | $-903.98 \pm 67.67$ | $142.52 \pm 58.00$ | $672.25 \pm 143.48$ | $898.40 \pm 379.02$ | $1968.86 \pm 118.37$ | $3.67 \pm 3.39$ |
| Reachability | $\mathbf{1671.62 \pm 77.84}$ | $78.82 \pm 63.39$ | $1939.28 \pm 138.34$ | $272.27 \pm 149.15$ | $2513.78 \pm 155.96$ | $13.15 \pm 17.28$ |
| MAS$^3$AC | $1221.79 \pm 50.45$ | $\mathbf{2.10 \pm 6.92}$ | $2177.90 \pm 714.48$ | $\mathbf{6.22 \pm 25.28}$ | $\mathbf{2596.56 \pm 105.95}$ | $\mathbf{0.03 \pm 0.17}$ |

## J.6 ABLATION: IMPACT OF INPUT-TO-STATE STABILITY (ISS) PART

To analyze how ISS helps to improve performance, we conducted an ablation study by completely removing the ISS part. Figure 19 and Table 16 show that the efficacy of the ISS part differs across tasks. In Ant, without ISS, the improvement in reward is slower than that of the complete version

Table 15: Reward and cost comparisons between different certificates on the non-cooperative Coupled HalfCheetah task.

| | Agent 0 | | Agent 1 | | Agent 2 | |
|---|---|---|---|---|---|---|
| **Algorithm** | **Reward** | **Cost** | **Reward** | **Cost** | **Reward** | **Cost** |
| IPPO | $902.29 \pm 386.41$ | $451.38 \pm 305.37$ | $904.99 \pm 389.86$ | $451.38 \pm 305.37$ | $905.91 \pm 388.62$ | $451.38 \pm 305.37$ |
| MACPO | $-140.28 \pm 276.72$ | $21.48 \pm 25.26$ | $-141.91 \pm 276.68$ | $21.48 \pm 25.26$ | $-142.75 \pm 276.65$ | $21.48 \pm 25.26$ |
| MAPPOL | $735.92 \pm 416.97$ | $174.96 \pm 143.68$ | $735.77 \pm 417.75$ | $174.96 \pm 143.68$ | $732.35 \pm 417.00$ | $174.96 \pm 143.68$ |
| MASAC | $2499.75 \pm 78.45$ | $25.25 \pm 20.76$ | $2503.90 \pm 84.95$ | $25.25 \pm 20.76$ | $2489.93 \pm 81.61$ | $25.25 \pm 20.76$ |
| Reachability | $2479.88 \pm 166.99$ | $14.22 \pm 26.00$ | $2472.27 \pm 161.99$ | $14.22 \pm 26.00$ | $2471.29 \pm 161.62$ | $14.22 \pm 26.00$ |
| MAS$^3$AC | $\mathbf{2531.32 \pm 136.89}$ | $\mathbf{6.35 \pm 14.12}$ | $\mathbf{2533.62 \pm 141.96}$ | $\mathbf{6.35 \pm 14.12}$ | $\mathbf{2529.90 \pm 142.89}$ | $\mathbf{6.35 \pm 14.12}$ |

| | Agent 3 | | Agent 4 | | Agent 5 | |
|---|---|---|---|---|---|---|
| **Algorithm** | **Reward** | **Cost** | **Reward** | **Cost** | **Reward** | **Cost** |
| IPPO | $898.48 \pm 395.57$ | $432.92 \pm 273.22$ | $900.26 \pm 397.68$ | $432.92 \pm 273.22$ | $903.87 \pm 397.26$ | $432.92 \pm 273.22$ |
| MACPO | $-215.51 \pm 277.43$ | $21.14 \pm 27.75$ | $-216.04 \pm 277.56$ | $21.14 \pm 27.75$ | $-215.42 \pm 277.29$ | $21.14 \pm 27.75$ |
| MAPPOL | $646.70 \pm 464.02$ | $150.76 \pm 134.02$ | $648.57 \pm 465.17$ | $150.76 \pm 134.02$ | $648.17 \pm 466.71$ | $150.76 \pm 134.02$ |
| MASAC | $2479.16 \pm 102.04$ | $35.51 \pm 26.71$ | $2502.30 \pm 96.44$ | $35.51 \pm 26.71$ | $2501.68 \pm 108.95$ | $35.51 \pm 26.71$ |
| Reachability | $\mathbf{2506.60 \pm 194.16}$ | $14.05 \pm 23.27$ | $\mathbf{2506.42 \pm 190.56}$ | $14.05 \pm 23.27$ | $\mathbf{2504.92 \pm 187.60}$ | $14.05 \pm 23.27$ |
| MAS$^3$AC | $2486.43 \pm 172.32$ | $\mathbf{7.48 \pm 20.11}$ | $2488.44 \pm 171.39$ | $\mathbf{7.48 \pm 20.11}$ | $2489.25 \pm 175.59$ | $\mathbf{7.48 \pm 20.11}$ |

where ISS is used, especially at the beginning of the training process. Moreover, regarding the final reward obtained, with ISS the final reward is higher than that achieved by the counterpart without ISS constraints. In the other two tasks, when ISS is removed, the reward cannot meaningfully increase and therefore remains at a low level. The reason is that with the barrier certificate for safety but without the ISS component to encourage convergence toward the desired state, the controlled agent becomes overly conservative during exploration. It is prevented from achieving high rewards by approaching the desired goal due to the strong repulsion imposed by the exponential safety constraint. In such settings, it is essential to incorporate an additional stimulus to encourage effective exploration while maintaining state-wise safety, and this is exactly the role of the ISS part.

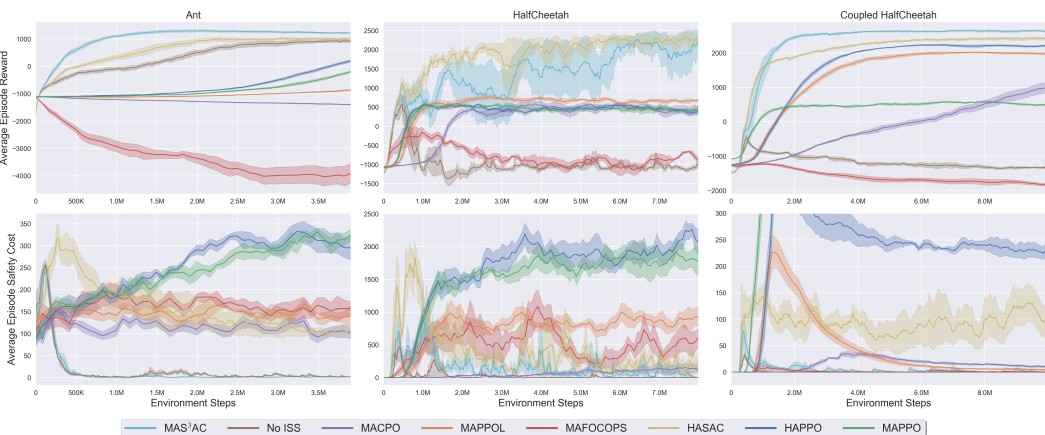

Figure 19: Ablation study on the ISS part.

## J.7 ABLATION: NO REWARD SIGNAL

Similarly, we considered completely removing the reward signal and only using the ISS stability term to guide the training. Based on these results, the reward signal also plays an important role in training: (1) It helps achieve a more efficient increase in reward, especially at the beginning of training, as demonstrated in Ant. Reward signals provide useful direction for policy improvement. (2) In some tasks such as HalfCheetah, without the reward signal, the training may totally fail to achieve an increase in reward. The likely reason is that, on some tasks, without reward to provide a stable improvement direction, the applied constraint terms may compound and be large in size, which leads to numerical issues in the training and thus the learning fails.

Table 16: Reward and cost comparisons for ISS ablation study

| Algorithm | Ant | | HalfCheetah | | Coupled | |
|---|---|---|---|---|---|---|
| | Reward | Cost | Reward | Cost | Reward | Cost |
| HAPPO | $41.29 \pm 215.89$ | $311.96 \pm 67.96$ | $373.64 \pm 200.58$ | $2131.21 \pm 483.13$ | $2197.22 \pm 112.77$ | $230.28 \pm 67.11$ |
| HASAC | $993.56 \pm 87.49$ | $107.49 \pm 22.64$ | $\mathbf{2229.41 \pm 335.17}$ | $211.07 \pm 300.58$ | $2410.71 \pm 106.28$ | $84.77 \pm 65.91$ |
| MACPO | $-1381.66 \pm 37.86$ | $105.24 \pm 50.66$ | $443.50 \pm 265.72$ | $134.58 \pm 142.09$ | $1104.05 \pm 356.15$ | $9.32 \pm 8.53$ |
| MAFOCOPS | $-3965.01 \pm 720.78$ | $156.47 \pm 77.84$ | $-781.55 \pm 302.59$ | $581.41 \pm 494.46$ | $-1865.72 \pm 117.10$ | $0.14 \pm 0.97$ |
| MAPPO | $-354.58 \pm 203.39$ | $322.31 \pm 64.85$ | $418.49 \pm 194.12$ | $1834.08 \pm 513.17$ | $480.86 \pm 195.30$ | $1764.08 \pm 431.53$ |
| MAPPOL | $-903.98 \pm 67.67$ | $142.52 \pm 58.00$ | $672.25 \pm 143.48$ | $898.40 \pm 379.02$ | $1968.86 \pm 118.37$ | $3.67 \pm 3.39$ |
| No-ISS | $921.66 \pm 104.09$ | $\mathbf{1.53 \pm 1.53}$ | $-1043.67 \pm 202.75$ | $\mathbf{0.00 \pm 0.00}$ | $-1317.85 \pm 163.06$ | $\mathbf{0.00 \pm 0.00}$ |
| MAS$^3$AC | $\mathbf{1221.79 \pm 50.45}$ | $2.10 \pm 6.92$ | $2177.90 \pm 714.48$ | $6.22 \pm 25.28$ | $\mathbf{2596.56 \pm 105.95}$ | $0.03 \pm 0.17$ |

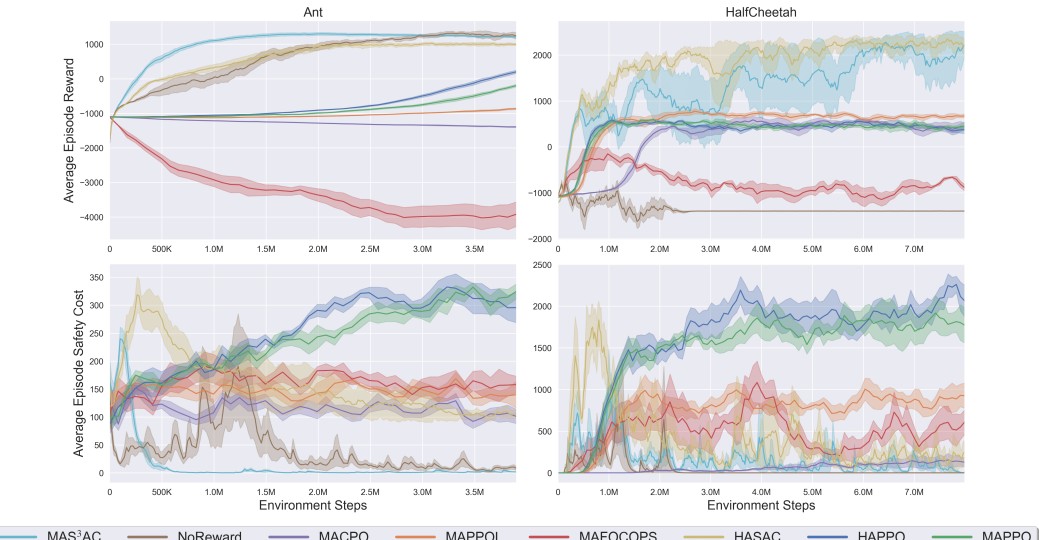

Figure 20: Ablation study on the reward signal.

Table 17: Reward and cost comparisons for ablation study on the reward signal.

| Algorithm | Ant | | HalfCheetah | |
|---|---|---|---|---|
| | Reward | Cost | Reward | Cost |
| HAPPO | $41.29 \pm 215.89$ | $311.96 \pm 67.96$ | $373.64 \pm 200.58$ | $2131.21 \pm 483.13$ |
| HASAC | $993.56 \pm 87.49$ | $107.49 \pm 22.64$ | $\mathbf{2229.41 \pm 335.17}$ | $211.07 \pm 300.58$ |
| MACPO | $-1381.66 \pm 37.86$ | $105.24 \pm 50.66$ | $443.50 \pm 265.72$ | $134.58 \pm 142.09$ |
| MAFOCOPS | $-3965.01 \pm 720.78$ | $156.47 \pm 77.84$ | $-781.55 \pm 302.59$ | $581.41 \pm 494.46$ |
| MAPPO | $-354.58 \pm 203.39$ | $322.31 \pm 64.85$ | $418.49 \pm 194.12$ | $1834.08 \pm 513.17$ |
| MAPPOL | $-903.98 \pm 67.67$ | $142.52 \pm 58.00$ | $672.25 \pm 143.48$ | $898.40 \pm 379.02$ |
| No-Reward | $\mathbf{1258.19 \pm 167.19}$ | $8.78 \pm 10.02$ | $-1396.78 \pm 5.49$ | $\mathbf{0.00 \pm 0.00}$ |
| MAS$^3$AC | $1221.79 \pm 50.45$ | $\mathbf{2.10 \pm 6.92}$ | $2177.90 \pm 714.48$ | $6.22 \pm 25.28$ |

## J.8 ABLATION: NECESSITY OF THE SQUARED TERM

We performed an experiment where the additional squared term was removed to observe the performance change. Based on the figures and statistics, it can be found that with the extra squared terms, the reward obtained will not experience noticeable deteriorations that are difficult to recover (for HalfCheetah, the reward curve finally falls down without efficient recovery when no extra squared terms are used. This squared term design also applies to stability constraints beyond safety constraints, and therefore, the reward curve also shows its efficacy). Regarding safety, the version with extra squared terms generally achieves better safety satisfaction during the learning process on both tasks. This design is specifically proposed as an effort to minimize the safety violations, even during the learning time.

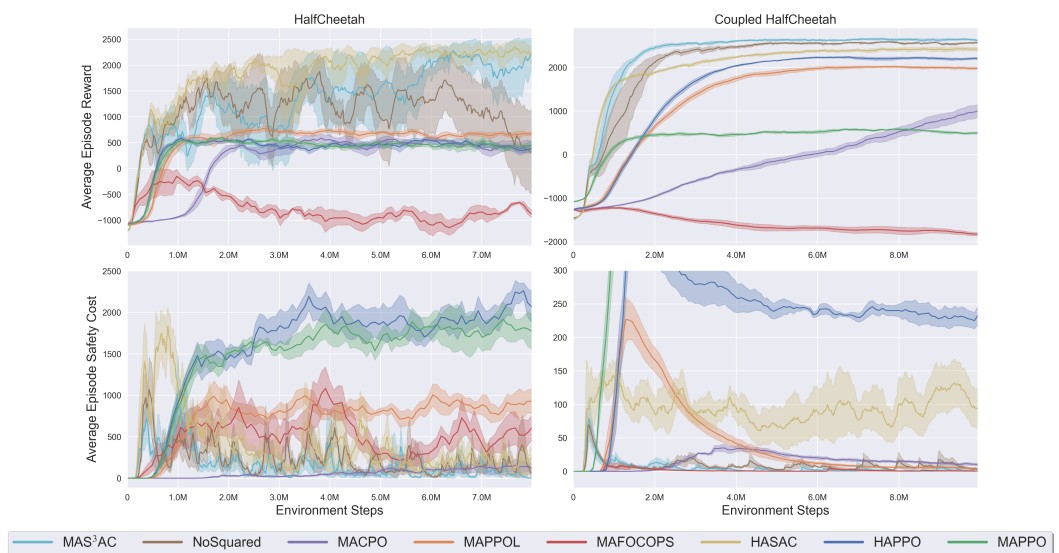

Figure 21: Ablation study on the additional squared term.

Table 18: Reward and cost comparisons for the ablation study on the additional squared term.

| | HalfCheetah | | Coupled HalfCheetah | |
|---|---|---|---|---|
| Algorithm | Reward | Cost | Reward | Cost |
| HAPPO | $373.64 \pm 200.58$ | $2131.21 \pm 483.13$ | $2197.22 \pm 112.77$ | $230.28 \pm 67.11$ |
| HASAC | $\mathbf{2229.41 \pm 335.17}$ | $211.07 \pm 300.58$ | $2410.71 \pm 106.28$ | $84.77 \pm 65.91$ |
| MACPO | $443.50 \pm 265.72$ | $134.58 \pm 142.09$ | $1104.05 \pm 356.15$ | $9.32 \pm 8.53$ |
| MAFOCOPS | $-781.55 \pm 302.59$ | $581.41 \pm 494.46$ | $-1865.72 \pm 117.10$ | $0.14 \pm 0.97$ |
| MAPPO | $418.49 \pm 194.12$ | $1834.08 \pm 513.17$ | $480.86 \pm 195.30$ | $1764.08 \pm 431.53$ |
| MAPPOL | $672.25 \pm 143.48$ | $898.40 \pm 379.02$ | $1968.86 \pm 118.37$ | $3.67 \pm 3.39$ |
| No-Squared | $342.48 \pm 1599.78$ | $161.24 \pm 447.73$ | $2564.67 \pm 72.10$ | $2.58 \pm 6.56$ |
| MAS$^3$AC | $2177.90 \pm 714.48$ | $\mathbf{6.22 \pm 25.28}$ | $\mathbf{2596.56 \pm 105.95}$ | $\mathbf{0.03 \pm 0.17}$ |

## J.9 LAGRANGIAN MULTIPLIERS EVOLUTION

Finally, we draw the initial and end values of the Lagrangian multipliers for the cooperative ant task. Figure 22 illustrates the reward and cost for the task, and Figure 23 illustrates the adaptation of the Lagrangian multipliers corresponding to the safety and stability constraints over the course of training.

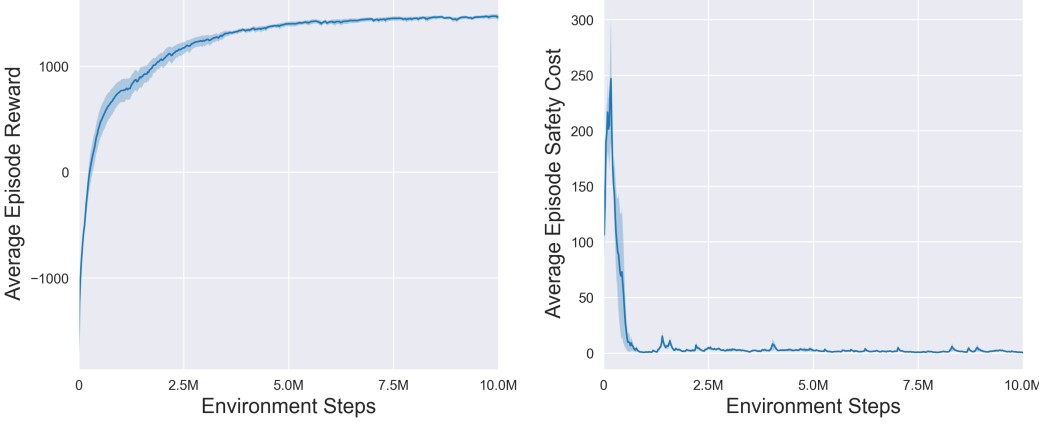

Figure 22: Reward and cost of the proposed MAS$^3$AC over the learning.

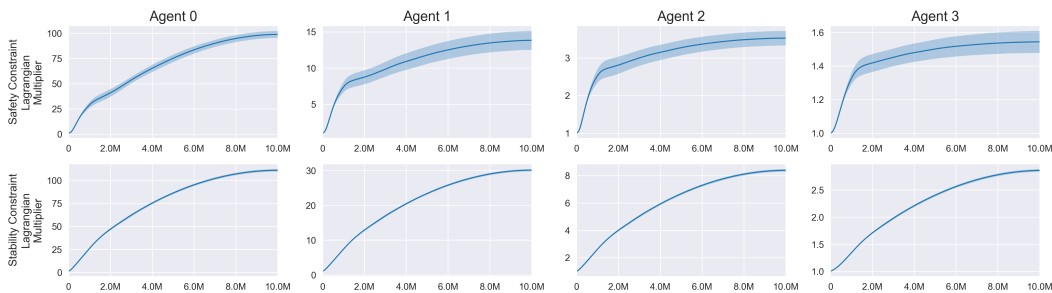

Figure 23: Evolution of the Lagrangian multipliers (corresponding to safety and stability constraints) of the proposed MAS$^3$AC over the learning.

### J.10 PLOTS OF DIFFERENCES BETWEEN THE DESIRED STATE

In this subsection, we provide some plots on the differences between the controlled system's current state and the desired state to investigate whether the proposed MAS$^3$AC effectively helps to minimize such differences and encourages convergence towards the pre-defined desired goal. Figure 24 clearly shows that the proposed algorithm can effectively train the policy to drive the controlled multi-agent system towards the desired goal without noticeable divergence, and Figure 25 shows that MAS$^3$AC consistently achieves better minimization of the difference compared to standard and safe MARL baselines.

## K THE USE OF LARGE LANGUAGE MODELS (LLMs)

We used a large language model (ChatGPT) solely for polishing the writing (e.g., grammar and phrasing). All research ideas, theoretical developments, experimental design, implementation, analysis, and substantive writing were entirely conducted by the authors.

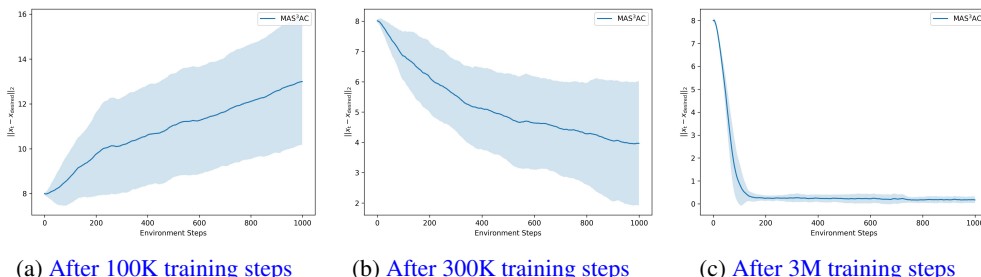

(a) After 100K training steps     (b) After 300K training steps     (c) After 3M training steps

Figure 24: Differences between the desired state under the execution of the joint policy obtained via MAS$^3$AC after different training steps.

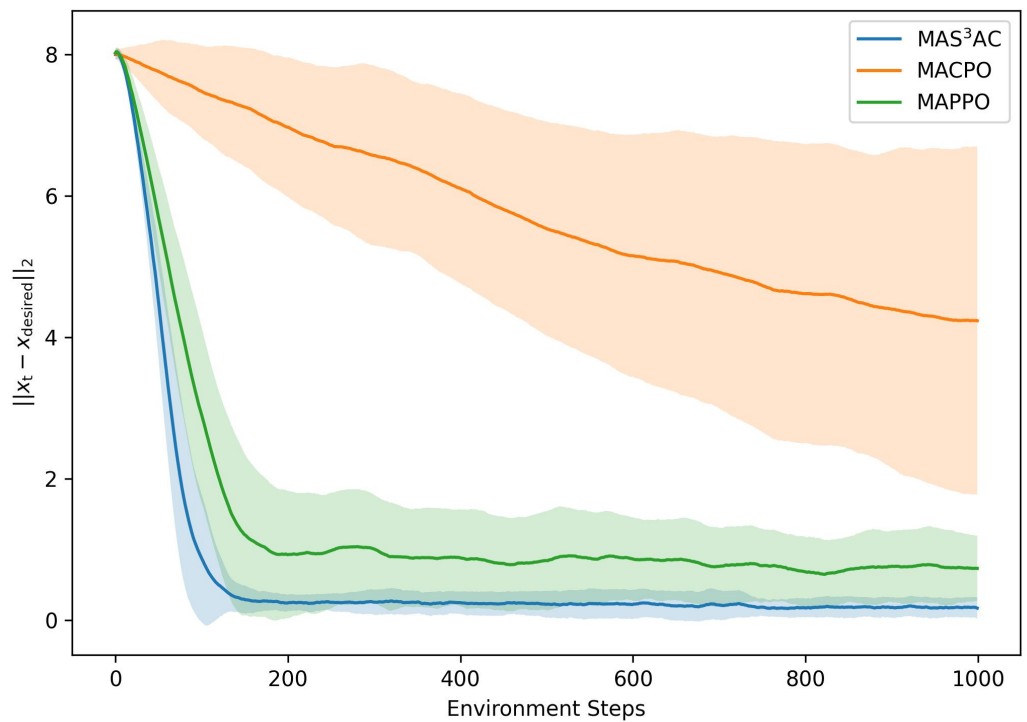

Figure 25: Comparisons among the differences between the desired state under the execution of joint policies obtained via MAS$^3$AC, MACPO, and MAPPO after 3M training steps.

