# OpenReview forum: "MAS$^3$AC: A Learning Framework for General Multi-Agent Safe and Stable Control with State-Wise Guarantees"
_ICLR.cc/2026/Conference — Submitted to ICLR 2026_

### Official Review · Reviewer_yTHj · 2025-10-27

**Soundness:** 1
**Presentation:** 3
**Contribution:** 2
**Rating:** 4
**Confidence:** 4

**Summary:**

This paper considers the problem of ensuring both state-wise safety and stability in multi-agent reinforcement learning. The paper proposes the multi-agent safe and stable soft actor-critic (MAS$^3$AC) method, which uses neural barrier functions for safety, and the concept of input-to-state stability to guarantee stability. Theoretical analyses are provided to show the convergence and feasibility of the proposed method. Empirical results on the safe multi-agent MuJoCo and an electrical bidding task show that MAS$^3$AC achieves a better balance between reward maximization and constraint satisfaction than baselines.

**Strengths:**

1. The paper considers an important problem.

2. The theoretical analysis is adequate.

3. The overall writing of the paper is good.

4. Both fully observable and partially observable environments are considered.

**Weaknesses:**

1. The paper includes many important results in the appendix (like Figures 8 and 11), and the main pages repeatedly refer to the appendix, making the paper difficult to read. Generally, the main pages should be **self-contained**, with only additional details provided in the appendix.

1. The baselines for safe MARL are not appropriate enough. All the safe MARL baselines in the paper consider the **CMDP** setting, while the paper considers **state-wise** safety. It would be better if other works addressing state-wise safety are considered, e.g., the works discussed in the related work section.

1. Some of the paper's claims are not fully justified. Please see the Questions section (Q1 - Q4).

1. Some important ablation studies are missing. Please see the Questions section (Q8 - Q10).

**Questions:**

1. The paper claims that the stability is **guaranteed** using the proposed method. How can this be true with learned value functions?

1. Can you prove that the safety is guaranteed with Definition 3?

1. The stability is defined as convergence in **states** in the main pages. How is this realized in the experiments? It seems that the experiments define cost functions as $|d_t - d^*|$.

1. Can you show some empirical analysis about the stability of the learned policies? For example, plot $\\|x_t - x_\mathrm{desired}\\|$.

1. Does the stochasticity in the problem formulation only come from the policy? This needs to be clearly stated.

1. What kind of convergence do you consider with a stochastic ${x}_t$?

1. Some previous works are criticised for "focusing on tasks with a joint team reward/cost". Why is this a problem, as it seems easy to adapt these works to the proposed framework (for example, MAPPO has already been adapted by the authors)?

1. In the Hamilton-Jacobi reachability community, it is popular to use a max-over-time safety signal as a neural barrier function. In contrast, this paper proposes to use a sum-over-time barrier signal as the neural barrier function. What is the motivation for doing this, and does this achieve better results? If so, it would be great to see some empirical results.

1. The efficacy of the ISS stability part is unclear. Can you consider completely removing this part and comparing the results? Also, maybe consider completely removing the reward signal and only using the ISS stability term to guide the training.

1. The paper considers an extra squared term to more strongly penalize safety violations. This needs an ablation study to demonstrate the improvement.

1. It seems that the learning rates are missing in Table 1. Can you provide this important hyperparameter? I am also curious about the initial and end values of the Lagrange multipliers (curves can be better).

1. Why can MAS$^3$AC achieve higher results than algorithms that do not consider safety?

---

> ### Author Response · Authors · 2025-11-25
> **Authors' response to Reviewer yTHj (1)**
>
> ### Weakness 1
>
> >  The paper includes many important results in the appendix (like Figures 8 and 11), and the main pages repeatedly refer to the appendix, making the paper difficult to read. Generally, the main pages should be self-contained, with only additional details provided in the appendix.
>
> **A1:** Thanks for the question. Due to the 9-page limit in the initial submission, the authors moved some experimental results and related discussions to the appendix. Since the page limit is now relaxed to 10 pages, we **have moved some key experimental results from the appendix to the main paper in the revised PDF** to improve readability. Thank you sincerely for the suggestion.
>
> ---
> ### Weakness 2
> >The baselines for safe MARL are not appropriate enough. All the safe MARL baselines in the paper consider the CMDP setting, while the paper considers state-wise safety. It would be better if other works addressing state-wise safety are considered, e.g., the works discussed in the related work section.
>
> **A2:** Thanks sincerely for the suggestion. Over the past few days, we **added a new baseline that enforces state-wise safety for more comprehensive comparisons**. This baseline is modified from [1], which is discussed in the related work section. To keep the overall response concise, we kindly ask the reviewer to refer to our detailed explanation in the response to Weakness 3 raised by Reviewer mUAV. Thank you for your understanding.
>
> ---
> ### Weakness 3
> >  Some of the paper's claims are not fully justified. Please see the Questions section (Q1 - Q4).
>
> **A3:** Thanks for the question. Please refer to the authors’ responses to Questions Q1–Q4 below.
>
> ---
> ### Weakness 4
> >Some important ablation studies are missing. Please see the Questions section (Q8 - Q10).
>
> **A4:** Thanks for the question. The authors kindly refer the reviewer to the responses to Questions Q8–Q10 below.
>
> ---
> ### Question 1
>
> >The paper claims that the stability is guaranteed using the proposed method. How can this be true with learned value functions?
>
> **A5:** Thanks for the question. The reviewer correctly mentions that if the learned critic networks (including action-value networks and certificates parameterized by corresponding parameters) do not precisely represent the true actions-value functions $Q^i_{\boldsymbol{\pi}}$ and certificates like $H^i_{\boldsymbol{\pi}}$ for corresponding controller, then **there can be errors when update the policy** using the loss function based on the Lagrangian function. To mitigate this issue, the multi-timescale approach is applied to make sure all critics (value networks) are learned at a timescale quicker than the update of any individual policy, such that conceptually when update one policy, corresponding critic networks are adequately updated and can approximate real critics well. Beyond setting different learning rates, in implementation, **several additional measures can be taken**, like bounding the norms of reward, barrier, and cost signals through careful design, or standardizing the reward, barrier, and cost signals using mean and variance, for easier learning of critic functions.

---

> ### Author Response · Authors · 2025-11-25
> **Authors' response to Reviewer yTHj (2)**
>
> ### Question 2
> >Can you prove that the safety is guaranteed with Definition 3?
>
> **A6:** Thanks for the question. The authors want to clarify that the safety is maintained via the constraint (2b) in the submission **when a new policy needs to be found**, namely, (2b) guarantees the safety during the update. For definition 3, the first two points are used to identify the safe set and unsafe set under the current joint policy $\boldsymbol{\pi}$ for agent $i$, and the third point is to state that the barrier certificate $H_{\boldsymbol{\pi}}^i$ can be used to certify the safety of agent $i$ **under $\boldsymbol{\pi}$** if the initial state is within $\boldsymbol{\mathcal{X}}\_{\text{safe}}^{\boldsymbol{\pi},i}$, and such guarantee is not for the update of policy (Proposition 1 is developed to show how the safe set changes when the policy is updated). To show how the safety is guaranteed under current $\boldsymbol{\pi}$, consider any $i \in \mathcal{N}$ and let current $\boldsymbol{x}\_t$ be a safe set under the current policy, then,  $H\_{\boldsymbol{\pi}}^i(\boldsymbol{x}\_t) =  0$ according to Definition 3 and the barrier signal design (cannot be positive) in Lemma 1. Since $H\_{\boldsymbol{\pi}}^i(\boldsymbol{x}\_t)=\mathbb{E}\_{\tau \thicksim \boldsymbol{\pi}}\big[\sum_{k=0}^{\infty}\gamma^{k}b^i(\boldsymbol{x}\_{t+k})\big]$ is a critic function, we have $H\_{\boldsymbol{\pi}}^i(\boldsymbol{x}\_t) = b^i(\boldsymbol{x}\_t) + \gamma\mathbb{E}\_{\boldsymbol{x}\_{t+1}}[ H\_{\boldsymbol{\pi}}^i(\boldsymbol{x}\_{t+1})] = \gamma\mathbb{E}\_{\boldsymbol{x}\_{t+1}}[ H\_{\boldsymbol{\pi}}^i(\boldsymbol{x}\_{t+1})] = 0$ given $b^i(\boldsymbol{x}\_t) = 0$ according to the definition of safe state and barrier signal. Given all barrier certificates are non-positive, we know with probability 1, $H\_{\boldsymbol{\pi}}^i(\boldsymbol{x}\_{t+1}) = 0$, and thus, the next state $\boldsymbol{x}\_{t+1}$ under the current joint policy is also safe based on Definition 2. The same analysis applies for all following timesteps and for all $i$, which shows the guarantee of safety if starting from a safe state under $\boldsymbol{\pi}$.
>
> ---
> ### Question 3
> >  The stability is defined as convergence in states in the main pages. How is this realized in the experiments? It seems that the experiments define cost functions as $|d_t - d^\ast|$.
>
> **A7:** Thanks for the question. In the cooperative HalfCheetah task, $d_t$ denotes the distance between the HalfCheetah’s torso and the movable wall along the x-axis. In the experiment, the wall’s x-position is denoted $w_t$, and the HalfCheetah’s torso x-position is $x_t$, which is part of the system state. Thus, the distance defined as $d_t:=w_t-x_t$ is a direct function of the controlled system's state.
>
> Requiring $d_t$ to converge to and maintain a desired target value $d^\ast$ serves as the **stability requirement** (analogous to standard tracking tasks where convergence is essential). The cost signal used in training is defined as $|d_t-d^\ast|$ (scaled by a factor 0.1 in experiments). This cost is collected during interaction and used to train the corresponding critic $V^i$ for each agent $i$.
>
> When constraint (5c) enforces the **monotonic decrease of the critic $V^i$** along time, this also forces the sequence $c_t=0.1\times|d_t-d^\ast|$ to decrease. Consequently, the system is driven such that $d_t\rightarrow d^\ast$, and the convergence is realized as $d_t$ is a function of state. For non-cooperative tasks, each agent has its own state-dependent cost signal, but the same logic applies.
>
> ---
> ### Question 4
> > 4. Can you show some empirical analysis about the stability of the learned policies? For example, plot $||x_t - x_{\text{desired}}||$.
>
> **A8:** Thanks for the question. We provide plots of $||x_t-x_{\text{desired}}||$ in Section J.10 of the revised PDF submission.
>
> First, we compare the joint controllers learned by MAS$^3$AC at different training stages. These plots clearly show that as training progresses, MAS$^3$AC increasingly improves the controller’s ability to **minimize** $||x_t-x_{\text{desired}}||$ along the trajectory.
>
> Second, we compare MAS$^3$AC with MACPO and MAPPO. The results show that MAS$^3$AC consistently maintains the **smallest deviation** $||x_t-x_{\text{desired}}||$. Although MAPPO can also reduce the tracking error (since its reward encourages a similar form of convergence), its deviation remains **noticeably larger** than that of MAS$^3$AC, and MAPPO does not reduce safety violations efficiently.

---

> ### Author Response · Authors · 2025-11-25
> **Authors' response to Reviewer yTHj (3)**
>
> ### Question 5
> > Does the stochasticity in the problem formulation only come from the policy? This needs to be clearly stated.
>
> **A9:** Thanks for the question. In the problem formulation, the stochasticity comes from the policy and several other common sources, including: (1) sampling data from the replay buffer during training, and (2) randomness introduced by packages such as PyTorch and NumPy. These sources of randomness are typically controlled by setting specific random seeds.
>
> Beyond these standard MARL implementation factors, the authors would also like to comment on the external disturbance $\boldsymbol{d}_t$ mentioned in Section 4.2. In the main formulation, such disturbances are treated as negligible, since this work focuses on **fully autonomous systems** where the control input is generated solely by the RL-based joint controller using the system’s own state. Nevertheless, it is possible to introduce disturbances in the task to examine their impact on MAS$^3$AC. To keep the overall response concise, we kindly ask the reviewer to refer to the response to Question 7 raised by Reviewer D2kU. Thank you for your understanding.
>
> ---
> ### Question 6
> > What kind of convergence do you consider with a stochastic $x_t$?
>
> **A10:** Thanks for the question. Without external disturbances, for a given $\boldsymbol{x}_t$, all corresponding signals (reward $r^i$, barrier $b^i$, and cost $c^i$) are deterministic and can be collected and stored for training. The associated critics (the action-value function $Q$, and certificates $H$ and $V$) represent **expected values** over trajectories induced by the current joint controller, and therefore inherently account for all stochasticity arising from the policy and the other sources mentioned in the response to Question 5. The convergence toward the desired state is then ensured via the critic $V$ and constraint (5c). When explicit external disturbances are introduced in a task, please refer to the response to Question 5 above for additional experimental results and analysis.

---

> ### Author Response · Authors · 2025-11-25
> **Authors' response to Reviewer yTHj (4)**
>
> ### Question 7
> > Some previous works are criticised for "focusing on tasks with a joint team reward/cost". Why is this a problem, as it seems easy to adapt these works to the proposed framework (for example, MAPPO has already been adapted by the authors)?
>
> **A11:** Thanks for the question. Although certain cooperative MARL algorithms, such as HAPPO, has been modified to general versions to be used as baselines in this submission, the authors would like to clarify the following points:
>
> 1. **Theoretical considerations:** Such extension is not supported by theoretical analysis. Many previous cooperative MARL algorithms, including HAPPO, HATRPO, and the following MACPO, MAPPO-Lagrangian are developed and supported by the **Multi-Agent Advantage Decomposition Lemma [1]**, which relies on the existence of a single action-value function $Q_{\boldsymbol{\pi}}$ for the whole team, due to the application of a joint team signal, to provide the direction of **performance improvements**. By allowing each agent to have its own signal, this lemma may not naturally apply, and therefore, even if can be practically implemented, the modified baselines are not naturally guaranteed by such theoretical analysis.
>
> 2. **Practical considerations:**
>
> (1) **Critic updates:** For critics, in non-cooperative tasks, each agent’s critic must be updated **at the fastest timescale**, so that critics accurately represent the environment when any individual policy is updated. However, in cooperative task, it is possible to update critics only **once per round of sequential policy updates [2]**, because all agents optimize the *same* reward and effectively perform a coordinated update toward improving the same critic function.
>
> (2) **Order of policy updates:** In cooperative MARL, the order of agents in a sequential policy update **can be permuted**, and the learning rates used for each agent are not specifically instructed to be different by certain theory. However, for general MARL, once the **multi-timescale approach** is required to guarantee convergence, this order of agents is pre-defined and fixed in each sequential update to align with the specific learning rate design (different timescale design). For brevity, we kindly refer the reviewer to the response to Question 5 by Reviewer D2kU for a more detailed explanation.
>
> (3) **Algorithmic components designed for cooperation:** Algorithms such as HAPPO include coordination-oriented components, e.g., shared ratio-based factors, that **do not naturally apply** in competitive or general-sum settings where agents may have only local information or conflicting objectives. Simply removing these terms can substantially change the algorithm, effectively turning it into a different method with no theoretical support.
>
> (4) **Future directions:**  As stated in the Conclusion and Limitations section, MAS$^3$AC provides a foundation for further research, including: theoretical scalability analysis for non-cooperative off-policy MARL, and designs that improve sample efficiency, etc. These directions represent important future work that goes beyond cooperative MARL extensions.

---

> ### Author Response · Authors · 2025-11-25
> **Authors' response to Reviewer yTHj (5)**
>
> ### Question 8
> > In the Hamilton-Jacobi reachability community, it is popular to use a max-over-time safety signal as a neural barrier function. In contrast, this paper proposes to use a sum-over-time barrier signal as the neural barrier function. What is the motivation for doing this, and does this achieve better results? If so, it would be great to see some empirical results.
>
> **A12:** Thanks for the question. There exists difference between these two methods. The reachability-based method [3] trains a safety value function in a **max-over-time** manner and enforces that its value remains non-negative. This is an **algebraic constraint** directly tied to the certificate's numerical value. In contrast, our barrier-based formulation, as in constraint (2b), follows a more **geometric interpretation**, requiring that along the trajectory, the next state becomes *safer* than the current one if the current state is unsafe, or maintains safety otherwise.
>
> We additionally implemented several new experiments during the rebuttal phase. For the reachability-based safety enforcements, it is possible to replace the current barrier function-based method with the reachability-based certificate, where the max-over-time safety certificates (with a different operator for update) are used instead of the sum-over-time certificates, and applied safety constraints requiring values of the reachability-based certificates to remain non-positive. The corresponding figures are provided in Section J.5 and the statistics are included below.
>
> **Table: Reward and cost comparisons between different certificates on cooperative tasks**
>
> | Algorithm     | Ant Reward                              | Ant Cost                               | HalfCheetah Reward                           | HalfCheetah Cost                           | Coupled-HalfCheetah Reward                           | Coupled-HalfCheetah Cost                           |
> | ------------- | --------------------------------------- | -------------------------------------- | -------------------------------------------- | ------------------------------------------ | ---------------------------------------- | -------------------------------------- |
> | HAPPO         | $41.29 \pm 215.89$                      | $311.96 \pm 67.96$                     | $373.64 \pm 200.58$                          | $2131.21 \pm 483.13$                       | $2197.22 \pm 112.77$                     | $230.28 \pm 67.11$                     |
> | HASAC         | $993.56 \pm 87.49$                      | $107.49 \pm 22.64$                     | $\textbf{2229.41}$ $\pm$ $\textbf{335.17}$                         | $211.07 \pm 300.58$                        | $2410.71 \pm 106.28$                     | $84.77 \pm 65.91$                      |
> | MACPO         | $-1381.66 \pm 37.86$                    | $105.24 \pm 50.66$                     | $443.50 \pm 265.72$                          | $134.58 \pm 142.09$                        | $1104.05 \pm 356.15$                     | $9.32 \pm 8.53$                        |
> | MAFOCOPS      | $-3965.01 \pm 720.78$                   | $156.47 \pm 77.84$                     | $-781.55 \pm 302.59$                         | $581.41 \pm 494.46$                        | $-1865.72 \pm 117.10$                    | $0.14 \pm 0.97$                        |
> | MAPPO         | $-354.58 \pm 203.39$                    | $322.31 \pm 64.85$                     | $418.49 \pm 194.12$                          | $1834.08 \pm 513.17$                       | $480.86 \pm 195.30$                      | $1764.08 \pm 431.53$                   |
> | MAPPOL        | $-903.98 \pm 67.67$                     | $142.52 \pm 58.00$                     | $672.25 \pm 143.48$                          | $898.40 \pm 379.02$                        | $1968.86 \pm 118.37$                     | $3.67 \pm 3.39$                        |
> | Reachability  | $\textbf{1671.62}$ $\pm$ $\textbf{77.84}$ | $78.82 \pm 63.39$                   | $1939.28 \pm 138.34$                         | $272.27 \pm 149.15$                        | $2513.78 \pm 155.96$                     | $13.15 \pm 17.28$                      |
> | MAS$^3$AC     | $1221.79 \pm 50.45$                     | $\textbf{2.10}$ $\pm$ $\textbf{6.92}$ | ${2177.90}$ $\pm$ ${714.48}$   | $\textbf{6.22}$ $\pm$ $\textbf{25.28}$     | $\textbf{2596.56}$ $\pm$ $\textbf{105.95}$ | $\textbf{0.03}$ $\pm$ $\textbf{0.17}$ |

---

> ### Author Response · Authors · 2025-11-25
> **Authors' response to Reviewer yTHj (6)**
>
> ### Question 8 (continued)
> **Table: Reward and cost comparisons between different certificates on the non-cooperative Coupled HalfCheetah task**
>
> | Algorithm     | Agent 0 Reward                           | Agent 0 Cost                           | Agent 1 Reward                           | Agent 1 Cost                           | Agent 2 Reward                           | Agent 2 Cost                           |
> | ------------- | ---------------------------------------- | -------------------------------------- | ---------------------------------------- | -------------------------------------- | ---------------------------------------- | -------------------------------------- |
> | IPPO          | $902.29 \pm 386.41$                      | $451.38 \pm 305.37$                    | $904.99 \pm 389.86$                      | $451.38 \pm 305.37$                    | $905.91 \pm 388.62$                      | $451.38 \pm 305.37$                    |
> | MACPO         | $-140.28 \pm 276.72$                     | $21.48 \pm 25.26$                      | $-141.91 \pm 276.68$                     | $21.48 \pm 25.26$                      | $-142.75 \pm 276.65$                     | $21.48 \pm 25.26$                      |
> | MAPPOL        | $735.92 \pm 416.97$                      | $174.96 \pm 143.68$                    | $735.77 \pm 417.75$                      | $174.96 \pm 143.68$                    | $732.35 \pm 417.00$                      | $174.96 \pm 143.68$                    |
> | MASAC         | $2499.75 \pm 78.45$                      | $25.25 \pm 20.76$                      | $2503.90 \pm 84.95$                      | $25.25 \pm 20.76$                      | $2489.93 \pm 81.61$                      | $25.25 \pm 20.76$                      |
> | Reachability  | $2479.88 \pm 166.99$                     | $14.22 \pm 26.00$                      | $2472.27 \pm 161.99$                     | $14.22 \pm 26.00$                      | $2471.29 \pm 161.62$                     | $14.22 \pm 26.00$                      |
> | MAS$^3$AC     | $\textbf{2531.32}$ $\pm$ $\textbf{136.89}$ | $\textbf{6.35}$ $\pm$ $\textbf{14.12}$ | $\textbf{2533.62}$ $\pm$ $\textbf{141.96}$ | $\textbf{6.35}$ $\pm$ $\textbf{14.12}$ | $\textbf{2529.90}$ $\pm$ $\textbf{142.89}$ | $\textbf{6.35}$ $\pm$ $\textbf{14.12}$ |
>
> Based on these results, we observe that **barrier certificates generally achieve better safety performance** than reachability-based certificates. We believe the key reason is the form of the constraint: barrier certificates enforce **exponential safety improvement** through constraint (2b), which explicitly requires the barrier value, if negative, to increase at an exponential rate to ensure safety. In contrast, the reachability-based constraint only requires non-positivity and is updated via standard gradient descent, lacking the additional mechanism that encourages rapid recovery from unsafe states.

---

> ### Author Response · Authors · 2025-11-25
> **Authors' response to Reviewer yTHj (7)**
>
> ### Question 9
> > The efficacy of the ISS stability part is unclear. Can you consider completely removing this part and comparing the results? Also, maybe consider completely removing the reward signal and only using the ISS stability term to guide the training.
>
> **A13:** Thanks for the question. The authors have implemented experiments to remove the ISS stability part to show its efficacy (figures are in Section J.6, and statistics are provided below):
>
> **Table: Reward and cost comparisons between different certificates on the non-cooperative Coupled HalfCheetah task**
>
> | Algorithm   | Ant Reward                              | Ant Cost                               | HalfCheetah Reward                           | HalfCheetah Cost                              | Coupled Reward                           | Coupled Cost                              |
> | ----------- | --------------------------------------- | -------------------------------------- | -------------------------------------------- | --------------------------------------------- | ---------------------------------------- | ----------------------------------------- |
> | HAPPO       | $41.29 \pm 215.89$                      | $311.96 \pm 67.96$                     | $373.64 \pm 200.58$                          | $2131.21 \pm 483.13$                          | $2197.22 \pm 112.77$                     | $230.28 \pm 67.11$                        |
> | HASAC       | $993.56 \pm 87.49$                      | $107.49 \pm 22.64$                     | $\textbf{2229.41}$ $\pm$ $\textbf{335.17}$   | $211.07 \pm 300.58$                           | $2410.71 \pm 106.28$                     | $84.77 \pm 65.91$                         |
> | MACPO       | $-1381.66 \pm 37.86$                    | $105.24 \pm 50.66$                     | $443.50 \pm 265.72$                          | $134.58 \pm 142.09$                           | $1104.05 \pm 356.15$                     | $9.32 \pm 8.53$                           |
> | MAFOCOPS    | $-3965.01 \pm 720.78$                   | $156.47 \pm 77.84$                     | $-781.55 \pm 302.59$                         | $581.41 \pm 494.46$                           | $-1865.72 \pm 117.10$                    | $0.14 \pm 0.97$                           |
> | MAPPO       | $-354.58 \pm 203.39$                    | $322.31 \pm 64.85$                     | $418.49 \pm 194.12$                          | $1834.08 \pm 513.17$                          | $480.86 \pm 195.30$                      | $1764.08 \pm 431.53$                      |
> | MAPPOL      | $-903.98 \pm 67.67$                     | $142.52 \pm 58.00$                     | $672.25 \pm 143.48$                          | $898.40 \pm 379.02$                           | $1968.86 \pm 118.37$                     | $3.67 \pm 3.39$                           |
> | No-ISS      | $921.66 \pm 104.09$                     | $\textbf{1.53}$ $\pm$ $\textbf{1.53}$ | $-1043.67 \pm 202.75$                        | $\textbf{0.00}$ $\pm$ $\textbf{0.00}$         | $-1317.85 \pm 163.06$                    | $\textbf{0.00}$ $\pm$ $\textbf{0.00}$    |
> | MAS$^3$AC   | $\textbf{1221.79}$ $\pm$ $\textbf{50.45}$ | $2.10 \pm 6.92$                      | $2177.90 \pm 714.48$                         | $6.22 \pm 25.28$                              | $\textbf{2596.56}$ $\pm$ $\textbf{105.95}$ | $0.03 \pm 0.17$                        |
>
>
> According to the results, the efficacy of the ISS part differs across tasks. In Ant, without ISS, the improvement in reward is slower than that of the complete version where ISS is used, especially at the beginning of the training process. Moreover, regarding the final reward obtained, with ISS the final reward is higher than that achieved by the counterpart without ISS constraints. In the other two tasks, when ISS is removed, the reward cannot meaningfully increase and therefore remains at a low level. The reason is that with the barrier certificate for safety but without the ISS component to encourage convergence toward the desired state, the controlled agent becomes overly conservative during exploration. It is prevented from achieving high rewards by approaching the desired goal due to the strong repulsion imposed by the exponential safety constraint (2b). In such settings, it is **essential to incorporate an additional stimulus to encourage effective exploration** while maintaining state-wise safety, and this is exactly the role of the ISS part.

---

> ### Author Response · Authors · 2025-11-25
> **Authors' response to Reviewer yTHj (8)**
>
> ### Question 9 (continued)
>
> The authors have also implemented experiments where the **reward signal is completely removed** (figures are in Section J.7, and statistics are provided below):
>
> **Table: Reward and cost comparisons for ablation study on the reward signal**
>
> | Algorithm   | Ant Reward   | Ant Cost   | HalfCheetah Reward   | HalfCheetah Cost    |
> | - | - | -| - | - |
> | HAPPO | $41.29 \pm 215.89$   | $311.96 \pm 67.96$   | $373.64 \pm 200.58$  | $2131.21 \pm 483.13$  |
> | HASAC | $993.56 \pm 87.49$ | $107.49 \pm 22.64$    | $\textbf{2229.41}$ $\pm$ $\textbf{335.17}$ | $211.07 \pm 300.58$                           |
> | MACPO  | $-1381.66 \pm 37.86$ | $105.24 \pm 50.66$   | $443.50 \pm 265.72$   | $134.58 \pm 142.09$    |
> | MAFOCOPS    | $-3965.01 \pm 720.78$  | $156.47 \pm 77.84$   | $-781.55 \pm 302.59$  | $581.41 \pm 494.46$      |
> | MAPPO       | $-354.58 \pm 203.39$                    | $322.31 \pm 64.85$                     | $418.49 \pm 194.12$                        | $1834.08 \pm 513.17$                          |
> | MAPPOL      | $-903.98 \pm 67.67$                     | $142.52 \pm 58.00$                     | $672.25 \pm 143.48$                        | $898.40 \pm 379.02$                           |
> | No-Reward   | $\textbf{1258.19}$ $\pm$ $\textbf{167.19}$ | $8.78 \pm 10.02$                    | $-1396.78 \pm 5.49$                        | $\textbf{0.00}$ $\pm$ $\textbf{0.00}$         |
> | MAS$^3$AC   | $1221.79 \pm 50.45$                     | $\textbf{2.10}$ $\pm$ $\textbf{6.92}$ | $2177.90 \pm 714.48$                       | $6.22 \pm 25.28$                              |
>
> Based on these results, the reward signal also plays an important role in training:
>
> (1) It helps achieve a more efficient increase in reward, especially at the beginning of training, as demonstrated in Ant. Reward signals provide useful direction for policy improvement.
>
> (2) In some tasks such as HalfCheetah, without the reward signal, the training may totally fail to achieve an increase in reward. The likely reason is that, on some tasks, without reward to provide a stable improvement direction, the applied constraint terms may compound and be large in size, which leads to numerical issues in the training and thus the learning fails.
>
> ---
> ### Question 10
> >The paper considers an extra squared term to more strongly penalize safety violations. This needs an ablation study to demonstrate the improvement.
>
> **A14:** Thanks for the question. The authors have implemented experiments to remove the extra squared terms to show their efficacy. To reduce the length of the overall response, we kindly ask the reviewer to refer to the central part of our response to Weakness 1 raised by Reviewer 37aY.
>
> In short, the results show that **including the extra squared terms prevents noticeable and difficult-to-recover reward deterioration** (e.g., in HalfCheetah, the reward curve eventually collapses when the squared terms are removed). Since this squared-term design also applies to the stability constraints beyond safety constraints, the reward curve also reflects its effectiveness.
>
> Regarding safety, the version **with** extra squared terms generally achieves **better safety satisfaction during training on both tasks**. This design is specifically introduced as an effort to reduce safety violations even **during** the learning process.
>
> ---
> ### Question 11
> > It seems that the learning rates are missing in Table 1. Can you provide this important hyperparameter? I am also curious about the initial and end values of the Lagrange multipliers (curves can be better).
>
> **A15:** Thanks for the question. The learning rates have been added to **Table 3 in Appendix H.7** in the newly updated PDF submission. Specifically, we use **linearly decaying learning rates** with different initial values to ensure $\eta_1(e)>\eta_{2,i}(e)>\eta_{3,i}(e)$ for all $e$ and all $i\in\mathcal{N}$, and  $\eta_{3,i-1}(e)>\eta_{2,i}(e)$ for all $e$ and all $i\in\{2,\dots,N\}$, where $e$ is the number of updates, to approximate the required relationships among the learning rates as required by **Assumption 2**, which has been **empirically validated** to be sufficient to support stable and efficient training across all benchmarks compared to all baselines.
>
> We also note that **other initial-value choices** for policy and Lagrangian multiplier learning rates can also lead to good performance. For example, when different initial values of learning rates for critics,  policies, and Lagrangian multipliers are applied, the algorithm still achieves strong reward maximization and safety satisfaction, as denoted in Section H.7, Figure 22, Section J.9.
>
> Finally, **the curves showing the evolution of the Lagrange multipliers** on Ant task have also been added, with figures provided in Section J.9. The results show that, for all four agents, the Lagrangian multipliers associated with both the safety and stability constraints exhibit stable and gradual convergence.

---

> ### Author Response · Authors · 2025-11-25
> **Authors' response to Reviewer yTHj (9)**
>
> ### Question 12
> > Why can MAS$^3$AC achieve higher results than algorithms that do not consider safety?
>
> **A16:** Thank you for the question. The authors believe the following factors may explain why MAS$^3$AC can outperform algorithms that do not consider safety:
>
> 1. **Incorporation of the ISS part.** The ISS part encourages the controlled multi-agent system to converge toward the desired state (goal). In the considered benchmarks, higher rewards are obtained when the system approaches and maintains this goal (consistent with common stabilization and tracking tasks). Thus, adding the ISS mechanism directly improves reward maximization. This effect is also evident when compared with safety-unaware baselines, which **do not include ISS terms** and therefore lack this source of guidance (see the response to Question 9 for empirical support).
>
> 2. **Algorithmic family differences.** MAS$^3$AC is an **off-policy** method, whereas several baselines such as HAPPO belong to the **on-policy** family. Off-policy algorithms can achieve better sample efficiency. When all methods are granted several million environment steps, it is possible that the off-policy algorithm can achieve more efficient reward maximization and safety satisfaction.
>
> **References**
>
> [1] Grudzien Kuba, J., Chen, R., Wen, M., Wen, Y., Sun, F., Wang, J., \& Yang, Y. (2022). Trust region policy optimisation in multi-agent reinforcement learning. In International Conference on Learning Representations.
>
> [2] Li, Z., \& Azizan, N. (2024). Safe multi-agent reinforcement learning with convergence to generalized nash equilibrium. arXiv preprint arXiv:2411.15036.
>
> [3] Yu, D., Ma, H., Li, S., \& Chen, J. (2022, June). Reachability constrained reinforcement learning. In International conference on machine learning (pp. 25636-25655). PMLR.

---

### Official Review · Reviewer_mUAV · 2025-10-31

**Soundness:** 2
**Presentation:** 2
**Contribution:** 2
**Rating:** 4
**Confidence:** 4

**Summary:**

This paper presents MAS3AC, a multi-agent learning framework that unifies state-wise safety and system stability on top of SAC. The method employs learned barrier certificates to transform step-wise violation risks into inequality penalties, and enforces stability via ISS/Lyapunov inequalities, both optimized jointly with returns using a Lagrangian with squared penalties and multi-time-scale updates under decentralized, low-communication settings. Theoretically, under stated assumptions, the algorithm converges to a feasible local Nash equilibrium and includes an analysis to detect and circumvent infeasible regions when safety and stability conflict. On Safe-MuJoCo and other cooperative/decentralized tasks, MAS3AC achieves a superior reward–safety trade-off over strong baselines, with ablations (e.g., neighborhood radius) corroborating its low-communication design. By making “state-wise safety + stability” trainable within a MARL backbone, MAS3AC offers a practical route to safe and stable control for resource-constrained multi-agent systems.

**Strengths:**

1. This paper proposes a decentralized and implementable MARL framework that unifies state-wise safety and ISS-based stability within a single learning paradigm.
2. It proves convergence to a feasible local Nash equilibrium under given assumptions and provides an infeasibility analysis when safety and stability objectives conflict.
3. The methodology section is written in a clear and well-structured manner, making the technical ideas accessible and coherent.

**Weaknesses:**

1. Related Work Section: The section only touches on domain-specific challenges at a high level and lacks clear, structured articulation. Please add a dedicated “Challenges” subsection that explicitly enumerates the key difficulties this paper addresses.

2. Method Section: In the multi–time-scale training of neural barrier certificates, the learning-rate decay coefficients for the Critic/Barrier, Policy, and Lagrangian multipliers (e.g., $η_1(e)>η_2,_i(e)>η_3,_i(e)$) are presented without a clear rationale. Please explain why this ordering is preferable and provide justification for these choices to enhance the method’s interpretability.

3. Experiments Section: The comparison baselines are missing recent decentralized constrained MARL methods, such as GCBF-PPO [1], which makes it difficult to fully demonstrate the advantage of MAS³AC over methods in the same category. It is recommended to include such baselines so that the claimed stability guarantees become more convincing. In addition, the only non-cooperative task evaluated is Coupled HalfCheetah. Please clarify whether this is sufficient to justify the “general” claim in the title, or provide additional non-cooperative benchmarks.

**Reference**

[1] Songyuan Zhang, Oswin So, Mitchell Black, and Chuchu Fan. Discrete GCBF proximal policy optimization for multi-agent safe optimal control. In The Thirteenth International Conference on Learning Representations, 2025

**Questions:**

Please refer to the “Weakness” section for related questions.

**Details Of Ethics Concerns:**

Regarding ethical review, I have no concerns.

---

> ### Author Response · Authors · 2025-11-25
> **Authors' response to Reviewer mUAV (1)**
>
> ### Weakness 1
>
> > Related Work Section: The section only touches on domain-specific challenges at a high level and lacks clear, structured articulation. Please add a dedicated “Challenges” subsection that explicitly enumerates the key difficulties this paper addresses.
>
> **A1:** Thanks for the question. We appreciate the reviewer’s suggestion. In the revised PDF, we have added the following content, as the “Challenges” subsection in the Related Work section, to explicitly summarize the key challenges addressed in this submission.
>
> **Challenges: Many methods applying pre-defined control barrier function for safe single-agent RL require a control-affine nominal model of the system [1], or further restrict the CBF to be affine with respect to the state [2]. Meanwhile, although neural barrier certificates have also shown promising results in reducing safety violations, they may also rely on system models [3], which could limit their applicability. In contrast, the neural barrier certificates used in our work can be easily integrated into RL training in a principled way without demanding system dynamics. Further, our work differs from these existing safe multi-agent RL studies based on CMDP [4] and/or developed for cooperative tasks [5]: first, our method considers state-wise safety requirements and can be applied to general MARL settings beyond cooperative ones. Second, we additionally incorporate stability conditions naturally into the training process, yielding a more comprehensive framework for learning safe and stable controllers for multi-agent tasks. Third, the application of multiple constraints can lead to the issue of infeasibility [6], which requires further study in learning-based settings. Our work provides a theoretical analysis of this issue for MARL. Empirical results demonstrate that our algorithm yields consistently high rewards with very few safety violations compared to baselines on both cooperative and non-cooperative MARL tasks with either centralized or decentralized training.**

---

> ### Author Response · Authors · 2025-11-25
> **Authors' response to Reviewer mUAV (2)**
>
> ### Weakness 2
> > Method Section: In the multi–time-scale training of neural barrier certificates, the learning-rate decay coefficients for the Critic/Barrier, Policy, and Lagrangian multipliers (e.g., $\eta_1(e) > \eta_{2,i}(e) > \eta_{3,i}(e)$) are presented without a clear rationale. Please explain why this ordering is preferable and provide justification for these choices to enhance the method’s interpretability.
>
> **A2:** Thanks for the question. Here we answer the questions in points:
>
> 1. **Critic/Barrier should be updated at the quickest timescale.** All policy networks and Lagrangian multipliers are updated based on the loss function detailed in Appendix A. To be more specific, when considering action-value functions and barrier certificates, it is essential that **parameters $\phi_i$, $\omega_i$ used in (8) detailed in Appendix A have been updated adequately to precisely represent the action-value function $Q^i_{\boldsymbol{\pi}}$ and barrier certificate $H^i_{\boldsymbol{\pi}}$**, respectively, such that the loss function (8) can **correctly align with the Lagrangian function (3)**. Otherwise, **large errors can occur when policy and Lagrangian multipliers are updated** via gradient information using critic/barrier, and the overall update may fail to converge to a good final joint policy.
>
> 2. **For each agent, policy should be updated at a faster timescale than its Lagrangian multiplier.** Given the critic/barrier have been updated to correct parameterization, the work now is to solve the constrained optimization problem, whose primal variable is the policy parameter $\theta_i$ and dual variable is the Lagrangian multiplier $\lambda_i$, with a Lagrangian function which loss function, e.g., (8) in Appendix A, aligns with. In common dual ascent update to solve such an optimization problem, it is set that the primal variable is updated at a faster rate. Following this convention, here we require that for each agent, its policy has a quicker update (reflected by a larger learning rate), than that of the Lagrangian multiplier.
>
> 3. **Ordering across agents.** For orders between different agents, to ensure the convergence of the update towards a god final joint policy, the **non-stationarity** that is a long-standing difficulty preventing smooth convergence of the MARL training should be mitigated. Therefore, the authors require that when updating one agent $i$, all other agents should follow a certain fixed joint policy $\boldsymbol{\pi}^{-i}$ such that the environment (including all other agents) appears stationary from the perspective of the single agent $i$ that is being updated. This essentially leads to the multi-timescale design, where at each timescale the corresponding single agent updates its policy $\theta_i$ and all other agents follow a certain joint policy from agent $i$'s perspective. Combined with the Lagrangian multiplier's update, we finally have the specific order of different learning rates  (which is essentially an order of timescales for training) described in Assumption 2. The update whose learning rate is larger is at a quicker update timescale.
>
> 4. **Implementation detail.** In implementation, within each round for a complete sequential update, **the order of agents for policy update is kept the same without permutations**. If the learning rates of policy updates are designed to satisfy $\eta_{2,i}> \eta_{2,i-1}$, $\forall i \in \{2, \dots, N\}$, in theoretical analysis, this indicates when agent $i$ updates its policy, all agents $j \in \{1, \dots, i-1\}$ should have updated their policies adequately and converged. The proposed implementation, correspondingly, aims to perform the update of all agents $j \in \{1, \dots, i-1\}$ **before the update of agent $i$** in every round of complete sequential update to align with the theoretical interpretation. Since the learning rates are pre-defined before running the experiments, the corresponding required order for policy update is therefore fixed and obeyed during the whole training.

---

> ### Author Response · Authors · 2025-11-25
> **Authors' response to Reviewer mUAV (3)**
>
> ### Weakness 3
> > Experiments Section: The comparison baselines are missing recent decentralized constrained MARL methods, such as GCBF-PPO [7], which makes it difficult to fully demonstrate the advantage of MAS$^3$AC over methods in the same category. It is recommended to include such baselines so that the claimed stability guarantees become more convincing. In addition, the only non-cooperative task evaluated is Coupled HalfCheetah. Please clarify whether this is sufficient to justify the “general” claim in the title, or provide additional non-cooperative benchmarks.
>
> **A3:** Thanks for the question. For the first part, regarding GCBF-PPO [7], after careful investigation, the authors found that this method has not been evaluated on Safe MAMujoco tasks [5], from which the tasks used in this submission are developed. Therefore, unfortunately, no hyperparameters, performance benchmarks, or implementation details exist for GCBF-PPO on Safe MAMujoco tasks. Adapting and tuning GCBF-PPO from scratch for these environments would require extensive experimentation **beyond the rebuttal timeline**.
>
> Hence, the authors add **Scal-MAPPO-L** [8] as a strong additional baseline, because:
> 1. It is a **recent decentralized constrained MARL method** accepted at NeurIPS 2024.
> 2. It has been tested on Safe MAMujoco tasks.
>
> Further, to address Reviewer yTHj’s concern in Weakness 2 simultaneously, the authors made the following modifications to Scal-MAPPO-L:
> 1. We modify Scal-MAPPO-L to use **max-over-time safety certificates** (similar to the DCBF used in GCBF-PPO [7]) instead of CMDP constraints, directly addressing Reviewer yTHj’s concern regarding **state-wise vs. CMDP** safety formulations.
> 2. We implement an **off-policy MASAC-based version** instead of the original on-policy version because:
>    (a) we already include MAPPO-Lagrangian as a baseline, and
>    (b) this enables controlled comparison with our off-policy MAS$^3$AC.
>
> The comparisons were conducted on several tasks, with the statistics provided below (the corresponding figures are in Section J.3 of the revised PDF submission).
>
> **Table: Reward and cost comparison with the additional modified Scal-MAPPO-L baseline (cooperative tasks)**
>
> | Algorithm | Ant Reward  | Ant Cost | HalfCheetah Reward  | HalfCheetah Cost |
> | -| -| - | - | - |
> | HAPPO| $41.29 \pm 215.89$  | $311.96 \pm 67.96$| $373.64 \pm 200.58$     | $2131.21 \pm 483.13$|
> | HASAC  | $993.56 \pm 87.49$  | $107.49 \pm 22.64$   | $2229.41 \pm 335.17$ | $211.07 \pm 300.58$ |
> | MACPO  | $-1381.66 \pm 37.86$  | $105.24 \pm 50.66$ | $443.50 \pm 265.72$   | $134.58 \pm 142.09$  |
> | MAFOCOPS | $-3965.01 \pm 720.78$  | $156.47 \pm 77.84$| $-781.55 \pm 302.59$   | $581.41 \pm 494.46$  |
> | MAPPO   | $-354.58 \pm 203.39$| $322.31 \pm 64.85$  | $418.49 \pm 194.12$ | $1834.08 \pm 513.17$  |
> | MAPPOL   | $-903.98 \pm 67.67$    | $142.52 \pm 58.00$   | $672.25 \pm 143.48$  | $898.40 \pm 379.02$   |
> | Scal\_MAPPOL  | $997.24 \pm 64.07$  | $86.24 \pm 12.39$  | $\textbf{2460.87}$ $\pm$ $\textbf{95.22}$    | $22.44 \pm 67.28$          |
> | MAS$^3$AC   | $\textbf{1221.79}$ $\pm$ $\textbf{50.45}$ | $\textbf{2.10}$ $\pm$ $\textbf{6.92}$ | $2177.90 \pm 714.48$       | $\textbf{6.22}$ $\pm$ $\textbf{25.28}$ |
>
> **Table: Reward and cost comparison with the additional modified Scal-MAPPO-L baseline (non-cooperative Coupled HalfCheetah task) – Agents 0, 1, 2**
>
> | Algorithm | Agent 0 Reward| Agent 0 Cost| Agent 1 Reward  | Agent 1 Cost  | Agent 2 Reward | Agent 2 Cost |
> | - | - | - | - | - | - | - |
> | IPPO   | $902.29 \pm 386.41$   | $451.38 \pm 305.37$  | $904.99 \pm 389.86$  | $451.38 \pm 305.37$   | $905.91 \pm 388.62$ | $451.38 \pm 305.37$     |
> | MACPO  | $-140.28 \pm 276.72$  | $21.48 \pm 25.26$   | $-141.91 \pm 276.68$| $21.48 \pm 25.26$   | $-142.75 \pm 276.65$    | $21.48 \pm 25.26$   |
> | MAPPOL  | $735.92 \pm 416.97$  | $174.96 \pm 143.68$   | $735.77 \pm 417.75$ | $174.96 \pm 143.68$  | $732.35 \pm 417.00$    | $174.96 \pm 143.68$   |
> | MASAC   | $2499.75 \pm 78.45$  | $25.25 \pm 20.76$  | $2503.90 \pm 84.95$  | $25.25 \pm 20.76$ | $2489.93 \pm 81.61$   | $25.25 \pm 20.76$   |
> | Scal\_MAPPOL  | $2524.31 \pm 84.52$    | $18.51 \pm 17.73$| $2516.04 \pm 88.89$  | $18.51 \pm 17.73$   | $2512.03 \pm 83.29$    | $18.51 \pm 17.73$    |
> | MAS$^3$AC     | $\textbf{2531.32}$ $\pm$ $\textbf{136.89}$ | $\textbf{6.35}$ $\pm$ $\textbf{14.12}$ | $\textbf{2533.62}$ $\pm$ $\textbf{141.96}$ | $\textbf{6.35}$ $\pm$ $\textbf{14.12}$ | $\textbf{2529.90}$ $\pm$ $\textbf{142.89}$ | $\textbf{6.35}$ $\pm$ $\textbf{14.12}$ |

---

> ### Author Response · Authors · 2025-11-25
> **Authors' response to Reviewer mUAV (4)**
>
> ### Weakness 3 (continued)
> The results show that the new baseline can also achieve good balance in reward maximization and safety satisfaction compared to some other baselines. However, on cooperative Ant, its performance in safety is not satisfying, leading to large safety violations until the end of training process, while the proposed algorithm works well. The author think the possible reason is the constraint to apply: MAS³AC uses barrier-certificate constraints (2b), which enforce an **exponential** increase in barrier values when they are negative, and thus aggressively push the system toward the safe set. However, the max-over-time safety certificate used in the modified baseline only requires its value to be **non-positive** and is updated via standard gradient descent, lacking this exponential enforcement mechanism.
>
> Regarding non-cooperative tasks, we want to clarify that the original submission **already includes a decentralized non-cooperative multi-agent electrical bidding task**, whose system dynamics are fundamentally different from robotic control tasks. Details are provided in **Appendix H.5**, with corresponding results in **Appendix I.1.1**.
>
> Additionally, in response to the reviewer’s suggestion, we provide a **new decentralized non-cooperative Ant task with 8 agents**, where **only local observations** are available. The task description is in Section H.6, and the corresponding statistics is provided below, with figures included in Section J.4.
>
> **Table: Reward and cost per agent for ablation study on new non-cooperative decentralized Ant task**
>
> | Algorithm | Agent 0 Reward  | Agent 0 Cost    | Agent 1 Reward  | Agent 1 Cost | Agent 2 Reward     | Agent 2 Cost                               | Agent 3 Reward   | Agent 3 Cost    |
> | - | -- | - | - | - | --| - | - | -- |
> | IPPO   | $-473.73 \pm 127.13$  | $111.25 \pm 56.83$   | $-474.56 \pm 124.97$  | $111.25 \pm 56.83$    | $-514.63 \pm 106.13$                     | $109.26 \pm 56.65$  | $-519.11 \pm 102.94$    | $109.26 \pm 56.65$    |
> | MACPO  | $-704.12 \pm 33.29$  | $\textbf{47.89}$ $\pm$ $\textbf{27.50}$  | $-690.74 \pm 26.51$    | $\textbf{59.28}$ $\pm$ $\textbf{32.32}$   | $-729.84 \pm 34.31$   | $60.08$ $\pm$ $\{32.47}$  | $-742.70 \pm 42.06$  | $60.08$ $\pm$ $32.47$  |
> | MAPPOL | $-552.92 \pm 50.26$    | $83.31 \pm 40.09$    | $-552.76 \pm 51.65$      | $83.31 \pm 40.09$    | $-587.23 \pm 42.12$    | $80.83 \pm 38.76$ | $-590.74 \pm 41.15$ | $80.83 \pm 38.76$   |
> | MASAC  | $-539.92 \pm 30.43$ | $116.42 \pm 50.14$  | $-551.35 \pm 29.67$  | $116.42 \pm 50.14$ | $-573.06 \pm 26.74$                     | $114.57 \pm 49.26$ | $-575.84 \pm 26.83$   | $114.57 \pm 49.26$  |
> | MAS$^3$AC | $\textbf{106.95}$ $\pm$ $\textbf{147.84}$ | $65.33 \pm 33.32$  | $\textbf{102.38}$ $\pm$ $\textbf{164.69}$ | $65.33 \pm 33.32$ | $\textbf{143.49}$ $\pm$ $\textbf{130.09}$ | $\textbf{40.42}$ $\pm$ $\textbf{26.09}$  | $\textbf{139.48}$ $\pm$ $\textbf{128.53}$ | $\textbf{40.42}$ $\pm$ $\textbf{26.09}$  |
>
>
>
> | Algorithm | Agent 4 Reward   | Agent 4 Cost  | Agent 5 Reward   | Agent 5 Cost    | Agent 6 Reward   | Agent 6 Cost    | Agent 7 Reward | Agent 7 Cost    |
> | - | - | -| - | - | - | - | - | - |
> | IPPO  | $-531.28 \pm 97.42$  | $110.10 \pm 57.07$ | $-529.98 \pm 95.81$| $110.10 \pm 57.07$  | $-488.03 \pm 116.87$                     | $112.29 \pm 56.79$   | $-493.59 \pm 113.75$ | $112.29 \pm 56.79$    |
> | MACPO     | $-733.88 \pm 35.31$   | $59.72 \pm 34.65$   | $-730.70 \pm 29.61$    | $59.72 \pm 34.65$  | $-721.77 \pm 39.74$                      | $58.83 \pm 34.20$    | $-741.17 \pm 51.10$   | $58.83 \pm 34.20$    |
> | MAPPOL    | $-590.14 \pm 41.02$                      | $80.37 \pm 37.84$                      | $-587.67 \pm 39.76$                      | $80.37 \pm 37.84$                      | $-554.36 \pm 48.57$                      | $83.16 \pm 38.62$                      | $-558.00 \pm 50.22$                      | $83.16 \pm 38.62$                      |
> | MASAC     | $-573.30 \pm 33.81$                      | $139.44 \pm 57.73$                     | $-593.74 \pm 23.85$                      | $115.38 \pm 50.61$                     | $-565.41 \pm 26.94$                      | $116.93 \pm 50.39$                     | $-564.21 \pm 25.39$                      | $116.93 \pm 50.39$                     |
> | MAS$^3$AC | $\textbf{56.88}$ $\pm$ $\textbf{119.08}$ | $\textbf{28.69}$ $\pm$ $\textbf{22.59}$ | $\textbf{4.71}$ $\pm$ $\textbf{115.05}$ | $\textbf{28.69}$ $\pm$ $\textbf{22.59}$ | $\textbf{16.19}$ $\pm$ $\textbf{129.88}$ | $\textbf{46.70}$ $\pm$ $\textbf{32.85}$ | $\textbf{30.22}$ $\pm$ $\textbf{137.99}$ | $\textbf{46.70}$ $\pm$ $\textbf{32.85}$ |
>
>
> The results show that MAS$^3$AC almost achieves the **highest rewards** and **lowest final safety violations** compared to baselines for all agents, which demonstrates the efficacy of the proposed algorithm on more decentralized non-cooperative task.

---

> ### Author Response · Authors · 2025-11-25
> **Authors' response to Reviewer mUAV (5)**
>
> ### Question
>
> >Please refer to the “Weakness” section for related questions.
>
> **A4:** The authors hope that the above responses help clarify the challenges and difficulties addressed in this work, and provide the necessary rationale, additional baselines, and more tasks. Please feel free to let us know if you have any further questions.
>
>
> **References**
>
> [1] Emam, Y., Notomista, G., Glotfelter, P., Kira, Z., \& Egerstedt, M. (2022). Safe reinforcement learning using robust control barrier functions. IEEE Robotics and Automation Letters.
>
> [2] Cheng, R., Orosz, G., Murray, R. M., \& Burdick, J. W. (2019, July). End-to-end safe reinforcement learning through barrier functions for safety-critical continuous control tasks. In Proceedings of the AAAI conference on artificial intelligence (Vol. 33, No. 01, pp. 3387-3395).
>
> [3] Dawson, C., Qin, Z., Gao, S., \& Fan, C. (2022, January). Safe nonlinear control using robust neural lyapunov-barrier functions. In Conference on Robot Learning (pp. 1724-1735). PMLR.
>
> [4] Altman, E. (2021). Constrained Markov decision processes. Routledge.
>
> [5] Gu, S., Kuba, J. G., Chen, Y., Du, Y., Yang, L., Knoll, A., \& Yang, Y. (2023). Safe multi-agent reinforcement learning for multi-robot control. Artificial Intelligence, 319, 103905.
>
> [6] Reis, M. F., Aguiar, A. P., \& Tabuada, P. (2020). Control barrier function-based quadratic programs introduce undesirable asymptotically stable equilibria. IEEE Control Systems Letters, 5(2), 731-736.
>
> [7] Zhang, S., So, O., Black, M., \& Fan, C. (2025). Discrete GCBF proximal policy optimization for multi-agent safe optimal control. In The Thirteenth International Conference on Learning Representations.
>
> [8] Zhang, L., Li, L., Wei, W., Song, H., Yang, Y., \& Liang, J. (2024). Scalable constrained policy optimization for safe multi-agent reinforcement learning. Advances in Neural Information Processing Systems, 37, 138698-138730.

---

### Official Review · Reviewer_37aY · 2025-11-01

**Soundness:** 3
**Presentation:** 3
**Contribution:** 3
**Rating:** 6
**Confidence:** 3

**Summary:**

Proposes MAS³AC, a model‑free MARL framework that jointly enforces state‑wise safety via a learned neural barrier certificate and stability via input‑to‑state stability (ISS) constraints, trained with a multi‑timescale actor–critic + Lagrangian scheme. Convergence to a feasible local Nash equilibrium is proved under assumptions. Experiments on cooperative Safe‑MuJoCo variants and non‑cooperative, decentralized tasks (incl. an electrical bidding market) show higher reward with fewer safety violations than baselines.

**Strengths:**

1. Unified treatment of state‑wise safety + stability in general‑sum MARL (not only cooperative), with clear formalization of barrier certificates (Def. 3) and ISS conditions.
2. Practical training objective: penalizes differences of consecutive barrier values (rather than CMDP expected costs) with an extra squared penalty for violations (Eq. (3)), which the authors argue improves learning stability.
3. Theoretical story: multi‑timescale updates to convergence to feasible local NE (Thm. 1), monotone non‑shrinking safe set (Prop. 1), and a global barrier via the minimum over agents.
4. Empirical breadth & clarity: consistent “high reward + low violations” across tasks; the electrical bidding benchmark broadens beyond robotics.

**Weaknesses:**

1. Safety is only guaranteed after convergence. The paper motivates “zero violations at every step” for real systems, but the formal guarantees apply to the limit policy, not to learning‑time behavior; exploration can still violate constraints (no model/backup/shield). The paper emphasizes the need for per‑step safety, yet Corollary 1 removes disturbances and applies post‑convergence.
2. Assumptions are strong and hard to check. Assumption 4 effectively requires that, from any safe state, there exists a policy for agent i satisfying the barrier inequality against any fixed policies of others, and that all initial states are safe; plus multiple boundedness/Lipschitz assumptions (Assumptions 1–6). Practical verification is unclear.
3. Barrier signal availability. The method relies on a known state‑wise violation signal $b_i$ (set to 0 or a negative constant; Lemma 1) to train the certificate. In many real tasks, safety sets are unknown or non‑Markov, making this labeling non‑trivial.
4. Infeasibility resolution is asymptotic. Proposition 2 assumes positive‑probability satisfaction of stability inside the infeasible set and infinite updates; it does not shield against violations during training.

**Questions:**

1. How can MAS³AC deliver state‑wise zero‑violation behavior during learning under unknown dynamics, without a model, shield, or backup controller? If this is impossible in general, please (i) clarify the claim/scope to “post‑convergence guarantees,” and (ii) outline a practical safety mechanism (e.g., action filtering, reachability shield, offline‑to‑online warm‑start) that would integrate with MAS³AC.
2. Barrier signal acquisition: When the unsafe set is unknown or partially observed, how do you obtain reliable $b_i$ labels? Can your certificate be trained from implicit signals (e.g., collision proximity, rule sets) or stochastic risk estimators?
3. Sample efficiency & scaling: Multi‑timescale updates can be sample‑hungry; can authors provide the report wall‑clock / environment steps and scaling to larger N (>10) with partial observability.

---

> ### Author Response · Authors · 2025-11-25
> **Authors' response to Reviewer 37aY (1)**
>
> ### Weakness 1
> > Safety is only guaranteed after convergence. The paper motivates “zero violations at every step” for real systems, but the formal guarantees apply to the limit policy, not to learning‑time behavior; exploration can still violate constraints (no model/backup/shield). The paper emphasizes the need for per‑step safety, yet Corollary 1 removes disturbances and applies post‑convergence.
>
> **A1:** Thank you for the question. The authors agree with the reviewer that the proposed algorithm only guarantees the safety **after convergence**, and therefore, the **safety during the training process is not theoretically ensured**. The "zero violations at every step" mentioned in the original submission means, usually, a joint controller that can ensure **state-wise** safety for the controlled multi-agent system is required in many real-world safety-critical multi-agent tasks, and this requirement motivates this work: by running the proposed algorithm, a MARL-based joint controller that guarantees state-wise safety can be **finally obtained as the limit joint policy** of the update, which is supported by the theoretical analysis. Explorations during the training can violate constraints, since no known model dynamics [1] or backup controllers are involved in the proposed algorithm, which is specifically designed for model-free MARL problems. The "step" mentioned here **does not mean each training step during the learning process**, but the timestep **during the execution of the finally converged limit joint policy** whose safety can be theoretically proven. Such a statement, as well as emphasis, on "state-wise" and "every step" is to highlight the **difference between the proposed MAS$^3$AC based on neural barrier certificates, and the previous studies based on constrained MDP** [2].
>
> However, in order to efficiently obtain a good joint control policy that can achieve high rewards while effectively avoiding safety violations via updates that are **not too sensitive or with deteriorations in the performance**, the proposed algorithm specifically uses the **modified dual ascent update**, where **extra squared terms** are incorporated in the Lagrangian function (3). To demonstrate its efficacy, during the rebuttal phase, we implemented the corresponding ablation study on two tasks, with the figures available in Section J.8 of the revised PDF submission, and the statistics are provided below.
>
> **Table: Reward and cost comparisons for the ablation study on the additional squared term**
> | Algorithm   | HalfCheetah Reward                         | HalfCheetah Cost                           | Coupled HalfCheetah Reward                           | Coupled HalfCheetah Cost                              |
> | ----------- | ------------------------------------------ | ------------------------------------------ | ---------------------------------------- | ----------------------------------------- |
> | HAPPO       | $373.64 \pm 200.58$                        | $2131.21 \pm 483.13$                       | $2197.22 \pm 112.77$                     | $230.28 \pm 67.11$                        |
> | HASAC       | $\textbf{2229.41}$ $\pm$ $\textbf{335.17}$ | $211.07 \pm 300.58$                        | $2410.71 \pm 106.28$                     | $84.77 \pm 65.91$                         |
> | MACPO       | $443.50 \pm 265.72$                        | $134.58 \pm 142.09$                        | $1104.05 \pm 356.15$                     | $9.32 \pm 8.53$                           |
> | MAFOCOPS    | $-781.55 \pm 302.59$                       | $581.41 \pm 494.46$                        | $-1865.72 \pm 117.10$                    | ${0.14}$ $\pm$ ${0.97}$    |
> | MAPPO       | $418.49 \pm 194.12$                        | $1834.08 \pm 513.17$                       | $480.86 \pm 195.30$                      | $1764.08 \pm 431.53$                      |
> | MAPPOL      | $672.25 \pm 143.48$                        | $898.40 \pm 379.02$                        | $1968.86 \pm 118.37$                     | $3.67 \pm 3.39$                           |
> | No-Squared  | $342.48 \pm 1599.78$                       | $161.24 \pm 447.73$                        | $2564.67 \pm 72.10$                      | $2.58 \pm 6.56$                           |
> | MAS$^3$AC   | $2177.90 \pm 714.48$                       | $\textbf{6.22}$ $\pm$ $\textbf{25.28}$     | $\textbf{2596.56}$ $\pm$ $\textbf{105.95}$ | $\textbf{0.03}$ $\pm$ $\textbf{0.17}$                        |

---

> ### Author Response · Authors · 2025-11-25
> **Authors' response to Reviewer 37aY (2)**
>
> ### Weakness 1 (continued)
>
> Based on the figures and statistics, it can be found that with the extra squared terms, the reward obtained will **not experience noticeable deteriorations that are difficult to recover** (for HalfCheetah, the reward curve finally falls down without efficient recovery when no extra squared terms are used. This squared term design also applies to stability constraints beyond safety constraints, and therefore, the reward curve also shows its efficacy). Regarding safety, the version with extra squared terms generally achieves better safety satisfaction **during the learning process on both tasks**. This design is specifically proposed as an effort to minimize the safety violations, even during the learning time.
>
> Further, regarding the disturbance, in this work, we are investigating **fully autonomous systems** where the control input is generated solely by the RL-based joint controller that uses the system's own state as the input. In such a setting, the $\boldsymbol{d}_t$ is considered as a pure disturbance if included, and as an early attempt to integrate ISS with MARL, we simplify the analysis and implementation by treating disturbances as negligible. However, during the rebuttal phase, we conducted an **additional experiment** to study how external disturbances affect system's performance in both reward and safety satisfaction. To avoid repetition and reduce the length of the overall responses, we kindly refer the reviewer to the response to Question 7 raised by Reviewer D2kU. The results show that even under strong disturbances compared to commonly-used baselines under no disturbance during the whole training, which demonstrates the good performance of MAS$^3$AC both in the training and post-convergence.
>
> ---
>
> ### Weakness 2
> > Assumptions are strong and hard to check. Assumption 4 effectively requires that, from any safe state, there exists a policy for agent i satisfying the barrier inequality against any fixed policies of others, and that all initial states are safe; plus multiple boundedness/Lipschitz assumptions (Assumptions 1–6). Practical verification is unclear.
>
> **A2:** Thanks for the reviewer's question regarding the assumptions. The authors put an emphasis on assumptions and use a section (Appendix B) to summarize all assumptions with their rationales in the original submission to be rigorous. These assumptions are strong, but needed for the proofs provided. Regarding Assumption 4, one small clarification is that, this assumption is imposed on **states along the trajectories**, rather than on *any* safe state in the entire state space which would indeed be stronger.
>
> For the remaining assumptions related to boundedness and Lipschitz conditions, while they can be difficult to check in practice, in implementation, several measures can be taken, and examples include:
>
> 1. Bounding the norms of reward, barrier, and cost signals through careful design, or standardizing the reward, barrier, and cost signals using mean and variance before using them for training, to bound all critic functions.
> 2. Applying techniques like gradient clipping to bound the update magnitude of the networks, which prevents exploding gradients and implicitly controls their Lipschitz constants.
> 3. Limiting the range of the Lagrangian multiplier if one of its appropriate ranges for the investigated task is known before training. Also, during the rebuttal phase, we ran experiments to plot the curves of Lagrangian multipliers, and showed that these multipliers are bounded. Corresponding figures can be found in Section J.9.

---

> ### Author Response · Authors · 2025-11-26
> **Authors' response to Reviewer 37aY (3)**
>
> ### Weakness 3
> > Barrier signal availability. The method relies on a known state‑wise violation signal $b^i$ (set to 0 or a negative constant; Lemma 1) to train the certificate. In many real tasks, safety sets are unknown or non‑Markov, making this labeling non‑trivial.
>
> **A3:** Thank you for the question. First, the authors want to discuss the "unknown" situation. Different from previous work using neural barrier certificates in a supervised-learning manner, where precise labeling of safe and unsafe states are required and such classifications will then be used to construct the loss function for the update of both the neural barrier certificate and the control policy [3], in the proposed algorithm, the neural barrier certificate is constructed as a **critic function**, and thus conceptually takes all **real future safety violations** (reflected by the barrier signal defined in Lemma 1) into consideration when being updated via the Bellman operator.
>
> Thus, this method **only requires precise detection of real safety violation occurrences**, namely agent $i$'s entering into $\boldsymbol{\mathcal{X}}\_{\text{vio}}^{\boldsymbol{\pi},i}$, which is usually easy to detect via sensors (e.g., detecting whether the car currently collides with another car via its sensors), and such information can be expressed as barrier signals collected during execution and used for training in a principled way in RL. Hence, MAS$^3$AC requires **precise labeling of $\boldsymbol{\mathcal{X}}\_{\text{vio}}^{\boldsymbol{\pi},i}$**, which is usually easy to detect via sensors since the violation actually happens and damages are caused, instead of $\boldsymbol{\mathcal{X}}\_{\text{unsafe}}^{\boldsymbol{\pi},i}$ which is the full unsafe set of agent $i$ under the joint policy $\boldsymbol{\pi}$ (Definition 2). The difficult part to detect may be the $\boldsymbol{\mathcal{X}\_{\text{irrec}}}$ (irrecoverable set), which is essentially unsafe since it eventually leads to inevitable safety violations even if violations do not currently occur, and thus should be classified as part of the unsafe set. In a supervised-learning manner, it is important to correctly classify such states within $\boldsymbol{\mathcal{X}_{\text{irrec}}}$ as unsafe, which may require prediction or human expertise beyond direct detection of current real violation occurrences, and thus sets higher requirements for sensors and demonstrations to be precise. However, the proposed method can **automatically learn the shape of the unsafe set $\boldsymbol{\mathcal{X}}_{\text{unsafe}}^{\boldsymbol{\pi},i}$**, instead of requiring knowledge of the precise shape of the unsafe set through prediction or additional demonstration, since the critic-function-based neural barrier certificates can **learn to classify irrecoverable states** (barrier ceritificate values at such states are negative), as long as the direct detection of current real violation occurrences is possible and collected as barrier signals.
>
> For "non-Markov", safe sets can exhibit non-Markovian properties in some settings. The authors are aware of these cases, and still have solutions. [4] introduced Recurrent DPG in continuous control, which uses a recurrent critic $Q(h, a)$ that explicitly conditions on history $h_t$ to solve tasks requiring memory. [5] extended this to the Deep Recurrent Q-Network, which replaces a fully-connected layer in a DQN with an LSTM to successfully integrate information through time from partial observation. [6] further isolated this factor in their R-MADDPG framework, finding that a recurrent critic is the crucial component for enabling learning in partially observable settings. Thus, a **recurrent critic** offers a clear solution to handle non-Markovian safety constraints by allowing the agent to learn a barrier certificate based on a history of observations. This further study is beyond the scope of this submission, and the authors want to investigate it in future work.
>
> ---
> ### Weakness 4
> > Infeasibility resolution is asymptotic. Proposition 2 assumes positive‑probability satisfaction of stability inside the infeasible set and infinite updates; it does not shield against violations during training.
>
> **A4:** Thank you for the question. As with earlier responses, the authors acknowledge that Proposition 2 proves a desired property for the **limit policy**, and the infeasibility resolution is asymptotic. In practice, however, once the agent encounters a **real safety violation**, the barrier certificate (as a critic function) quickly learns to assign a negative value to that infeasible state and identifies it as unsafe. The application of safety constraints will effectively prevent the controlled system from visiting such infeasible states, which will lead to a good balance between reward maximization and safety cost reduction with good sample efficiency. This has been verified through extensive experiments on various tasks, and the ablation study on the efficacy of extra squared terms.

---

> ### Author Response · Authors · 2025-11-26
> **Authors' response to Reviewer 37aY (4)**
>
> ### Question 1
>
> > How can MAS$^3$AC deliver state‑wise zero‑violation behavior during learning under unknown dynamics, without a model, shield, or backup controller? If this is impossible in general, please (i) clarify the claim/scope to “post‑convergence guarantees,” and (ii) outline a practical safety mechanism (e.g., action filtering, reachability shield, offline‑to‑online warm‑start) that would integrate with MAS$^3$AC.
>
> **A5:** MAS$^3$AC is developed for model-free MARL under unknown dynamics, and therefore **cannot** provide zero-violation behavior during learning without any model, shield, or backup controller.
> 1. We thank the reviewer for pointing out the need to clarify the scope to **post‑convergence guarantees**, and **corresponding statements** have been made in the "Contributions" part in the Introduction of the revised PDF.
> 2. The authors add new experiments during the rebuttal period. For the reachability-based safety enforcements, it is possible to replace the current barrier function-based method with the reachability-based certificate, where the max-over-time safety certificates (with a different operator for update) are used instead of the sum-over-time certificates, and the constraints are to require the value of the reachability-based certificate to be non-positive. Corresponding figures are provided in Section J.5, and **statistics are summarized in the next block**.
>
> Based on the figures and statistics, it can be observed that for safety, **using barrier certificates usually achieves better safety satisfaction than the counterpart with reachability-based certificates**. The author think the possible reason is the constraint to apply: barrier certificates require safety constraints shown as (2b) in the submission that enforce safety in an exponential manner (the barrier certificate value, if negative, is explicitly required to increase at an exponential rate to achieve safety), enabling stronger and faster safety enforcement, while the constraints using barrier certificates only requires the value to be non-positive and updated via standard gradient descent; the additional exponential mechanism for rapid safety enforcement is missing.
>
>
> For the additional efficacy of the squared terms, please see our response to Weakness 1.
>
> ---
> ### Question 2
> > Barrier signal acquisition: When the unsafe set is unknown or partially observed, how do you obtain reliable $b^i$ labels? Can your certificate be trained from implicit signals (e.g., collision proximity, rule sets) or stochastic risk estimators?
>
> **A6:** For the first part, please refer to our response to **Weakness 3**, where we explain how MAS$^3$AC only requires detection of **actual safety violation occurrences** in $\boldsymbol{\mathcal{X}}_{\text{vio}}^{\boldsymbol{\pi},i}$ (which sensors can reliably detect given damage is caused), rather than full knowledge of the unsafe set.
>
> For the second part, in our setting, there is **no stochasticity** in the barrier signal, since this signal will be a specific negative value when real safety violations actually happen, and 0 otherwise. Even being very close to the state where safety violations really occur (collision proximity), such information is not passed through the detection or via the barrier signal (therefore avoids requiring sensors capable of prediction or risk estimation, but only requires them to correctly detect the real safety violations), but automatically learned by the barrier certificate as a critic function (stochasticity is not involved in the signal, but exists in the barrier certificate as a critic function, defined in Lemma 1). Further, many previous studies apply the distribution RL [7] to represent a distribution instead of the expected value for critics (including the action-value functions and certificates) to achieve better performance; however, in our setting, regarding safety, since the values of barrier certificates for safe states are strictly 0 theoretically, and negative for unsafe states due to the setting of barrier signals (Lemma 1), use a common RL manner where certificates output the expected value suffices to identify safe states. Extending MAS$^3$AC using distributional RL or conformal prediction [8] is possible and constitutes an interesting future direction.

---

> ### Author Response · Authors · 2025-11-26
> **Authors' response to Reviewer 37aY (5)**
>
> ### Question 1 (statistics)
>
> **Table: Reward and cost comparisons between different certificates on cooperative tasks**
>
> | Algorithm     | Ant Reward                              | Ant Cost                               | HalfCheetah Reward                           | HalfCheetah Cost                           | Coupled-HalfCheetah Reward                           | Coupled-HalfCheetah Cost                           |
> | ------------- | --------------------------------------- | -------------------------------------- | -------------------------------------------- | ------------------------------------------ | ---------------------------------------- | -------------------------------------- |
> | HASAC         | $993.56 \pm 87.49$                      | $107.49 \pm 22.64$                     | $2229.41 \pm 335.17$                         | $211.07 \pm 300.58$                        | $2410.71 \pm 106.28$                     | $84.77 \pm 65.91$                      |
> | MACPO         | $-1381.66 \pm 37.86$                    | $105.24 \pm 50.66$                     | $443.50 \pm 265.72$                          | $134.58 \pm 142.09$                        | $1104.05 \pm 356.15$                     | $9.32 \pm 8.53$                        |
> | MAFOCOPS      | $-3965.01 \pm 720.78$                   | $156.47 \pm 77.84$                     | $-781.55 \pm 302.59$                         | $581.41 \pm 494.46$                        | $-1865.72 \pm 117.10$                    | $0.14 \pm 0.97$                        |
> | MAPPO         | $-354.58 \pm 203.39$                    | $322.31 \pm 64.85$                     | $418.49 \pm 194.12$                          | $1834.08 \pm 513.17$                       | $480.86 \pm 195.30$                      | $1764.08 \pm 431.53$                   |
> | MAPPOL        | $-903.98 \pm 67.67$                     | $142.52 \pm 58.00$                     | $672.25 \pm 143.48$                          | $898.40 \pm 379.02$                        | $1968.86 \pm 118.37$                     | $3.67 \pm 3.39$                        |
> | Reachability  | $\textbf{1671.62}$ $\pm$ $\textbf{77.84}$ | $78.82 \pm 63.39$                   | $1939.28 \pm 138.34$                         | $272.27 \pm 149.15$                        | $2513.78 \pm 155.96$                     | $13.15 \pm 17.28$                      |
> | MAS$^3$AC     | $1221.79 \pm 50.45$                     | $\textbf{2.10}$ $\pm$ $\textbf{6.92}$ | $\textbf{2177.90}$ $\pm$ $\textbf{714.48}$   | $\textbf{6.22}$ $\pm$ $\textbf{25.28}$     | $\textbf{2596.56}$ $\pm$ $\textbf{105.95}$ | $\textbf{0.03}$ $\pm$ $\textbf{0.17}$ |
>
> **Table: Reward and cost comparisons between different certificates on the non-cooperative Coupled HalfCheetah task**
>
> | Algorithm     | Agent 0 Reward                           | Agent 0 Cost                           | Agent 1 Reward                           | Agent 1 Cost                           | Agent 2 Reward                           | Agent 2 Cost                           |
> | ------------- | ---------------------------------------- | -------------------------------------- | ---------------------------------------- | -------------------------------------- | ---------------------------------------- | -------------------------------------- |
> | IPPO          | $902.29 \pm 386.41$                      | $451.38 \pm 305.37$                    | $904.99 \pm 389.86$                      | $451.38 \pm 305.37$                    | $905.91 \pm 388.62$                      | $451.38 \pm 305.37$                    |
> | MACPO         | $-140.28 \pm 276.72$                     | $21.48 \pm 25.26$                      | $-141.91 \pm 276.68$                     | $21.48 \pm 25.26$                      | $-142.75 \pm 276.65$                     | $21.48 \pm 25.26$                      |
> | MAPPOL        | $735.92 \pm 416.97$                      | $174.96 \pm 143.68$                    | $735.77 \pm 417.75$                      | $174.96 \pm 143.68$                    | $732.35 \pm 417.00$                      | $174.96 \pm 143.68$                    |
> | MASAC         | $2499.75 \pm 78.45$                      | $25.25 \pm 20.76$                      | $2503.90 \pm 84.95$                      | $25.25 \pm 20.76$                      | $2489.93 \pm 81.61$                      | $25.25 \pm 20.76$                      |
> | Reachability  | $2479.88 \pm 166.99$                     | $14.22 \pm 26.00$                      | $2472.27 \pm 161.99$                     | $14.22 \pm 26.00$                      | $2471.29 \pm 161.62$                     | $14.22 \pm 26.00$                      |
> | MAS$^3$AC     | $\textbf{2531.32}$ $\pm$ $\textbf{136.89}$ | $\textbf{6.35}$ $\pm$ $\textbf{14.12}$ | $\textbf{2533.62}$ $\pm$ $\textbf{141.96}$ | $\textbf{6.35}$ $\pm$ $\textbf{14.12}$ | $\textbf{2529.90}$ $\pm$ $\textbf{142.89}$ | $\textbf{6.35}$ $\pm$ $\textbf{14.12}$ |

---

> ### Author Response · Authors · 2025-11-26
> **Authors' response to Reviewer 37aY (6)**
>
> ### Question 3
> >Sample efficiency \& scaling: Multi‑timescale updates can be sample‑hungry; can authors provide the report wall‑clock / environment steps and scaling to larger N ( $\geq$ 10) with partial observability.
>
> **A7:** Thanks for the question. The authors agree that sample efficiency and scalability are two essential aspects of MARL algorithms. The statistics regarding computation time on several tasks are provided below.
>
> **Table: Wall-clock time per 1000 environment steps**
> | Benchmark / Algorithm       | MAS³AC | MACPO | MAPPOL | MAFOCOPS | HASAC | HAPPO | MAPPO |
> |-----------------------------|--------|-------|--------|----------|--------|--------|--------|
> | **Cooperative Ant**         | 6.82 s | 3.69 s | 2.33 s | 7.93 s   | 6.16 s | 1.58 s | 1.34 s |
> | **Cooperative HalfCheetah** | 4.27 s | 2.53 s | 1.61 s | 7.24 s   | 3.72 s | 0.85 s | 1.22 s |
>
>
> | Benchmark / Algorithm                 | MAS³AC | MACPO | MAPPOL | MASAC | IPPO |
> |---------------------------------------|--------|-------|--------|--------|-------|
> | **Non-cooperative Coupled HalfCheetah** | 8.45 s | 3.42 s | 2.18 s | 8.26 s | 0.85 s |
>
> Based on these statistics, MAS$^3$AC requires longer wall-clock time compared to many on-policy baselines, but achieves comparable computation time to off-policy baselines (HASAC and MASAC). This observation indicates that the kind of algorithm, whether it is on-policy or off-policy, influences the computational time required. Notably, while many baselines require less training time per 1000 steps during training, they generally underperform in reward maximization and/or safety satisfication compared to the proposed MAS$^3$AC. In practice, **early stopping** may be applied to the proposed MAS$^3$AC since it achieves better performance with fewer training steps, and thus the overall time can be reduced to be shorter.
>
>
> For scalability to larger $N$, here we modify the original non-cooperative Coupled HalfCheetah task to make it include **12 agents** in the whole multi-agent system. The results are summarized below, with corresponding figures in Section J.2.
>
> **Table: Reward and Cost per Agent (12-Agent non-cooperative decentralized Coupled HalfCheetah task) – Agents 0, 1, 2**
>
> | Algorithm | Agent 0 Reward                           | Agent 0 Cost              | Agent 1 Reward                           | Agent 1 Cost              | Agent 2 Reward                           | Agent 2 Cost              |
> | --------- | ---------------------------------------- | ------------------------- | ---------------------------------------- | ------------------------- | ---------------------------------------- | ------------------------- |
> | IPPO     | $1255.74 \pm 372.60$                      | $242.76 \pm 175.59$       | $1246.24 \pm 366.04$                     | $242.76 \pm 175.59$       | $1255.41 \pm 374.03$                     | $242.76 \pm 175.59$       |
> | MACPO    | $31.32 \pm 313.10$                        | $18.64 \pm 22.76$         | $31.77 \pm 313.63$                       | $18.64 \pm 22.76$         | $27.10 \pm 312.90$                       | $18.64 \pm 22.76$         |
> | MAPPOL   | $1154.25 \pm 397.81$                      | $46.82 \pm 32.72$         | $1152.97 \pm 398.61$                     | $46.82 \pm 32.72$         | $1153.45 \pm 398.27$                     | $46.82 \pm 32.72$         |
> | MASAC    | $\textbf{2541.37}$ $\pm$ $\textbf{63.98}$ | $26.23 \pm 21.01$         | $\textbf{2542.25}$ $\pm$ $\textbf{63.80}$ | $26.23 \pm 21.01$       | $\textbf{2541.03}$ $\pm$ $\textbf{65.82}$ | $26.23 \pm 21.01$       |
> | MAS$^3$AC | $2312.98 \pm 227.15$                     | $\textbf{10.07}$ $\pm$ $\textbf{22.83}$ | $2318.55 \pm 234.76$      | $\textbf{10.07}$ $\pm$ $\textbf{22.83}$ | $2317.08 \pm 233.75$      | $\textbf{10.07}$ $\pm$ $\textbf{22.83}$ |

---

> ### Author Response · Authors · 2025-11-26
> **Authors' response to Reviewer 37aY (7)**
>
> ### Question 3 (continued)
>
> **Table: Reward and Cost per Agent (12-Agent non-cooperative decentralized Coupled HalfCheetah task) – Agents 7, 8, 9**
>
> | Algorithm | Agent 7 Reward                           | Agent 7 Cost              | Agent 8 Reward                           | Agent 8 Cost              | Agent 9 Reward                           | Agent 9 Cost              |
> | --------- | ---------------------------------------- | ------------------------- | ---------------------------------------- | ------------------------- | ---------------------------------------- | ------------------------- |
> | IPPO     | $1277.15 \pm 332.06$                      | $218.90 \pm 168.98$       | $1290.05 \pm 337.91$                     | $218.90 \pm 168.98$       | $1289.84 \pm 335.09$                     | $218.90 \pm 168.98$       |
> | MACPO    | $0.75 \pm 268.65$                         | $17.85 \pm 21.58$         | $-2.33 \pm 268.52$                       | $17.85 \pm 21.58$         | $2.16 \pm 268.55$                        | $17.85 \pm 21.58$         |
> | MAPPOL   | $1318.95 \pm 346.96$                      | $43.45 \pm 34.27$         | $1321.36 \pm 347.10$                     | $43.45 \pm 34.27$         | $1318.80 \pm 346.23$                     | $43.45 \pm 34.27$         |
> | MASAC    | $\textbf{2513.88}$ $\pm$ $\textbf{88.55}$ | $43.84 \pm 33.44$         | $\textbf{2510.71}$ $\pm$ $\textbf{94.22}$ | $43.84 \pm 33.44$       | $\textbf{2515.29}$ $\pm$ $\textbf{89.45}$ | $43.84 \pm 33.44$       |
> | MAS$^3$AC | $2219.22 \pm 240.95$                     | $\textbf{8.63}$ $\pm$ $\textbf{19.38}$  | $2208.60 \pm 238.10$      | $\textbf{8.63}$ $\pm$ $\textbf{19.38}$  | $2220.00 \pm 238.44$      | $\textbf{8.63}$ $\pm$ $\textbf{19.38}$  |
>
>
> Consistent with earlier findings, MAS$^3$AC continues to achieve a superior balance between reward maximization and constraint satisfaction for each agent, demonstrating its effectiveness when scaling to larger multi-agent systems.
>
> We also provide results for an additional decentralized Ant task with **8 agents**. To avoid redundancy and reduce the length of this response, we kindly ask the reviewer to refer to our answer to **Weakness 3** raised by Reviewer mUAV. Thank you for your understanding.
>
> **References**
>
> [1] Cheng, R., Orosz, G., Murray, R. M., \& Burdick, J. W. (2019, July). End-to-end safe reinforcement learning through barrier functions for safety-critical continuous control tasks. In Proceedings of the AAAI conference on artificial intelligence (Vol. 33, No. 01, pp. 3387-3395).
>
> [2] Altman, E. (2021). Constrained Markov decision processes. Routledge.
>
> [3] Dawson, C., Qin, Z., Gao, S., \& Fan, C. (2022, January). Safe nonlinear control using robust neural lyapunov-barrier functions. In Conference on Robot Learning (pp. 1724-1735). PMLR.
>
> [4] Heess, N., Hunt, J. J., Lillicrap, T. P., \& Silver, D. (2015). Memory-based control with recurrent neural networks, arXiv:1512.04455v1.
>
> [5] Hausknecht, M. J., \& Stone, P. (2015, November). Deep Recurrent Q-Learning for Partially Observable MDPs. In AAAI fall symposia (Vol. 45, p. 141).
>
> [6] Wang, R. E., Everett, M., \& How, J. P. (2020). R-MADDPG for Partially Observable Environments and Limited Communication, arXiv:2002.06684v2.
>
> [7] Bellemare, M. G., Dabney, W., \& Munos, R. (2017, July). A distributional perspective on reinforcement learning. In International conference on machine learning (pp. 449-458). PMLR.
>
> [8] Angelopoulos, A. N., \& Bates, S. (2021). A gentle introduction to conformal prediction and distribution-free uncertainty quantification. arXiv preprint arXiv:2107.07511.

---

### Official Review · Reviewer_D2kU · 2025-11-01

**Soundness:** 3
**Presentation:** 3
**Contribution:** 2
**Rating:** 4
**Confidence:** 3

**Summary:**

In the paper under review, the authors propose a model-free MARL framework for safe and stable control of general multi-agent tasks. The state-wise safety constraint is handled by neural barrier functions. The stability of the multi-agent system is ensured based on control theory. Both the convergence and feasibility of the proposed framework are established theoretically. Based on the framework, a practical MARL algorithm is proposed, whose effectiveness in dealing with reward maximization and constraint satisfaction is empirically demonstrated through extensive simulation experiments.

**Strengths:**

(1) The proposed safe MARL framework can deal with state-wise safety constraints, while also addressing the stability issue in multi-agent systems.

(2) Extensive simulation experiments are conducted, and advanced baselines are compared.

**Weaknesses:**

(1) The problem setting of this work is unclear.

(2) The contributions of this work are unclear.

**Questions:**

(1) Regarding the first contribution, there exist other neural barrier certificate-based safe RL methods which are also model-free (e.g., [1]). In addition, it seems that your method also relies on the precise labeling of safe and unsafe states. What is the novelty of using neural barrier certificates in this work?

(2) It is mentioned in the related work section that one drawback of existing MARL works is the assumption of the existence of a stationary distribution. Nevertheless, the stochastic approximation theory used in this work for convergence analysis also requires this assumption.

(3) It seems that Definition 1 is only used in Problem 1. What is the relationship between Definition 1 and Definition 3?

(4) Could the authors provide a task example to explain the notations including $\delta_{ij}$, $x_{\text{desire}}$, and $d_t$?

(5) In (2a)–(2b), it seems that all agents need to update their local policies in a certain order, which is similar to HATRPO. Could the authors provide some insights here? In addition, does the feasibility of (2a)–(2b) hold naturally?

(6) Under the setting of coupled dynamics in Section 3, it is apparent that $V^i$ is a function of the joint policy instead of the local policy. Hence, it is not straightforward to obtain the results in Section 4.2, and whether the proposed framework is suitable for the decentralized training setting is questionable.

(7) It is strange that the term $\kappa(\Vert d_t \Vert)$ only appears in (4), which is not considered in the rest of the manuscript because external disturbances are assumed to be negligible.

(8) The simulation result shown in Fig. 1 is not normal. It can be found that HASAC shows similar safety satisfaction ability as safe MARL algorithms like MAPPOL and MAFOCOPS, which do not consider safety in fact. In addition, it is reported in [2] that MAFOCOPS performs well in the ant task. However, it shows the worst learning performance in this figure.

References:

[1] Yang, Y., Jiang, Y., Liu, Y., Chen, J., & Li, S. E. (2023). Model-free safe reinforcement learning through neural barrier certificate. IEEE Robotics and Automation Letters, 8(3), 1295–1302.

[2] Zhao, Y., Yang, Y., Lu, Z., Zhou, W., & Li, H. (2023). Multi-agent first-order constrained optimization in policy space. Advances in Neural Information Processing Systems, 36, 39189–39211.

---

> ### Author Response · Authors · 2025-11-25
> **Authors' response to Reviewer D2kU (1)**
>
> Here, the authors first address the reviewer’s questions, and then move to the weaknesses.
>
> ### Question 1
> > Regarding the first contribution, there exist other neural barrier certificate-based safe RL methods which are also model-free (e.g., [1]). In addition, it seems that your method also relies on the precise labeling of safe and unsafe states. What is the novelty of using neural barrier certificates in this work?
>
> **A1:** Thank you for the question. For the first part, rather than claiming to be the first to introduce the concept of “neural barrier certificates” for safe model-free learning, the main point of the first contribution is the extension of this concept to **general** (not limited to cooperative) **model-free multi-agent systems**. Different from previous studies (e.g., [1]):
>
> 1. The presence of multiple agents and evolving joint policy motivates the development of definitions such as $\boldsymbol{\mathcal{X}}\_{\text{irrec}}^{\boldsymbol{\pi},i}$ and $\boldsymbol{\mathcal{X}}\_{\text{safe}}^{\boldsymbol{\pi},i}$ for each agent $i$ (explained in more detail in the answer to Question 3) to **formalize neural barrier certificates in general MARL settings**. These definitions further enable us to analyze **the issue of infeasibility in multi-agent systems** (Proposition 2), a topic seldom investigated in prior barrier-certificate-based safe RL work.
>
> 2. The non-stationarity inherent in MARL requires developing methods that can be **theoretically proven to guarantee convergence** of the update to a **safe local Nash equilibrium**, which is a key challenge in general MARL and has been barely studied in prior barrier-certificate-based safe RL work that focuses mainly on single-agent or cooperative multi-agent settings. MARL also typically requires large amounts of data for training and can be sensitive, and thus we introduce a modified dual-ascent update for better constraint enforcement. A corresponding ablation study added during the rebuttal phase is provided in Figure 21 and Table 18, Section J.8 of the revised PDF submission.
>
> All the above points, taken together, constitute the first contribution claimed in the paper.

---

> ### Author Response · Authors · 2025-11-25
> **Authors' response to Reviewer D2kU (2)**
>
> ### Question 1 (continued)
>
> For the second part: in previous work such as [1] and [2], data is first collected from environment interaction, or human demonstrations as training samples, and then classified into safe set $X_{\mathrm{safe}}$ and unsafe set $X_{\mathrm{unsafe}}$ (notations in those studies). A neural barrier certificate is then trained in a **supervised-learning manner, where the loss function depends on this classification** of samples ($X_{\mathrm{safe}}$ or $X_{\mathrm{unsafe}}$). In contrast, the definition of neural barrier certificates in this submission:
>
> 1. Is constructed as a **critic function**, and thus conceptually takes all **real future safety violations** (reflected by the barrier signal in Lemma 1) into consideration when being updated via the Bellman operator, instead of being a normal neural network updated to purely minimize a loss function based on existing classifications in a supervised-learning manner, and thus;
>
> 2. The proposed method **only requires precise detection of real safety violation occurrences**, i.e., when agent $i$ enters $\boldsymbol{\mathcal{X}}_{\text{vio}}^{\boldsymbol{\pi},i}$, which is usually easy to detect through sensors (e.g., detecting whether the car currently collides with another car via its sensors). Such information can be expressed as barrier signals collected during execution and used for training in a principled way in RL.
>
> 3. Thus, MAS$^3$AC requires **precise labeling of $\boldsymbol{\mathcal{X}}_{\text{vio}}^{\boldsymbol{\pi},i}$ only**, which is easy to detect because the violation actually occurs and causes damage, instead of $\boldsymbol{\mathcal{X}}\_{\text{unsafe}}^{\boldsymbol{\pi},i}$ which is the full unsafe set of agent $i$ under joint policy $\boldsymbol{\pi}$ (Definition 2). $\boldsymbol{\mathcal{X}}_{\text{irrec}}$ (the irrecoverable set) is essentially unsafe because it inevitably leads to future safety violations even if no violation occurs now, and therefore should be labeled as unsafe. To train the barrier certificates in a supervised-learning manner, it is important to correctly classify such irrecoverable states within $\boldsymbol{\mathcal{X}}\_{\text{irrec}}$ as unsafe, which may require prediction or human expertise beyond direct detection of current and instantaneous real violation occurrences, placing higher demands on sensors or demonstrations. In contrast, the proposed method can **automatically learn the shape of $\boldsymbol{\mathcal{X}}\_{\text{unsafe}}^{\boldsymbol{\pi},i}$**, instead of requiring knowledge of the precise shape of the unsafe set through prediction or additional demonstration, because the critic-based neural barrier certificate can **learn to classify irrecoverable states** (the barrier certificate values at such states are negative), as long as current real violation occurrences are detectable and collected as barrier signals (more explanations of the relationships between sets are provided in the answer to Question 3.)
>
> 4. Additionally, since the neural barrier certificate is a critic function, the Bellman operator is used for its updates, and the contraction mapping theorem can be applied to establish convergence of the barrier certificate update, which further supports the overall convergence analysis of the proposed algorithm.
>
> In summary, the reviewer is correct that our method also relies on precise labeling. Although this labeling is required not for the whole unsafe set, but only the easier-to-detect part ($\boldsymbol{\mathcal{X}}\_{\text{vio}}^{\boldsymbol{\pi},i}$) to enable the automatic shaping of the complete unsafe set $\boldsymbol{\mathcal{X}}_{\text{unsafe}}^{\boldsymbol{\pi},i}$ via critic-function-based barrier certificates, to be rigorous, we removed the original claim regarding precise labeling in the revised PDF. We also note that some works use reachability-based safety certificates for safe RL; we added experiments comparing barrier certificate-based and reachability-based approaches in the rebuttal (results provided in Figure 18 and Table 14 and 15, Section J.5).

---

> ### Author Response · Authors · 2025-11-25
> **Authors' response to Reviewer D2kU (3)**
>
> ### Question 2
> >It is mentioned in the related work section that one drawback of existing MARL works is the assumption of the existence of a stationary distribution. Nevertheless, the stochastic approximation theory used in this work for convergence analysis also requires this assumption.
>
> **A2:**  After rereading the original submission, we agree that some sentences were imprecise and potentially misleading. By stating “this relies on the assumption of ergodicity that requires a unique stationary distribution” and “however, the assumption on the existence of a final unique stationary distribution may not easily hold in MARL” in the original submission, we were not questioning the assumption of ergodicity itself, which requires a unique stationary distribution under a certain joint policy $\boldsymbol{\pi}$ [3], but stating that, in [4], the stability guarantee **implicitly relies on the assumption that learning converges to a fixed final joint policy**. Once learning converges, since the MDP under this final joint policy is ergodic according to the assumption, a corresponding unique stationary distribution exists and is used for stability analysis. However, in MARL, non-stationarity often prevents the multi-agent system from converging to a final fixed joint policy. Without convergence, the final unique stationary distribution in general multi-agent settings may not be well-defined.
>
> Thus, it will be helpful if certain mechanisms can be designed to ensure a final stationary distribution can be obtained by guaranteeing the learning converges to a feasible and locally optimal joint policy for the multi-agent system, without suffering from the non-stationarity and therefore failing to converge smoothly. Hence, this work applies the multi-timescale approach for the update, supported by a theoretical guarantee of convergence to a **feasible local Nash equilibrium**, which can induce the corresponding final unique stationary distribution.
>
> The reviewer is correct that stochastic approximation theory also requires the existence of a stationary distribution under a certain policy. Our point is that **non-stationarity may prevent convergence to such a fixed final joint policy**, not that the ergodicity assumption is problematic. In our stochastic-approximation-based analysis, the multi-timescale design helps reduce the multi-agent analysis to multiple single-agent analyses, thereby mitigating the issue of non-stationarity.
>
> We have revised the corresponding sentences in the updated PDF to be more precise:
> (1) “RL has been integrated with Lyapunov functions to help guarantee stability for single-agent tasks [3], but this relies on a final unique stationary distribution, which may limit its extension to MARL tasks where non-stationarity is a well-known issue that may prevent the learning from converging to a final fixed joint policy.”
> (2) “[4] apply Lyapunov functions in RL to provide stability guarantees; however, the requirement of a final unique stationary distribution may not easily be satisfied in MARL since non-stationarity may make it difficult for the update to converge towards a final fixed joint policy” for rigor.
>
> ---
> ### Question 3
> >It seems that Definition 1 is only used in Problem 1. What is the relationship between Definition 1 and Definition 3?
>
> **A3:** Definition 1 provides the strict definition of unsafe states for the whole multi-agent system: besides the states within $\boldsymbol{\mathcal{X}\_{\text{vio}}}$ where the real safety violation occurs instantaneously for at least one agent, only states where there exists no feasible joint controller within $\boldsymbol{\Pi}$, which denotes the set of all joint policies, that can be used to avoid future safety violations for all agent $i \in \mathcal{N}$, will be classified as $\boldsymbol{\mathcal{X}\_{\text{irrec}}}$. Thus, $\boldsymbol{\mathcal{X}\_{\text{unsafe}}}=\boldsymbol{\mathcal{X}\_{\text{vio}}}\cup\boldsymbol{\mathcal{X}\_{\text{irrec}}}$ is the union set of all states where safety cannot be guaranteed under any controller, and therefore is the most conservative description of the unsafe set for the multi-agent system (for any specific joint policy $\boldsymbol{\pi}$, the set of states where the safety cannot be maintained under this $\boldsymbol{\pi}$ will include $\boldsymbol{\mathcal{X}\_{\text{unsafe}}}$ as one subset). Therefore, $\boldsymbol{\mathcal{X}\_{\text{safe}}} = \boldsymbol{\mathcal{X}} \setminus \boldsymbol{\mathcal{X}\_{\text{unsafe}}}$ is **the most optimistic description** of the safe set for the multi-agent systems. Hence, concepts like $\boldsymbol{\mathcal{X}\_{\text{safe}}}$ provided in Definition 1 denote the largest possible safe space allowing exploration and interaction, and thus the corresponding feasible set (policy is the decision variable) is the largest, which enables us to find the optimal feasible joint policy that achieves higher rewards. Therefore, concepts proposed in Definition 1 are used in Problem 1.

---

> ### Author Response · Authors · 2025-11-25
> **Authors' response to Reviewer D2kU (4)**
>
> ### Question 3 (continued)
>
> However, since this algorithm is developed by training the safety certificate as a critic function jointly with the controller of each individual agent $i$, here concepts in Definition 2, like $\boldsymbol{\mathcal{X}}\_{\text{safe}}^{\boldsymbol{\pi},i}$, are proposed to define the safe and unsafe sets of each agent $i$ under certain joint policy $\boldsymbol{\pi}$, and it can be known that $\cap\_{i \in \mathcal{N} }\boldsymbol{\mathcal{X}}\_{\text{safe}}^{\boldsymbol{\pi},i}$ is the safe set of the whole multi-agent system under $\boldsymbol{\pi}$. Based on the above analysis, we also have $\cap\_{i \in \mathcal{N} }\boldsymbol{\mathcal{X}}\_{\text{safe}}^{\boldsymbol{\pi},i} \subseteq \boldsymbol{\mathcal{X}\_{\text{safe}}}$ where $\boldsymbol{\mathcal{X}\_{\text{safe}}}$ is the largest possible safe set of the problem. Nevertheless, since this $\boldsymbol{\mathcal{X}\_{\text{safe}}}$ is **not precisely provided** by prior knowledge or external demonstrations, the algorithm cannot use $\boldsymbol{\mathcal{X}\_{\text{safe}}}$ in the training, and this $\boldsymbol{\mathcal{X}\_{\text{safe}}}$ is therefore only used in problem definition. The proposed method needs to gradually expand  $\boldsymbol{\mathcal{X}}\_{\text{safe}}^{\boldsymbol{\pi},i}, i\in \mathcal{N}$ to encourage $\cap\_{i \in \mathcal{N} }\boldsymbol{\mathcal{X}}\_{\text{safe}}^{\boldsymbol{\pi},i}$  to expand and approach $\boldsymbol{\mathcal{X}\_{\text{safe}}}$ such that the feasible set can be enlarged and better feasible solution can be found to achieve better reward maximization while maintaining the safety of the multi-agent system.
>
> Thus, two requirements arise:
>
> 1. How to approximate $\boldsymbol{\mathcal{X}}\_{\text{safe}}^{\boldsymbol{\pi},i}, i\in \mathcal{N}$ and investigate the forward invariance of each $\boldsymbol{\mathcal{X}}\_{\text{safe}}^{\boldsymbol{\pi},i}$ to maintain state-wise safety for each $i\in \mathcal{N}$, which leads to the safety of the whole multi-agent system.
>
> 2. How to expand $\boldsymbol{\mathcal{X}}\_{\text{safe}}^{\boldsymbol{\pi},i}$ for each $i\in \mathcal{N}$ to maximize $\cap\_{i \in \mathcal{N} }\boldsymbol{\mathcal{X}}\_{\text{safe}}^{\boldsymbol{\pi},i}$, which leads to a larger safe space for exploration and results in better rewards while maintaining safety.
>
> Thus, we propose Definition 3, since:
>
> 1. For all $i\in \mathcal{N}$, after obtaining its barrier certificate $H\_{\boldsymbol{\pi}}^i$ via training as a critic function (via Bellman operator, based on Lemma 1), we can approximate $\boldsymbol{\mathcal{X}}\_{\text{safe}}^{\boldsymbol{\pi},i}$ as the set of states where the barrier certificate value is non-negative ($\forall \boldsymbol{x}\_t \in \boldsymbol{\mathcal{X}}\_{\text{unsafe}}^{\boldsymbol{\pi},i}$, $H\_{\boldsymbol{\pi}}^i(\boldsymbol{x}\_t) <0$. $\forall \boldsymbol{x}\_t \in \boldsymbol{\mathcal{X}}\_{\text{safe}}^{\boldsymbol{\pi},i}$, $H\_{\boldsymbol{\pi}}^i(\boldsymbol{x}\_t) \geq0$, as in Definition 3), and the forward invariance of $\boldsymbol{\mathcal{X}}_{\text{safe}}^{\boldsymbol{\pi},i}$ is naturally ensured under the current joint policy $\boldsymbol{\pi}$ according to the third point in Definition 3.
>
> 2. Based on the barrier certificate proposed in Definition 3, it can be shown in Proposition 1 that $\boldsymbol{\mathcal{X}}\_{\text{safe}}^{\boldsymbol{\pi},i}, i\in \mathcal{N}$ can be guaranteed to be monotonically non-shrinking, which helps the expansion (at least non-shrinking) of $\cap\_{i \in \mathcal{N} }\boldsymbol{\mathcal{X}}\_{\text{safe}}^{\boldsymbol{\pi},i}$, which leads to a larger feasible set of policies.

---

> ### Author Response · Authors · 2025-11-25
> **Authors' response to Reviewer D2kU (5)**
>
> ### Question 4
> > Could the authors provide a task example to explain the notations including $\delta_{ij}$, $\boldsymbol{x}_{\text{desired}}$, and $\boldsymbol{d}_t$?
>
> **A4:** Thank you for the question. We use the **cooperative Coupled HalfCheetah** task as an example. $\delta_{ij}\in \mathcal{K} \cup\{0\}$, $\forall i, j \in \mathcal{N}$, denote the gain functions representing inter-agent couplings and will be used in the local stability constraint of each agent $i$, as shown in (4) in the paper. Formally, when agent $j$ directly influences agent $i$, $\delta_{ij} \in \mathcal{K}$, and agent $j$'s certificate $V^j$ affects the agent $i$'s local stability constraint via the term $\delta_{ij}(V^j)$. Otherwise, no direct influence exists from $j$ to $i$, and $\delta_{ij}=0$.
>
> In the cooperative Coupled HalfCheetah task, we choose **proportional functions** as the gain functions, and $\delta_{ij}$ are coefficients treated as hyperparameters. In experiments, we set all $\delta_{ij}=0.0005$ (a small value since certificates $V^j$ can have large values as critic functions, and the training can have numerical issues if $\delta_{ij}$ are also large), and the experiment results are good compared to baselines. Notably, instead of pre-defining these $\delta_{ij}$, it is also **possible to make them tunable** (e.g., make them as outputs of the policy network) and thus they can be trained via loss functions like $\left[\sum\_{j\neq i}\delta\_{ij}(V^j(x^j\_t)) + \beta_i\left(\sum\_{j\neq i}\delta\_{ij}(V^j(x^j_t))\right) - \alpha_i(V^i(x^i\_t))\right]\_+,$ alongside a second constraint
> $\left[\dot{V}^i(x^i_t) + \alpha\_i(V^i(x^i_t)) - \sum_{j\neq i}\delta_{ij}(V^j(x^j_t))\right]\_+,$ where $[\cdot]_+ = \max(0, \cdot)$ (similar to Appendix F of the original submission) and assume no exogenous signal $\boldsymbol{d}_t$ exists. This makes the algorithm further flexible without corresponding hyperparameter tuning, although the computational burden will increase.
>
> The term $\boldsymbol{x}_{\text{desired}}$ is the **desired state (goal destination)** where the controlled multi-agent system is required to finally arrive and maintain. Such requirements are common in navigation and trajectory tracking tasks. In the cooperative Coupled HalfCheetah task, the controlled multi-agent system (Coupled HalfCheetah) is required to reach and maintain a desired distance $d^\ast = 12.0$ from the movable wall along the x-axis. During the rebuttal phase, we **plot the difference** between the current state and the desired state (goal) along the environment steps, which can be found in Figure 24 and Figure 25, Section J.10. These plots clearly show that the proposed algorithm can **effectively** train the policy to drive the controlled multi-agent system towards the desired goal without noticeable divergence, and achieves better minimization of the difference compared to standard and safe MARL baselines.
>
> Finally, $\boldsymbol{d}_t$ is the exogenous signal. A more detailed discussion of $\boldsymbol{d}_t$ is provided in the response to Question 7.
>
> ---
> ### Question 5
> > In (2a)–(2b), it seems that all agents need to update their local policies in a certain order, which is similar to HATRPO. Could the authors provide some insights here? In addition, does the feasibility of (2a)–(2b) hold naturally?
>
> **A5:** Thank you for the question. Regarding the sequential update:
>
> 1. **HATRPO [5]:** This method is developed for **cooperative** multi-agent problems and uses a scalar team reward signal for the whole multi-agent system, enabling multi-agent advantage decomposition. At the beginning of each new round for a complete sequential update, the order of agents for policy update **can be permuted**, given all agents collaborate to maximize the same reward. No explicit safety constraints are imposed.
>
> 2. **Our method** is designed for **general** multi-agent problems, and no restrictions are imposed on the reward signal of each individual agent, which motivates the application of the multi-timescale approach. Within each round for a complete sequential update, **the order of agents for policy update is kept the same without permutations**. This design is due to the multi-timescale requirement, which is described as Assumption 2 in Appendix B. If the learning rates of policy updates are designed to satisfy $\eta_{2,i}> \eta_{2,i-1}$, $\forall i \in \{2, \dots, N\}$, in theoretical analysis, this indicates when agent $i$ updates its policy, all agents $j \in \{1, \dots, i-1\}$ should have updated their policies adequately and converged. The proposed implementation, correspondingly, aims to perform the update of all agents $j \in \{1, \dots, i-1\}$ **before the update of agent $i$** in every round of complete sequential update to align with the theoretical interpretation. Since the learning rates are pre-defined before running the experiments, the corresponding required order for policy update is therefore fixed and obeyed during the whole training.

---

> ### Author Response · Authors · 2025-11-25
> **Authors' response to Reviewer D2kU (6)**
>
> ### Question 5 (continued)
>
> Regarding the feasibility of (2a)–(2b): It is **not held naturally**. In the original submission, the authors proposed Assumption 4 in Appendix B, which requires the feasibility of Problem (2) when agent $i$ updates its policy while all other agents follow a certain joint policy $\boldsymbol{\pi}^{-i}$. This assumption rules out scenarios where the environment, including other agents’ fixed policies, renders safety of agent $i$ impossible, namely situations where no admissible action can help to satisfy safety requirements for agent $i$. It ensures the problem is well-posed and that a safe policy is learnable and achievable. The requirement that all initial states lie within the safe set is also standard in constrained control setups [2], as starting from unsafe states would inevitably lead to safety constraint violations.
>
> ---
> ### Question 6
> > Under the setting of coupled dynamics in Section 3, it is apparent that $V^i$ is a function of the joint policy instead of the local policy. Hence, it is not straightforward to obtain the results in Section 4.2, and whether the proposed framework is suitable for the decentralized training setting is questionable.
>
> **A6:** Thank you for the question. The reviewer is correct that $V^i(x^i_t) = \mathbb{E}\_{\tau^i \sim \pi^i}\big[\sum_{k=0}^{\infty}\gamma^{k}c^i(x^i_{t+k})\big]$, where $\tau^i$ is the trajectory of agent $i$ starting from $x^i_t$, is affected by and therefore a function of the joint policy. The influence from other agents is due to the policy $\pi^i$ that can use either global information or local observations as inputs, and therefore states and actions of other agents, which is driven by the joint policy $\boldsymbol{\pi}^{-i}$ (or part of it when only local observations are possible), will be incorporated into agent $i$'s decision making, leading to various following trajectories and thus influence $V^i$ via different sequence of $c^i(x^i_{t+k})$ and Bellman operator-based updates. However, the authors want to make the following clarifications:
>
> 1.  Given a certain joint policy $\boldsymbol{\pi}$, similar to some previous work including HATRPO [5], we have $V^i(x^i_t)=\mathbb{E}_{\boldsymbol{x}_t^{-i} \sim \boldsymbol{\pi}^{-i}}\big[  V^i(x^i_t,\boldsymbol{x}_t^{-i})  \big]$ where $\boldsymbol{x}_t^{-i}$ is the **joint state of all other agents** and driven by $\boldsymbol{\pi}^{-i}$. This means the Lyapunov certificate of agent $i$, when only using its own state $x^i_t$ as the input and the joint control policy is a fixed $\boldsymbol{\pi}$, can be understood as an **expected value** over the distribution of all other agents. Further;
>
> 2. Theorem 2 in Section 4.2 aims to propose stability constraints **for a certain given joint policy $\boldsymbol{\pi}$**: if the given fixed $\boldsymbol{\pi}$ satisfies the constraint shown as (4) in the submission for each agent $i$, then the controlled multi-agent system is input-to-state stable, which is the desired property. All $V^i$s used in (4) can be interpreted as the expected values (described above) over all other agents' stationary distributions induced by such a fixed joint policy (as pointed out in the response to Question 2, ergodicity is also required for this work). This is also reflected by the phrase "under certain joint policy $\boldsymbol{\pi}$" in Theorem 2.
>
> 3. The question is how to obtain such $\boldsymbol{\pi}$ satisfying these stability constraints through training. One key consideration is to **clearly obtain $V^i$s to correctly have the constraints**, and this is achieved by the application of the multi-timescale approach: when updating the Lyapunov certificates, all policy updates are at a slower timescale and thus these $V^i$s can be updated under the current fixed joint policy. With the cost signals $c^i$ collected from the distribution induced by this current joint policy (in implementation, even if the setting is decentralized, each agent can collect its own $c^i$ since this is agent $i$'s own local signal, and the sequence of $c^i$ along the time encodes other agents' behavior during the execution), the obtained Lyapunov certificates $V^i$s, trained via Bellman operator-based updates and use $x^i_t$ as the input, are the expected values over all other agents' stationary distributions given the collected data is rich. Such $V^i$s are then utilized to update the policy, and this iteration continues until convergence under the multi-timescale setting.

---

> ### Author Response · Authors · 2025-11-25
> **Authors' response to Reviewer D2kU (7)**
>
> ### Question 6 (continued)
>
> 4. Notably, when the collected data is not rich enough, the critic updates may not result in the correct expected values needed, and there **exists an error between** the desired correct $V^i$ and the utilized surrogates $V_{\chi_i}$ where $\chi_i$ is the parameter. **This can further cause errors in the policy update, and is usually common in decentralized training settings**. We also pointed it out in the final section of the original submission as one limitation. The authors are currently working on this issue, trying to **provide an upper bound** on the error. Author's ongoing investigation indicates that there are several possible measures to make the upper bound smaller, including standardizing the cost signals using the mean and variance, applying techniques like gradient clipping to bound the update magnitude of the networks, etc. The authors hope to finish this further investigation promptly.
>
> Finally, the authors agree that, as pointed out by the reviewer, despite the aforementioned analysis,  even if written as $\tau^i \sim \pi^i$ in the definition of $V^i$, policy $\pi^i$ **can incorporate other agents' information**, which makes this $V^i$ a function of the joint policy. In fact, this has been reflected in $5(c)$ where the joint policy is used as the subscript. However, there was an inconsistency in the expression of $V^i$, and therefore, we modified its definition in the new PDF submission. Thank you for pointing this out.
>
> ---
>
> ### Question 7
> > It is strange that the term $\kappa_i(||\boldsymbol{d}_t ||)$ only appears in (4), which is not considered in the rest of the manuscript because external disturbances are assumed to be negligible.
>
> **A7:** Thank you for the question. In Theorem 2, we specifically include the external disturbance $\boldsymbol{d}_t$, and its influence on stability via terms $\kappa_i(||\boldsymbol{d}_t ||)$, **for completeness**. Classical input-to-state stability (ISS) analysis uses the external signal as the input to the system, and thus, it is necessary to analyze the influence of such external input on the system's stability. Nevertheless, in this work, we are investigating **fully autonomous systems** where the control input is generated solely by the RL-based joint controller that uses the system's own state as the input. In such a setting, the $\boldsymbol{d}_t$ is considered as a pure disturbance if included, and as an early effort to combine ISS with MARL, the authors chose to simplify the analysis and implementation by treating this external disturbance $\boldsymbol{d}_t$ as negligible.
>
> Nevertheless, during the rebuttal phase, we **implement an additional experiment to investigate how the external disturbance affects** the system's performance reflected by the reward. We use the cooperative Ant task, and apply external disturbances (random torques) on its 8 rotors. Give the control input's maximum magnitude is 1 for this task, four settings are tested, where the external disturbance's maximum magnitude $||\boldsymbol{d}||_{\infty} = \text{sup}\{ ||\boldsymbol{d}_t||,t\geq 0  \}$ are set to be 0 (no external disturbances), 0.1 (small external disturbances), 0.25 (medium external disturbances) and 0.4 (strong external disturbance). **Terms $\kappa_i(||\boldsymbol{d}_t ||)$ are also incorporated into the training process**. Baselines under no external disturbances are also presented for comparison. Corresponding figures are in Section J.1, with the statistics provided below.
>
> **Table: Reward and cost for ablation study on Disturbance in Ant task**
>
> | Algorithm                     | Reward                      | Cost                      |
> | ---------------------------- | --------------------------- | ------------------------- |
> | MAPPO                        | $-834.56 \pm 317.29$        | $249.46 \pm 87.03$        |
> | MAPPOL                       | $-1039.23 \pm 95.85$        | $145.60 \pm 58.37$        |
> | Strong Disturbance  (MAS$^3$AC)          | $-73.35 \pm 100.57$         | $7.86 \pm 12.35$          |
> | Medium Disturbance (MAS$^3$AC)           | $550.07 \pm 194.18$         | $4.02 \pm 8.27$           |
> | Small Disturbance (MAS$^3$AC)            | $1152.68 \pm 58.21$         | $1.69$ $\pm$ $5.90$      |
> | No Disturbance (MAS$^3$AC)   | $1232.87$ $\pm$ $57.45$    | $2.02 \pm 5.45$           |
>
>
>
>
> It can be found that as the disturbance becomes stronger, the algorithm's performance inevitably deteriorates in both reward maximization and safety satisfaction. However, even under the strong disturbances, the proposed algorithm still achieves better rewards and fewer safety violations compared to commonly-used baselines under no disturbance, which demonstrates the good performance of MAS$^3$AC.

---

> ### Author Response · Authors · 2025-11-25
> **Authors' response to Reviewer D2kU (8)**
>
> ### Question 8
> > The simulation result shown in Fig. 1 is not normal. It can be found that HASAC shows similar safety satisfaction ability as safe MARL algorithms like MAPPOL and MAFOCOPS, which do not consider safety in fact. In addition, it is reported in [2] that MAFOCOPS performs well in the ant task. However, it shows the worst learning performance in this figure.
>
> **A8:** Thank you for the question. For the first part, whether HASAC can have good safety satisfaction usually depends on the specific setting of the problem. The utilized tasks, as described in detail in Appendix H, are different from the original safe MAMujoco tasks provided in [6], because the stability requirement also needs to be incorporated and relates closely to the reward. For example, in the cooperative HalfCheetah task, maximizing reward is not strictly opposed to maintaining safety, and HASAC's ability to effectively explore may allow it to quickly find a good policy that can achieve high rewards and low safety violations at the same time. This is also partially because HASAC is an off-policy method that can have better sample efficiency. However, for some other tasks, HASAC failed to achieve good safety satisfaction.
>
> Regarding MAPPO-Lagrangian, prior work such as [7] notes that “It is noteworthy that the performance of MAPPO-L highly relies on the Lagrangian coefficient, rendering it more sensitive to the hyperparameters" and "This may be due to that MAPPO-Lagrangian being built upon Lagrangian multiplier combined with standard MARL algorithms, leading to a performance more similar to safety-unaware MARL algorithms”. In our experiments, we similarly observed that MAPPOL sometimes fails to reduce safety violations effectively, while in some other runs it produces smooth and efficient reductions in safety violations. This variability may arise from hyperparameter choices (e.g., safety violation threshold, learning rates) and from MAPPO-Lagrangian’s relatively low sample efficiency due to its on-policy nature, which can become pronounced when compared to off-policy methods.
>
> For MAFOCOPS, first, our experimental setup utilizes a modified reward signal: instead of the environment's original reward in [6], we incorporate a stability metric for evaluation. We figure out that this new reward structure, particularly the difficulty in maximizing stability-related returns during exploration, compromises the validity of MAFOCOPS's first-order approximation within its trust region. As a result, the second step finding the parameterized policy closest to the non-parametric optimal solution is no longer equivalent to finding the true optimal policy within the parametric space, introducing a significant projection error. Compounding this, the use of the acceptance indicator function to enforce the trust region may lead to a low effective sample size, high variance in gradient estimates, and highly unstable training.
>
> Furthermore, the algorithm's performance is highly sensitive to this new reward definition, which would require a careful hyperparameter search for the dual variables, $\lambda_j$ and $\nu_{max}$ (used in [7]). A new reward structure means the optimal $\lambda_j$ (which is fixed as a hyperparameter in MAFOCOPS) would shift significantly. Finally, the algorithm relies heavily on the heuristic for the $\nu_j$ update (Equation 10), which assumes $E_{a \sim \pi^{i_h*}}[A_j] \approx 0$. If exploration of our stability metric induces large cost-advantage fluctuations, this approximation breaks down, rendering the $\nu_j$ update rule invalid. Overall, these factors could explain the poor reward performance of MAFOCOPS under our modified environment setup.
>
> Nevertheless, after examining MAFOCOPS’s methodology in detail and rerunning its experiments on the original tasks, the authors acknowledge that MAFOCOPS is a highly effective MARL algorithm with strong performance in both reward and safety satisfaction. It therefore serves as a solid baseline and contributes meaningfully to advancing research in safe MARL.
>
> ---
> ### Weaknesses
> > (Both 1 and 2) The problem setting of this work is unclear. The contributions of this work are unclear.
>
> **A9:** The authors hope that the aforementioned responses to questions regarding the application of neural barrier certificates, the assumption of the existence of a stationary distribution, the relationship between Definition 1 and Definition 3, notations used in the ISS constraint, order of policy update, application of $V^i$, discussions on the external disturbances $\boldsymbol{d}_t$, and the discussion on certain baselines can help clarify the overall problem setting and the contributions of this work. Please feel free to raise any additional questions.

---

> ### Author Response · Authors · 2025-11-25
> **Authors' response to Reviewer D2kU (9)**
>
> **References:**
>
> [1] Yang, Y., Jiang, Y., Liu, Y., Chen, J., \& Li, S. E. (2023). Model-free safe reinforcement learning through neural barrier certificate. IEEE Robotics and Automation Letters, 8(3), 1295–1302.
>
> [2] Dawson, C., Qin, Z., Gao, S., \& Fan, C. (2022, January). Safe nonlinear control using robust neural lyapunov-barrier functions. In Conference on Robot Learning (pp. 1724-1735). PMLR.
>
> [3] Han, M., Zhang, L., Wang, J., \& Pan, W. (2020). Actor-critic reinforcement learning for control with stability guarantee. IEEE Robotics and Automation Letters, 5(4), 6217-6224.
>
> [4] Yao, J., Han, M., \& Yin, X. (2025). Lyapunov-based distributed reinforcement learning control with stability guarantee. Computers \& Chemical Engineering, 195, 108979.
>
> [5] Grudzien Kuba, J., Chen, R., Wen, M., Wen, Y., Sun, F., Wang, J., \& Yang, Y. (2022). Trust region policy optimisation in multi-agent reinforcement learning. In International Conference on Learning Representations.
>
> [6] Gu, S., Kuba, J. G., Chen, Y., Du, Y., Yang, L., Knoll, A., \& Yang, Y. (2023). Safe multi-agent reinforcement learning for multi-robot control. Artificial Intelligence, 319, 103905.
>
> [7] Zhao, Y., Yang, Y., Lu, Z., Zhou, W., \& Li, H. (2023). Multi-agent first-order constrained optimization in policy space. Advances in Neural Information Processing Systems, 36, 39189–39211.

---

### Author Response · Authors · 2025-12-01
**The Summary Comment to the AC (2)**

Below, we summarize our responses to each reviewer.

**Reviewer D2kU** mainly asks questions regarding concepts and interpretations. To respond, the authors clearly explain the **concepts** used in the submission, including the notations and definitions for the general multi-agent safe and stable control setting with state-wise guarantees, and the **role and contribution of neural barrier certificates in enforcing such guarantees in multi-agent systems**. The concepts related to the input-to-state stability (**ISS**, for Questions 4, 6, and 7), which is **new to MARL studies**, are also clarified carefully. We respond through multiple approaches, including **illustrative examples** (for Questions 1 and 4), **theoretical analysis and interpretation** (for Question 2, 3, and 6), **key insights and comparisons with previous studies** (for Questions 1, 5, and 8), together with **specific additional experiments and visualizations** (for Questions 4 and 7) that support the claim made in the submission.


**Reviewer 37aY** provides positive evaluations from many perspectives, but asks the authors to clearly clarify that the constraint satisfaction guarantee holds after the convergence. The reviewer also raises the question regarding assumptions, and the factual implementation issue of how the barrier signal is obtained. In response, the authors **clearly state the scope (post‑convergence guarantees)** in the revised submission, emphasize the extensive **summary of all utilized assumptions together with their rationales** in Appendix B of the original submission for rigor, and include **practical measures** that can be taken in implementation to satisfy the assumptions as much as possible. Different "unknown" and "non‑Markov" situations are also thoroughly discussed. Additional ablation studies showing that MAS$^3$AC can **improve constraint satisfaction efficiently during training** to minimize safety violations, even before convergence, are included, and an additional task is provided to demonstrate that the algorithm can **scale to larger multi-agent systems** to address Question 3.

**Reviewer mUAV** mainly asks the authors to provide an **extra baseline** and additional non-cooperative tasks; otherwise, the generality claim may not hold. In response, the authors add a **modified recent decentralized constrained MARL method** accepted at NeurIPS 2024 as an additional strong baseline. Notably, Reviewer mUAV commented that "the only non-cooperative task evaluated is Coupled HalfCheetah". The authors want to **clarify that the original submission already included a decentralized non-cooperative multi-agent electrical bidding task**, whose system dynamics are fundamentally different from robotic control tasks. This task serves as an effective complementary non-cooperative benchmark to the non-cooperative Coupled HalfCheetah, as supported by Point 4 of the Strengths in Reviewer 37aY's original comments. Additionally, the authors **introduce a decentralized non-cooperative Ant task** as an extra benchmark to further evaluate MAS$^3$AC under non-cooperative settings.

**Reviewer yTHj**'s suggestions regarding **more experimental results** further helps consolidate the submission and highlight the effectiveness of various proposed design components. The authors have completed **all required tasks**, providing corresponding plots and statistics to strengthen the claims and make the submission more comprehensive. All the added experiments can be found in **Section J in the appendix of the revised submission**.

Overall, the reviewers’ comments helped the submission **strengthen its clarity and add further ablation studies to consolidate its contributions, and none of the reviewers questioned the importance of the investigated problem or proposed that the overall proposed MAS$^3$AC framework lacks novelty (clearer clarifications and explanations on certain parts were requested)**. The authors answered all concerns directly, thoroughly, and transparently, supported by theoretical analysis, comprehensive new experiments, and relevant extensions where appropriate. **The authors hope that these responses satisfactorily address all questions, and we sincerely thank the area chair for the time and effort devoted to evaluating this submission**.

---

### Author Response · Authors · 2025-12-01
**The Summary Comment to the AC (1)**

We sincerely thank the area chair for handling this submission, and all reviewers for their feedback. **Since the discussion period was cut short, reviewers did not have the opportunity to reply to our responses, but we have made considerable efforts to address all concerns comprehensively and provide structured and clear explanations and experiments that are easy to follow. We appreciate the area chair’s time in considering these responses, which we believe help clarify and consolidate the contributions of our work**. Below, we summarize:
1. The main content of the submission,
2. The main questions raised by the reviewers, and
3. How our responses and revisions address these questions.

In this submission, we propose **MAS$^3$AC, the first general multi-agent reinforcement learning (MARL) framework that enforces state-wise safety via neural barrier certificates and stability via input-to-state stability (*ISS*)-based *local* conditions, a concept that is *new* to MARL studies, for both cooperative and non-cooperative settings**. We also conduct an investigation of the **issue of infeasibility** that can arise when enforcing simultaneous state-wise safety and stability constraints. Empirically, we show that MAS$^3$AC **consistently achieves a favorable balance between reward maximization and safety satisfaction** compared to baselines, across cooperative and non-cooperative MARL tasks, under centralized and decentralized schemes.

This work combines substantial theoretical analysis with extensive empirical evaluation. The submission provides **detailed algorithmic analysis with theoretical proofs** (as acknowledged by **all** reviewers), together with **broad empirical validation across different training schemes** (centralized and decentralized) and **application domains** (robotics tasks and a new multi-agent electrical bidding task), which is **recognized positively by most** reviewers (D2kU, 37aY, and yTHj). The presentation quality is also generally **regarded as good by most reviewers**, and no reviewer questioned the importance of the investigated problem.

Generally, reviewers' feedback focused on **clarifications (mainly by Reviewer D2kU) regarding the problem setting and concepts used to demonstrate contributions, and adding supplementary experiments and ablations (mainly by Reviewer mUAV and Reviewer yTHj) to better support the claims** backed by theoretical guarantees. To respond, we addressed these points with clear explanations and additional experiments to better clarify the framework and consolidate the original claims. The length of our responses reflects our effort to address all questions as clearly as possible through direct explanations and extensive empirical evidence, while keeping the original core contributions and theoretical guarantees unchanged. **In sum, the authors addressed the reviewers’ comments through the following efforts:**

1. Clarified **the notations and definitions for the *general* multi-agent safe and stable control problem setting with state-wise guarantees**, and clarified the **contribution of extending neural barrier certificates to multi-agent systems, supported by theoretical analysis on formulation, convergence, and the issue of infeasibility**. The post-convergence nature of the constraint satisfaction guarantee is also made explicit.

2. **Expanded conceptual explanations**, including barrier signals and the role of ISS-based conditions, with clearer interpretations and examples.

3. Highlighted that the **original submission already evaluated MAS$^3$AC on a decentralized non-cooperative electrical bidding task** (also supported by Reviewer 37aY, Point 4 of the Strengths), whose dynamics are fundamentally different from those of robot control. **Thus, the concern, raised as a weakness by Reviewer mUAV, that "the only non-cooperative task evaluated is Coupled HalfCheetah", *does not apply***.

4. **Added extensive supplementary experiments and ablations in Appendix J**, covering robustness to external disturbances, barrier versus reachability for safety, the impact of input-to-state stability conditions on MAS$^3$AC, the necessity of the squared terms in the modified dual ascent update, and the evolution of Lagrangian multipliers, to further support the claims made in the original submission.

5. **Included an additional strong decentralized constrained MARL baseline adapted from a recent NeurIPS 2024 method** (therefore **seven** baselines for cooperative tasks and **five** baselines for non-cooperative tasks), and **added a decentralized non-cooperative Ant task** for broader empirical coverage.

6. Introduced an additional **multi-agent task with more agents ($N>10$)** to illustrate the **scalability** of MAS$^3$AC.

7. Made **minor clarifications and organizational improvements** in the introduction and related sections (e.g., highlighting the challenges in the field of constrained RL) to more clearly convey the scope and contributions of the work.

---

### Meta-Review · Area_Chair_rC9d · 2026-01-06

**Summary:**

The reviewers acknowledge the importance of the problem addressed—unifying state-wise safety and stability within a general-sum Multi-Agent Reinforcement Learning (MARL) framework. The proposed MAS³AC algorithm utilizes neural barrier certificates and Input-to-State Stability (ISS) constraints, backed by a multi-timescale convergence analysis. However, the initial submission faced questions regarding its clarity and the appropriateness of its experimental setup. Major concerns included: (1) the lack of clarity regarding the theoretical novelty compared to existing single-agent barrier methods; (2) the highly restrictive nature of the multi-timescale assumptions required for convergence; (3) a disconnect between the "state-wise safety" claims and the use of CMDP-based baselines; and (4) the lack of safety guarantees during the training phase, which contradicts the motivation for many safety-critical applications.

Therefore, the consensus decision on this paper is a (borderline) rejection.

**Reviewer Concerns:**

The authors provided an exceptionally thorough rebuttal, including extensive new experiments (e.g., 12-agent scaling, 8-agent non-cooperative tasks, and new baselines like Scal-MAPPO-L). They successfully clarified the scope of their safety guarantees as being post-convergence and provided ablation studies for the additional squared Lagrangian terms. However, several core concerns remain outstanding. Most notably, the multi-timescale update requirement (where each agent’s policy must converge before the next agent updates) remains a significant practical bottleneck that limits the decentralized claim's real-world utility. Furthermore, while the authors defended the use of sum-over-time barrier signals, the conceptual critique from reviewers regarding whether this formulation truly achieves state-wise safety better than existing max-over-time (Hamilton-Jacobi) or standard CMDP approaches was not definitively resolved. The reliance on Assumption 4—which essentially assumes the existence of a safe policy—is viewed by the committee as a strong and hard-to-check requirement that limits the paper's theoretical impact in model-free settings.

**Reviewer Scores:**

The initial ratings were highly polarized (6, 4, 4, 4).Reviewer yTHj raised fundamental concerns about the sum-over-time formulation and the validity of stability claims with learned value functions. While the authors' empirical responses were strong, this reviewer’s score would likely have stayed at 4, as the readability issues and conceptual disagreement on the barrier signal remain. Reviewer mUAV and Reviewer D2kU were concerned with the lack of challenges articulation and the ordering of learning rates. Given the authors' detailed response to the learning rate rationale and the addition of the Challenges section, these scores likely would have improved to 5 or 6. Reviewer 37aY was the most positive but noted the lack of during-training safety. Despite the authors' impressive effort during the rebuttal, the paper is recommended for rejection. The core theoretical framework relies on assumptions that are too restrictive for the general MARL claim, and the fundamental shift from the CMDP paradigm to state-wise safety remains empirically blurred. The paper would benefit from a significant improvement of presentation that integrates the rebuttal's findings into the main text and relaxes the rigid multi-timescale requirements.

---

### Decision · Program_Chairs · 2026-01-26

Reject